# Finite-Time Convergence Rates in Stochastic Stackelberg Games with Smooth Algorithmic Agents

Eric Frankel [1]   Kshitij Kulkarni [2]   Dmitriy Drusvyatskiy [1]   Sewoong Oh [1]   Lillian J. Ratliff [1]

## Abstract

Decision-makers often adaptively influence downstream competitive agents' behavior to minimize their cost, yet in doing so face critical challenges: $(i)$ decision-makers might not *a priori* know the agents' objectives; $(ii)$ agents might *learn* their responses, introducing stochasticity and non-stationarity into the decision-making process; and $(iii)$ there may be additional non-strategic environmental stochasticity. Characterizing convergence of this complex system is contingent on how the decision-maker controls for the tradeoff between the induced drift and additional noise from the learning agent behavior and environmental stochasticity. To understand how the learning agents' behavior is influenced by the decision-maker's actions, we first consider a decision-maker that deploys an arbitrary sequence of actions which induces a sequence of games and corresponding equilibria. We characterize how the drift and noise in the agents' stochastic algorithms decouples from their optimization error. Leveraging this decoupling and accompanying finite-time efficiency estimates, we design decision-maker algorithms that control the induced drift relative to the agent noise. This enables efficient finite-time tracking of game theoretic equilibrium concepts that adhere to the incentives of the players' collective learning processes.

## 1. Introduction

Decision-making not only under uncertainty but also in environments with competitive learning agents arises quite naturally and frequently in machine learning applications (Cai et al., 2015; Dean et al., 2024; Kim & Perdomo, 2023;

Westenbroek et al., 2019). For example, recommendation systems deploy predictive models of engagement to encourage user interactions (Calvano & Polo, 2021; Hardt et al., 2022), crowd-sourcing markets leverage incentives to elicit responses (Dasari et al., 2020; Hu et al., 2018; Scheid et al., 2024; Shah & Zhou, 2016; Xie et al., 2014), and in multi-agent systems, control policies influence outcomes of agent competition (Ho et al., 1982; Ratliff & Fiez, 2020; Yang et al., 2022). Common to these domains is the assumption that agent preferences are *fully known* and their behavior is *stationary*: the decision-maker knows how agents will react and expects them to react the same over time. Yet, in practice this assumption is frequently violated, particularly when competitive agents *adapt* their response to a decision-maker's action (Fiez et al., 2020; Liu & Ratliff, 2024; Miller et al., 2021; Narang et al., 2023; Perdomo et al., 2020).

A natural abstraction between the decision-maker and agents is a Stackelberg game, where the decision-maker takes the role of *leader* and the agents take the role of *followers* (Stackelberg et al., 1952). Stackelberg games can be viewed as a bilevel optimization problem wherein the decision-maker seeks to optimize its objective subject to a variational inequality that captures the equilibrium behavior of the agents. As a class of convex structured problems, monotone variational inequalities have garnered much attention in machine learning due to their application to classical (Combettes et al., 2023; Ghadimi & Lan, 2015) and modern problems such as adversarial learning (Fiez et al., 2020; 2021a; Gidel et al., 2019; Goodfellow et al., 2014), robust and multi-agent reinforcement learning (Foerster et al., 2018; Pinto et al., 2017; Zheng et al., 2022), auction theory (Syrgkanis et al., 2015), and recently to fine-tune large language models (Amini et al., 2024; Yang et al., 2024).

In the setting we consider, the agents are *learning* and therefore adapting their behavior for any deployed decision-maker action. This alone makes the decision-maker's objective time-varying: indeed, even for a fixed decision-maker's action deployed over a time horizon, the agents' learning process may not have stabilized at an equilibrium, much less for a sequence of decision-maker actions. Moreover, the *environment* itself is also stochastic and is unknown to the decision-maker beyond query access. Ignoring that the

---

[1]University of Washington, Seattle [2]University of California, Berkeley. Correspondence to: Eric Frankel <ericsf@cs.washington.edu>.

*Proceedings of the 42nd International Conference on Machine Learning*, Vancouver, Canada. PMLR 267, 2025. Copyright 2025 by the author(s).

distribution shift arises from competing agents, the decision-maker's objective is reminiscent of stochastic tracking problems (Borkar, 2009; Cutler et al., 2023; Kusher & Yin, 1997). Similarly, if the decision-maker faces a single, non-learning agent, this setting is reminiscent of performative prediction (Perdomo et al., 2020). A natural challenge is to design and analyze the combined learning processes of multiple competing agents *and* the decision-maker ensuring convergence to game-theoretically meaningful equilibria.

**Contributions.** We analyze scenarios that reflect the decision-maker's ability to reason about the agents' behavior via different estimates of how it impacts their gradient. The agents update their actions by playing any one of a class of stochastic $\rho$-contracting algorithms. Experiments illustrating our theoretical results in practical scenarios are in Appendix D.

*Characterizing Drift-to-Noise Ratio in Agent Play.* To understand how the learning agents' behavior is influenced by the decision-maker's actions, we first analyze a decision-maker that obliviously deploys an arbitrary sequence of actions $u_t$ that cause *drift* in the agents' stochastic algorithms and induce a sequence of games and corresponding equilibria. In Section 3, we identify regimes governed by the drift-to-noise ratio and characterize the agents' optimal play via *non-asymptotic* convergence guarantees. In particular, we bound the equilibrium tracking error in each of the regimes and give high probability tracking bounds (Appendix H.3) that provide guarantees that hold in settings with irreversible drift such as learning with adaptive agents that strategically respond. These efficiency estimates expose how the equilibrium error decouples from noise in the agents' learning process and time drift in the game; it is also integral to controlling the induced drift in the decision-maker dynamics (Section 4).

*Controlling Induced Drift in Agent Play.* Given this decoupling, we then design decision-maker $\tau$ epoch-based algorithms that *control* the induced drift relative to the agent noise and therefore enable efficient tracking of game theoretic equilibrium concepts dependent upon the gradient information available. The key theoretical challenge is designing the epoch length such that the stochastic tracking error induced in the agent game can be bounded efficiently; for this we much leverage the novel analysis from Section 3 in combination with analysis of the bias-variance trade-off in the combined dynamics. We examine two natural settings, wherein for each setting we set the epoch length—$\tau \asymp \mathcal{O}(\log(1/\epsilon_\tau) + \sigma_{\mathtt{a}}^2/\epsilon_\tau)$, where $\sigma_{\mathtt{a}}^2$ is the agents' noise parameter—by cleverly setting the per-epoch agent tolerance $\epsilon_\tau$ using the drift-to-noise analysis.

In the first setting, a *naïve decision-maker* that recognizes there is distribution shift and opts for a stochastically queryable gradient estimator (i.e., stochastic repeated re-

training) that is biased due to ignoring the reaction of agents. We show (cf. Section 4.1) convergence to an approximate performatively stable equilibrium in $\mathcal{O}(\log(1/\varepsilon) + \sigma^2/\varepsilon)$ epochs where $\sigma^2$ and $\varepsilon$ are the decision-maker gradient estimator variance and target accuracy, respectively. Increasing the gradient information but at a computational cost, *strategic decision-maker* recognizes the agents are dynamically responding, yet does not know the agents objectives. We devise a derivative-free method (cf. Section 4.2) that converges to an approximate Stackelberg equilibrium in $\mathcal{O}(d^2/\varepsilon^2)$ epochs. In both cases, the epoch complexity matches the optimal rate for their single player counterparts.[1] A key challenge we address is designing the epoch length to get an efficient an overall iteration complexity, where the per epoch rate is optimal for tracking problems.

## 2. Preliminaries

Throughout, we use $\mathbb{R}^d$ to denote a $d$-dimensional space with inner product $\langle \cdot, \cdot \rangle$ and the corresponding induced norm is given by $\|x\| = \sqrt{\langle x, x \rangle}$. For any set $\mathcal{X} \subset \mathbb{R}^d$, we denote the projection of a vector $y$ onto $\mathcal{X}$ as $\operatorname{proj}_{\mathcal{X}}(y) = \operatorname{argmin}_{x \in \mathcal{X}} \|x - y\|$. We also set $[n] := \{1, \ldots, n\}$.

### 2.1. Stackelberg Game Abstraction

The interaction between agents and the decision-maker is a *Stackelberg game*: a decision-maker takes actions which influences the behavior of $n$ competitive agents.

**Induced Agent Game.** Given a decision-maker's action $u \in \mathcal{U} \subseteq \mathbb{R}^d$, where $\mathcal{U}$ is some closed convex set, each player $i \in [n]$ seeks to solve the problem $\min_{x_i \in \mathcal{X}_i} f_i^u(x_i, x_{-i})$ where $\mathcal{X}_i \subseteq \mathbb{R}^{m_i}$ is the set of agent $i$'s actions and $f_i^u(x_i, x_{-i})$ denotes a $C^2$-smooth loss function of agent $i$ induced by the decision-maker's action $u$. We use the standard notation $x := (x_i, x_{-i}) \in \mathcal{X} := \prod_i \mathcal{X}_i \subset \mathbb{R}^m$, where $x_i$ is the action of agent $i$, $x_{-i}$ is the joint action of all other agents, and $m = \sum_{i \in [n]} m_i$.

The tuple $\mathcal{G}_u := (f_1^u, \ldots, f_n^u)$ denotes the *game* induced by $u \in \mathcal{U}$. We say that $\mathcal{G}_u$ is a $C^1$-*smooth convex game* if, for each $i \in [n]$, the set $\mathcal{X}_i$ is closed and convex, the function $f_i^u(\cdot, x_{-i})$ is convex in $x_i$ for all fixed $(u, x_{-i}) \in \mathcal{U} \times \mathcal{X}_{-i}$, and the partial gradient $\nabla_i f_i^u(x_i, x_{-i})$ with respect to $x_i$ exists and is continuous. A $C^1$-smooth convex game is called $\mu$-*strongly monotone* for $\mu > 0$ if the inequality $\langle \omega_u(x) - \omega_u(x'), x - x' \rangle \geq \mu \|x - x'\|^2$ holds for all $x, x' \in \mathcal{X} \subseteq \mathbb{R}^m$, where the map $\omega_u(x) := (\nabla_1 f_1^u, \ldots, \nabla_n f_n^u)$ is the *vector of individual gradients*. Strongly monotone games arise in economics (e.g., Kelly auctions) as well as

---

[1] Duchi et al. (2013) improve the dependence on $d$ with a two-point estimator using two queries of the *exact same* environment; this is impossible in our setting, as agents irrevocably update their play in response to queries.

in engineering and machine learning systems as highlighted in Section 1; further examples and commentary on the challenges to relaxing monotonicity are in Appendix C.

The most natural solution concept for the induced game is a *Nash equilibrium*. Given a fixed action $u \in \mathcal{U}$, a strategy $x^* \in \mathcal{X}$ is a *Nash equilibrium for* $\mathcal{G}_u$ if the condition holds: $f_i^u(x_i^*, x_{-i}^*) \leq f_i^u(x_i, x_{-i}^*)$ for all $x_i \in \mathcal{X}_i$ and all $i \in [n]$. A Nash equilibrium $x^* \in \text{Eq}(\mathcal{G}_u)$ satisfies $\langle -\omega_u(x^*), x - x^* \rangle \leq 0$ for all $x \in \mathcal{X}$ (Rockafellar, 2018). Denote the set of Nash equilibria for $\mathcal{G}_u$ as $\text{Eq}(\mathcal{G}_u)$. To ensure existence and uniqueness (Rosen, 1965), we adopt the following assumption (cf. Appendix F for more insight).

**Assumption 2.1.** We assume the following for each $u \in \mathcal{U}$: $(i)$ the induced game $\mathcal{G}_u$ is $\mu$–strongly monotone, $(ii)$ the mappings $x_i \mapsto \nabla_i f_i^u(x_i, x_{-i})$ are $L_i$–Lipschitz continuous, and $(iii)$ $\|\nabla_u \omega_u(x)\|_{\text{op}}$ is bounded.

Notably, Assumption 2.1 implies that the agents' equilibrium response $x^*(\cdot)$ is Lipschitz continuous; let the Lipschitz constant be $L_{\text{eq}}$, and define $L_{\text{a}} := \max_{i \in [n]} L_i$.

**Decision-Maker's Problem.** The decision-maker seeks to minimize loss $\ell : \mathcal{U} \times \mathcal{Z} \to \mathbb{R}$ that depends on their action $u \in \mathcal{U}$ and the decision-dependent environment $z = (x, \xi) \in \mathcal{Z} := \mathcal{X} \times \Xi$ which comprises the joint action of the agents $x \in \mathcal{X}$ and a non-strategic random variable $\xi \in \Xi$. We write $z \sim \mathcal{D}(u) := \mathcal{D}_x(u) \times \mathcal{D}_e(u)$ where $\mathcal{D}_x(\cdot)$ captures the decision-dependent stochasticity of the agents' reactions and $\mathcal{D}_e(\cdot)$ captures *non-strategic* decision-dependent stochasticity. The latter arise in economics, e.g., based on external market factors such as seasonality (Miron, 1990) or economic growth (Davis et al., 2010). Further, the expected loss is denoted $\mathcal{L}(u) = \mathbb{E}_{z \sim \mathcal{D}(u)}[\ell(u, z)]$.

The decision-maker aims to find an equilibrium of the stochastic hierarchical game wherein the agents are playing a Nash equilibrium of the induced game. The most salient solution concept here is a *Stackelberg equilibrium*: namely, $(u^*, x^*(u^*)) \in \mathcal{U} \times \mathcal{X}$ such that $u^* \in \text{argmin}_{u \in \mathcal{U}} \left\{ \mathbb{E}_{\xi \sim \mathcal{D}_e(u)} \ell(u, (x^*(u), \xi)) \middle| x^*(u) \in \text{Eq}(\mathcal{G}_u) \right\}$. However, for any deployed action $u$, the agents are not necessarily playing behavior consistent with $x^*(u) \in \text{Eq}(\mathcal{G}_u)$. The agents are trying to *adaptively learn* the induced equilibrium $x^*(u)$ and may be employing a stochastic algorithm to do so. This necessitates the design of algorithms for the decision-maker that can *control the induced drift and the stochasticity* from the environment.

*Remark* 2.2. In the main body of the paper, we primarily concern ourselves with the setting in which $\mathcal{D}_e(u) \equiv \mathcal{D}_e$ (i.e., a stationary distribution) in an attempt to reduce notational overhead. Throughout, we remark on the decision-dependent non-strategic setting (i.e. where $\mathcal{D}_e(u)$ is non-stationary) and defer proofs to the appendix.

## 2.2. Smooth Algorithmic Agents

Returning to the agents, we now define the class of stochastic algorithms the agents are employing. The decision-maker deploys an action $u_t$ for a number of iterations $\tau \in \mathbb{N}$ within epoch $t$. For a fixed action $u_t$, each agent $i \in [n]$ independently updates their action according to a stochastic algorithm $\mathcal{A}_i$—i.e., $x_{i,t+1}^{k+1} = \mathcal{A}_i(x_t^0, u_{t+1})$ for $k \in [\tau]$ where the epoch-$t$ initial condition is $x_t^0 := x_t$, and by a slight abuse of notation, $x_t := x_{t-1}^\tau$. That is $x_t = \mathcal{A}(x_{t-1}, u_t)$ is the agents' collective response after running $\mathcal{A} = (\mathcal{A}_1, \ldots, \mathcal{A}_n)$ for $\tau$ timesteps. We consider a broad class of algorithms for the agents that adhere to the following definition.

**Definition 2.3.** Fix constants $\rho \in (0, 1)$, $\sigma_{\text{a}} \in [0, \infty)$ and $c > 0$. A stochastic algorithm is $\rho$-*contracting* if, for fixed $u \in \mathcal{U}$, the inequality holds: $\mathbb{E} \|x_t^{k+1} - x^*(u)\|^2 \leq \rho^2 \mathbb{E} \|x_t^k - x^*(u)\|^2 + c^2 \cdot (\rho \sigma_{\text{a}})^2$, where $x^*(u) \in \text{Eq}(\mathcal{G}_u)$.

As we show in Appendix G, there are many examples of algorithms that satisfy Definition 2.3, including stochastic gradient play, asynchronous stochastic gradient play, best response dynamics, and even momentum-based gradient play in strongly convex-strongly concave zero-sum games.

## 2.3. Challenges in Equilibrium Tracking

Let us start by providing some intuition for the technical challenges. A decision-maker's action sequence $\{u_t\}$ induces a time-varying game $\mathcal{G}_{u_t}$ and are trying to learn a time-vary equilibrium $x_t^\star \in \text{Eq}(\mathcal{G}_{u_t})$. For any $\rho$-contracting stochastic algorithm executed by the agents, Proposition 3.1 shows that their *equilibrium tracking error* decomposes as

$$\mathbb{E} \|x_t - x_t^\star\|^2 \lesssim \left(1 - \frac{1-\rho^2}{2}\right)^t \|x_0 - x_0^\star\|^2 + \frac{\sigma_{\text{a}}^2}{1-\rho^2} + \left(\frac{\Delta_{\text{a}}}{1-\rho^2}\right)^2,$$

where $\Delta_{\text{a}} := \max_k \{\|x_k^\star - x_{k-1}^\star\|^2\}$ is the *drift* and $\sigma_{\text{a}}$ characterizes the *noise*.[2] The first term is exponentially decaying so that, as $t \to \infty$, we are left with the drift and noise terms. This raises the following challenge:

Challenge 1: *Given the decision-maker induced drift in the agents' stochastic game, what is the contraction $\rho$ that optimizes the target accuracy?*

Section 3 contains results that characterizes the drift-to-noise regimes and corresponding optimal contraction rate for stochastic gradient play. These efficiency rates are crucial for analysis of decision-maker algorithms in Section 4.

Turning now to the decision-maker's perspective, even if the agents have equilibrated such that for any given $u$ the agents play a "best response" $x^*(u)$, the expected loss $\mathbb{E}_{z \sim \mathcal{D}(u)}[\ell(u, z)]$ still depends on the *a priori* unknown preferences of the agents. In practice, this means that the second

---

[2] Here, $\asymp$ and $\lesssim$ indicate an equality and inequality, respectively, holding up to a constant.

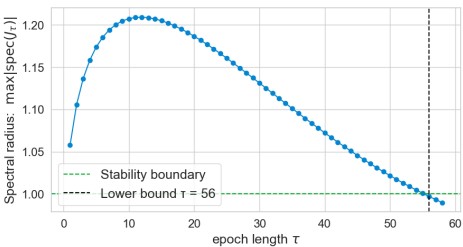

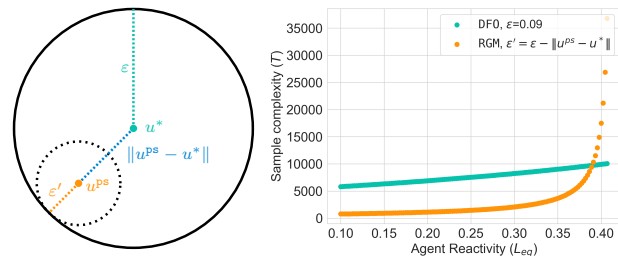

*Figure 1.* The spectral radius of the local linearization $J_\tau(x^*, u^*)$—parameterized by $\tau$—of the combined update. Stability is attained only when $\tau \geq \min\{\tau' \mid \text{spec}(J_{\tau'}(x^*, u^*)) \subset D[0,1] \subset \mathbb{C}\}$. See Appendix D.4 for further details.

term in the decision maker's gradient

$$\mathbb{E}_{z \sim \mathcal{D}(u)} \nabla_u \ell(u, z) + \frac{d}{dv} \mathbb{E}_{z \sim \mathcal{D}(v)} \ell(u, z)|_{v=u}, \quad (1)$$

is not directly computable.

Challenge 2: *Can the decision-maker algorithmically exploit available gradient information to converge to a game-theoretically meaningful equilibrium?*

To address this nontrivial problem, we set up a hierarchy of reasonable interaction models that account for progressively more gradient information. There are two natural settings: the decision-maker $(i)$ naïvely employs repeated retraining that does not account for the second term in (1), and $(ii)$ strategically employs a query-based method to estimate this decision-dependent term. In Section 4, we characterize which equilibrium concepts are achievable under these two approaches: namely, performatively stable equilibrium in $(i)$ and Stackelberg equilibrium in $(ii)$.

Further exacerbating the aforementioned challenge, because the agents' *learned* play $x_t$ also evolves with the decision-maker's actions, the decision-maker also faces a time-varying objective. To illustrate this point, consider a decision-maker with $\alpha$-strongly convex loss $\ell(\cdot, z)$ and define $u_t^\star \in \arg\min_u \{\mathbb{E}_{\xi \sim \mathcal{D}_e(u)} \ell(u, (x_t, \xi))\}$. Analogous to the agent decomposition above, if the decision-maker uses a stochastic gradient-based algorithm with stepsize $\eta$, then

$$\mathbb{E} \|u_t - u_t^\star\|^2 \lesssim \left(1 - \frac{\eta\alpha}{4}\right)^t \|u_0 - u_0^\star\|^2 + \frac{\eta\sigma^2}{\alpha} + \left(\frac{\Delta}{\alpha\eta}\right)^2, \quad (2)$$

where $\sigma^2$ is the gradient estimator variance ("noise") and $\Delta := \max_k \|u_k^\star - u_{k-1}^\star\|^2$ parameterizes the drift induced by the agents time-varying actions $\{x_t\}$. The aim is to optimize the right-hand side of (2) by controlling $\Delta/\sigma$. However, recall that the decision-maker does not have *a priori* knowledge of the agents' objectives nor their update rules.

Challenge 3: *Can we design algorithms to control the induced drift such that $(u_t, x_t)$ reaches an $\varepsilon$-equilibrium in finite time?*

The question of controlling the drift in *finite-time* reveals interesting algorithm design questions: how does the decision-

*Figure 2.* (left) A scenario where the performatively stable equilibrium $u^{\text{ps}}$ lies in an $\varepsilon$-ball around $u^*$; (right) Approximate big-$\mathcal{O}$ sample complexity (Theorems 4.4 and 4.9) needed by the repeated-gradient (RGM) and derivative-free (DFM) methods to bring the decision-maker within $\varepsilon$ of $u^*$ as agent reactivity $L_{\text{eq}}$ grows. Because RGM ignores the implicit term of the decision-maker gradient, the gap $\|u^{\text{ps}} - u^*\|$ widens as $L_{\text{eq}}$ grows, shrinking its effective tolerance $\varepsilon' = \varepsilon - \|u^{\text{ps}} - u^*\|$ and pushing its sample complexity past that of DFM. See Appendix D.5 for additional details.

maker ensure that the agents equilibrate *fast enough* so as to control $\Delta$ relative to $\sigma$? To address this, we introduce time-scale separation via *epoch-based* algorithms in the two aforementioned gradient information settings. To obtain finite-time rather than asymptotic convergence, it is necessary to allow the agents to update multiple times while the decision-maker remains relatively stationary. This is illustrated by analyzing the local stability of the dynamical system representing the combined update. For fixed stepsizes $(\eta, \gamma)$ and gradient updates for the decision-maker and agents, respectively, at each fixed action $u_t$, the agents update $\tau$ steps and then the decision-maker updates. As illustrated in Figure 1, it is required that $\tau \geq 1$ for equilibrium convergence, meaning that the epoch-based algorithm structure is necessary for finite-time guarantees. We describe the epoch length lower bound in more detail in Appendix D.4. In Section 4, for each of the gradient information settings, we optimize the epoch length to obtain non-asymptotic convergence by controlling $\Delta/\sigma$.

A last natural question centers on efficiency versus optimality. Recall that the iterates equilibrate at different equilibrium concepts dependent upon the decision-maker's estimated gradient information. We show that the sample efficiency of the gradient method decreases as more of the gradient (1) is estimated (under the same information assumptions on the agents behavior), but is there an effect on the optimality of achievable equilibrium? Consider the scenario shown in Figure 2: a decision-maker could use a derivative-free method to reach within $\varepsilon > 0$ of the Stackelberg equilibrium $u^*$, *or* use the more efficient repeated gradient method to reach within $\varepsilon' = \varepsilon - \|u^{\text{ps}} - u^*\|$ of the performatively stable equilibrium $u^{\text{ps}}$; in either case, the *worst-case* expected distance from $u^*$ is the same. This poses a problem for the decision-maker: *how to assess the*

*tradeoff between performance degradation and sample complexity?* With this in mind, in Appendix I, we characterize the performance gap (Proposition I.4) in terms of properties of the game—in particular, how dynamic agents are in reaction to the decision-maker.

## 3. Characterizing Drift-to-Noise Ratio

To understand the drift versus noise, we consider the setting in which a decision-maker obliviously deploys a sequence of actions $\{u_t\}$ and passively observes how the agents' response $\{x_t\}$ as generated by some set of algorithms $\mathcal{A}$. Examples include pricing or recommendations where periodic changes are made to interventions. The decision-maker's sequence of actions induces drift in the agents' equilibrium: $x_t^\star := x^*(u_t)$. Prior work has focused only on asymptotic guarantees for stochastic tracking in strongly monotone games assuming that the time-varying sequence of games being tracked equilibrates *a priori* (Duvocelle et al., 2023). Given the induced drift $\Delta$ as identified in (1), a natural question is *can we obtain a finite-time bound on the time to reach a target equilibrium, $\mathbb{E}\|x_t - x_t^\star\|^2$?* With the design questions outlined in Section 2.3, taking a different tack than prior asymptotic work, we extend recent work in stochastic optimization, namely Cutler et al. (2023), to stochastic monotone games; this requires novel analysis as games generally *do not admit* a single cost function to which we can appeal in the analysis.

### 3.1. Bounding the Equilibrium Tracking Error

Suppose the decision-maker deploys actions $u_t$ such that $x_t^\star$ is the induced equilibrium for the $\mu$–strongly monotone game $\mathcal{G}_{u_t}$, which depends on $u_t$. As long as the agents employ a $\rho$-contracting stochastic method as in Definition 2.3, then it is possible to bound the expected equilibrium tracking error with a notable dependence on the noise $\sigma_a$ and induced drift $\Delta_a := \max_t\{\|x_{t+1}^\star - x_t^\star\|\}$.

**Proposition 3.1** (Informal). *Under Assumption 2.1, suppose agents employ a $\rho$-contracting stochastic algorithm in the regime $\rho \in [0, 1)$. Then the estimate holds:*

$$\mathbb{E}\|x_t - x_t^\star\|^2 \lesssim \left(1 - \frac{(1-\rho^2)}{2}\right)^t \|x_0 - x_0^\star\|^2 + \frac{(c\sigma_a)^2}{1-\rho^2} + \left(\frac{\Delta_a}{1-\rho^2}\right)^2.$$

This result is formally stated in Proposition H.2 (Appendix H). We exploit this decomposition of the tracking error to obtain efficiency estimates, thereby laying the foundation for efficient decision-maker algorithms.

Indeed, *do last-iterate convergent algorithms exist for the agents in this time-varying setting?* To provide insight into the difficulty of the problem, we focus on a natural learning rule, *stochastic gradient play*:

$$x_{t+1} = \underset{\mathcal{X}}{\mathrm{proj}}(x_t - \gamma\widehat{\omega}_t): \quad \widehat{\omega}_t := (\widehat{\nabla}_i f_i^{u_t}(x_t))_{i=1}^n. \quad \text{(SGP)}$$

Proposition 3.1 reduces to the following corollary, the formal version of which is given in Corollary H.4.

**Corollary 3.2** (Informal). *Under Assumption 2.1, suppose agents are running stochastic gradient play (SGP) with step-size $\gamma \le \mu/(2L_a^2)$, and an unbiased estimator $\widehat{\omega}_t$ satisfying $\mathbb{E}[\|\widehat{\omega}_t - \mathbb{E}_t[\widehat{\omega}_t]\|^2] \le \sigma_a^2$ for $\sigma_a \in [0, \infty)$. Then $\rho^2 = 1/(1 + \gamma\mu)$ and $c = \sqrt{2}\gamma$, respectively, so that*

$$\mathbb{E}_t\|x_t - x_t^\star\|^2 \lesssim \left(1 - \frac{\mu\gamma}{4}\right)^t\|x_0 - x_0^\star\|^2 + \frac{\gamma\sigma_a^2}{\mu} + \left(\frac{\Delta_a}{\gamma\mu}\right)^2.$$

Letting $t \to \infty$, the optimization error tends to zero, leaving only the noise and drift terms. Optimizing with respect to $\gamma$ leads to the optimal learning rate $\gamma_\star := \min\{\mu/(2L_a^2), (2\Delta_a^2/(\mu\sigma_a^2))^{1/3}\}$ and asymptotic tracking error $\varepsilon_\star := \min_{\gamma\in(0,\mu/(2L_a^2)]}\{\gamma\sigma_a^2/\mu + (\Delta_a/(\mu\gamma))^2\}$. Here $\gamma_\star$ determines the interesting regimes. Indeed, setting $\mu/(2L_a^2) = (2\Delta_a^2/(\mu\sigma_a^2))^{1/3}$ and rearranging, we have two regimes for the drift-to-noise ratio: the *low* regime if $\Delta_a/\sigma_a < \mu^2/(4L_a^3)$, and otherwise the *high* regime.

In the high drift-to-noise regime, if agents run stochastic gradient play with $\gamma_\star \asymp \mu/(2L_a^2)$, we have that

$$\mathbb{E}\|x_t - x_t^\star\|^2 \lesssim \varepsilon_\star \text{ in } t \lesssim \frac{L_a^2}{\mu^2}\log\left(\frac{\|x_0 - x_0^\star\|^2}{\varepsilon_\star}\right) \text{ time steps.}$$

With high drift-to-noise the problem is essentially deterministic, and the rate in fact matches the deterministic setting (see, e.g., Chasnov et al. (2020a)). The low drift-to-noise regime is decidedly more interesting as the rate can be improved by controlling the combined drift and noise (cf. Section 4). Setting the step-size $\gamma_\star \asymp (2\Delta_a^2 \cdot \frac{1}{\mu\sigma_a^2})^{1/3}$, informally we have that

$$\mathbb{E}\|x_t - x_t^\star\|^2 \lesssim \varepsilon_\star \text{ in } t \lesssim \frac{\sigma_a^2}{\mu^2\varepsilon_\star}\log\left(\frac{\|x_0 - x_0^\star\|^2}{\varepsilon_\star}\right) \text{ time steps.}$$

The following proposition shows that if the agents employ stochastic gradient play in stages, then much like the single player time-invariant optimization problem (Kulunchakov & Mairal, 2019), the agents tracking error can be improved.

**Proposition 3.3** (Informal). *Suppose that induced time-varying agent problem is in the low drift-to-noise regime. There is an algorithm that proceeds by running stochastic gradient play (SGP) in $K$ stages with $T_k$ steps in each of the $k \in [K]$ stages such that the total time satisfies $T = \sum_{k=0}^{K-1} T_k \lesssim \frac{L_a^2}{\mu^2}\log\left(\frac{2\|x_0 - x_0^\star\|^2}{\varepsilon}\right)^+ + \frac{\sigma_a^2}{\mu^2\varepsilon}$, and the expected tracking error satisfies $\mathbb{E}\|x_K - x_K^\star\|^2 \lesssim \varepsilon$. Further, if the agents employ $\gamma_\star$ then $\varepsilon = \varepsilon_\star$.*

In Appendix H, we detail the construction of this algorithm, and give the formal statement in Proposition H.5. Essentially, the agents employ an algorithm that repeatedly runs stochastic gradient play in stages by reinitializing the algorithm at the previous stage and adjusting the stepsize by

**Algorithm 1** Epoch-Based Drift-to-Noise Control

> **Input**: $\texttt{Alg}, T, \eta_t, x_{-1}, u_0, \tau_t$
> **for** $t = 1, \ldots, T$ **do**
>     **for** $k = 0, \ldots, \tau_t - 1$: query agents with $u_t$
>     observe $z_t = (\mathcal{A}(x_{t-1}, u_t), \xi_t)$
>     update $u_{t+1} = \texttt{Alg}(x_t, \eta_t, \widehat{g}_t)$
> **end for**

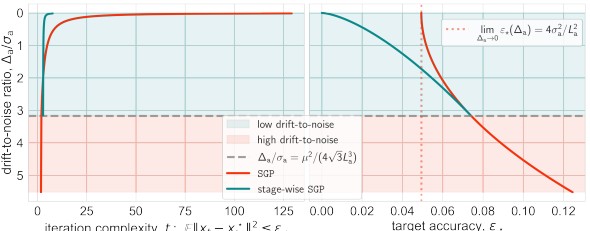

*Figure 3.* Iteration complexity and target accuracy of stochastic gradient play as a function of drift-to-noise ratio. Switching to the optimal learning rate $\gamma_\star$ (Section 3) and using a stage based algorithm improves iteration complexity and lowers the achievable target error in the low drift-to-noise regime by decreasing the variance, while SGP is limited by the noise floor.

a factor of $2^{-k}$. This progressively reduces the constant variance term. The iteration complexity clearly is composed of two terms: the classical deterministic complexity $\mathcal{O}((L_a^2/\mu^2)\log(1/\varepsilon_\star))$ of gradient play, and the optimal complexity for stochastic gradient play in $\mathcal{O}(\sigma_a^2/(\mu^2\varepsilon_\star))$. Crucially, the regime is *controlled by the decision-maker's algorithm*—since $\mathbb{E}\|x_{t+1}^\star - x_t^\star\| \leq L_{eq}\,\mathbb{E}\|u_{t+1} - u_t\|$ so that $\Delta_a \leq L_{eq}\Delta$—*as well as* the agents' step-size which determines proximity to optimal. Bounds such as those in Proposition 3.3 are exploited (cf. Section 4) to design the epoch length in algorithms to control the agents'—and therefore the decision-maker's–error.

*Remark* 3.4 (Beyond Worst Case Tracking Bounds). The preceding results focus on the *worst-case equilibrium expected tracking error*. In Appendix H.2, by assuming contraction rate on the sequence $\{u_t\}$, we give a time-varying equilibrium tracking error. However, expected equilibrium tracking guarantees are only meaningful if the dynamics run *many* times. Hence, in Appendix H.3, we give *high probability bounds* on the tracking error. Due to the strategic nature of agents, if the $u_t$'s are deployed in *real-time* there will be *irreversible drift*, and thus high-probability efficiency results are more meaningful as they characterize the performance if dynamics were executed only once.

## 4. Controlling for the Drift-to-Noise Ratio

Recall from Section 2 that the decision-maker seeks to solve a stochastic optimization problem with both uncertainty due to the response of the agents and stochasticity of the environment—namely, they seek to minimize the loss $\mathcal{L}(u) = \mathbb{E}_{z \sim \mathcal{D}(u)}[\ell(u, z)]$ with respect to $u$ where $z \sim \mathcal{D}(u)$ is the stochastic observation that the decision-maker receives from the environment and abstracts the agents' (stochastic) decision process. Ultimately for $u_t$ to stabilize around some (appropriate) equilibrium $u^\star$, the decision-maker needs to control $\|x_t - x_t^\star\|^2$ and $\|x_t^\star - x_{t-1}^\star\|^2$ as these terms drive the induced drift and noise (and therefore bias and variance). Key questions center on 1) what types of algorithms enable controlling these two sources of error, and 2) given a particular algorithm, how long before the agents reach the low drift-to-noise regime, wherein the optimal target accuracy can be achieved? In Figure 3, for stochastic gradient play, we show the iteration complexity and optimal target accuracy for the agents as a function of the drift-to-noise. There

is a clear transition at $\Delta_a/\sigma_a = \mu^2/(4\sqrt{3}L_a^2)$ after which, if the agents adopt a stage-based algorithm, then the target accuracy can be set arbitrarily small with modest impact on the iteration complexity. On the other hand, if a stage based algorithm is not employed, the agents risk hitting the noise floor $\sigma_a$ which will lead to a constant error in the decision-maker's update.

We address these design and analysis questions in the subsequent sections, focusing on a hierarchy of different gradient information and epoch-based algorithms (Algorithm 1). We use bounds from Section 3 to set epoch lengths and the decision-maker's step-size to control the two aforementioned two errors, respectively.

### 4.1. Naïve Decision-Maker

A common naïve approach in machine learning is a *stochastic repeated gradient method* wherein the decision-maker periodically retrains given new data from the environment. Since the decision-maker does not have *a priori* access to the agents' response mapping, it updates its loss using a stochastic gradient estimate of $\mathbb{E}_{z \sim \mathcal{D}(u)} \nabla_u \ell(u, z)$—i.e., only the part of the total gradient of $\mathcal{L}(u)$ with explicit dependence on $u$. We make the following regularity assumption.

**Assumption 4.1.** *a.* The loss $\ell(\cdot, z)$ is $C^1$-smooth and $\alpha$–strongly convex for any $z$; *b.* The maps $u \mapsto \nabla_u \ell(u, z)$ and $z \mapsto \nabla_u \ell(u, z)$ are $L_u$ and $L_z$ Lipschitz continuous, respectively.

**Equilibrium Baseline.** The appropriate notion of an equilibrium in this case is that of a *performatively stable equilibrium* of the "lifted" $(n + 1)$–player game $(\ell, f_1, \ldots, f_n)$ (Narang et al., 2023).

**Definition 4.2.** The joint action $(u^{ps}, x^*(u^{ps}))$ is a *performatively stable equilibrium* if $x^*(u^{ps}) \in \text{Eq}(\mathcal{G}_{u^{ps}})$ and $u^{ps} = \text{argmin}_{u \in \mathcal{U}} \mathbb{E}_{\xi \sim \mathcal{D}_e(u^{ps})} \ell(u, (x^*(u^{ps}), \xi))$.

In Appendix I.1, we show the equilibrium exists and

is unique when $\alpha < L_z(L_{\mathsf{en}} + L_{\mathsf{eq}})$ where $L_{\mathsf{en}}$ is the Lipschitz parameter for the environment—namely, $W_1(\mathcal{D}_e(u), \mathcal{D}_e(w)) \leq L_{\mathsf{en}}\|u - w\|$ (cf. Assumption I.2). In the setting where $\mathcal{D}_e(u) \equiv \mathcal{D}_e$ existence is guaranteed with $\alpha < L_z L_{\mathsf{eq}}$, and the performatively stable equilibrium is a Nash equilibrium of the game $(\ell, f_1, \ldots, f_n)$: the decision-maker is playing a best response to the equilibrium response of the agents in expectation.[3] In Appendix I.2, we analyze the *performative gap* which is defined as $\|u^* - u^{\mathsf{ps}}\| + \|x^*(u^*) - x^*(u^{\mathsf{ps}})\|$—the distance between the Stackelberg and performatively stable equilibrium. Depending on the problem parameters, the performatively stable equilibrium is sub-optimal, yet the gap may be small (cf. Proposition I.4). It is interesting to examine regimes in which the gap is small so that obtaining the Stackelberg is *not worth* extra sample complexity (cf. Figure 2).

**Stochastic Repeated Gradient Method.** The repeated stochastic gradient method is given by

$$u_{t+1} = \underset{\mathcal{U}}{\mathrm{proj}}(u_t - \eta g_t) : g_t = \nabla_u \ell(u_t, (x_t, \xi_t)). \quad \text{(RGM)}$$

**Assumption 4.3** (Finite Variance). Suppose there exists a filtered probability space $(\Omega, \mathcal{F}, \mathbb{F}, \mathbb{P})$ with filtration $\mathbb{F} = (\mathcal{F}_t)_{t \geq 0}$ such that $\mathcal{F}_0 = \{\emptyset, \Omega\}$, $g_t$ is $\mathcal{F}_{t+1}$-measurable, and there exists a constant $\sigma > 0$ satisfying $\mathbb{E}_t \|g_t - \mathbb{E}_t[g_t]\|^2 \leq \sigma^2$ where $\mathbb{E}_t = \mathbb{E}[\cdot \mid \mathcal{F}_t]$ denotes the conditional expectation.

Suppose that the decision-maker fixes its action $u_t$ in round $t$ for $\tau$ time-steps, and the agents run a stage-based $\rho$-contracting algorithm (cf. Corollary G.2) $\mathcal{A}$. Agents are incentivized to run stage-based algorithms: they aim to stabilize within the smallest neighborhood of $x_t^\star$ as possible, thereby requiring control of their own variance $\sigma_{\mathsf{a}}^2$. In the remainder of this subsection, fix constants $\bar{\alpha} := \alpha - L_z L_{\mathsf{en}} > 0$ (for existence and uniqueness, Theorem I.3), and $L^2 := L_u^2 + L_z^2 L_{\mathsf{eq}}^2$ (abusing notation).

**Theorem 4.4** (Informal). *Suppose Assumptions 2.1, 4.1, and 4.3 hold, and that constants $R > \|x_{-1} - x^*(u_0)\|$ and $B > \|u_0 - u^{\mathsf{ps}}\|^2$ are available. Further, suppose we are in the regime where $\alpha > L_z L_{\mathsf{en}}$ and the decision-maker runs Algorithm 1 with $\mathtt{Alg} := \mathtt{RGM}$ using step-size $\eta \leq \frac{\bar{\alpha}}{4L^2}$, and the agents employ a stage-based $\rho$–contracting algorithm $\mathcal{A}$ with $\rho \in [0, 1)$ and $\sigma_{\mathsf{a}} \in (0, \infty)$. Set $\bar{R} := R + \frac{2c^2\sigma_{\mathsf{a}}^2}{1-\rho^2} + 6L_{\mathsf{eq}}^2(4B + \frac{\sigma^2}{L^2}))/(1-\rho^2)^2$, the tolerance $\epsilon_\tau \asymp \eta^2 \sigma^2$ and epoch length $\tau \asymp \mathcal{O}\left(\frac{1}{(1-\rho^2)} \log\left(\frac{2\bar{R}^2}{\epsilon_\tau}\right) + \frac{c^2 \sigma_{\mathsf{a}}^2 \rho^2}{(1-\rho^2)^2 \epsilon_\tau}\right)$. Then, the following estimate holds:*

$$\mathbb{E}_t \|u_t - u^{\mathsf{ps}}\|^2 \leq \left(1 - \frac{\bar{\alpha}\eta}{2}\right)^t \|u_0 - u^{\mathsf{ps}}\|^2 + \frac{4\eta\sigma^2}{\bar{\alpha}}.$$

---

[3]Performatively stable points in the single player setting (Perdomo et al., 2020) are equally interpretable as Nash when the environment comprises a stochastic best responding agent whose reaction is determined by a utility function.

Proposition I.5 contains the formal statement and proof. A key technical difficulty is bounding the induced drift (i.e., agents' equilibrium tracking error). We leverage results from Section 3 to set the epoch length via bounding the sequential (epoch to epoch) initialization error $\mathbb{E}\|x_{t-1} - x_t^\star\|^2$, and combine this with within epoch efficiency estimates for $\rho$-contracting algorithms (Appendix G). We also provide a specialization to the case where the $\rho$-contracting algorithm is stochastic gradient play (Proposition I.6). To provide more intuition on the effect of drift vs noise, we also specialize to the case where the $\rho$-contracting algorithm is deterministic (Proposition I.11); indeed, here agents do not need to run a stage-based algorithm as there is no extra bias from their stochasticity. Finally, we also leverage the high probability tracking error results (Appendix H.3) to give stronger guarantees for irreversible induced drift (cf. Theorem I.9, Appendix I.3.1).

Given this $t$-step bound, the decision-maker may employ a staged method to obtain convergence to an approximate performatively stable equilibrium.

**Corollary 4.5** (Informal). *Fix a target accuracy $\varepsilon > 0$. Under the assumptions of Theorem 4.4, suppose the decision maker runs the stochastic repeated gradient method in $k = 0, \ldots, K$ super-epochs, for $T_k$ epochs each with constant step-size $\eta_k = 2^{-k}\eta_0$, and such that the last iterate of each super-epoch is the first iterate of the next. Fix constants $\eta_0 = \frac{\bar{\alpha}}{(4L^2)}$, $T_0 = \lceil \frac{2}{\bar{\alpha}\eta_0} \log\left(\frac{2B^2}{\varepsilon}\right)\rceil$, $T_k = \lceil \frac{2\log(4)}{\bar{\alpha}\eta_k}\rceil$, and $K = \lceil 1 + \log_2\left(\frac{\sigma^2}{L^2 \varepsilon}\right)\rceil$. Then $\mathbb{E}\|u_T - u^{\mathsf{ps}}\|^2 \leq \varepsilon$ and $\mathbb{E}\|x_T - x^*(u^{\mathsf{ps}})\|^2 \leq 2(\epsilon_\tau + L_{\mathsf{eq}}\varepsilon)$ in*

$$T = \sum_{k=1}^K T_k \lesssim \mathcal{O}\left(\frac{L^2}{\bar{\alpha}^2} \log\left(\frac{2B^2}{\varepsilon}\right) + \frac{\sigma^2}{\bar{\alpha}^2 \varepsilon}\right) \text{ epochs.}$$

The formal statement and proof is in Corollary I.7. Note that since we run $\tau$ iterations within each epoch, the total number of iterations is $T \cdot \tau$. Analogous to Proposition 3.3, we progressively decrease the step-size to control the bias—namely, the drift in the agent game $\mathbb{E}\|x_{t-1}^\star - x_t^\star\|^2$ and noise from their update $\mathbb{E}\|x_t - x_t^\star\|^2$—until the desired accuracy is achieved. Reflecting back to Figure 3, the target accuracy can be better optimized if the agents switch their step-size to the optimal $\gamma_\star$ once in the low drift-to-noise regime. Hence, it is interesting to characterize the time $T$ after which $\max_{k \leq T} \mathbb{E}\|u_{k-1} - u_k\|^2$ ensures the agents are in the low drift-to-noise regime in expectation.

**Proposition 4.6** (Informal). *Under the assumptions of Corollary 4.5, the estimate $\max_{k \leq T} \mathbb{E}\|u_k - u_{k-1}\|^2 \lesssim \left(\frac{\mu^2 \sigma_{\mathsf{a}}}{4 \cdot L_{\mathsf{eq}} L_{\mathsf{a}}^2}\right)^2$ holds after $T = \sum_{k=1}^K T_k \lesssim \mathcal{O}\left(\frac{L^2}{\bar{\alpha}^2} \log\left(\frac{2B^2}{\varepsilon}\right) + \frac{\sigma^2}{\bar{\alpha}^2 \varepsilon}\right)$ epochs where $\varepsilon = \frac{1}{6}\left(\mu^2 \sigma_{\mathsf{a}}/(4 \cdot L_{\mathsf{eq}} L_{\mathsf{a}}^2)\right)^2$.*

We give the formal statement in Proposition I.10. Once in this region the agents are naturally incentivized to optimize their learning rates (i.e., selecting $\gamma_\star$) as it will enable them to more effectively stabilize the learning process.

*Remark* 4.7 (Non-Strategic Environment Decision-Dependence). As we show in Appendix I.3.4, even if $\mathcal{D}_e(u)$ depends on $u$ the results immediately extend. Indeed, suppose there exists $L_{\text{en}} < \infty$ such that $W_1(\mathcal{D}_e(u), \mathcal{D}_e(w)) \leq L_{\text{en}}\|u - w\|$ and that $\alpha > L_z(L_{\text{en}} + L_{\text{eq}})$ so an equilibrium exists (cf. Proposition I.3). Then, the results immediately apply replacing $\bar{\alpha}$ with $\alpha - L_z(L_{\text{en}} + L_{\text{eq}})$.

### 4.2. Strategic Decision-Maker

A more *strategic* approach in low-information settings—i.e., where the decision-maker knows that the agents are responding to $u_t$, yet still does not know the agents' objectives nor the algorithms they employ—is to estimate the effect of the agents' time-varying behavior via carefully designed stochastic queries to the environment.

**Equilibrium Baseline.** The natural equilibrium baseline in this case is the Stackelberg equilibrium (cf. Section 2.1) since the decision-maker aims to optimize through the reaction of the environment including the agents' collective response. To do so via gradient methods, the decision-maker needs to estimate the *full gradient* of its loss as given in (1). Derivative free methods are one type of approach that enable estimation of the second term in (1) via stochastic samples of the loss at the queried environment state. The resulting gradient estimator tends to be inherently biased, prone to high variance, and is nominally sample inefficient; nonetheless, Stackelberg equilibrium convergence is possible, as opposed to suboptimal performatively stable equilibrium.

**Derivative Free Method.** Given that the environment is time-varying and responsive to the decision-maker's queries, we adapt a single point derivative free method (see, e.g., (Agarwal et al., 2010; Drusvyatskiy et al., 2022)) to the epoch-based framework outlined in Algorithm 1. Multipoint methods tend to have better complexity in terms of $d$, yet agents are *responding* and therefore it may not be possible to query the same static population repeatedly.

For a fixed query radius $\delta > 0$ and epoch length $\tau \geq 1$, the decision-maker updates its action via

$$u_{t+1} = \underset{(1-\delta)\mathcal{U}}{\text{proj}} \, (u_t - \eta_t g_t), \qquad \text{(DFM)}$$

where, for a uniformly sampled vector $v_t \sim \mathbb{S}^d$, the gradient estimate is $g_t = \frac{d}{\delta}\ell(u_t + \delta v_t, (\mathcal{A}(x_{t-1}, u_t + \delta v_t), \xi_t))v_t$ with $\xi \sim \mathcal{D}_e$. The challenge compared to classical analysis is accounting for the *additional* bias from the agents' *drift* and *noise*. We require some regularity assumptions.

**Assumption 4.8.** The following hold: $a$. the loss $\ell(u, z)$ is bounded with $\ell_* := \sup_{(u,z) \in \mathcal{U} \times \mathcal{Z}} |\ell(u, z)|$; $b$. the map $u \mapsto \nabla^2 \mathcal{L}(u)$ is $L_{\text{H}}$-Lipschitz continuous; $c$. the expected loss $\mathcal{L}(u)$ is $\bar{\alpha}$–strongly convex; $d$. $\exists \, b, B > 0$ such that $b\mathbb{B} \subseteq \mathcal{U} \subseteq B\mathbb{B}$ where $\mathbb{B} = \{u \in \mathbb{R}^d | \, \|u\| \leq 1\}$.

Assumption 4.8.$d$ is common (cf. Agarwal et al. (2010)), and implies the convex set $\mathcal{U}$ is compact with a non-empty interior; otherwise, we can map $\mathcal{U}$ to a lower dimensional space. Assumption 4.8.$c$ requires the composition of the algorithm the agents' play and the loss $\ell$ to be convex. We give examples of how this criteria may be met in Appendix J, and discussion of relevant literature (Dong et al., 2018; Miller et al., 2021; Perdomo et al., 2020; Ray et al., 2022). This assumption allows *convergence to global optima*; when it is not met, convergence is possible to local solutions; yet, as with non-monotone games, what constitutes an *interesting* solution is widely debated (Fiez et al., 2020; Jin et al., 2020; Mangoubi & Vishnoi, 2021). We explore local convergence numerically (Appendix D), and leave theory to future work.

Let $(u^*, x^*(u^*))$ be the Stackelberg equilibrium. In the remainder of this subsection, let $L := L_u + L_z L_{\text{eq}}$.

**Theorem 4.9** (Informal). *Suppose that Assumptions 2.1, 4.1, and 4.8 hold, and that a constant $R > \|x_{-1} - x^*(u_0)\|^2$ is available. Further, suppose the decision-maker runs Algorithm 1 with* Alg := DFM *using step-size $\eta_t = \frac{4}{\bar{\alpha}(t+1)}$ and query radius $\delta < \min\{b, \frac{\bar{\alpha}}{L_{\text{H}}}\}$, and that the agents employ a $\rho$–contracting algorithm $\mathcal{A}$ with $\rho \in [0, 1)$ and $\sigma_a \in (0, \infty)$. There exists a constant $\bar{R} < \infty$ such that if the tolerance is set to $\epsilon_\tau = (\delta(t + 1))^{-1}$ with epoch length $\tau \asymp \mathcal{O}\left(\frac{1}{(1-\rho^2)}\log\left(\frac{2\bar{R}}{\epsilon_\tau}\right) + \frac{c^2\sigma_a^2\rho^2}{(1-\rho^2)^2\epsilon_\tau}\right)$, then the estimate holds: $\mathbb{E}\|u_t - u^*\|^2 \leq \frac{\max\{2\bar{\alpha}^2\delta^2 B, 16(\ell_*^2 d^2 + 1)\}}{\delta^2\bar{\alpha}^2(t+1)} + 2\delta^2\left(\left(1 + \frac{L}{\bar{\alpha}}\right)\|u^*\| + \frac{L}{\bar{\alpha}}\right)$.*

For simplicity, the formal definition of $\bar{R}$, which determines the required epoch length, is given in Appendix J.2; it is analogous to the constant in Theorem 4.4 in that it is derived from bounding the tracking error $\mathbb{E}\|x_t - x_t^\star\|^2$ using the drift-to-noise decomposition analysis in Section 3. We now characterize the iteration complexity to reach an approximate Stackelberg equilibrium.

**Corollary 4.10.** *Suppose the assumptions of Theorem 4.9 hold. Fix target accuracy $\varepsilon < 4b^2\left(\left(1 + \frac{L}{\bar{\alpha}}\right)B + \frac{L}{\bar{\alpha}}\right)^2$ and set $\delta = \bar{\alpha}\sqrt{\varepsilon/4}/((\bar{\alpha} + L)B + L)$ and $\eta_t = 4/(\bar{\alpha}(t + 1))$. The iterates $(u_t, x_t)$ converge to an approximate Stackelberg equilibrium: $\mathbb{E}[\|u_t - u^*\|^2] \leq \varepsilon$ and $\mathbb{E}[\|x_t - x^*(u^*)\|^2] \leq 2(\epsilon_\tau + L_{\text{eq}}\varepsilon)$ hold for all $t \geq 16\max\{\bar{\alpha}^4\varepsilon B^2, 8(\ell_*^2 d^2 + 1)((\bar{\alpha} + L)B + L)^2\}/(\bar{\alpha}^4\varepsilon^2)$.*

The formal statement is given in Theorem J.6 (Appendix J). An analogous statement to Proposition 4.6 which identifies the transition from high to low regimes as the decision-maker stabilizes is also provided. Observe that $\varepsilon$ may be selected arbitrarily small to control the agents' locality relative to the equilibrium. Corollary 4.10 provides a bound in terms of the number of epochs; the total iteration complexity is $\sum_{s=1}^{t}\tau_s \asymp \mathcal{O}\left(\frac{d^2}{\varepsilon^2}\left(\log\left(\frac{1}{\epsilon_\tau}\right) + \frac{\sigma_a^2}{\epsilon_\tau}\right)\right)$. If the agents are deterministic ($\sigma_a = 0$), this rate matches the rate of single

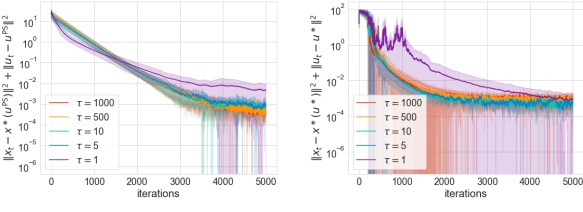

(a) `RGM` with `SGP` agents.    (b) `DFM` with `SGP` agents.

*Figure 4.* Effects of decision-maker's choice of $\tau$ on equilibrium convergence in a convex quadratic game. As expected, larger $\tau$ produces smaller equilibrium error.

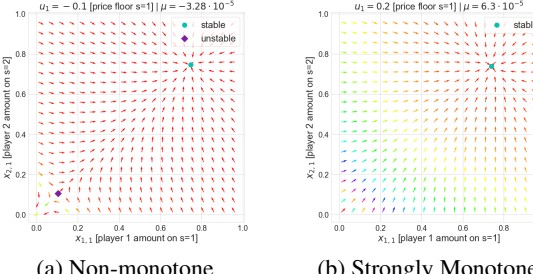

(a) Non-monotone    (b) Strongly Monotone

*Figure 5.* Phase portraits for the dynamics of a decision-maker influenced Kelly auction. Markers indicate where $\omega_u(x) = 0$ on $\text{int}(\mathcal{X})$, and arrows represent the gradient flow. From (a) to (b), the agents' game transitions from non-monotone (admitting a local Nash and saddle point) to strongly monotone (single stable Nash).

point derivative-free convex optimization (Agarwal et al., 2010) up to log factors, where the log factor is precisely due to the agents running their algorithms for $\tau$ time-steps. The rate for the derivative free method is decidedly worse than for the stochastic repeated gradient method, owing to the extra estimator bias, yet the latter converges to a suboptimal equilibrium.

*Remark* 4.11 (Non-Strategic Decision-Dependence). As we show in Appendix J.3, if $\mathcal{D}_e(u)$ depends on $u$, the results in this section apply with minor changes to constants as long as $\ell$ satisfies mixture dominance (cf. Assumption J.10) and $W_1(\mathcal{D}_e(u), \mathcal{D}_e(w)) \leq L_{\mathsf{en}}\|u - w\|$ (cf. Assumption I.2).

## 5. Empirical Vignettes

In this section, we present some empirical results targeting some of the technical assumptions. These results elucidate the effect of the decision-maker's algorithm design space on the "shape" of the agent game and convergence behavior. A broader set of experiments is contained in Appendix D.

**Selecting $\tau$.** In many real-world settings the optimal $\tau$ is unknown as some of the constants on which it depends are properties of the agents' algorithms. We study a representative quadratic game to examine the effects of a decision-maker's choice of $\tau$. Indeed, many real-world problems (e.g. Bertrand/Cournot markets, multi-agent linear–quadratic games, and revenue-maximization models (Narang et al., 2023)) can be cast as *quadratic games* (cf. Appendix C). Here, the $i$-th agent's cost formulated as

$$f_i(x_i, x_{-i}) = \tfrac{1}{2}x_i^\top A_i x_i + x_i^\top x_{-i} + c_i^\top x_i + \phi_i(x, u),$$

where $\phi_i$ belongs to a quadratic family of incentives by which the decision-maker tries to nudge agents toward a desired outcome $(x^{\mathsf{d}}, u^{\mathsf{d}})$ (cf. Ratliff & Fiez (2020)). Figure 6 illustrates that several modest choices of $\tau$ suffice: larger $\tau$ results in faster convergence, yet the benefit of increasing $\tau$ is marginal. See Appendix D.1 for further details.

**Shaping the Landscape.** Another assumption made throughout is that the induced game $\mathcal{G}_u$ is strongly mono-

tone. It is natural to question whether or not the decision-maker can induce strong monotonicity. We explore this via a Kelly auction with $m$ resources, where $n$ agents submit bids $x_i \in \mathbb{R}_+^m$ for the resources, with the joint set of bids given by $x = (x_1, \ldots, x_n) \in \mathbb{R}^{m \cdot n}$. The $i$-th agent receives $\rho_{ij}^u(x) = \frac{q_j x_{ij}}{u_j + \sum_{l=1}^n x_{lj}}$ units of resource $j$ proportionate to their bid, and minimizes their loss $f_i^u(x_i, x_{-i}) = -\sum_{j=1}^m (a_i \rho_{ij}^u(x) - x_{ij})$ over $\mathcal{X}_i = \{x_i \in \mathbb{R}_+^m : \sum_{j=1}^m x_{ij} \leq b_i\}$. A common auction control mechanism is price floor $u = (u_1, \ldots, u_m) \in \mathbb{R}^m$ regulation. Shown in Appendix C.5, if $\mu = \max_i a_i \cdot \frac{\min_s\{q_s u_s\}}{\left(\sum_{s=1}^m u_s + \sum_{j=1}^n b_j\right)^3} > 0$, then the agents' game is $\mu$-strongly monotone. Figure 7 shows the decision-maker's action changes the equilibrium landscape from multiple stationary points to a single stable Nash. It is interesting to consider incorporating a design constraint on the shape of the agent game. See Appendix D.2 for details.

## 6. Discussion

We consider a novel class of stochastic time-varying Stackelberg games. We present finite-time efficiency estimates that are governed by the drift-to-noise ratio for the decision-maker influenced agent updates. We also identify two epoch-based algorithms that find two different notions of equilibria, the performatively stable Stackelberg equilibrium and the true Stackelberg equilibrium, and establish finite-time convergence rates. This work enables ample opportunities for future work. First, parameters intrinsic to the theoretical bounds are oft unavailable in practice, hence adaptive algorithms tuned to game theoretic settings may be especially useful. Second, future work might better capture the trade-offs in the performative gap and sample complexity. Finally, a particularly interesting direction is to estimate agent dynamics using techniques from adaptive control. Further discussion of these proposed directions are in Appendix B.

## Impact Statement

This paper presents work whose goal is to advance the field of Machine Learning. There are many potential societal consequences of our work, none which we feel must be specifically highlighted here.

## Acknowledgements

We graciously thank (in alphabetical order) Gavin Brown, Scott Geng, Tiffany Liu, and Rui Xin for their helpful comments and feedback. SO and EF were supported in part by NSF Awards 2019844, 2112471, and 2229876, and Microsoft Grant for Customer Experience Innovation. LJR and EF were supported in part by NSF CPS-1844729, and ONR YIP N00014-20-1-2571. EF is also supported in part by the NSF Graduate Research Fellowship Program. DD was supported by NSF DMS-2306322, NSF DMS-2023166, and AFOSR FA9550-24-1-0092 awards.

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

## Organization of Appendix

The appendix contains many sections, so here we provide a contents list to help the reader navigate the material.

- Appendix §A: **Related Work.** This section contains a discussion of related work

- Appendix §B: **Extended Discussion.** An extended discussion section containing implications for future work and proposed directions.

- Appendix §C: **Examples of Monotone Games.** This section contains examples of monotone games.

- Appendix §D: **Numerical Experiments.** Illustrative numerical experiments exploring both the limits of the theoretical results as well as semi-synthetic real-world simulations.

- Appendix §E: **Technical Lemmas.** Technical lemmas used to prove the theoretical results.

- Appendix §F: **Regularity of the Equilibrium Response.** Exposition on the regularity assumption on the equilibrium response of agents.

- Appendix §G: **Contracting Agent Updates.** Examples (and proofs) of $\rho$–contracting learning rules.

- Appendix §H: **Proofs for Oblivious Decision-Maker Setting.** Proofs for all the theoretical results for the setting in which the decision-maker is obliviously deploying a sequence of actions.

- Appendix §I: **Proofs for the Naïve Decision-Maker Setting.** Proofs for all the theoretical results for the setting in which the decision-maker is naïvely deploying a sequence of actions generated by running a repeated stochastic gradient method.

- Appendix §J: **Proofs for the Strategic Decision-Maker Setting.** Proofs for all the theoretical results for the setting in which the decision-maker is strategically deploying a sequence of actions that are selected via a derivative free stochastic method that allows the decision-maker to optimize through the smooth algorithmic response of the agents.

## A. Extended Related Work

Asymptotic equilibrium tracking is a long studied problem in single player stochastic optimization and stochastic approximation; see (Borkar, 2009; Kusher & Yin, 1997) and references therein. Our work focuses on obtaining convergence rates when the decision-maker faces a time-varying stochastic optimization problem subject to equilibrium constraints that are themselves varying in time. Below we highlight the most relevant work in this broad field, focusing on recent developments.

**Static Performative Prediction.**    The decision-maker's problem is analogous to the setting of performative prediction, first introduced in Perdomo et al. (2020), in the sense that the decision-maker faces a stochastic optimization problem where the distribution describing the environment depends on the actions of the decision-maker. Performative prediction, in turn, shares many features with the earlier work on stochastic optimization with decision-dependent probabilities (Hellemo et al., 2018) and strategic classification (Hardt et al., 2016; Mendler-Dünner et al., 2020). Numerous recent papers have developed algorithms and convergence guarantees in different performative prediction settings (Brown et al., 2022; Cutler et al., 2024; Drusvyatskiy & Xiao, 2023; Maheshwari et al., 2022; Mendler-Dünner et al., 2020; Miller et al., 2021; Narang et al., 2022).

In particular, Mendler-Dünner et al. (2020) develops the first stochastic optimization algorithms within the performative prediction setting. The subsequent work by Drusvyatskiy & Xiao (2023) reveals that all the typical stochastic optimization algorithms used in the classical static setting extend directly to the performative setting with no loss in efficiency. The work Cutler et al. (2024) moreover shows that the basic stochastic gradient method asymptotically achieves the best possible sample complexity among any estimation procedures.

Recent work by Conger et al. (2023) extends the specific sub-problem known as strategic classification to functional spaces by way of optimal transport in order to analyze the effects of the entire distribution (as compared to the mean) as a function of the decision-maker's action. Another interesting direction is explored in Narang et al. (2022) wherein the authors extend the performative prediction problem to multiple players and characterize the Nash equilibrium. Finally, work from Wood & Dall'Anese (2022), which finds equilibrium points that are analogs to performatively stable points.

**Time-Varying Stochastic Optimization & Performative Prediction.** Of the recent work on performative prediction, the most closely related work focuses on performative prediction problems that change dynamically over time in response to exogenous changes in the environment. This is in contrast to classical online tracking work (Fujita & Fukao, 1972; Kushner & Yin, 1997; Tsypkin & Nikolic, 1971; Tsypkin & Polyak, 1992), which, despite using several modern techniques and objectives such as accelerated gradients (Madden et al., 2020) and dynamic regret (Besbes et al., 2015) respectively, does not capture decision-dependence in the environment's drift.

Brown et al. (2022) introduced the notion of dynamics in the performative prediction problem through repeated risk minimization. Ray et al. (2022) introduce novel epoch-based algorithms for performative prediction when the environment is subject to geometrically decaying dynamics.

There has also been a recent surge on the empirical front in related fields such as recommendation design when the decision-maker recognizes that the user pool may be reactive (Cen et al., 2024). Cutler et al. (2023) provides convergence rates for gradient-based stochastic optimization methods over time-varying decision-dependent distributions. Wood & Dall'Anese (2023) develop a similar analysis for zero sum games, and provide bounds on tracking stochastic saddle point equilibrium. Finally, (Cen et al., 2024) studies performative and strategic effects in recommendation systems, and provides a theoretical model to study user strategization along with an empirical study.

Of these, the analysis in (Cutler et al., 2023) is most closely related to our work, especially in the oblivious decision-maker setting. We extend the analysis in that paper to strongly monotone games. Further, none of these works considers the Stackelberg setting in which the decision-maker (leader) faces multiple competing agents (followers) who are themselves learning and adapting.

**Bilevel Optimization & Stackelberg Games.** There is vast work on bilevel optimization and Stackelberg games (Başar & Olsder, 1998; Bracken & McGill, 1973; Colson et al., 2007; Dempe & Zemkoho, 2020; Stackelberg et al., 1952); the specific work most related to this paper focuses on settings where the agent problem is an equilibrium problem or variational inequality. In this setting, the literature is specialized to mathematical programming with equilibrium constraints. Prominent examples include settings where a leader optimizes over the outcome of a Cournot game (Sherali et al., 1983), or Stackelberg congestion games (Wardrop, 1952).

Typically it is assumed that the decision-maker has full knowledge of the agent game or can control the agent game through multiple specialized queries such as in recent work (Li et al., 2023; Maheshwari et al., 2023). There has also been work on incentive design when facing multiple adaptive agents such as (Ratliff & Fiez, 2020; Yang et al., 2020; 2022); however, the majority of this work makes the assumption that the decision-maker can estimate the preferences of the agents, can compute the *a priori* optimal solution to use as a benchmark, gives asymptotic convergence guarantees, or provides empirical results. In contrast, our work does not assume that the decision-maker has any knowledge of the agent preferences or update methods beyond belonging to a broad contractive update class.

## B. Extended Discussion, Implications for Future Work, and Proposed Directions

We consider a novel class of stochastic Stackelberg games, where updates from the decision-maker and the agents induces a time-varying game for both parties. We present finite-time efficiency estimates that are governed by the drift-to-noise ratio for the agents' updates for settings where the decision-maker sequentially deploys actions. The results motivate future work on better characterization of the tradeoffs in the performative gap and in sample complexity as this helps determine the most efficient class of algorithms to run in information limited settings.

We also identify two epoch-based algorithms that find two different notions of equilibria, the performatively stable Stackelberg equilibrium and the true Stackelberg equilibrium. We characterize the existence of the former equilibrium, and establish convergence rates. Illustrative numerical examples explore the theoretical assumptions and suggest many interesting directions for future work.

**Performative Gap vs Sample Complexity.** Indeed, the results motivate future work that captures the interplay between game theory, optimization, and learning. Better characterizing the tradeoffs in the performative gap, both in $\mathcal{U}$ as well as in cost, and in sample complexity is essential across a number of performative prediction settings. Additionally, having a better characterization the extent of performativity exists in a stochastic optimization system would enable decision-makers to determine which algorithmic approach (i.e., computationally expensive derivative free methods versus sub-optimal repeated

stochastic methods) is beneficial given the reactivity of its user base.

**Designing Adaptive Parameter Estimation Methods.** Our theoretical results also depend on a number of intrinsic parameters such as Lipschtiz constants which may not be readily available in practice. This suggests developing adaptive algorithms for learning in game-theoretic settings such as the ones explored in this paper. In the game theory literature, there are few adaptive methods in part because extending traditional methods from single-player optimization would require coordination, however, there is recent work on adaptive methods in distributed setting for monotone variational inequalities which can be leveraged.

**Methods for Estimating Opponent/Agent Models using Adaptive Control.** Finally, we examine the two extremes in terms of estimating the performative effects on the loss of the decision-maker. Indeed, recall the gradient decomposition from Equation 1:

$$\mathbb{E}_{z\sim\mathcal{D}(u)} \nabla_u \ell(u, z) + \frac{d}{dv} \mathbb{E}_{z\sim\mathcal{D}(v)} \ell(u, z)\big|_{v=u},$$

where the second term is the derivative of the loss through the reaction of the agents (and the non-strategic decision-dependent component of the environment). We examine what happens when the decision-maker updates by ignoring the reaction of the agents in its gradient and, at the other end of the spectrum, when the decision-maker estimates—via a derivative free method—the second term in (1). There is a natural intermediate formulation in which the decision-maker models the $\mathcal{D}(u)$ with a sufficiently rich function class. For instance, in prior work by Ratliff & Fiez (2020), in the context of adaptive incentive design the authors model best responding agents via reproducing kernel Hilbert spaces (RKHS)—which are known to have desirable properties such as persistence of excitation. In particular, the cost functions of the agents' are estimated via RKHS. Then, using the estimated costs, they constrain the incentive design problem to ensure that the (estimated) game amongst the agents has a positive definite Jacobian ($D_x\omega(x, u) > 0$) which is a sufficient condition for strong monotonicity. This requires an assumption on the model class being expressive enough to capture the true cost functions; the challenge here is that the model could of course be miss-specified, and this assumption is not *a priori* verifiable. Hence, what is needed is some uncertainty quantification or distributional robustness on top in order to give guarantees. This is an interesting direction of future research.

# C. Examples of Monotone Games

In this section, we provide several examples of strongly monotone games. Before diving into some examples, let us give some intuition for strong monotonicity via sufficient conditions, and discuss challenges related to relaxing this assumption.

## C.1. Strong Monotonicity of Agent Game

Recall that Assumption 2.1.i. Here we explore the strength of this assumption. In particular, this assumption requires that the decision maker is only deploying $u$'s such that the agents game is strongly monotone. A sufficient condition for this game to be strongly monotone for a given $u$ is that the game Jacobian is positive definite:

$$J_u(x) := \begin{bmatrix} \nabla_1^2 f_1^u & \nabla_{12} f_1^u & \cdots & \nabla_{1n} f_n^u \\ \nabla_{21} f_2^u & \ddots & \ddots & \vdots \\ \vdots & \ddots & \ddots & \nabla_{(n-1)n} f_{n-1}^u \\ \nabla_{n1} f_n^u & \cdots & \nabla_{n(n-1)} f_n^u & \nabla_n^2 f_n^u \end{bmatrix} \succ 0.$$

It is instructive to see what this implies via an example. Consider a revenue maximization game amongst the players

$$f_i^u(x_i, x_{-i}) = (A_{ii}x_i + A_{i,-i}x_{-i} + \xi_i + u_i)^\top x_i + \frac{\lambda_i}{2}\|x_i\|^2$$

here $u_i$ is some demand signal to correct for the implicit bias agent $i$ has about the demand.

For simplicity let's consider the two player case:

$$\omega_u(x) = (\nabla_1 f_1^u, \nabla_2 f_2^u) = ((2A_{11} + \lambda_1 I)x_1 + \xi_1 + u_1 + A_{12}x_2, (2A_{22} + \lambda_2 I)x_2 + \xi_2 + u_2 + A_{21}x_1)$$

$$J_u(x) = \begin{bmatrix} 2A_{11} + \lambda_1 I & A_{12} \\ A_{21} & 2A_{22} + \lambda_2 I \end{bmatrix}$$

Then,

$$\frac{1}{2}(J_u(x) + J_u^\top(x)) = \begin{bmatrix} 2A_{11} + \lambda_1 I & A_{12} + A_{21}^\top \\ A_{12}^\top + A_{21} & 2A_{22} + \lambda_2 I \end{bmatrix}$$

Then as long as the game when $u = 0$ is strongly monotone, so is any induced game.

Prior work by Ratliff & Fiez (2020) on adaptive incentive design with simultaneous utility estimation incorporates this condition as a constraint on the optimization problem (for choosing the next $u_t$). One direction for future work is specifying the space of $\mathcal{U}$ via a similar radial basis function method as in Ratliff & Fiez (2020), and then characterizing the additional constraint to be added to the epoch-based algorithms we propose herein. If this expansion results in a closed, convex set $\mathcal{U}$, then our results will apply and the decision-maker will only be choosing actions $u$ that retain strong monotonicity of the agents' game.

## C.2. Challenges to Relaxing Strong Monotonicity

Relaxing the monotonicity assumption leads to issues of not just non-uniqueness but **especially non-existence** of solutions (Facchinei & Pang, 2003; Rockafellar, 2018). To our knowledge there are not solution concepts for Stackelberg problems over non-monotone games. The rich literature on non-convex, non-concave zero sum games, especially in machine learning, is illustrative of the challenge which is only exacerbated by the lack of structure in general sum settings. In particular, there are many different local solution concepts and *no one in particular* that is widely accepted in zero sum settings (Daskalakis et al., 2023; Fiez et al., 2021a;b; Jin et al., 2020; Mangoubi & Vishnoi, 2021), and very few in general sum settings (Fiez et al., 2020).

There are two natural methods of defining solutions:

1. assume a locally monotone structure, or
2. define a solution concept with respect to the algorithm class (or regularizer) adopted by the agents.

The first option *demands the novel analysis in this paper for monotone settings* as that is what is exploited locally around the equilibrium to obtain convergence. The main technical concerns beyond the analysis in this paper are then ensuring saddle point avoidance and characterizing the lock in probability guaranteeing that the combined learning behavior remains in the appropriate local neighborhood—see the appendix of (Fiez et al., 2020) for asymptotic analysis that is illustrative. Essentially, it is necessary to bound the probability that iterates will get locked into a neighborhood around the equilibrium which can be more difficult in general sum settings without a lack of the equilibrium landscape outside the region of attraction, itself a difficult concept to characterize without local structure (e.g., bounds on the game Jacobian $D\omega_u(x)$).

The second option arises from the fact that most methods for solving non-monotone inequalities derive algorithms that leverage regularization (e.g., Tikhonov regularization is popular (Tatarenko & Kamgarpour, 2019)), and then define a solution concept *relative to a performance gap notion that depends on this choice of regularizer*. This is dissatisfying in a game theoretic sense as it requires the agents to coordinate on the choice of regularizer and then it gives no guarantees with respect to the incentive structure of the agents objectives. That being said, there may be interesting future research on understanding the relationship between algorithms adopted by strategic or learning agents and their underlying objectives and incentives.

## C.3. Quadratic Games

Consider the game defined by costs

$$f_i(x_i, x_{-i}) = \frac{1}{2} \begin{bmatrix} x_i \\ x_{-i} \end{bmatrix}^\top \begin{bmatrix} A_i & B_i^\top \\ B_i & D_i \end{bmatrix} \begin{bmatrix} x_i \\ x_{-i} \end{bmatrix} + \begin{bmatrix} a_i \\ b_i \end{bmatrix}^\top \begin{bmatrix} x_i \\ x_{-i} \end{bmatrix}, \tag{3}$$

where $A_i \in \mathbb{R}^{d_i \times d_i}$, $D_i \in \mathbb{R}^{d_{-i} \times d_{-i}}$, $B_i \in \mathbb{R}^{d_{-i} \times d_i}$, $a_i \in \mathbb{R}^{d_i}$ and $b_i \in \mathbb{R}^{d_{-i}}$ with $A_i = A_i^\top$ and $D_i = D_i^\top$. Further, we assume that $A_i \succ 0$ for each $i = 1, 2$. The $D_i$ matrices penalize player $i$ based solely on $x_{-i}$ and may often be negative or zero. Quadratic games are a useful approximation of the behavior of more complex games around an equilibrium. This game is strongly monotone if there exists $\mu \in (0, \infty)$ such that

$$\langle \omega(x) - \omega(x'), x - x' \rangle \geq \mu \|x - x'\|^2,$$

where
$$\omega(x) = (\nabla_1 f_1(x), \dots, \nabla_n f_n(x)) \quad \text{with} \quad \nabla_i f_i(x_i, x_{-i}) = A_i x_i + B_i^\top x_{-i} + a_i.$$

A sufficient condition for strong monotonicity is

$$J(x) = \nabla \omega(x) = \begin{bmatrix} A_1 & B_{12} & \cdots & & B_{1n} \\ B_{21} & A_2 & \ddots & & \vdots \\ \vdots & \ddots & \ddots & & B_{(n-1)n} \\ B_{n1} & \cdots & B_{n(n-1)} & & A_n \end{bmatrix} \succ 0, \quad \text{where} \quad B_i = \begin{bmatrix} B_{i1} \\ \vdots \\ B_{i(i-1)} \\ B_{i(i+1)} \\ \vdots \\ B_{in} \end{bmatrix}.$$

There are many important examples of quadratic games in economics and control theory. Below we highlight a few.

### C.3.1. OPEN LOOP LINEAR QUADRATIC DYNAMIC GAMES

One important class in control theory is that of linear quadratic dynamic games in open loop strategies. For simplicity, we write out the details for a $n = 2$ player game; however, these derivations easily extend to arbitrary but finite $n$. To that end, consider a two player linear quadratic dynamic game with open loop policies $\mathbf{v}_i = (v_{i,0}, \dots, v_{i,T-1})$ with costs

$$f_i(\mathbf{v}_1, \mathbf{v}_2) = \sum_{t=0}^{T-1} \frac{1}{2} z_t^\top Q_i z_t + \frac{1}{2} v_{i,t}^\top R_i v_{i,t} + v_{i,t}^\top R_{i,-i} v_{-i,t} + \frac{1}{2} z_T^\top Q_{i,f} z_T$$

$$z_{t+1} = F z_t + G_1 v_{1,t} + G_2 v_{2,t}, \ z_t \in \mathbb{R}^m.$$

Unfolding the dynamics and letting $Z = [z_0^\top, \dots, z_T^\top]^\top$, we have that $Z = W_1 \mathbf{v}_1 + W_2 \mathbf{v}_2 + \mathbf{F} z_0$ where

$$W_i = \begin{bmatrix} 0 & & \cdots & & 0 \\ G_i & 0 & \cdots & & 0 \\ FG_i & G_i & 0 & \cdots & 0 \\ \vdots & \vdots & \ddots & \ddots & \vdots \\ F^{T-2}G_i & F^{T-3}G_i & \cdots & G_i & 0 \\ F^{T-1}G_i & F^{T-2}G_i & \cdots & FG_i & G_i \end{bmatrix}, \quad i = 1, 2,$$

and $\mathbf{F} = \begin{bmatrix} I & F^\top & \cdots & (F^{T-1})^\top & (F^T)^\top \end{bmatrix}^\top$. Define the following cost matrices:

$$\mathbf{Q}_i := \text{diag}(Q_i, \dots, Q_i, Q_{i,f}) \in \mathbb{R}^{m(T+1) \times m(T+1)},$$
$$\mathbf{R}_i := \text{diag}(R_i, \dots, R_i) \in \mathbb{R}^{d_i T \times d_i T},$$
$$\mathbf{R}_{i,-i} := \text{diag}(R_{i,-i}, \dots, R_{i,-i}) \in \mathbb{R}^{d_i T \times d_{-i} T}.$$

Player $i$'s cost is

$$f_i(\mathbf{v}_i, \mathbf{v}_{-i}) = \tfrac{1}{2} \mathbf{v}_i^\top \mathbf{R}_i \mathbf{v}_i + \mathbf{v}_i^\top \mathbf{R}_{i,-i} \mathbf{v}_{-i} + \tfrac{1}{2}(W_1 \mathbf{v}_1 + W_2 \mathbf{v}_2 + \mathbf{F} z_0)^\top \mathbf{Q}_i (W_1 \mathbf{v}_1 + W_2 \mathbf{v}_2 + \mathbf{F} z_0).$$

Mapping back to the original quadratic cost form in (3), we have that

$$A_i = \mathbf{R}_i + W_i^\top \mathbf{Q}_i W_i, \quad B_i = \left( \mathbf{R}_{i,-i} + W_i^\top \mathbf{Q}_i W_{-i} \right)^\top, \quad D_i = W_{-i}^\top \mathbf{Q}_i W_{-i}$$
$$a_i^\top = z_0^\top \mathbf{F}^\top \mathbf{Q}_i W_i, \quad b_i^\top = z_0^\top \mathbf{F}^\top \mathbf{Q}_i W_{-i}.$$

The game Jacobian is given by

$$J(x) = \begin{bmatrix} A_1 & B_1^\top \\ B_2^\top & A_2 \end{bmatrix} = \begin{bmatrix} \mathbf{R}_1 + W_1^\top \mathbf{Q}_1 W_1 & \left( \mathbf{R}_{1,2} + W_1^\top \mathbf{Q}_1 W_2 \right) \\ \left( \mathbf{R}_{2,1} + W_2^\top \mathbf{Q}_2 W_1 \right) & \mathbf{R}_2 + W_2^\top \mathbf{Q}_2 W_2 \end{bmatrix}$$

In a typical linear quadratic regulator problem it is assumed that $\mathbf{R}_i \succ 0$ and $Q_i \succeq 0$ in order for solutions to exist (there are conditions that weaken these assumptions), and hence $A_i \succ 0$. In this case $A_i$ is non-degenerate for $i = 1, 2$, and hence

a sufficient condition for the game Jacobian to be positive definite is checking that either Schur complement is positive definite.

The goal of a decision-maker here might be to design 'pricing mechanisms' to influence the equilibrium; e.g., they may optimize over the matrices $(R_i, R_{i,-i})$ (Coogan et al., 2013; Ratliff et al., 2012).

### C.3.2. COURNOT AND BERTRAND COMPETITION

Both Cournot and Bertrand oligopoly models are monotone games under certain conditions on the cost parameters. In the Cournot model, firms choose quantities in non-cooperative competition, and the market determines the price of each good. On the other hand, in a Bertrand competition, firms set prices, and the market determines its demand for each type of good.

To see that these games are both strongly monotone, consider a setting with $n$ firms.

**Cournot Competition.** Each firm supplies the market with a quantity $x_i \in [0, B_i]$ of some good or service where $B_i > 0$ is firm $i$'s capacity for production. The market determines the price $P(x)$ for the good, where the pricing mechanism here $P(\cdot)$ is typically a decreasing function of the total supply to the market $\mathbf{1}^\top x = \sum_{i=1}^n x_i$. For example, a commonly adopted model is a linear pricing function of the form

$$P(x) = r - q \sum_{i=1}^n x_i, \quad \text{where } r, q > 0. \tag{4}$$

The $i$–th firm aims to maximize their utility which is given by $U_i(x) = x_i P(x) - c_i x_i$. Here, the first term $x_i P(x)$ is the revenue generated from selling $x_i$ goods in the market and the second term $c_i x_i$ is the cost of production.

Consider the game with costs $f_i(x) = -U_i(x)$ over strategy spaces $\mathcal{X}_i = [0, B_i]$. The game is strongly monotone if there exists $\mu > 0$ such that

$$
\langle \omega(x) - \omega(x'), x - x' \rangle = \left\langle \begin{bmatrix} -x_1 \nabla_1 P(x) - P(x) + c_1 \\ \vdots \\ -x_n \nabla_n P(x) - P(x) + c_n \end{bmatrix} - \begin{bmatrix} -x'_1 \nabla_1 P(x') - P(x') + c_1 \\ \vdots \\ -x'_n \nabla_n P(x') - P(x') + c_n \end{bmatrix}, x - x' \right\rangle
$$

$$
= \left\langle \begin{bmatrix} -x_1 \nabla_1 P(x) + x'_1 \nabla_1 P(x') - P(x) + P(x') \\ \vdots \\ -x_n \nabla_n P(x) + x'_n \nabla_n P(x') - P(x) + P(x') \end{bmatrix}, x - x' \right\rangle
$$

$$
\geq \mu \|x - x'\|^2.
$$

Using the linear form of $P(x)$ from (4), it is easy to compute

$$
\langle \omega(x) - \omega(x'), x - x' \rangle = q \left\langle \begin{bmatrix} x_1 - x'_1 \\ \vdots \\ x_n - x'_n \end{bmatrix}, \begin{bmatrix} x_1 - x'_1 \\ \vdots \\ x_n - x'_n \end{bmatrix} \right\rangle + q \left\langle \begin{bmatrix} \sum_i x_i - \sum_i x'_i \\ \vdots \\ \sum_i x_i - \sum_i x'_i \end{bmatrix}, \begin{bmatrix} x_1 - x'_1 \\ \vdots \\ x_n - x'_n \end{bmatrix} \right\rangle
$$

$$
= q \sum_i (x_i - x'_i)^2 + q \sum_i \left( \sum_j (x_j - x'_j)(x_i - x'_i) \right)
$$

$$
= q \|x - x'\|^2 + q \left( \sum_i (x_i - x'_i) \right)^2
$$

$$
\geq q \|x - x'\|^2
$$

so that the game is strongly monotone with $\mu = q > 0$. In a Cournot competition, the market reaches an equilibrium where all firms choose a quantity that is their *best response* to their competitors' quantities. This turns out to be an inefficient equilibrium, in that the equilibrium price is above the price in perfect competition and therefore firms earn a profit. A third party (such as a government entity) may intervene in the market by modulating the price $P(x)$ or by taxing individual firms (thereby increasing the cost of production $c_i x_i$) in order to move the market to an efficient equilibrium.

**Bertrand Competition.** In a Bertrand competition, where prices are the strategic variable, firms are incentivized to set their price slightly lower than the competition. Since all firms are so incentivized, they repeatedly drop the price until the price reaches the price in perfect competition wherein firms do not earn a profit. A third party may intervene in this market to improve uncertainties related to forecasting demand. For example, often times demand depends on exogenous time varying quantity or signal, such as gross domestic product, for which an individual firm may not have a good (low variance) forecaster.

To illustrate this, consider again $n$ firms, but now the strategies $x_i \in [0, B_i]$ are the prices instead of quantities of production. The firms seek to maximize their revenue in this setting which is given by $R_i(x_i, x_{-i}, u) = x_i F_i(x_i, x_{-i}, u)$ where $F_i$ is the marginal revenue function or demand curve given prices $x = (x_i, x_{-i})$. Here $u$ is some exogenous signal as described above. Then, for a fixed $u$, we have that

$$\langle \omega(x, u) - \omega(x', u), x - x' \rangle = \langle -x_i \nabla_i F_i(x, u) - F_i(x, u) - (-x'_i \nabla_i F_i(x', u) - F_i(x', u)), x - x' \rangle$$

so that, just like with the Cournot competition, if the marginal revenue is an affine function with parameters $(r, q)$ then the game is strongly monotone with $\mu = q$. The marginal revenue function, however, does not have to be linear for the game to be strongly monotone. Indeed a common form for the marginal revenue includes logarithmic terms (Bertsimas et al., 2015; Ratliff & Fiez, 2020). For instance consider marginal revenue function given by

$$F_i(x, u) = \log(x_i) + \theta_i^\top x + \xi_i + u_i,$$

where $(\theta_i, \xi_i)$ are parameters. Therefore we have that

$$\langle \omega(x, u) - \omega(x', u), x - x' \rangle = \left\langle \begin{bmatrix} -(\log(x_1) + 2\theta_{1,1}x_1 + \theta_{1,-1}x_{-1}) + (\log(x'_1) + 2\theta_{1,1}x'_1 + \theta_{1,-1}x'_{-1}) \\ \vdots \\ -(\log(x_n) + 2\theta_{1,1}x_1 + \theta_{1,-1}x_{-1}) + (\log(x'_n) + 2\theta_{n,n}x'_n + \theta_{n,-n}x'_{-n}) \end{bmatrix}, x - x' \right\rangle.$$

A sufficient condition for strong monotonicity is that the game Jacobian is positive definite. The game Jacobian is given by

$$\nabla \omega(x, u) = \begin{bmatrix} -2\theta_{1,1} - \frac{1}{x_1} & -\theta_{1,2} & \cdots & -\theta_{1,n} \\ -\theta_{2,1} & -2\theta_{2,2} - \frac{1}{x_2} & \ddots & \vdots \\ \vdots & \ddots & \ddots & \vdots \\ -\theta_{n,1} & \cdots & -\theta_{n,(n-1)} & -2\theta_{n,n} - \frac{1}{x_n} \end{bmatrix}$$

For the game Jacobian to be positive definite the prices have to be strictly positive, and there are constraints on the parameters $\theta$. One interesting question is if there is a natural mechanism that a third party could use to shape the game in order to ensure it is positive definite; in the example above, since $u_i$ enters linearly in the marginal revenue, it does not directly shape the game. However, e.g., if $u_i$ was a tariff affine in $x_i$, such as $a_i x_i + b_i$, then the diagonal terms of the Jacobian would be $-2\theta_{i,i} - 2a_i - \frac{1}{x_i}$ and the third party could design $a_i$ to ensure monotonicity.

## C.4. Strongly Convex Potential Game

A game $\mathcal{G} = (f_1, \ldots, f_n)$ is called a potential game (Monderer & Shapley, 1996) if there exists a potential function $\Phi : \mathcal{X} \to \mathbb{R}$ such that

$$f_i(x_i, x_{-i}) - f_i(x'_i, x_{-i}) = \Phi(x_i, x_{-i}) - \Phi(x'_i, x_{-i}), \quad \forall\, i \in [n], \, \forall\, x \in \mathcal{X}, \, x'_i \in \mathcal{X}_i.$$

If the potential function $\Phi$ is $\mu$–strongly convex, it follows from convex analysis that the game is $\mu$–strongly monotone (Rockafellar, 1970). The following is an example of such a game where there is a natural decision-maker influencing the outcomes.

**Example: Power Control in Shared Wireless Channel.** Another interesting class which has a similar structure to the Kelly auction is power control for shared wireless channels (d'Oro et al., 2015; Duvocelle et al., 2023; Facchinei & Kanzow, 2007; Tse & Viswanath, 2005). Consider $n$ wireless users that aim to transmit a set of packets to a common receiver over a set $\mathcal{S}$ of shared wireless channels (subcarriers). The aggregate received signal $y_s$ over the $s \in \mathcal{S}$ subcarrier is

$$y_s = \sum_{i=1}^n h_{i,s}\xi_{i,s} + z_s$$

where $\xi_{i,s}$ is the transmitted signal of user $i$ over the $s$-th subcarrier, $h_{i,s}$ is the corresponding channel coefficient, and $z_s$ is the aggregate interference-plus-noise received from all sources not in $[n]$ and for which we have that $z_s \sim \mathcal{N}(0, \sigma_s^2)$ is a Gaussian random variable. The average transmit power of user $i$ on subcarrier $s$ is $x_{i,s} = \mathbb{E}|\xi_{i,s}|^2$ and each users total power $x_i$ satisfies $x_i = \sum_s x_{i,s} \leq P_i$ for some $P_i > 0$. Then the strategy space of user $i$ is

$$\mathcal{X}_i = \left\{ x_i \in \mathbb{R}^{|\mathcal{S}|} \,\middle|\, x_{i,s} \geq 0 \quad \text{and} \quad \sum_{s \in \mathcal{S}} x_{i,s} \leq P_i \right\}.$$

Each users transmission rate is given by Shannon's formula:

$$R_i(x_i, x_{-i}) = \sum_{s \in \mathcal{S}} \log\left(1 + \varphi_{i,s}(x)\right) = \sum_{s \in \mathcal{S}} \left( \log\left(\sigma_s^2 + w_s(x)\right) - \log\left(\sigma_s^2 + \sum_{j \neq i} v_{j,s} x_{j,s}\right) \right),$$

where $w_s(x) = \sum_{i \in [n]} |h_{i,s}|^2 \cdot x_{i,s}$ for each $s \in \mathcal{S}$ and such that $|h_{i,s}|^2$ is the channel gain of user $i$ over subcarrier $s$. Additionally, the term

$$\varphi_{i,s}(x) = \frac{|h_{i,s}|^2 \cdot x_{i,s}}{\sigma_s^2 + \sum_{j \neq i} |h_{j,s}|^2 \cdot x_{j,s}}$$

is the signal-to-inference-and-noise ratio. The network operator (decision-maker) aims to design a pricing scheme for the channel so as to induce an efficient equilibrium. For example, in a cognitive radio scenario the users described above are *secondary users* that are free riding on the network and cause interference on the primary users and therefore the the network operator needs to ensure that the system's users meet the quality of service guarantees that they have already paid for—typically in the form of minimum rate requirements or maximum interference tolerance per subcarrier. How this is achieved is by designing a pricing mechanism that consists of a flat spectrum access price $\pi_0 : \mathbb{R}^{|\mathcal{S}|} \to \mathbb{R}$ and a user specific price $\pi_i : \mathcal{X}_i \to \mathbb{R}$. Thus user $i$'s utility is given by

$$U_i(x) = R_i(x) - (\pi_0(w) + \pi_i(x_i)) \quad \text{where} \quad w = (w_1, \ldots, w_s).$$

This game admits an exact potential function (d'Oro et al., 2015):

$$\Phi(x) = \sum_{s \in \mathcal{S}} \log(\sigma_s^2 + w_s) - \pi_0(w) - \sum_{i \in [n]} \pi_i(x_i).$$

To align with economic considerations on diminishing returns, it is common to assume that the pricing functions $\pi_0$ and each $\pi_i$ is non-decreasing and convex in each of its arguments, and they are Lipschitz continuous. This ensures that $\Phi(x)$ is concave (though not necessarily strongly). Some regularization of the potential function would ensure its strongly concave but would induce a different set of Nash equilibrium than optimizing $\Phi$. It is interesting to see how much regularization is introduced impacts the difference between the induced sets of equilibrium.

### C.5. Kelly Auction: Resource Allocation Mechanisms

Resource allocation problems are another interesting class of games that can be strongly monotone games. Consider a service provider with a number of divisible resources $s \in \mathcal{S} = \{1, 2, \ldots, m\}$. These resources could be things like server time, bandwidth, ad space, amongst many other divisible resources. Now, suppose there are $n$ agents to which these resources can be leased. Each agent $i$ submits a (monetary) bid $x_i = (x_{i1}, \ldots, x_{im}) \in \mathbb{R}_+^m$ for the resources; call the joint set of bids $x = (x_1, \ldots, x_n) \in \mathbb{R}^{m \cdot n}$. The bids are non-negative and satisfy a budget constraint for each agent: $\sum_{s \in \mathcal{S}} x_{i,s} \leq b_i$ for some $b_i \in [0, \infty)$.

A common mechanism for allocating resources in this setting is the *Kelly mechanism* (Kelly et al., 1998). Under this mechanism, each agent receives an amount of resource $s$ in proportion to their bid, as a pro-rata percentage of the other agents' bids. That is, each agent receives

$$\rho_{is}^u(x) = \frac{q_s x_{is}}{u_s + \sum_{j=1}^n x_{js}}$$

units of the resource $s$. Here, the parameter $q_s$ is the total available units of resource $s$ and $u_s$ is the *barrier to entry* for $u = (u_1, \ldots, u_m) \in \mathbb{R}^m$.

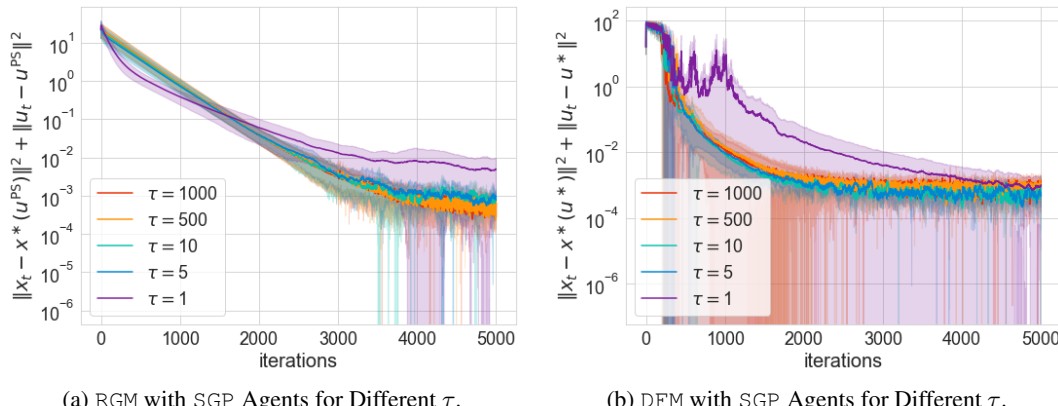

(a) `RGM` with `SGP` Agents for Different $\tau$.  (b) `DFM` with `SGP` Agents for Different $\tau$.

*Figure 6.* Effects of $\tau$ on equilibrium convergence: we explore the effect of the decision-maker's choice of $\tau$ on the ultimate equilibrium in a convex quadratic game. As expected the larger $\tau$ is, the smaller the equilibrium error.

Altogether, we can define the loss for each agent as

$$f_i^u(x_i, x_{-i}) = -\sum_{s=1}^m (a_i \rho_{is}^u(x) - x_{is}).$$

The strategy space of the $i$–th agent is $\mathcal{X}_i = \{x_i \in \mathbb{R}_+^m : \sum_{s=1}^m x_{is} \le b_i\}$. Using analogous analysis to (Lin et al., 2021), it is easy to see that this game is $\mu$-strongly monotone with

$$\mu = \max_i a_i \cdot \frac{\min_s\{q_s u_s\}}{\left(\sum_{s=1}^m u_s + \sum_{j=1}^n b_j\right)^3}.$$

If the choice of the decision-maker is the floor prices $u = (u_s)_s$, then they can control whether or not the game is strongly monotone.

# D. Numerical Experiments

Code is available at https://github.com/SewoongLab/stoch-stackelberg.

In this section, we present numerical examples that are aimed at exploring the limits of the theory, and numerical examples that explore applying the theory to semi-synthetic simulations based on real-world data. In particular, we use real-world data to create semi-synthetic experiments for price setting in ride-share markets. We explore two decision-maker actions: providing a demand signal to shape the equilibrium outcome, and modulating the price (via taxes or incentives) to estimate the price elasticities.

In the subsections that follow, we first explore the choice of $\tau$ which is a parameter that the decision-maker gets to set, but depends on instance dependent quantities related to the agents' game which in practice may not be *a priori* known. We show that even if $\tau$ is set modestly, convergence is still possible. A future direction of research is developing adaptive methods to estimate key quantities. Next, we explore relaxing the regularity assumptions made on the agents' game (monotonicity) and on the decision-maker's loss (convexity). We show that the decision-maker can control whether or not the agents' game is strongly monotone through their action, and that local convergence is possible in the absence of convexity. Future directions consider designing constraints on the decision-maker problem to ensure monotonicity, and local convergence results in the absence of convexity. We also comment on the key challenges that arise theoretically when these assumptions on regularity are relaxed. Finally, we explore semi-synthetic real world examples that leverage data from ride-share markets.

### D.1. Effect of Choosing $\tau$ without a priori Knowledge of Agent Game Parameters

In practice, the decision-maker may not have access to the precise constants that determine the theoretically correct choice of the epoch length $\tau$ since some of these constants are determined by the private cost functions of the agents. In this section,

we explore the choice of $\tau$ on the tracking error of the agents' equilibrium relative to the appropriate equilibrium—namely, the performatively stable equilibrium if the decision-maker employs the repeated gradient method (RGM) and the Stackelberg equilibrium if the decision-maker employs the derivative free method (DFM). To conduct this exploration, we generate a random quadratic game instance where the decision-maker is deploying actions to with the objective of tracking a desired equilibrium. The decision-maker may not know the agent game-related hyperparameters *a priori* and thus may not be able to set $\tau$ optimally as noted. Each player has cost

$$f_i(x_i, x_{-i}) = \frac{1}{2} x_i^\top A_i x_i + x_i^\top B_i x_{-i} + c_i^\top x_i + \phi_i(x, u) \quad \text{for } i \in [n],$$

and where $\phi_i(x, u)$ belongs to a quadratic family of incentives, e.g., of the form $\phi_i(x, u) = x^\top Q_i x + x^\top R_i u + q_i^\top x$. The goal of the decision maker is to design the input $u = (u_1, \ldots, u_n) \in \mathbb{R}^d$ such that the agents converge to a desired equilibrium $(x^{\mathrm{d}}, u^{\mathrm{d}})$. For instance, if the decision-maker's loss is $\mathbb{E}_{\xi \sim \mathcal{D}_e(u)} \ell(u, (x, \xi))$, then $(x^{\mathrm{d}}, u^{\mathrm{d}})$ may be defined as the globally optimal tuple for the decision-maker's loss as if they were able to control both $u$ and $x$. In the examples we consider in this section, we let $(x^{\mathrm{d}}, u^{\mathrm{d}})$ be so defined for a randomly generated convex loss $\ell$ and we let the decision-maker optimize the auxiliary loss $\mathbb{E}_{\xi \sim \mathcal{D}_e}[\|x - x^{\mathrm{d}}\|^2 + \|u - u^{\mathrm{d}}\|^2]$ where $x$ is generate by the agents' algorithmic response to $u$. In the accompanying code-base, the parameters of the game can be changed.

Figure 6 demonstrates that even if the decision-maker does not know the constants that define the optimal $\tau$ (since these are related to the agent game instance), there are choices of $\tau$ for which the decision-maker's algorithm still converges. Perhaps surprisingly, it shows more specifically for the quadratic game instance we consider, that it suffices to pick $\tau = 1$, meaning the agents only perform one update of their action in each round. That being said, the choice of $\tau$ impacts the convergence rate as can be seen in the plots: larger $\tau$ results in faster convergence as expected, but the benefit of increasing $\tau$ is marginal after a certain point. The reader can explore different game instances by modifying the provided code base.

## D.2. Relaxing Regularity Assumptions

We also consider numerical examples from Kelly auctions from economics. Here, we explore a synthetic Kelly auction between two players participating in the auction, where the amount bid is influenced by the marginal utilities of the competitors as well as the actions from the decision-maker.

### D.2.1. KELLY AUCTION GAME FORMULATION

Consider a service provider with a number of divisible resources $s \in \mathcal{S} = \{1, 2, \ldots, m\}$. These resources could be things like server time, bandwidth, ad space, amongst many other divisible resources. Now, suppose there are $n$ agents to which these resources can be leased. Each agent $i$ submits a (monetary) bid $x_i = (x_{i1}, \ldots, x_{im}) \in \mathbb{R}_+^m$ for the resources; call the joint set of bids $x = (x_1, \ldots, x_n) \in \mathbb{R}^{m \cdot n}$. The bids are non-negative and satisfy a budget constraint for each agent: $\sum_{s \in \mathcal{S}} x_{i,s} \leq b_i$ for some $b_i \in [0, \infty)$.

A common mechanism for allocating resources in this setting is the *Kelly mechanism* (Kelly et al., 1998). Under this mechanism, each agent receives an amount of resource $s$ in proportion to their bid, as a pro-rata percentage of the other agents' bids. That is, each agent receives

$$\rho_{is}^u(x) = \frac{q_s x_{is}}{u_s + \sum_{j=1}^n x_{js}}$$

units of the resource $s$. Here, the parameter $q_s$ is the total available units of resource $s$ and $u_s$ is the *barrier to entry* or *floor price* for $u = (u_1, \ldots, u_m) \in \mathbb{R}^m$.

Altogether, we can define the loss for each agent as

$$f_i^u(x_i, x_{-i}) = -\sum_{s=1}^m (a_i \rho_{is}^u(x) - x_{is}).$$

The strategy space of the $i$–th agent is $\mathcal{X}_i = \{x_i \in \mathbb{R}_+^m : \sum_{s=1}^m x_{is} \leq b_i\}$. Using straightforward analysis of the eigenstructure of the game Jacobian $D\omega_u(x)$, it is easy to see that this game is $\mu$-strongly monotone with

$$\mu = \max_i a_i \cdot \frac{\min_s \{q_s u_s\}}{\left( \sum_{s=1}^m u_s + \sum_{j=1}^n b_j \right)^3}.$$

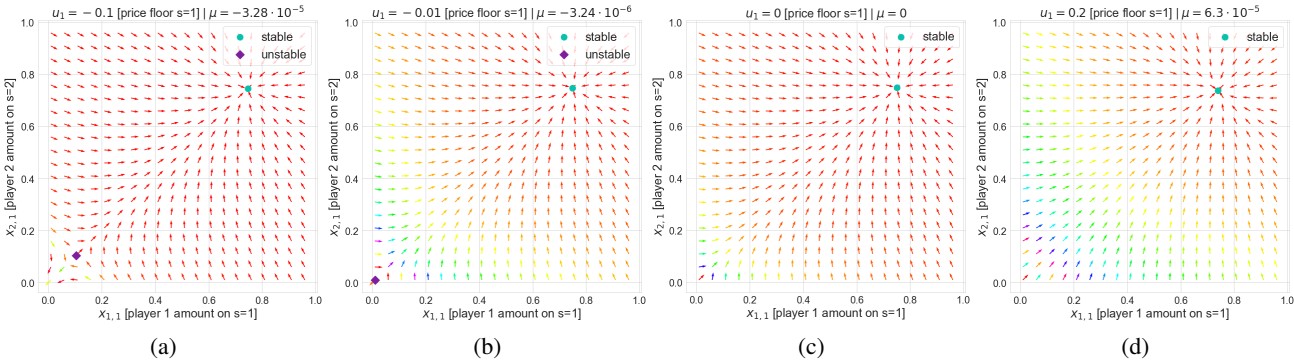

*Figure 7.* Decision-maker equilibrium landscape design in a two-player Kelly auction: projected phase plot for the dynamics of the induced game. The markers indicated different candidate equilibrium—namely, where $\omega_u(x) = 0$ for $x \in \text{int}(\mathcal{X})$. Arrows indicate the direction of the gradient flow relative to the equilibrium—namely, if the arrows are pointing towards an equilibrium, then those are gradient directions which are attracted to the equilibrium. The magnitude is indicated by the color spectrum. In this example, we explore *saddle node bifurcation* in terms of varying $u_1$. In (a-b), $u_1$ is negative, making the agents' game not strongly monotone; here, we find two local optima, a single stable Nash and a saddle point. In (c-d), as $u_1$ becomes non-negative, the problem becomes strongly monotone, with only a single stable Nash. This demonstrates how the choice of decision-maker action can control the equilibrium landscape.

This has been shown in prior works including that of Lin et al. (2021). If the choice of the decision-maker is the floor prices $u = (u_s)_s$, then they can control whether or not the game is strongly monotone.

On the decision-maker side, there are many potential cost functions that are reasonable. For example, the auctioneer may care about maximizing their revenue and therefore their cost is

$$\mathcal{L}_{\text{rev}}(u) = -\sum_{i=1}^{n}\sum_{j=1}^{m} x_{is}\rho_{is}^{u}(x).$$

Alternatively, the decision-maker may care about the total agent welfare in which case their cost is given by

$$\mathcal{L}_{\text{welf}}(u) = \sum_{i=1}^{n} f_i^{u}(x) = -\sum_{i=1}^{n}\sum_{s=1}^{m} (a_i\rho_{is}^{u}(x) - x_{is}).$$

We note that in general, the decision-maker's objective is *not* strongly convex in $u$; accordingly, this makes simple Kelly auctions a useful framework for characterizing the strength of our various assumptions.

### D.2.2. EFFECT OF DECISION-MAKER ON MONOTONICITY AND AGENTS' EQUILIBRIUM LANDSCAPE

A decision-maker may alter the structure of the auction—for example, changing the barrier to entry or levying a subsidy or tax on the agents—to encourage behavior that aligns with a desired outcome, such as revenue maximization for the auctioneer or welfare maximization for the participants. Indeed, let us demonstrate how a decision-maker's action can induce monotonicity. Within the notation of our example, consider a setting where the decision-maker selects $u$; in other words, the decision-maker intervenes by changing the barrier to entry (floor price) in the auction. In this synthetic example, we consider a Kelly auction of $m = 2$ resource types and define the auction's parameters as

$$a = \begin{bmatrix} 1 & 1 \end{bmatrix}, \quad q = \begin{bmatrix} 1 & 1 \end{bmatrix}, \quad \text{and} \quad b = \begin{bmatrix} 10 & 10 \end{bmatrix}.$$

Accordingly, the constraints on the agents' action space can be characterized as $\sum_s x_{i,s} \leq b_i$ and $x_{i,s} \geq 0 \ \forall i, s$. Recall that the Kelly auction is strongly monotone when

$$\mu = \max_i a_i \cdot \frac{\min_s\{q_s u_s\}}{(u_1 + u_2 + b_1 + b_2)^2} > 0.$$

Define the decision-maker's action $u$ as $u = \begin{bmatrix} u_1 & 1 \end{bmatrix}$, and note that $\min_s\{q_s u_s\} = \min\{u_1, 1\}$. Accordingly, the decision-maker's action $u_1$ controls strong monotoncity. Figure 7 demonstrates the degree to which the game monotonicity (and thus,

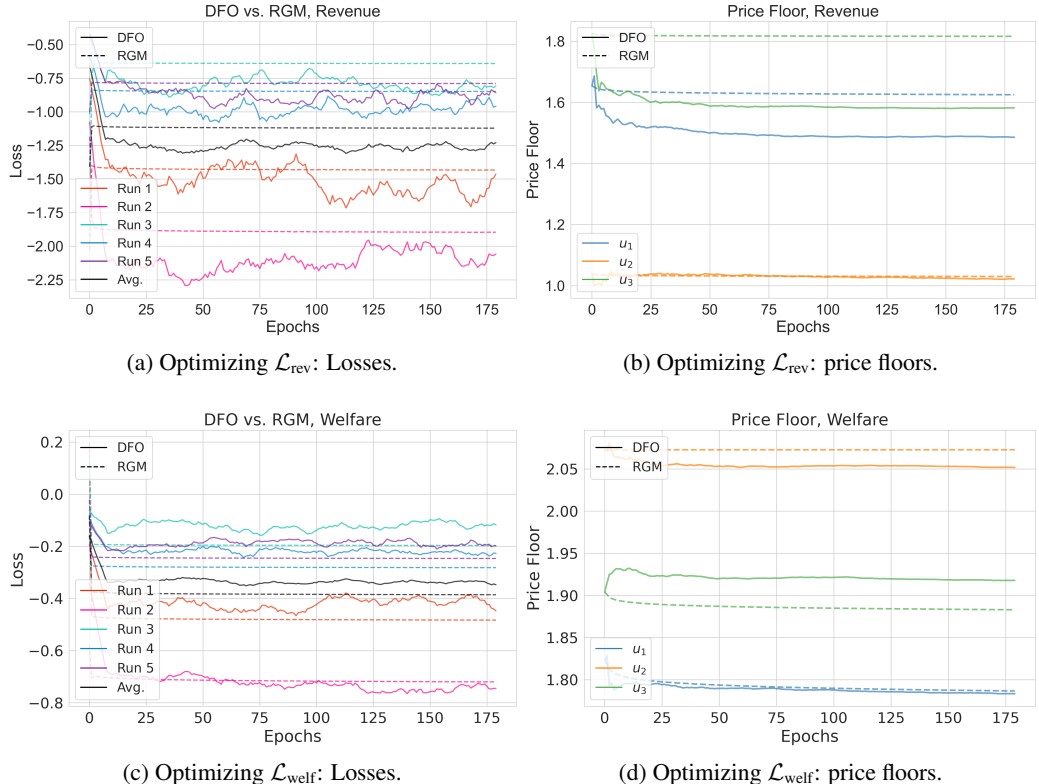

(a) Optimizing $\mathcal{L}_{\text{rev}}$: Losses.

(b) Optimizing $\mathcal{L}_{\text{rev}}$: price floors.

(c) Optimizing $\mathcal{L}_{\text{welf}}$: Losses.

(d) Optimizing $\mathcal{L}_{\text{welf}}$: price floors.

*Figure 8.* Local convergence in non-convex Kelly Auction optimizing for agent welfare or auctioneer revenue. We simulate convergence over five distinct random seeds. In (a), we find that DFO tends to find local Stackelberg equilibria with lower cost than local performatively stable Stackelberg equilibria found by RGM when optimizing $\mathcal{L}_{\text{rev}}$, and in (2) confirm that these equilibria are distinct. For similar experiments optimizing the social welfare $\mathcal{L}_{\text{welf}}$, (c-d) demonstrate that while again finding distinct solutions, RGM finds local performatively stable Stackelberg equilibria with lower cost than local Stackelberg equilibria found by DFO.

the equilibrium landscape), even in a simple setting, depends on the decision-maker's action. We see that as $u_1$ varies from $-0.1$ to $0.2$ we start from having a single stable Nash and a saddle point and eventually only a single stable Nash so that through the choice of $u$ the decision-maker can control the equilibrium landscape.

An interesting direction for future work would be including a constraint in the decision-maker's optimization problem that ensures the induced game amongst agents is strongly monotone. However, in this paper, we consider settings where the decision-maker a priori has no information on the cost functions of the agents. Prior work in the asymptotic regime has exampled the use of methods from adaptive control to estimate the cost functions and then use the estimated cost functions to incorporate a constraint that the game is strongly monotone (Ratliff & Fiez, 2020). A natural question, therefore, is whether or not we can develop utility estimation techniques in the non-asymptotic regime and whether its worth the additional sample complexity. In fact, such techniques might interpolate between the naïve and strategic settings considered in this paper.

### D.2.3. LOCAL CONVERGENCE IN NON-CONVEX DECISION-MAKER LOSS

In general, the decision-maker's cost could be non-convex in $u$. As a result, while the auction played by the agents might be strongly monotone with a single stable Nash equilibrium, our established convergence and sample complexity guarantees for the *Stackelberg game* need not hold. Accordingly, we use this setting to empirically investigate the convergence properties of our algorithms when the decision-maker has a non-convex cost. In our synthetic example, we consider a Kelly auction over $m = 3$ resource types and auction parameters

$$q = \begin{bmatrix} 4 & 2 & 3 \end{bmatrix}, \, a = \begin{bmatrix} 1 & 1 & 1 \\ 1 & 1 & 1 \end{bmatrix}, \, b = \begin{bmatrix} 1.5 & 2 \end{bmatrix}$$

Here, we restrict the decision-maker's action space to $\mathcal{U} = \{u \in \mathbb{R}^3 : 1 \leq u_s \leq 2.5 \ \forall s \in [3]\}$ and compare the repeated gradient method (RGM) and the derivative free method (DFO) in two settings: minimizing $\mathcal{L}_{\text{rev}}$ and $\mathcal{L}_{\text{welf}}$. We display our results in Figure 8. Here, in simulations run over five distinct seeds, we find that our algorithms nonetheless converge at the predicted rate, but to distinct *local* Stackelberg equilibria. In Figure 8(a), when optimizing $\mathcal{L}_{\text{rev}}$, DFO generally outperforms RGM though takes longer to converge, a finding consistent with our results in the strongly convex case; Figure 8(b) likewise demonstrates that RGM and DFO converge to distinct local solutions in the decision-maker's objective. In contrast, Figure 8(c-d) indicates that optimizing $\mathcal{L}_{\text{welf}}$ with RGM can result in *lower* cost than DFO. In other words, the cost at local performatively stable equilibria can be lower than that of local Stackelberg equilibria obtained from the same initialization. Formal theoretical characterizations of the equilibrium dynamics and convergence rates in non-convex settings remains an interesting line of future work.

### D.3. Quadratic Ride-Sharing Game: Semi-Synthetic Simulations

We consider an example from ride-sharing markets. Demand signals may be used to create more efficient ride share markets without reducing individual revenue streams by enabling information-limited firms to recover latent demand. Using an analogous set up as in Narang et al. (2022), we explore semi-synthetic competition between two ride-share platforms seeking to maximize their revenue given that the demand they experience is influenced by their own prices as well as their competitors. The data we use is from a prior Kaggle competition.[4]

**Game Formulation.**   Each firm divides rides into \$5 price bins ranging from \$10 to \$30, and then chooses a additive surge on top of that price as described in Narang et al. (2022). The social cost is given by

$$\mathcal{L}(u) = \mathop{\mathbb{E}}_{\xi \sim \mathcal{D}_0} [f_1^u(x_1, x_2) + f_2^u(x_1, x_2)]$$

where each firm $i$'s cost is given by

$$f_i^u(x) = \mathop{\mathbb{E}}_{z_i \sim \mathcal{D}_i(x_i, x_{-i}, u)} \left[ -\frac{1}{2} z_i^\top x_i + \frac{\lambda_i}{2} \|x_i\|^2 \right].$$

Notice that each firm's cost is decision-dependent in that the distribution on $z_i$ depends on not only $x_i$ but also the actions of the other firms and the decision-maker $(x_{-i}, u)$. The action $x_i$ is a vector of additive surge prices to each \$5 dollar bin and across the eleven different physical locations, and the term $z_i^\top x_i$ represents the added revenue across bins achieved via surge pricing. The term $\frac{\lambda_i}{2}\|x_i\|^2$ is a regularizer: namely, firm $i$ does not want to charge too high of surge prices and prefers to spread the surge prices it charges across the locations. The random variable capturing the demand vector for firm $i$'s service is modeled as

$$z_i := \xi_i + A_{i,i} x_i + A_{i,-i} x_{-i} + u_i$$

where the vector $A_i := \begin{bmatrix} A_{i,i} & A_{i,-i} \end{bmatrix}$ contains the price elasticity parameters for firm $i$; in particular, $A_{i,i}$ is the price elasticity of demand for firm $i$'s service given changes in firm $i$'s price $x_i$ and $A_{i,-i}$ is the price elasticity of demand for firm $i$'s service given changes in all other firms' prices $x_{-i}$. In the simulations, we set these in the same manner as described in Narang et al. (2023). Here, the decision-maker's action is given by $u = (u_1, u_2)$ where $u_i$ acts as a demand signal informing the firm $i$ about latent demand. The regularization parameter $\lambda_i$ serves to reduce the surge multiplier; that is, the firm does not want to inadvertently set the price too high.

#### D.3.1. ESTIMATION OF PRICE ELASTICITIES

In many applications, a decision-maker may want to estimate the reactivity of agents. For example, a local government may seek to estimate the price elasticity of agents—in this case ride-share companies—in a ride-share market so that they can then subsequently set taxes or subsidies on these agents or even the other side of the market (passengers).

In the context of the ride-share market example above, if a decision-maker aims to estimate each of the $A_i$'s—i.e., the price elasticities of players—then they can run online least squares where in each round $t$ they first query the environment and observe

$$z_{t,i} = \xi_{t,i} + A_i x_t \quad \text{where} \quad A_i = \begin{bmatrix} A_{i,i} & A_{i,-i} \end{bmatrix}.$$

---

[4]Data is publicly available: https://www.kaggle.com/datasets/brllrb/uber-and-lyft-dataset-boston-ma

Then, they perturb the prices with actions $u_i$ for each player, and observe

$$q_{t,i} = A_i(x_t + u_{t,i}) + \xi'_{t,i}.$$

With these two queries, the decision maker updates their estimate of the price elasticities as follows:

$$\hat{A}_{t+1,i} = \hat{A}_{t,i} + \nu_t(q_{t,i} - z_{t,i} - \hat{A}_{t,i}u_{t,i})u_{t,i}^\top,$$

where $\nu_t$ is the step size.

In Narang et al. (2022), the authors show that if multiple firms are running a stochastic gradient method while simultaneously estimating their own price elasticities, then the joint strategy of the firms converges to the Nash equilibrium and the estimates of the price elasticities converge to the true values as long as the firms inject noise satisfying the following assumption.

**Assumption D.1.** The sequence $u_t = (u_{t,1}, \ldots, u_{n,t}) \in \mathbb{R}^d$ is a zero-mean random vector that is independent of $x_t$, and independent of the previous random vectors $\{u_s \mid s < t\}$. Moreover, there exists constants $c_l, R > 0$ and $c_{u,i} > 0$ for each $i \in [n]$ such that for all $t \geq 0$ and $i \in [n]$ the random vector $v_i := u_{i,t}$ satisfies

$$0 \prec c_l \cdot I \preceq \mathbb{E}[v_i v_i^\top], \quad \mathbb{E}\|v_i\|^2 \leq c_{u,i}, \quad \text{and} \quad \mathbb{E}[\|v_i\|^2 v_i v_i^\top] \preceq R^2 \mathbb{E}[v_i v_i^\top].$$

In our setting, it is an external third party that is injecting "noise" (which can be interpreted here as a random demand signal) and they are decaying that noise over time with the goal of obtaining an approximate estimate of the price elasticities and then leaving the base system close to the nominal Nash equilibrium. The firms in this case are assumed to know their price elasticities $A_i$.[5] There is a tradeoff between how quickly the noise is decaying and the accuracy of the price elasticity estimates as well as where the agents actions end up relative to the nominal Nash equilibrium $x^\star$—i.e., the Nash equilibrium of the game $\mathcal{G}_u$ where $u = 0$.

Indeed, fixing a $\lambda \in (0, 1)$ and some horizon $T$, suppose that the decision-maker samples $u_t$ from $\mathcal{N}(0, \sigma_t \cdot I_d)$ where $\sigma_{t+1} = \lambda \sigma_t$. Then, in Assumption D.1 we have

$$c_l = \lambda^T \sigma_0, \quad c_{u,i} = d_i, \quad \text{and} \quad R^2 = 3 \max_{i \in [n]} d_i.$$

From Lemma 21 of Narang et al. (2022), we have that

$$\mathbb{E}\|\hat{A}_T - A\|_F^2 \leq \frac{\max\left\{\left(1 + \frac{2R^2}{\lambda^T \sigma_0}\right)\|\hat{A}_0 - A\|_F^2, \frac{8}{\lambda^{2T}\sigma_0^2}\sum_{i=1}^n \mathrm{Tr}(\Sigma_{0,i})c_{u,i}\right\}}{T + \frac{2R^2}{\lambda^T \sigma_0}}$$

where $\Sigma_{0,i} = \mathrm{diag}(\sigma_{0,1}, \ldots, \sigma_{0,n})$.

To bound the effect on the firms we analyze how injecting $u_t$ impacts the convergence to the nominal Nash equilibrium. In this case, the firms are running stochastic gradient play with noise

$$\zeta_{t,i} = \xi_{i,t} + A_i u_{t,i}$$

where $\xi_{i,t} \sim \mathcal{D}_0$ and $u_{t,i} \sim \mathcal{N}(0, \sigma_{t,i} \cdot I_d)$. Hence, in the proof of the one step contraction for stochastic gradient play (cf. Lemma G.4), we have that

$$P_1 \leq \frac{(\sigma_a + \sigma_t)^2}{2\nu_1} + \frac{\nu_1 \mathbb{E}_t\|x_{t+1} - x_t\|^2}{2} = \frac{(\sigma_a + \lambda^t \sigma_0)^2}{2\nu_1} + \frac{\nu_1 \mathbb{E}_t\|x_{t+1} - x_t\|^2}{2}$$

so that, after some algebra, we have that

$$\mathbb{E}_t\|x_{t+1} - x^*\|^2 \leq \frac{1}{1 + \gamma\mu}\|x_t - x^*\|^2 + \frac{2\gamma^2(\sigma_a + \lambda^t \sigma_0)^2}{1 + \gamma\mu}.$$

---

[5]Another interesting example would be having firms that are simultaneously estimating $A_i$ and looking at the combined effects of injecting noise by the decision-maker and the firms. If multiple entities are injecting noise it could be the case that the combined effect reduces the time for convergence or makes it worse depending on the injected noise distributions.

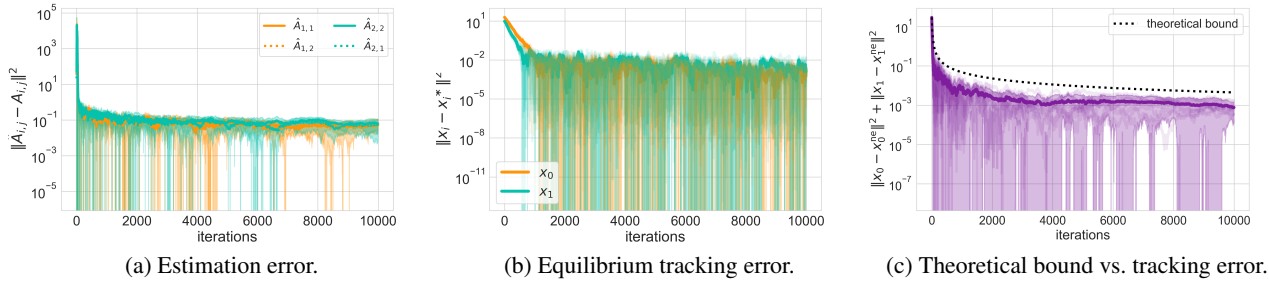

(a) Estimation error.       (b) Equilibrium tracking error.       (c) Theoretical bound vs. tracking error.

*Figure 9.* (a) Estimation error for the price elasticities $\hat{A}_{1,1}$, $\hat{A}_{1,2}$, $\hat{A}_{2,1}$, and $\hat{A}_{2,2}$. (b) Agent tracking error. (c) The black dashed line is the theoretical bound on tracking error versus number of iterations, and the purple trajectories are the actual tracking error. For each of the plots, we run ten different random seeds and show the mean, the mean $\pm 1$ standard deviation, and the actual trajectories using lower opacity. Note that although the magnitude of noise injected into the game by the decision-maker decays over time, the decision-maker still has sufficient information to estimate the price elasticities.

We know that if the firms run stage-wise stochastic gradient play with some target accuracy $\varepsilon > 0$, then the agents obtain a Nash equilibrium in a total of $T$ iterations where $T$ is given in Corollary G.5. Here $T$ is fixed *a priori*, so in order to obtain an estimate for $\varepsilon$, consider that the total number of iterations satisfies

$$
\begin{aligned}
T &= \sum_{k=0}^{K} T_k \\
&= \left\lceil \left(1 + \frac{2L_{\mathsf{a}}^2}{\mu^2}\right) \log\left(\frac{2R}{\varepsilon}\right) \right\rceil + \sum_{k=1}^{K} \left\lceil \left(1 + \frac{2^{k+1}L_{\mathsf{a}}^2}{\mu^2}\right) \log(4) \right\rceil \\
&= \left\lceil \left(1 + \frac{2L_{\mathsf{a}}^2}{\mu^2}\right) \log\left(\frac{2R}{\varepsilon}\right) \right\rceil + \frac{\left(-4\varepsilon L_{\mathsf{a}}^2 + \varepsilon\mu^2 + 8(\sigma_{\mathsf{a}} + \lambda^T\sigma_0)^2\right)\log(2) + \varepsilon\mu^2\log\left(\frac{(\sigma_{\mathsf{a}} + \lambda^T\sigma_0)^2}{\varepsilon L_{\mathsf{a}}^2}\right)}{\varepsilon\mu^2\log(2)}\log(4) \\
&= \left\lceil \left(1 + \frac{2L_{\mathsf{a}}^2}{\mu^2}\right) \log\left(\frac{2R}{\varepsilon}\right) \right\rceil + \left(1 - \frac{4L_{\mathsf{a}}^2}{\mu^2} + \frac{8(\sigma_{\mathsf{a}} + \lambda^T\sigma_0)^2}{\varepsilon\mu^2} + \frac{\log((\sigma_{\mathsf{a}} + \lambda^T\sigma_0)^2/(\varepsilon L_{\mathsf{a}}^2))}{\log(2)}\right) \\
&= \mathcal{O}\left(\frac{L_{\mathsf{a}}^2}{\mu^2}\log\left(\frac{2R}{\varepsilon}\right) + \frac{(\sigma_{\mathsf{a}} + \lambda^T\sigma_0)^2}{\mu^2\varepsilon}\right)
\end{aligned}
$$

Hence, in terms of fixed $T$, the firms achieve an $\varepsilon_T$ Nash equilibrium where

$$
\varepsilon_T \asymp \frac{(\sigma_{\mathsf{a}} + \lambda^T\sigma_0)^2}{\mu^2 T}.
$$

### D.3.2. SOCIALLY OPTIMAL DEMAND SIGNAL PROVISIONING

Recall that the social cost is $\mathcal{L}(u) = \mathbb{E}_{\xi\sim\mathcal{D}_0}[f_1^u(x) + f_2^u(x)]$. Define the socially optimal intervention to be

$$
u^{\mathsf{so}} := \operatorname*{argmin}_{u\in\mathcal{U}} \ \mathcal{L}(u).
$$

In this numerical example, we explore the effect of the decision-maker intervening with the socially optimal demand signal versus no intervention. To this end, we compute $u^{\mathsf{so}}$ using a symbolic solver (e.g., Mathematica) and then simulate stochastic gradient play on the player objectives $f_i^{u^{\mathsf{so}}}(x)$.

Figure 10 illustrates the players' behavior when there is no intervention by the decision-maker (i.e., $u = 0$) and when the decision-maker intervenes with the socially optimal intervention (i.e., $u = u^{\mathsf{so}}$). In particular, Figure 10(a) shows that stochastic gradient play converges to the Nash equilibrium under $u = 0$ and to the social optimum under $u = u^{\mathsf{so}}$. We plot this for a single location and price bin, both of which can be changed in the code base. The simulations show that when provided with the optimal demand signal, the firms are induced to increase their prices which indicates that user are

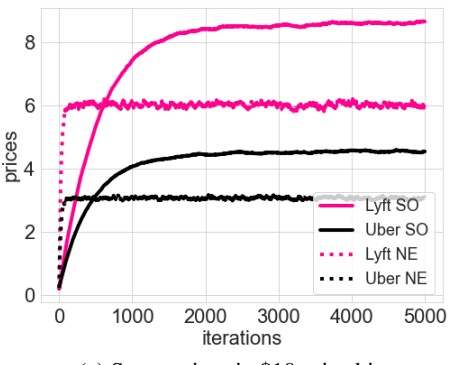

(a) Surge prices in $10 price bin.

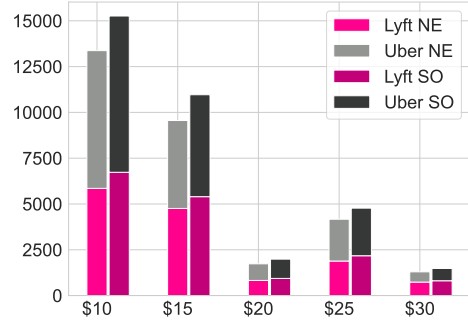

(b) Utilities at social optimum vs. Nash.

*Figure 10.* (a) Surge prices in the $10 price bin given no intervention—i.e., $u = 0$—and the socially optimal intervention—i.e. the optimal demand signal $u^{\text{so}} = \arg\min_u \mathcal{L}(u)$. With no intervention the agents converge to the Nash equilibrium (NE) of their nominal game; with the socially optimal intervention, the agents converge to the socially optimal (SO) equilibrium. (b) Welfare for each firm in each of the price bins under the Nash equilibrium and socially optimal equilibrium.

willing to pay more for the service and competition between services actually drives prices down. One interesting question pertains exploring a social cost that includes the cost to users, incorporating elements such as cost of alternative means of transportation such as public transit; unfortunately this data set does not include such information and would need to be augmented so we leave that to future work.

Figure 10(b) shows that the welfare for each firm is higher in all the price bins under the social optimum, though more marginally as the price increases. This demonstrates that a decision-maker who is able to provide optimal and informative demand signals may be able to improve the welfare of the ridesharing marketplace, even under competition by the firms, since both the prices are lower (thereby increasing the demand and cost to passengers) and the revenue is higher for all players. It is also interesting to observe that the smaller player in the market—namely Lyft which has less demand in the data set—has a larger marginal gain than the larger player (Uber) in the market.

### D.4. Lower Bound on Epoch Length

To construct the example seen in Figure 11 (and Figure 1), we build the following quadratic, two-player game. Let $x_1, x_2 \in \mathbb{R}^d$ be the leader's and follower's actions respectively. For simplicity, consider a two player quadratic Stackelberg game with costs

$$f_1(u, x) = \frac{1}{2}u^\top A u + u^\top B x, \quad \text{and} \quad f_2(u, x) = \frac{1}{2}x^\top D x + x^\top C u.$$

We demonstrate that in order to obtain finite time convergence guarantees, in many games it is necessary to have an epoch length $\tau \geq 1$. Indeed, it is possible to randomly generate quadratic games of the form described above such that the following hold:

1. There exists unique performatively stable equilibrium—i.e., $D_1^2 f_1(u^*, x^*(u^*)) = A \succ 0$ and $D_2 f_2(u^*, x^*(u)) = D \succ 0$.

2. The gradient update is not a strongly monotone operator: i.e., for some $((x, u), (x', u'))$, we have that

$$\langle g(x, u) - g(x', u'), (x, u) - (x', u') \rangle \leq 0 \quad \text{where} \quad g(x, u) = (D_1 f_1(u, x), D_2 f_2(u, x)).$$

3. The equilibrium is not stable when the agents and decision-maker simultaneously update–i.e., $\text{spec}(I - \eta J) \not\subset D[0, 1] \subset \mathbb{C}$ for any $(\eta, \gamma)$ pair and $\text{Re}(\text{spec}(-J)) \subset \mathbb{R}_{>0}$ where

$$J = \begin{bmatrix} A & B \\ \frac{\gamma}{\eta}C & \frac{\gamma}{\eta}D \end{bmatrix}.$$

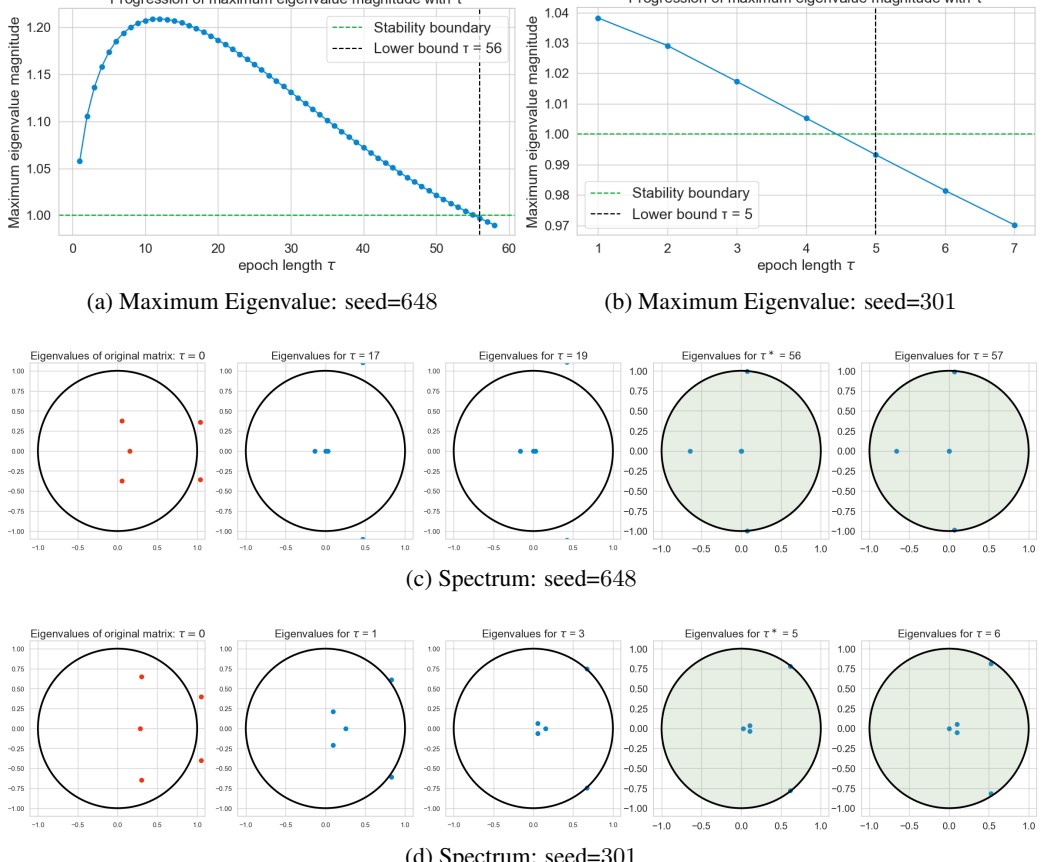

*Figure 11.* Lower bound examples: (a-b) Maximum eigenvalue for the combined dynamics as the epoch length $\tau$ increases; (c-d) the spectrum of the combined dynamics for selected values of $\tau$. From (a-b), we see that the shape of the curve tracing out the maximum eigenvalue of the combined dynamics as a function of $\tau$ is highly non-linear and depends upon the game structure. Panels (c-d) illustrate the eigenvalues relative to the unit disc in the complex plane, where eigenvalues inside the unit disc indicate the system is stable. Future work would construct a guard map for the unit disc in the complex plane, and determine an instance dependent lower bound on $\tau$ ensuring stability of the combined dynamics.

4. The epoch based dynamics, which take the form

$$\begin{bmatrix} x_{t+1} \\ u_{t+1} \end{bmatrix} = \left( I - \eta \begin{bmatrix} A & B \\ 0 & 0 \end{bmatrix} \right) \left( I - \gamma \begin{bmatrix} 0 & 0 \\ C & D \end{bmatrix} \right)^{\tau} \begin{bmatrix} x_t \\ u_t \end{bmatrix},$$

are stable for some $\tau \geq 1$.

This means that there are many instances of games such that it is necessary to have time-scale separation introduced via an epoch-based algorithms in order to obtain convergence rates without making asymptotic assumptions on the learning rate schedules. The precise construction of the instance-dependent lower bound on $\tau$ remains an open question; we conjecture that tools analogous to the "guard map" for stability used by Fiez & Ratliff (2021) can be leveraged to derive this construction and moreover extend the lower bound analysis to even non-convex settings.

### D.5. Interplay Between Sample Complexity & the Performative Gap

This section summarizes the setup for Figure 2, wherein we characterize the relationship between the expected tracking error, sample complexity, and the performative gap as parameterized by the agents' reactivity parameter $L_{\text{eq}}$. To generate this plot, we fix constants related to the decision-maker's problem as follows:

$$L_u = 10, \ L_z = 1, \ \alpha = 5, \ R = 1.0, \ \sigma = 1, \ d = 2, B = 1 \quad \text{and } \varepsilon = 0.09.$$

---

**Algorithm 2** Geometric Decay Schedule

---

1: **Input:** $y_0 \in \mathbb{R}^d$, $C, D > 0$, $\delta_0 \in (0, 1)$, estimate $\Delta \geq h(y_0)$, accuracy $\epsilon > 0$, algorithm $\mathcal{A}(y, \delta, T)$ satisfying (7)
2: **Initialize:** Set $y_0 = \mathcal{A}(y_0, \delta_0, T_0)$ with $T_0 = \frac{1}{\psi(\delta_0)} \cdot \log\left(\frac{2C\Delta}{\epsilon}\right)$;
3: Set $K = \left\lceil 1 + \log_2\left(\frac{D\delta_0}{\epsilon}\right) \right\rceil$.
4: **for** $k = 1, \ldots, K$ **do**
5:     Set $y_k = \mathcal{A}(y_{k-1}, \delta_k, T_k)$   with   $\delta_k = 2^{-k}\delta_0$, $T_k = \left\lceil \frac{1}{\psi(\delta_k)} \cdot \log(4C) \right\rceil$.
6: **end for**
7: **Return:** $y_K$.

---

We then vary $L_{\mathsf{eq}}$, and check that for each value the following are satisfied (otherwise we throw out that value of $L_{\mathsf{eq}}$ as there would not exist games meeting these criteria and our assumptions):

1. **Existence of performatively stable equilibrium**: $L_{\mathsf{eq}} L_z < \alpha$

2. **Existence of convex, smooth decision-maker cost**: $\alpha < L_u + L_z L_{\mathsf{eq}}$

Then for each valid value of $L_{\mathsf{eq}}$, we compute the "big-$\mathcal{O}$" sample complexity and plot this on the $y$-axis for each method RGM and DFO. Here, we fix $\varepsilon := 0.09$ while defining $\varepsilon' := \|u^{\mathsf{ps}} - u^*\|$. In doing so, we characterize the worst-case sample complexity needed for an $\varepsilon$ or $\varepsilon'$ approximation, defined as in Figure 2.

What we can see from this numerical example is that there is a tradeoff between sample complexity and performance for the decision-maker as a function of how reactive the agent is (as measured by $L_{\mathsf{eq}}$). If the decision-maker had access to $L_{\mathsf{eq}}$ they could determine whether or not it is worth it to run a more sample efficient algorithm like RGM versus alternatives.

## E. Technical Lemmas

Throughout, we use $\mathbb{R}^d$ to denote a $d$-dimensional space with inner product $\langle \cdot, \cdot \rangle$ and the corresponding induced norm is given by $\|x\| = \sqrt{\langle x, x \rangle}$. For any set $\mathcal{X} \subset \mathbb{R}^d$, we denote the projection of a vector $y$ onto $\mathcal{X}$ as $\mathrm{proj}_{\mathcal{X}}(y) = \mathrm{argmin}_{x \in \mathcal{X}} \|x - y\|$. Finally, for a convex set $\mathcal{X}$, we denote its normal cone at $x \in \mathcal{X}$ as $N_{\mathcal{X}}(x) = \{v \in \mathbb{R}^d : \langle v, y - x \rangle \leq 0 \ \forall y \in \mathcal{X}\}$. To simplify notation, we set $[n] := \{1, \ldots, n\}$.

### E.1. Technical Lemmas for Convergence Analysis

We need the following standard technical lemma for convergence of sequences and high-probability guarantees.

**Lemma E.1.** *Consider a sequence $w_t \geq 0$ for $t \geq 1$ and constants $t_0 \geq 0$, $a > 0$ satisfying*

$$a_{t+1} \leq \left(1 - \frac{2}{t + t_0}\right) a_t + \frac{c}{(t + t_0)^2} \tag{5}$$

*Then the following estimate holds:*

$$a_t \leq \frac{\max\{(1 + t_0)a_1, c\}}{t + t_0} \quad \forall t \geq 1. \tag{6}$$

We also restate the following Lemma, adapted from (Drusvyatskiy & Xiao, 2023).

**Lemma E.2** (Lemma B.2, (Drusvyatskiy & Xiao, 2023)). *Suppose we have a stochastic algorithm $\mathcal{A}(y_0, \delta, T)$ such that as long as $\delta < \delta_0$, the method generates a point satisfying*

$$\mathbb{E}[h(Y_T)] \leq C(1 - \psi(\delta))^T h(y_0) + D\delta, \tag{7}$$

*where $h$ is a non-negative function, $C, D > 0$, and $\delta_0 \in (0, 1)$ are constants specific to the algorithm, and $\psi$ is a function mapping $[0, \delta_0)$ into $(0, 1)$. The point $y$ returned by Algorithm 2 satisfies $\mathbb{E}[h(y_K)] \leq \epsilon$ with the efficiency estimate*

$$\sum_{k=0}^{K} T_k = \left\lceil \frac{1}{\psi(\delta_0)} \cdot \log\left(\frac{2C\Delta}{\epsilon}\right) \right\rceil + \sum_{k=1}^{K} \left\lceil \frac{\log(4C)}{\psi(2^{-k}\delta_0)} \right\rceil$$

Note that this does not immediately apply to our general case: $h$ is generally a fixed non-negative function, such as $h(y) = \|y - \bar{x}\|^2$ the distance to some fixed equilibrium $\bar{x}$. However, in our case, our target equilibrium is changing depending on the action $u_t$ from the decision-maker. A short corollary, however, gives us the desired result.

**Lemma E.3.** *Suppose we have a stochastic algorithm $\mathcal{A}(y_0, \delta, T)$ such that as long as $\delta < \delta_0$, the method generates a point satisfying*

$$\mathbb{E}\, \|y_T - y_T^*\|^2 \leq C(1 - \psi(\delta))^T \|y_0 - y_0^*\|^2 + D\delta$$

*for $C, D, \delta_0$, and $\psi$ as defined above. Then the point $y$ returned by Algorithm 2 satisfies $\mathbb{E}\, \|y_K - y_K^*\| \leq \epsilon$ with the efficiency estimate*

$$\sum_{k=0}^{K} T_k = \left\lceil \frac{1}{\psi(\delta_0)} \cdot \log\left(\frac{2C\Delta}{\epsilon}\right) \right\rceil + \sum_{k=1}^{K} \left\lceil \frac{\log(4C)}{\psi(2^{-k}\delta_0)} \right\rceil$$

*Proof.* For posterity, we include a short proof. Suppose that our target accuracy is $2D\delta$. Then note that it would be sufficient to run our algorithm for $T_0$ iterations such that

$$C(1 - \psi(\delta))^{T_0} \|y_0 - y_0^*\|^2 \leq D\delta.$$

Note that since $\psi(\delta) \in (0, 1)$, we have that $-\log(1 - \psi(\delta)) > \psi(\delta)$, so

$$\frac{1}{\psi(\delta)} \log\left(\frac{C\|y_0 - y_0^*\|^2}{D\delta}\right) \leq T_0.$$

By the concavity of $\log$, it is in fact sufficient to chose

$$\frac{1}{\psi(\delta)} \log\left(\frac{C\|y_0 - y_0^*\|^2}{\epsilon}\right) \leq T_0.$$

Then we proceed just as that of Drusvyatskiy & Xiao (2023, Lemma B.2). $\qquad\square$

## E.2. Technical Lemma for Decomposing the Decision-Maker's Tracking Error

Recall in Section 2.3, we made the claim that the decision-maker's tracking error could be decomposed into a optimization error, drift and noise term. In this section, we formally prove this claim.

**Lemma E.4.** *Consider a setting in which the decision-maker's loss is $\alpha$–strongly convex in $u$, define $u_t^\star \in \arg\min_u\{\mathbb{E}_{(x_t, \xi) \sim \mathcal{D}(u)}\, \ell(u, (x_t, \xi))\}$, where $x_t$ is the agents' response from a stochastic algorithm $\mathcal{A}$, and define $g(u) = \nabla_u\left(\mathbb{E}_{(x_t, \xi) \sim \mathcal{D}(u)}\, \ell(u, (x_t, \xi))\right)$. Suppose $g$ is $L$–Lipshcitz continuous. If the decision-maker employs a stochastic gradient-based algorithm with stepsize $\eta$ and unbiased gradient estimator for , then*

$$\mathbb{E}\, \|u_t - u_t^\star\|^2 \;\leq\; \left(1 - \frac{\eta\alpha}{4}\right)^t \|u_0 - u_0^\star\|^2 + 8\frac{\eta\sigma^2}{\alpha} + 20\left(\frac{\Delta}{\alpha\eta}\right)^2,$$

*where $\sigma^2$ is the gradient estimator variance and $\Delta := \max_k \|u_k^\star - u_{k-1}^\star\|^2$ is the worst-case drift.*

Note that the decision-maker could also be employing a repeated gradient method where we alternatively define $u_t^\star \in \arg\min_u\{\mathbb{E}_{(x_t, \xi) \sim \mathcal{D}(u_t^\star)}\, \ell(u, (x_t, \xi))\}$ and replace in the statement above $\nabla_u\left(\mathbb{E}_{(x_t, \xi) \sim \mathcal{D}(u)}\, \ell(u, (x_t, \xi))\right)$ with $\nabla_u\left(\mathbb{E}_{(x_t, \xi) \sim \mathcal{D}(u_t)}\, \ell(u, (x_t, \xi))\right)$. An analogous decomposition argument holds.

*Proof of Lemma E.4.* For a stochastic gradient method, we have that

$$\mathbb{E}_t\|u_{t+1} - u_t^\star\|^2 \leq \frac{1}{1 + \alpha\eta}\|u_t - u_t^\star\|^2 + \frac{2\eta^2\sigma^2}{1 + \alpha\eta}.$$

Indeed, let $\hat{g}_t$ be the unbiased gradient estimator, and $g(u)$ be the expected gradient evaluated at $u \in \mathcal{U}$. Then, we have that the map $u \mapsto \frac{1}{2}\|u_t - \eta\hat{g}_t - u\|^2$ is a 1-strongly convex function over $\mathcal{U}$. Hence, we have that

$$\begin{aligned}
\frac{1}{2}\|u_{t+1} - u_t^\star\|^2 &\leq \frac{1}{2}\|u_t - \eta\hat{g}_t - u_t^\star\|^2 - \frac{1}{2}\|u_t - \eta\hat{g}_t - u_{t+1}\|^2 \\
&\leq \frac{1}{2}\|u_t - u_t^\star\|^2 - \eta\langle \hat{g}_t, u_{t+1} - u_t^\star\rangle - \frac{1}{2}\|u_{t+1} - u_t\|^2 \\
&= \frac{1}{2}\|u_t - u_t^\star\|^2 - \eta\langle \hat{g}_t, u_t - u_t^\star\rangle - \frac{1}{2}\|u_{t+1} - u_t\|^2 - \eta\langle \hat{g}_t, u_{t+1} - u_t\rangle.
\end{aligned}$$

Taking expectations, we have that

$$
\begin{aligned}
\frac{1}{2}\mathbb{E}_t\|u_{t+1} - u_t^\star\|^2 &\le \frac{1}{2}\|u_t - u_t^\star\|^2 - \eta\langle\mathbb{E}_t\hat{g}_t, u_t - u_t^\star\rangle - \frac{1}{2}\mathbb{E}_t\|u_{t+1} - u_t\|^2 - \eta\mathbb{E}_t\langle\hat{g}_t, u_{t+1} - u_t\rangle \\
&\le \frac{1}{2}\|u_t - u_t^\star\|^2 - \eta\langle g(u_t), u_t - u_t^\star\rangle - \frac{1}{2}\mathbb{E}_t\|u_{t+1} - u_t^\star\|^2 - \eta\mathbb{E}_t\langle\hat{g}_t, u_{t+1} - u_t\rangle \\
&= \frac{1}{2}\|u_t - u_t^\star\|^2 - \eta\mathbb{E}_t\langle g(u_{t+1}), u_{t+1} - u_t^\star\rangle - \frac{1}{2}\mathbb{E}_t\|u_{t+1} - u_t\|^2 \\
&\quad + \eta\underbrace{\mathbb{E}_t\langle\hat{g}_t - g(u_t), u_t - u_{t+1}\rangle}_{=:P_1} + \eta\underbrace{\mathbb{E}_t\langle g(u_t) - g(u_{t+1}), u_t^\star - u_{t+1}\rangle}_{=:P_2}.
\end{aligned}
$$

Since the game is $\alpha$–strongly convex, we have that

$$
\langle g(u_{t+1}), u_{t+1} - u_t^\star\rangle \ge \langle g(u_{t+1}) - g(u_t^\star), u_{t+1} - u_t^\star\rangle \ge \alpha\|u_{t+1} - u_t^\star\|^2.
$$

This in turn implies that

$$
\frac{1 + 2\eta\alpha}{2}\|u_{t+1} - u_t^\star\|^2 \le \frac{1}{2}\|u_t - u_t^\star\|^2 - \frac{1}{2}\mathbb{E}_t\|u_{t+1} - u_t\|^2 + \eta(P_1 + P_2).
$$

Employing Young's inequality, we upper bound $P_1$ as follows:

$$
P_1 \le \frac{\sigma_{\mathsf{a}}^2}{2\nu_1} + \frac{\nu_1\mathbb{E}_t\|u_{t+1} - u_t\|^2}{2}.
$$

Applying Young's inequality, we bound $P_2$ as follows:

$$
\begin{aligned}
P_2 &\le \frac{\mathbb{E}_t\|g(u_t) - g(u_{t+1})\|^2}{2\nu_2} + \frac{\nu_2\mathbb{E}_t\|u_{t+1} - u_t^\star\|^2}{2}, \\
&\le \frac{L^2\mathbb{E}_t\|u_t - u_{t+1}\|^2}{2\nu_2} + \frac{\nu_2\mathbb{E}_t\|u_{t+1} - u_t^\star\|^2}{2},
\end{aligned}
$$

so that

$$
\frac{1 + 2\eta\alpha - \eta\nu_2}{2}\mathbb{E}_t\|u_{t+1} - u_t^\star\|^2 \le \frac{1}{2}\|u_t - u_t^\star\|^2 + \frac{\sigma_{\mathsf{a}}^2}{2\nu_1} - \frac{(1 - \eta L^2\nu_2^{-1} - \eta\nu_1)}{2}\mathbb{E}_t\|u_{t+1} - u_t\|^2.
$$

Setting $\nu_2 = \alpha$ and $\nu_1 = \eta^{-1} - L^2/\alpha$, we have that the last term on the right hand side is zero, and since $\eta \le \frac{\alpha}{2L^2}$ we have that $\nu_1 \ge \frac{1}{2\eta}$; indeed, $-\frac{1}{2\eta} \le -\frac{2L^2}{\alpha}$ so that $\nu_1 = \frac{1}{\eta} - \frac{2L^2}{\alpha} \ge \frac{1}{\eta} - \frac{1}{2\eta} = \frac{1}{2\eta}$. Therefore

$$
\mathbb{E}_t\|u_{t+1} - u_t^\star\|^2 \le \frac{1}{1 + \eta\alpha}\|u_t - u_t^\star\|^2 + \frac{2\eta^2\sigma^2}{1 + \alpha\eta},
$$

as claimed.

Now observe that

$$
\begin{aligned}
\|u_{t+1} - u_{t+1}^\star\|^2 &= \|u_{t+1} - u_t^\star\|^2 + \|u_t^\star - u_{t+1}^\star\|^2 + 2\langle u_{t+1} - u_t^\star, u_t^\star - u_{t+1}^\star\rangle \\
&\le \|u_{t+1} - u_t^\star\|^2 + \|u_t^\star - u_{t+1}^\star\|^2 + 2\|u_{t+1} - u_t^\star\|\|u_t^\star - u_{t+1}^\star\| \\
&\le \left(1 + \frac{\alpha\eta}{4}\right)\|u_{t+1} - u_t^\star\|^2 + \left(1 + \frac{4}{\alpha\eta}\right)\|u_t^\star - u_{t+1}^\star\|^2
\end{aligned}
$$

where the last inequality follows from Young's inequality. Since $1 - \frac{\eta\alpha}{1+\eta\alpha} \le 1 - \frac{\alpha\eta}{2}$, we have that

$$
\begin{aligned}
\mathbb{E}_t\|u_{t+1} - u_{t+1}^\star\|^2 &\le \left(1 + \frac{\eta\alpha}{4}\right)\left(\left(1 - \frac{\eta\alpha}{2}\right)\|u_t - u_t^\star\|^2 + 2\eta^2\sigma^2\left(1 - \frac{\eta\alpha}{2}\right)\right) + \left(1 + \frac{4}{\eta\alpha}\right)\|u_t^\star - u_{t+1}^\star\|^2 \\
&\le \left(1 - \frac{\eta\alpha}{4}\right)\|u_t - u_t^\star\|^2 + 2\eta^2\sigma^2\left(1 - \frac{\eta\alpha}{4}\right) + \left(1 + \frac{4}{\eta\alpha}\right)\Delta^2.
\end{aligned}
$$

Iterating, we have that

$$\mathbb{E}_t \|u_{t+1} - u_{t+1}^\star\|^2 \leq \left(1 - \frac{\eta\alpha}{4}\right)^{t+1} \|u_0 - u_0^\star\|^2 + 8\frac{\eta\sigma^2}{\alpha} + 20\left(\frac{\Delta}{\alpha\eta}\right)^2,$$

which concludes the proof. $\qquad\square$

## F. Regularity of the Equilibrium Response

Recall that $\omega_u(x) := (\nabla_1 f_1^u, \ldots, \nabla_n f_n^u)$. Strong metric regularity allows for Lipschitz continuity of solutions to $\omega_u(x) \in N_{\mathcal{X}}(x)$ to be Lipschitz continuous. The following proposition is a formal statement of the discussion in Section 2.

**Proposition F.1** (Inner Problem Regularity: Polyhedral Constraints). *Under Assumption 2.1.i–iii, suppose that, for any fixed $u \in \mathcal{U}$, the Jacobian of $\omega_u(x)$ with respect to $x$ is non-degenerate and the Jacobian with respect to $u$ has finite operator norm—i.e., $\det(D_x\omega_u(x)) \neq 0$ and $\|D_u\omega_u(x)\|_{\mathsf{op}} < \infty$. Then $\omega_u$ is $\kappa$-metrically regular with with $\kappa := \frac{1}{\mu} \sup_{(u,x)\in\mathcal{X}\times\mathcal{Y}} \|D_u\omega_u(x)\|_{\mathsf{op}}$.*

This proposition follows precisely from Dontchev et al. (2009, Chapter 2.F, Chapter 3); indeed, the strong metric regularity parameter in this case is equivalent to the Lipschitz continuity parameter of the implicit function.

To give some intuition, we can consider the case where $\mathcal{X}$ is the whole Euclidean space $\mathbb{R}^m$. In this section alone, we define $\omega(u, x) := \omega_u(x)$ for the purpose of clarity on the derivatives herein. In this case, by the fact that the joint strategy space $\mathcal{X}$ is unconstrained, for any fixed $u \in \mathcal{U}$, assuming $\det(D_x\omega(u, x)) \neq 0$, the Nash equilibrium $x^*(u)$ is defined as an implicit function (cf. Abraham et al. (2012)) that solves $\omega(u, x^*(u)) = 0$. By the implicit function theorem, the derivative $Dx^*(u)$ is given by

$$Dx^*(u) = -D_x\omega(u, x^*(u))^{-1} D_u\omega(u, x^*(u)).$$

We have the following lemma which provides sufficient conditions for $x^*(u)$ to be Lipschitz by assuming suitable bounds on $\|D_u\omega(u, x)\|_{\mathsf{op}}$ and $\|D_x\omega(u, x)\|_{\mathsf{op}}$.

**Lemma F.2.** *Suppose that $\|D_x\omega(u, x)\|_{\mathsf{op}} \geq \mu_1$ and $\|D_u\omega(u, x)\|_{\mathsf{op}} \leq \mu_2$ for all $x \in \mathcal{X}$ and set $L_{\mathsf{eq}} := \frac{\mu_2}{\mu_1}$. Then $x^*(u)$ is $L_{\mathsf{eq}}$-Lipschitz.*

*Proof.* We realize that by the mean value theorem,

$$\|x^*(u) - x^*(u')\| \leq \left\|\left(\int_0^1 Dx^*((1-\lambda)u + \lambda u')\lambda\right)(u - u')\right\|$$
$$\leq \sup_{\lambda\in[0,1]} \|Dx^*((1-\lambda)u + \lambda u')\|_{\mathsf{op}} \|u - u'\|$$

By the assumption, we have $\|D_x\omega(u, x^*(u))\| \geq \mu_1$ and $\|D_u\omega(u, x^*(u))\| \leq \mu_2$ for all $u \in \mathcal{U}$, then we have that

$$\sup_{\lambda\in[0,1]} \|Dx^*((1-\lambda)x + \lambda x')\| \leq \frac{\mu_2}{\mu_1} := L_{\mathsf{eq}},$$

which concludes the proof. $\qquad\square$

Again, in the polyhedral constraint case, the analysis above is almost identical; see, e.g., Dontchev et al. (2009).

## G. Contracting Agent Learning Algorithms

In this section, we show that several natural learning dynamics are $\rho$-contracting for some $\rho \in [0, 1)$. The following is a modified version of Assumption 2.1 where we remove the decision-maker for simplicity.

**Assumption G.1.** The following hold:

1. The game $\mathcal{G} := (f_1, \ldots, f_n)$ is a $C^1$-smooth convex game and $\mu$–strongly monotone;

2. The mappings $x_i \mapsto \nabla_i f_i(x_i, x_{-i})$ are $L_i$–Lipschitz continuous;

3. The game $\mathcal{G}$ is $\kappa$–strongly metrically regular.

In this section, set $\omega(x) := (\nabla_1 f_1(x), \ldots, \nabla_n f_n(x))$, and recall that we have set $L_{\mathsf{a}} := \max_{i\in[n]} L_i$.

## G.1. General $\rho$-contracting Updates

Before getting into specific examples, let us analyze stage-based algorithms such as Algorithm 2 applied to a general $\rho$-contracting algorithms (cf. Definition 2.3).

First, we iterate on the contraction for $t$ steps to get

$$
\begin{aligned}
\mathbb{E}_t \|x_t - x^*\|^2 &\leq \rho^2 \|x_{t-1} - x^*\|^2 + \rho^2 c^2 \sigma_{\mathsf{a}}^2 \\
&\leq \rho^2 \left( \rho^2 \|x_{t-2} - x^*\|^2 + \rho^2 c^2 \sigma_{\mathsf{a}}^2 \right) + \rho^2 c^2 \sigma_{\mathsf{a}}^2 \\
&\leq \rho^{2 \cdot t} \|x_0 - x^*\|^2 + c^2 \sigma_{\mathsf{a}}^2 \sum_{k=1}^{t} \rho^{2 \cdot k} \\
&\leq \rho^{2 \cdot t} \|x_0 - x^*\|^2 + c^2 \sigma_{\mathsf{a}}^2 \frac{\rho^2}{1 - \rho^2}.
\end{aligned}
$$

The following corollary shows how Lemma E.2 (and Algorithm 2) applies to $\rho$-contracting algorithms.

**Corollary G.2** (Stage-wise Stochastic $\rho$-Contracting $\mathcal{A}$). *Consider some target accuracy $\varepsilon > 0$ and suppose we have a constant $R \geq \|x_0 - x^*\|^2$. Define $\psi(\gamma) := 1 - \gamma$, $C := 1$, and $D := \frac{c^2 \sigma_{\mathsf{a}}^2}{1 - \rho^2}$. Set $\gamma_0 := \rho^2$. Then running Algorithm 2 with a stochastic $\rho$ contracting algorithm as $\mathcal{A}$, guarantees that $\mathbb{E} \|x_t - x^*\|^2 \leq \varepsilon$ after*

$$
T = \sum_{k=0}^{K} T_k = \left\lceil \frac{1}{1 - \rho^2} \cdot \log \left( \frac{2R}{\varepsilon} \right) \right\rceil + \sum_{k=1}^{K} \left\lceil \left( \frac{1}{1 - 2^{-k} \rho^2} \right) \log(4) \right\rceil
$$

*total iterations where $K := \left\lceil 1 + \log_2 \left( \frac{\rho^2 c^2 \sigma_{\mathsf{a}}^2}{(1 - \rho^2) \varepsilon} \right) \right\rceil$.*

## G.2. Stochastic Gradient Play and Asynchronous Gradient Play

Consider first players updating according to the stochastic gradient method given by

$$
x_{t+1} = \operatorname*{proj}_{\mathcal{X}} (x_t - \gamma \widehat{\omega}(x_t)), \tag{8}
$$

where $\mathbb{E}[\widehat{\omega}(x_t)] = \omega(x_t)$. This is *stochastic gradient play*.

**Assumption G.3.** Suppose that there exists a constant $\sigma_{\mathsf{a}} > 0$ satisfying

$$
\mathbb{E}[\|\widehat{\omega}(x_t) - \omega(x_t)\|^2] \leq \sigma_{\mathsf{a}}^2.
$$

Given the above assumption on the variance of the estimator $\widehat{\omega}$ of the vector of individual gradients, we have the following lemma showing that stochastic gradient play is $\rho$–contracting.

**Lemma G.4.** *Under Assumptions G.1 and G.3, suppose that players update according to stochastic gradient play with $\gamma \leq \frac{\mu}{2 L_{\mathsf{a}}^2}$. Then, the dynamics satisfy*

$$
\mathbb{E} \|x_{t+1} - x^*\|^2 \leq \frac{1}{1 + \mu\gamma} \mathbb{E} \|x_t - x^*\|^2 + \frac{2\gamma^2 \sigma_{\mathsf{a}}^2}{1 + \gamma\mu},
$$

*so that (8) is $\rho$-contracting with $\rho^2 = \frac{1}{1 + \mu\gamma}$ and $c = \sqrt{2}\gamma$.*

*Proof.* Observe that $x \mapsto \frac{1}{2} \|x_t - \gamma \widehat{\omega}(x_t) - x\|^2$ is a 1-strongly convex function over $\mathcal{X}$. Hence we deduce that

$$
\begin{aligned}
\frac{1}{2} \|x_{t+1} - x^*\|^2 &\leq \frac{1}{2} \|x_t - \gamma \widehat{\omega}(x_t) - x^*\|^2 - \frac{1}{2} \|x_t - \gamma \widehat{\omega}(x_t) - x_{t+1}\|^2 \\
&\leq \frac{1}{2} \|x_t - x^*\|^2 - \gamma \langle \widehat{\omega}(x_t), x_{t+1} - x^* \rangle - \frac{1}{2} \|x_{t+1} - x_t\|^2 \\
&= \frac{1}{2} \|x_t - x^*\|^2 - \gamma \langle \widehat{\omega}(x_t), x_t - x^* \rangle - \frac{1}{2} \|x_{t+1} - x_t\|^2 - \gamma \langle \widehat{\omega}(x_t), x_{t+1} - x_t \rangle.
\end{aligned}
$$

Taking expectations, we have that

$$\frac{1}{2}\mathbb{E}_t\|x_{t+1} - x^*\|^2 \leq \frac{1}{2}\|x_t - x^*\|^2 - \gamma\langle\mathbb{E}_t\widehat{\omega}(x_t), x_t - x^*\rangle - \frac{1}{2}\mathbb{E}_t\|x_{t+1} - x_t\|^2 - \gamma\mathbb{E}_t\langle\widehat{\omega}(x_t), x_{t+1} - x_t\rangle$$

$$\leq \frac{1}{2}\|x_t - x^*\|^2 - \gamma\langle\omega(x_t), x_t - x^*\rangle - \frac{1}{2}\mathbb{E}_t\|x_{t+1} - x^*\|^2 - \gamma\mathbb{E}_t\langle\widehat{\omega}(x_t), x_{t+1} - x_t\rangle$$

$$= \frac{1}{2}\|x_t - x^*\|^2 - \gamma\mathbb{E}_t\langle\omega(x_{t+1}), x_{t+1} - x^*\rangle - \frac{1}{2}\mathbb{E}_t\|x_{t+1} - x_t\|^2$$

$$+ \gamma\underbrace{\mathbb{E}_t\langle\widehat{\omega}(x_t) - \omega(x_t), x_t - x_{t+1}\rangle}_{=:P_1} + \gamma\underbrace{\mathbb{E}_t\langle\omega(x_t) - \omega(x_{t+1}), x^* - x_{t+1}\rangle}_{=:P_2}.$$

Since the game is $\mu$–strongly monotone, we have that

$$\langle\omega(x_{t+1}), x_{t+1} - x^*\rangle \geq \langle\omega(x_{t+1}) - \omega(x^*), x_{t+1} - x^*\rangle \geq \mu\|x_{t+1} - x^*\|^2.$$

This in turn implies that

$$\frac{1 + 2\gamma\mu}{2}\|x_{t+1} - x^*\|^2 \leq \frac{1}{2}\|x_t - x^*\|^2 - \frac{1}{2}\mathbb{E}_t\|x_{t+1} - x_t\|^2 + \gamma(P_1 + P_2).$$

Employing Young's inequality, we upper bound $P_1$ as follows:

$$P_1 \leq \frac{\sigma_a^2}{2\nu_1} + \frac{\nu_1\mathbb{E}_t\|x_{t+1} - x_t\|^2}{2}.$$

Applying Young's inequality, we bound $P_2$ as follows:

$$P_2 \leq \frac{\mathbb{E}_t\|\omega(x_t) - \omega(x_{t+1})\|^2}{2\nu_2} + \frac{\nu_2\mathbb{E}_t\|x_{t+1} - x^*\|^2}{2},$$

$$\leq \frac{L_a^2\mathbb{E}_t\|x_t - x_{t+1}\|^2}{2\nu_2} + \frac{\nu_2\mathbb{E}_t\|x_{t+1} - x^*\|^2}{2},$$

so that

$$\frac{1 + 2\gamma\mu - \gamma\nu_2}{2}\mathbb{E}_t\|x_{t+1} - x^*\|^2 \leq \frac{1}{2}\|x_t - x^*\|^2 + \frac{\gamma\sigma_a^2}{2\nu_1} - \frac{(1 - \gamma L_a^2\nu_2^{-1} - \gamma\nu_1)}{2}\mathbb{E}_t\|x_{t+1} - x_t\|^2.$$

Setting $\nu_2 = \mu$ and $\nu_1 = \gamma^{-1} - L_a^2/\mu$, we have that the last term on the right hand side is zero, and since $\gamma \leq \frac{\mu}{2L_a^2}$ we have that $\nu_1 \geq \frac{1}{2\gamma}$; indeed, $-\frac{1}{2\gamma} \leq -\frac{2L_a^2}{\mu}$ so that $\nu_1 = \frac{1}{\gamma} - \frac{2L_a^2}{\mu} \geq \frac{1}{\gamma} - \frac{1}{2\gamma} = \frac{1}{2\gamma}$. Therefore

$$\mathbb{E}_t\|x_{t+1} - x^*\|^2 \leq \frac{1}{1 + \gamma\mu}\|x_t - x^*\|^2 + \frac{2\gamma^2\sigma_a^2}{1 + \mu\gamma},$$

which concludes the proof. $\qquad\square$

The following corollary demonstrates how agents could use a stage-based algorithm to decrease the bias in their tracking error estimate and achieve a target accuracy.

**Corollary G.5** (Stage-wise Stochastic Gradient Play). *Consider some target accuracy $\varepsilon > 0$ and suppose we have a constant $R \geq \|x_0 - x^*\|^2$. Define $\psi(\gamma) := 1 - \frac{1}{1+\mu\gamma}$, $C := 1$, and $D := \frac{2\sigma_a^2}{\mu}$. Set $\gamma_0 := \frac{\mu}{2L_a^2}$. Then running Algorithm 2 with stochastic gradient play as $\mathcal{A}$, guarantees that $\mathbb{E}\|x_t - x^*\|^2 \leq \varepsilon$ after*

$$T = \sum_{k=0}^{K} T_k = \left\lceil\left(1 + \frac{2L_a^2}{\mu^2}\right)\log\left(\frac{2R}{\varepsilon}\right)\right\rceil + \sum_{k=1}^{K}\left\lceil\left(1 + \frac{2^{k+1}L_a^2}{\mu^2}\right)\log(4)\right\rceil$$

*total iterations where $K := \left\lceil 1 + \log_2\left(\frac{\sigma_a^2}{L_a^2\varepsilon}\right)\right\rceil$.*

*Proof.* Note that from Lemma G.4, we can iterate further to get that

$$
\begin{aligned}
\mathbb{E}_t \|x_t - x^*\|^2 &\leq \frac{1}{1+\mu\gamma} \|x_{t-1} - x^*\|^2 + \frac{2}{1+\mu\gamma}\gamma^2\sigma_{\mathtt{a}}^2 \\
&\leq \frac{1}{1+\mu\gamma}\left(\frac{1}{1+\mu\gamma}\|x_{t-2} - x^*\|^2 + \frac{2}{1+\mu\gamma}\gamma^2\sigma_{\mathtt{a}}^2\right) + \frac{2}{1+\mu\gamma}\gamma^2\sigma_{\mathtt{a}}^2 \\
&\leq \left(\frac{1}{1+\mu\gamma}\right)^t \|x_0 - x^*\|^2 + 2\sigma_{\mathtt{a}}^2\gamma^2 \sum_{s=1}^{t}\frac{1}{(1+\mu\gamma)^s} \\
&\leq \left(\frac{1}{1+\mu\gamma}\right)^t \|x_0 - x^*\|^2 + \gamma\frac{2\sigma_{\mathtt{a}}^2}{\mu} \\
&\leq \left(1 - \left(1 - \frac{1}{1+\mu\gamma}\right)\right)^t \mathbb{E}\|x_0 - x^*\|^2 + \gamma\frac{2\sigma_{\mathtt{a}}^2}{\mu}.
\end{aligned}
$$

Letting $\psi(\gamma) = 1 - \frac{1}{1+\mu\gamma}$, $C = 1$, and $D = \frac{2\sigma_{\mathtt{a}}^2}{\mu}$, invoking Corollary E.3 and running Algorithm 2 with the specified parameters gives us the desired result.

$\square$

**Asynchronous Updates.** In practice, it may not be the case that the agents observe data or actions synchronously, and as a result they may not have the requisite information to update their action in every time step. A natural model to capture asynchronous updates is one in which agent $i$ receives sufficient information to update its decision $y_i$ with probability $p_i$. For instance, this means that

$$
x_{i,t+1} = \begin{cases} \text{proj}_{\mathcal{X}_i}\left(x_{i,t} - \gamma\nabla_i f_i(x_{i,t}, x_{-i.t})\right), & \text{w.p. } p_i \\ x_{i,t}, & \text{w.p. } (1 - p_i) \end{cases} \tag{9}
$$

Let $p_{\mathtt{max}} := \max_{i \in [n]} p_i$ and $p_{\mathtt{min}} := \min_{i \in [n]} p_i$. Then as described in (Narang et al., 2022), this can be dealt with using techniques from preconditioning in optimization—see, e.g., (Chasnov et al., 2020b; Huo & Huang, 2017; Lian et al., 2015; Recht et al., 2011; Zhou et al., 2018) and references therein.

The analysis in Lemma G.4 does not change much; the primary difference is that the Lipschitz constant $L_{\mathtt{a}}$ is rescaled by $p_{\mathtt{max}}$ and the strong monotonicity constant $\mu$ is rescaled by $p_{\mathtt{min}}$. The reason this works out is that we can simply perform the exact same analysis using a modified inner product as has been performed in prior literature—i.e., we simply perform the analysis in the inner product $[x, y] = \langle P^{-1}x, y \rangle$ where $P = \text{diag}(p_1, \ldots, p_n)$.

### G.3. Momentum Updates: Strongly Convex-Strongly Concave Zero-Sum Games

Consider a strongly convex, strongly concave zero sum game $(f, -f)$ where player one seeks to minimize $f(x_1, x_2)$ with respect to $x_1$ and player two seeks to maximize $f$ with respect to $x_2$. It is known that such games are strongly monotone. Momentum based updates such as optimistic gradient descent-ascent (OGDA) and negative momentum are $\rho$ contracting for such games. This family of updates is given by

$$
x_{t+1} = (1 + \beta)x_t - \beta x_{t-1} - \gamma((1 + \alpha)\omega(x_t) - \alpha\omega(x_{t-1})), \tag{10}
$$

where $\alpha$ is the extrapolation parameter, $\beta$ is the momentum parameter, and

$$
\omega(x) = \begin{bmatrix} \nabla_1 f(x_1, x_2) \\ -\nabla_2 f(x_1, x_2) \end{bmatrix}.
$$

For example, standard gradient descent-ascent is equivalent to setting $(\alpha, \beta) = (0, 0)$. OGDA is given by $(\alpha, \beta) = (1, 0)$ and negative momentum is given by $(\alpha, \beta) = (0, \beta)$ for some $\beta < 1$.

Let $\kappa := L_{\mathtt{a}}/\mu$. Gradient decent ascent is a commonly studied update and has been shown to be $\rho$-contracting with $\rho = O(1 - \kappa^{-2})$ (Ryu & Boyd, 2016). Mokhtari et al. (2020) study both OGDA and proximal point methods for this

class of games. They show that OGDA is $\rho$-contracting with $\rho = O(1 - \kappa^{-1})$, and that proximal point methods are also $\rho$-contracting with $\rho = 1/(1 + \gamma\mu)$ and $c = 0$ in both cases. Zhang & Wang (2021) show that the negative momentum based update is $\rho$-contracting for strongly convex, strongly concave zero sum games, which are known to be strongly monotone. Specifically they say that negative momentum is suboptimal, but nonetheless, still $\rho$-contracting with $\rho = 1 - \Theta(\kappa^{-1.5})$.

### G.4. Best Response Dynamics

Now, we show that the best response dynamics converge linearly to the Nash equilibrium of the game. This result is commonly known and the proof is analogous to Theorem 1 in Narang et al. (2023), with one exception where we obtain a tighter bound on the regime where linear convergence is guaranteed. Nevertheless, we include it for convenience. Define

$$\mathsf{BR}(x) := \{x' \in \mathcal{X} : \ x_i' \text{ is a best response to } \ x_{-i}' \ \forall i \in [n]\}.$$

That is, unrolling notation, given a current decision vector $x_t$, the updated decision vector $x_{t+1}$ is such that

$$x_{i,t+1} = \underset{x_i \in \mathcal{X}_i}{\text{argmin}} \, f_i(x_i, x_{-i,t}) \quad \forall \, i \in [n]. \tag{11}$$

**Lemma G.6.** *Under Assumption G.1, set* $\rho := \frac{L_a \sqrt{n-1}}{\mu}$ *and suppose that we are in the regime where* $\rho < 1$, *and that players update according to* (11). *Then, the game admits a unique Nash equilibrium* $x^\star \in \mathcal{X}$ *and the best response process converges linearly:*

$$\|x_{t+1} - x^\star\| \leq \rho\|x_t - x^\star\| \quad \forall \, t \geq 0.$$

*Proof.* Since the game is $\mu$ strongly monotone, we have that

$$\sum_{i=1}^n \langle \nabla_i f_i(u) - \nabla_i f_i(u'), u_i - u_i' \rangle \geq \mu\|u - u'\|^2. \tag{12}$$

We will show the map $\mathsf{BR}(\cdot)$ is Lipschitz continuous with parameter $\rho$. To this end, consider a point $w \in \mathcal{X}$ and set $x := \mathsf{BR}(w)$. For each $i \in [n]$, first order optimality conditions for $x_i$ guarantee that

$$\langle \nabla_i f_i(x_i, w_{-i}), x_i - x_i' \rangle \leq 0 \quad \forall \, x_i' \in \mathcal{X}_i.$$

Strong monotonicity implies that, for any $x_i, x_i' \in \mathcal{X}_i$, we have that

$$\langle \nabla_i f_i(x_i, w_{-i}) - \nabla_i f_i(x_i', w_{-i}), x_i - x_i' \rangle \geq \mu\|x_i - x_i'\|^2 \quad \text{for each } i \in [n].$$

Indeed this follows from (12) bx letting $u = (x_i, w_{-i})$ and $u' = (x_i', w_{-i})$ for each $i \in [n]$. Hence

$$\mu\|x_i - x_i'\|^2 \leq \langle \nabla_i f_i(x_i, w_{-i}) - \nabla_i f_i(x_i', w_{-i}), x_i - x_i' \rangle \leq \langle -\nabla_i f_i(x_i', w_{-i}), x_i - x_i' \rangle.$$

Since this holds for any $x_i'$ we can replace $x_i'$ with $x_i^\star$ to get that

$$\mu \sum_{i=1}^n \|x_i - x_i^\star\|^2 \leq \sum_{i=1}^n \langle \nabla_i f_i(x_i^\star, w_{-i}), x_i^\star - x_i \rangle$$

Since $x^\star$ is a Nash equilibrium, we have that

$$\langle \omega(x^\star), x - x^\star \rangle \leq 0 \quad \forall \, x \in \mathcal{X}.$$

Hence, we deduce that

$$\begin{aligned}
\mu\|x - x^\star\|^2 &\leq \sum_{i=1}^n \langle \nabla_i f_i(x_i^\star, w_{-i}), x_i^\star - x_i \rangle, \\
&\leq \langle \omega(x_i^\star, w_{-i}) - \omega(x^\star), x^\star - x \rangle, \\
&\leq \|\omega(x_i^\star, w_{-i}) - \omega(x^\star)\|\|x^\star - x\|, \\
&\leq L_a\|x^\star - x\| \sum_{i=1}^n \|w_{-i} - x_{-i}^\star\|, \\
&= L_a\langle u_1, u_2 \rangle,
\end{aligned}$$

where $u_1 = (\|x_1^\star - x_1\|, \ldots, \|x_n^\star - x_n\|)$ and $u_2 = (\|w_{-1} - x_{-1}^\star\|, \ldots, \|w_{-n} - x_{-n}^\star\|)$. Letting $\zeta = \|w - x^\star\|$, we have that $\zeta^2 = \|w_{-i} - x_{-i}^\star\|^2 + \|w_i - x_i^\star\|^2$ for each $i \in [n]$. Observe that

$$
\begin{aligned}
\|u_2\| &= \left( \sum_{i=1}^n \|w_{-i} - x_{-i}^\star\|^2 \right)^{1/2} \\
&= \left( \sum_{i=1}^n \zeta^2 - \|w_i - x_i^\star\|^2 \right)^{1/2} \\
&= |\zeta| \left( n - \frac{1}{\zeta^2} \|w_i - x_i^\star\|^2 \right)^{1/2} \\
&= |\zeta|\sqrt{n-1}.
\end{aligned}
$$

Therefore, we deduce that

$$
\mu\|x - x^\star\|^2 \leq L_{\mathsf{a}} \langle u_1, u_2 \rangle \leq L_{\mathsf{a}} \|u_1\|\|u_2\| = L_{\mathsf{a}}\sqrt{n-1}\|w - x^\star\|\|x^\star - x\|
$$

so that by dividing through we have

$$
\|x - x^\star\| \leq \frac{L_{\mathsf{a}}\sqrt{n-1}}{\mu}\|w - x^\star\|.
$$

Since this holds for any $w$ and corresponding $y := \mathsf{BR}(w)$ we have that

$$
\|x_{t+1} - x^\star\| \leq \rho \cdot \|x_t - x^\star\| \quad \text{where} \quad \rho := \frac{L_{\mathsf{a}}\sqrt{n-1}}{\mu},
$$

as claimed. The rest of the result follows immediately from the Banach fixed point theorem. $\qquad\square$

# H. Oblivious Decision-Maker

In this section we prove the bounds on the equilibrium tracking error and the dynamic regret of the agents given an oblivious decision-maker who is deploying a sequence of actions $\{x_s\}_{s=1}^t$ that are of bounded variation and sufficiently contracting.

## H.1. Worst-Case Expected Equilibrium Tracking Error.

The first natural question in this setting relates to bounding the time to track the time-varying equilibrium in expectation given the drift $\Delta_t = \|x_t^\star - x_{t+1}^\star\|$ where $x_t^\star \in \mathsf{Eq}(\mathcal{G}_{u_t})$ where $\mathcal{G}_{u_t} := (f_1^{u_t}, \ldots, f_n^{u_t})$ That is, given that the decision-maker is obliviously deploying sequence $\{u_s\}_{s=0}^t$,

*how long does it take for $\|x_t - x_t^\star\|^2$ to be less than some error tolerance $\varepsilon$ and how to we optimize that error?*

We make the following assumption on the stochasticity.

**Assumption H.1.** There exists a probability space $(\Omega, \mathcal{F}, \mathbb{P})$ with filtration $(\mathcal{F}_t)_{t \geq 0}$ such that $\mathcal{F}_0 = \{\emptyset, \Omega\}$. Iterates and corresponding Nash equilibrium, $x_t, x_t^\star : \Omega \to \mathbb{R}^m$, are $\mathcal{F}_t$-measurable.

**Proposition H.2** (Formal Statement of Proposition 3.1)**.** *Suppose that Assumption H.1 holds, that agents employ a $\rho$-contracting update (Definition 2.3), and we are in the regime $\rho \in [0, 1)$. Then, the expected equilibrium tracking error satisfies*

$$
\mathbb{E}_t\|x_{t+1} - x_{t+1}^\star\|^2 \leq \left( 1 - \frac{(1 - \rho^2)}{2} \right)^{t+1} \|x_0 - x_0^\star\|^2 + \frac{2(c\sigma_{\mathsf{a}})^2}{1 - \rho^2} + 6 \left( \frac{\Delta_{\mathsf{a}}}{1 - \rho^2} \right)^2,
$$

*where $\Delta_{\mathsf{a}} := \max\{\|x_{t+1}^\star - x_t^\star\|\}$.*

*Proof.* Since the agents' updates are $\rho$-contracting with constants $(c, \sigma_{\mathsf{a}})$, following the analysis in Appendix G.1, we have that

$$
\mathbb{E}_t\|x_{t+1} - x_t^\star\|^2 \leq \rho^2\|x_t - x_t^\star\| + \rho^2(\tilde{c}\sigma_{\mathsf{a}})^2, \quad \text{where} \quad \tilde{c} := 1/(1 - \rho^2).
$$

Now observe that

$$\mathbb{E}_t \|x_{t+1} - x_{t+1}^\star\|^2 = \|x_{t+1} - x_t^\star\|^2 + \|x_t^\star - x_{t+1}^\star\|^2 + 2\mathbb{E}_t \langle x_{t+1} - x_t^\star, x_t^\star - x_{t+1}^\star \rangle$$

$$\leq \mathbb{E}_t \|x_{t+1} - x_t^\star\|^2 + \|x_t^\star - x_{t+1}^\star\|^2 + 2\mathbb{E}_t \|x_{t+1} - x_t^\star\| \|x_t^\star - x_{t+1}^\star\|$$

$$\leq (1 + \lambda)\, \mathbb{E}_t \|x_{t+1} - x_t^\star\|^2 + (1 + \lambda^{-1})\, \mathbb{E}_t \|x_t^\star - x_{t+1}^\star\|^2$$

$$\leq (1 + \lambda)\left(\rho^2 \|x_t - x_t^\star\|^2 + \rho^2 (\tilde{c}\sigma_{\mathsf{a}})^2\right) + (1 + \lambda^{-1})\, \mathbb{E}_t \|x_t^\star - x_{t+1}^\star\|^2,$$

where the second to last inequality holds since $\|a\|\|b\| \leq \lambda \|a\|^2 + \lambda^{-1}\|b\|^2$ for any $\lambda > 0$. For algebraic convenience, let $\rho^2 = (1 - \tau)$ for some variable $\tau > 0$. Setting $\lambda = \frac{\tau}{2}$, we have that

$$\mathbb{E}_t \|x_{t+1} - x_{t+1}^\star\|^2 \leq \left(1 + \frac{\tau}{2}\right)\left((1-\tau)\|x_t - x_t^\star\|^2 + (1-\tau)(\tilde{c}\sigma_{\mathsf{a}})^2\right) + \left(1 + \frac{2}{\tau}\right)\mathbb{E}_t \|x_t^\star - x_{t+1}^\star\|^2,$$

$$\leq \left(1 - \frac{\tau}{2}\right)\|x_t - x_t^\star\|^2 + \left(1 - \frac{\tau}{2}\right)(\tilde{c}\sigma_{\mathsf{a}})^2 + \left(1 + \frac{2}{\tau}\right)\Delta_{\mathsf{a}}^2$$

where $\Delta_{\mathsf{a}}^2 := \max\{\|x_t^\star - x_{t+1}^\star\|^2\}$. Iterating this expression, we have that

$$\mathbb{E}_t \|x_{t+1} - x_{t+1}^\star\|^2 \leq \left(1 - \frac{\tau}{2}\right)\left(\left(1 - \frac{\tau}{2}\right)\|x_{t-1} - x_{t-1}^\star\|^2 + \left(1 - \frac{\tau}{2}\right)(\tilde{c}\sigma_{\mathsf{a}})^2 + \left(1 + \frac{2}{\tau}\right)\Delta_{\mathsf{a}}^2\right)$$

$$+ \left(1 - \frac{\tau}{2}\right)(\tilde{c}\sigma_{\mathsf{a}})^2 + \left(1 + \frac{2}{\tau}\right)\Delta_{\mathsf{a}}^2$$

$$\leq \left(1 - \frac{\tau}{2}\right)^{t+1}\|x_0 - x_0^\star\|^2 + \sum_{k=1}^{t+1}(\tilde{c}\sigma_{\mathsf{a}})^2\left(1 - \frac{\tau}{2}\right)^k + \Delta_{\mathsf{a}}^2\left(1 + \frac{2}{\tau}\right)\sum_{k=0}^{t}\left(1 - \frac{\tau}{2}\right)^k$$

$$\leq \left(1 - \frac{\tau}{2}\right)^{t+1}\|x_0 - x_0^\star\|^2 + \frac{2(\tilde{c}\sigma_{\mathsf{a}})^2}{\tau} + \Delta_{\mathsf{a}}^2\left(1 + \frac{2}{\tau}\right)\frac{2 - 2(1 - \frac{\tau}{2})^t + (1 - \frac{\tau}{2})^t \tau}{\tau}$$

$$\leq \left(1 - \frac{\tau}{2}\right)^{t+1}\|x_0 - x_0^\star\|^2 + \frac{2(\tilde{c}\sigma_{\mathsf{a}})^2}{\tau} + \Delta_{\mathsf{a}}^2\left(1 + \frac{2}{\tau}\right)\frac{2}{\tau}.$$

Observe that $(1 + \frac{2}{w})\frac{2}{w} \leq \frac{2(2+w)}{w^2} \leq \frac{6}{w^2}$ for $w \in (0, 1]$. In the regime where $\rho \leq 1$ we have that $\tau = 1 - \rho^2 \leq 1$. Hence, we have that

$$\mathbb{E}_t \|x_{t+1} - x_{t+1}^\star\|^2 \leq \left(1 - \frac{\tau}{2}\right)^{t+1}\|x_0 - x_0^\star\|^2 + \frac{2(\tilde{c}\sigma_{\mathsf{a}})^2}{\tau} + \Delta_{\mathsf{a}}^2\left(1 + \frac{2}{\tau}\right)\frac{2}{\tau}$$

$$\leq \left(1 - \frac{\tau}{2}\right)^{t+1}\|x_0 - x_0^\star\|^2 + \frac{2(\tilde{c}\sigma_{\mathsf{a}})^2}{\tau} + 6\left(\frac{\Delta_{\mathsf{a}}}{\tau}\right)^2.$$

Since $\tau := (1 - \rho^2)$, we have that

$$\mathbb{E}_t \|x_{t+1} - x_{t+1}^\star\|^2 \leq \left(1 - \frac{(1 - \rho^2)}{2}\right)^{t+1}\|x_0 - x_0^\star\|^2 + \frac{2(\tilde{c}\sigma_{\mathsf{a}})^2}{1 - \rho^2} + 6\left(\frac{\Delta_{\mathsf{a}}}{1 - \rho^2}\right)^2.$$

This concludes the proof. $\qquad\square$

Let us specialize to the stochastic gradient play setting. Set $\omega_{i,t} := \nabla_i f_i^{u_t}(x_t)$, $\zeta_{i,t} := \omega_{i,t} - \widehat{\omega}_{i,t}$ and $\zeta_t := (\zeta_{1,t}, \ldots, \zeta_{n,t})$. Define also $\omega_t := (\omega_{1,t}, \ldots, \omega_{n,t})$. Here $\zeta_t$ is the noise of the vector of individual gradients.

**Assumption H.3.** Suppose there exists $\Delta_{\mathsf{a}}, \sigma_{\mathsf{a}} > 0$ such that the following hold:

a. The drift $\Delta_{\mathsf{a},t} := \|x_{t+1}^\star - x_t^\star\|$ is such that $\mathbb{E}\,\Delta_{\mathsf{a},t}^2 \leq \Delta_{\mathsf{a}}^2$ for all $t$;

b. The gradient noise $\zeta_t$ satisfies $\mathbb{E}\,\|\zeta_t\|^2 \leq \sigma_{\mathsf{a}}^2$;

c. The gradient noise $\zeta_t : \Omega \to \mathbb{R}^d$ is $\mathcal{F}_{t+1}$-measurable with $\mathbb{E}[\zeta_t | \mathcal{F}_t] = 0$.

Recall that $\Delta := \max\{\|u_{t+1} - u_t\|\}$ and $\Delta_{\mathsf{a}} \leq L_{\mathsf{eq}} \cdot \Delta$.

---

**Algorithm 3** Projected Stochastic Gradient Play

---

1: **Input:** Step-size $\gamma \leq \frac{\mu}{2L_a^2}$; initial condition $x_0$; decision-maker input sequence $\{u_s\}_{k=0}^t$
2: **for** $k = 1, \ldots, t-1$ **do**
3:    **for** $i \in [n]$ **do**
4:       Set $x_{i,k+1} = \underset{\mathcal{X}_i}{\text{proj}} (x_{i,k} - \gamma \widehat{\omega}_{i,k}) \quad \forall i \in [n] \quad$ where $\widehat{\omega}_{i,k} := \widehat{\nabla}_i f_i^{u_k}(x_k)$
5:    **end for**
6: **end for**

---

**Corollary H.4** (Formal Statement of Corollary 3.2). *Under the assumptions of Proposition 3.1 and Assumption H.3, suppose that the agents are running stochastic gradient play (Algorithm 3) with $\gamma \leq \mu/(2L_a^2)$. Then $\rho^2 = \frac{1}{1+\gamma\mu}$ and $c = \sqrt{2}\gamma$ so that*

$$\mathbb{E}_t \|x_{t+1} - x_{t+1}^\star\|^2 \leq \left(1 - \frac{\mu\gamma}{4}\right)^{t+1} \|x_0 - x_0^\star\|^2 + \frac{8\gamma\sigma_a^2}{\mu} + 24\left(\frac{L_{eq}\Delta}{\gamma\mu}\right)^2.$$

Given Lemma G.4, the proof of the above corollary follows an identical argument to Proposition 3.1.

*Proof.* From Lemma G.4, we have that

$$\mathbb{E}_t \|x_{t+1} - x^*\|^2 \leq \frac{1}{1+\gamma\mu} \|x_t - x^*\|^2 + \frac{2\gamma^2\sigma_a^2}{1+\mu\gamma}.$$

Now observe that

$$\begin{aligned}
\mathbb{E}_t \|x_{t+1} - x_{t+1}^\star\|^2 &= \|x_{t+1} - x_t^\star\|^2 + \|x_t^\star - x_{t+1}^\star\|^2 + 2\mathbb{E}_t \langle x_{t+1} - x_t^\star, x_t^\star - x_{t+1}^\star \rangle \\
&\leq \mathbb{E}_t \|x_{t+1} - x_t^\star\|^2 + \|x_t^\star - x_{t+1}^\star\|^2 + 2\mathbb{E}_t \|x_{t+1} - x_t^\star\| \|x_t^\star - x_{t+1}^\star\| \\
&\leq (1+\lambda) \mathbb{E}_t \|x_{t+1} - x_t^\star\|^2 + \left(1 + \lambda^{-1}\right) \mathbb{E}_t \|x_t^\star - x_{t+1}^\star\|^2 \\
&\leq (1+\lambda) \left(\frac{1}{1+\mu\gamma} \|x_t - x_t^\star\|^2 + \frac{2}{1+\mu\gamma} (\gamma\sigma_a)^2\right) + \left(1 + \lambda^{-1}\right) \mathbb{E}_t \|x_t^\star - x_{t+1}^\star\|^2,
\end{aligned}$$

where the second to last inequality holds since $\|a\|\|b\| \leq \lambda\|a\|^2 + \lambda^{-1}\|b\|^2$ for any $\lambda > 0$. Observe that

$$\frac{1}{1+\mu\gamma} = 1 - \frac{\mu\gamma}{1+\mu\gamma} \leq 1 - \frac{\mu\gamma}{2}.$$

Using this fact and setting $\lambda = \frac{\mu\gamma}{4}$, we have that

$$\begin{aligned}
\mathbb{E}_t \|x_{t+1} - x_{t+1}^\star\|^2 &\leq \left(1 + \frac{\mu\gamma}{4}\right) \left(\left(1 - \frac{\mu\gamma}{2}\right) \|x_t - x_t^\star\|^2 + \left(1 - \frac{\mu\gamma}{2}\right) (\sqrt{2}\gamma\sigma_a)^2\right) + \left(1 + \frac{4}{\mu\gamma}\right) \mathbb{E}_t \|x_t^\star - x_{t+1}^\star\|^2, \\
&\leq \left(1 - \frac{\mu\gamma}{4}\right) \|x_t - x_t^\star\|^2 + 2\left(1 - \frac{\mu\gamma}{4}\right) (\gamma\sigma_a)^2 + \left(1 + \frac{4}{\mu\gamma}\right) \Delta_a^2
\end{aligned}$$

where $\Delta_a^2 := \max\{\|x_t^\star - x_{t+1}^\star\|^2\}$. Iterating this expression, we have that

$$\begin{aligned}
\mathbb{E}_t \|x_{t+1} - x_{t+1}^\star\|^2 &\leq \left(1 - \frac{\mu\gamma}{4}\right) \|x_t - x_t^\star\|^2 + 2\left(1 - \frac{\mu\gamma}{4}\right) (\gamma\sigma_a)^2 + \left(1 + \frac{4}{\mu\gamma}\right) \Delta_a^2 \\
&\leq \left(1 - \frac{\mu\gamma}{4}\right) \left(\left(1 - \frac{\mu\gamma}{4}\right) \|x_{t-1} - x_{t-1}^\star\|^2 + 2\left(1 - \frac{\mu\gamma}{4}\right) (\gamma\sigma_a)^2 + \left(1 + \frac{4}{\mu\gamma}\right) \Delta_a^2\right) \\
&\quad + 2\left(1 - \frac{\mu\gamma}{4}\right) (\gamma\sigma_a)^2 + \left(1 + \frac{4}{\mu\gamma}\right) \Delta_a^2 \\
&\leq \left(1 - \frac{\mu\gamma}{4}\right)^{t+1} \|x_0 - x_0^\star\|^2 + \sum_{k=1}^{t+1} (\sqrt{2}\gamma\sigma_a)^2 \left(1 - \frac{\mu\gamma}{4}\right)^k + \Delta_a^2 \left(1 + \frac{4}{\mu\gamma}\right) \sum_{k=0}^{t} \left(1 - \frac{\mu\gamma}{4}\right)^k \\
&\leq \left(1 - \frac{\mu\gamma}{4}\right)^{t+1} \|x_0 - x_0^\star\|^2 + 8\frac{\gamma\sigma_a^2}{\mu} + 24\left(\frac{\Delta_a}{\mu\gamma}\right)^2,
\end{aligned}$$

which concludes the proof. □

The preceding corollary shows that in order to obtain last iterate convergence guarantees there is a clear tradeoff between the step-size and the drift-to-noise ratio. Using Corollary 3.2, we define the *asymptotic tracking error* of Algorithm 3 and the optimal step-size as follows:

$$\varepsilon_\star := \min_{\gamma \in (0,\, \mu/(2L_\mathsf{a}^2)]} \left\{ \frac{8\gamma\sigma_\mathsf{a}^2}{\mu} + 24\left(\frac{\Delta_\mathsf{a}}{\mu\gamma}\right)^2 \right\} \quad \text{and} \quad \gamma_\star := \min\left\{ \frac{\mu}{2L_\mathsf{a}^2},\, \left(\frac{6\Delta_\mathsf{a}^2}{\mu\sigma_\mathsf{a}^2}\right)^{1/3} \right\}.$$

so that the high and low regimes are determined by

$$\frac{\mu}{2L_\mathsf{a}^2} = \left(\frac{6\Delta_\mathsf{a}^2}{\mu\sigma_\mathsf{a}^2}\right)^{1/3} \iff \frac{\mu^4}{6 \cdot 2^3 L_\mathsf{a}^6} = \frac{\Delta_\mathsf{a}^2}{\sigma_\mathsf{a}^2} \iff \left(\frac{\mu^4}{3 \cdot 2^4 L_\mathsf{a}^6}\right)^{1/2} = \frac{\mu^2}{4\sqrt{3} \cdot L_\mathsf{a}^3} = \frac{\Delta_\mathsf{a}}{\sigma_\mathsf{a}}.$$

Therefore the high drift-to-noise regime is $\frac{\Delta_\mathsf{a}}{\sigma_\mathsf{a}} > \frac{\mu^2}{4\sqrt{3}\cdot L_\mathsf{a}^3}$ and otherwise we are in the low drift-to-noise regime[6]. Plugging $\gamma_\star$ into $\varepsilon_\star$, we have that

$$\varepsilon_\star = \begin{cases} 96 \cdot \dfrac{L_\mathsf{a}^4\Delta_\mathsf{a}^2}{\mu^4} + 4 \cdot \dfrac{\sigma_\mathsf{a}^2}{L_\mathsf{a}^2} & \text{if } \dfrac{\Delta_\mathsf{a}}{\sigma_\mathsf{a}} \geq \dfrac{\mu^2}{4\sqrt{3}\cdot L_\mathsf{a}^3}, \\[2ex] 12 \cdot 6^{1/3} \cdot \left(\dfrac{\Delta_\mathsf{a}\sigma_\mathsf{a}^2}{\mu^2}\right)^{2/3} & \text{otherwise.} \end{cases}$$

**High drift-to-noise regime.** When $\frac{\Delta_\mathsf{a}}{\sigma_\mathsf{a}} \geq \frac{\mu^2}{4\sqrt{3}\cdot L_\mathsf{a}^3}$, the decision-maker is in the *high drift-to-noise regime*. In this case, running Algorithm 3 with $\gamma_\star \asymp \mu/(2L_\mathsf{a}^2)$ results in a point $x_t$ such that

$$\mathbb{E}\,\|x_t - x_t^\star\|^2 \lesssim \varepsilon_\star \quad \text{in } t \lesssim \frac{L_\mathsf{a}^2}{\mu^2}\log\left(\frac{\|x_0 - x_0^\star\|^2}{\varepsilon_\star}\right) \quad \text{time steps.}$$

This case is less interesting as the decision-maker is deploying a sequence $\{u_k\}_{k=0}^t$ such that the drift (i.e., change in the equilibrium corresponding to the induced sequence of games) is higher than the stochastic noise in the game. Here, the agents must use a learning rate $\gamma$ that is as large as the deterministic setting and therefore, achieve a expected tracking error within a constant factor of

$$\frac{L_\mathsf{a}^4\Delta_\mathsf{a}^2}{\mu^4} + \frac{\sigma_\mathsf{a}^2}{L_\mathsf{a}^2}.$$

**Low drift-to-noise regime.** The more interesting regime results when the deployed sequence causes low drift relative to the noise level—i.e. when $\Delta_\mathsf{a}/\sigma_\mathsf{a} < \mu^2/\left(4\sqrt{3}\cdot L_\mathsf{a}^2\right)$. In this case, it is possible that the agents can choose a step-size such that the tracking error is within a constant factor of $\varepsilon_\star$. Indeed, with $\gamma_\star \asymp \left(6\Delta_\mathsf{a}^2/\left(\mu\sigma_\mathsf{a}^2\right)\right)^{1/3}$, its straightforward to show that stochastic gradient play (Algorithm 3) finds a point $x_t \in \mathcal{X}$ such that

$$\mathbb{E}\,\|x_t - x_t^\star\|^2 \lesssim \varepsilon_\star \quad \text{in } t \lesssim \frac{\sigma_\mathsf{a}^2}{\mu^2\varepsilon_\star}\log\left(\frac{\|x_0 - x_0^\star\|^2}{\varepsilon_\star}\right) \quad \text{time steps.}$$

The following proposition (formal statement of Proposition 3.3 from the main body) shows that there is an algorithm (a super algorithm that consumes stochastic gradient play) that proceeds in stages to obtain a stronger convergence guarantee.

**Proposition H.5** (Formal Statement of Proposition 3.3: Expected Tracking Error for Induced Time-Varying Game.)**.** *Suppose that Assumptions 2.1, H.1, and H.3 hold and that the decision-maker deploys a sequence $\{u_s\}_{s=0}^t$ satisfying $\Delta_\mathsf{a}/\sigma_\mathsf{a} < \mu^2/\left(4\sqrt{3}\cdot L_\mathsf{a}^2\right)$. Set constants*

$$\gamma_\star := \left(6\Delta_\mathsf{a}^2/(\mu\sigma_\mathsf{a}^2)\right)^{1/3} \quad \text{and} \quad \varepsilon_\star := (\Delta_\mathsf{a}\sigma_\mathsf{a}^2/\mu^2)^{2/3}.$$

---

[6]Observe that this bound is in terms of the induced equilibrium drift; it is equivalent to restate it in terms of the decision-maker action sequence drift as follows: the high drift-to-noise regime is $\frac{\Delta}{\sigma_\mathsf{a}} > \frac{\mu^2}{4\sqrt{3}\cdot L_\mathsf{a}^3 L_\mathsf{eq}}$ and otherwise we are in the low drift-to-noise regime, where we have used the bound $\Delta_\mathsf{a} \leq L_\mathsf{eq}\Delta$.

*Suppose that there is a constant $R$ available such that $R \geq \|x_0 - x_0^\star\|^2$. Further, set constants $\gamma_0 = \frac{\mu}{2L_a^2}$ and*

$$K = 1 + \left\lceil \log_2 \left( \frac{\mu}{L_a^2} \cdot \left( \frac{\sigma_a^2 \mu}{\Delta_a^2} \right)^{1/3} \right) \right\rceil, \quad T_0 = \left\lceil \frac{8L_a^2}{\mu^2} \log \left( \frac{L_a^2 R}{\sigma_a^2} \right)^+ \right\rceil, \quad \gamma_k = \frac{\gamma_{k-1} - \gamma_\star}{2}, \quad T_k = \left\lceil \frac{4\log(4)}{\mu \gamma_k} \right\rceil,$$

*for all $k \geq 1$. Consider running stochastic gradient play (Algorithm 3) in $k = 0, \ldots, K-1$ stages. Then $T = T_0 + \cdots + T_{K-1}$ satisfies*

$$T \lesssim \frac{L_a^2}{\mu^2} \log \left( \frac{L_a^2 R}{\sigma_a^2} \right)^+ + \frac{\sigma_a^2}{\mu^2 \varepsilon_\star} \leq \frac{L_a^2}{\mu^2} \log \left( \frac{R}{\varepsilon_\star} \right)^+ + \frac{\sigma_a^2}{\mu^2 \varepsilon_\star},$$

*and the expected tracking error satisfies $\mathbb{E} \|x_K - x_K^\star\|^2 \lesssim \varepsilon_\star$.*

In the above corollary $(\cdot)^+ := \max\{(\cdot), 0\}$. We use this operator since some of the logarithmic terms can be negative depending on the size of constants.

*Proof.* Set $t_0 := 0$ and for each stage index $k$, let $t_k := \sum_{s=0}^{k-1} T_s$ be the total cumulative time up to stage $k$. Let $x_k^\star$ be the Nash equilibrium of the induced game $\mathcal{G}_{u_{t_k}}$, and set

$$\epsilon_k := \frac{8}{\mu} \left( \gamma_k \sigma_a^2 + 3 \frac{\Delta_a^2}{\mu \gamma_\star^2} \right).$$

Recall that $\gamma_k \geq \gamma_\star$. Corollary 3.2 implies that

$$\mathbb{E} \|x_{k+1} - x_{k+1}^\star\|^2 \leq \left( 1 - \frac{\mu \gamma_k}{4} \right)^{T_k} \mathbb{E} \|x_k - x_k^\star\|^2 + \frac{8}{\mu} \left( \gamma_k \sigma_a^2 + 3 \frac{\Delta_a^2}{\mu \gamma_k^2} \right)$$

$$\leq e^{-\frac{\mu \gamma_k}{4} T_k} \mathbb{E} \|x_k - x_k^\star\|^2 + \epsilon_k.$$

We claim that $\mathbb{E} \|x_k - x_k^\star\|^2 \leq 2\epsilon_{k-1}$ for all $k \geq 1$. The argument proceeds by induction. The base case holds since

$$T_0 = \left\lceil \frac{8L_a^2}{\mu^2} \log \left( \frac{L_a^2 R}{\sigma_a^2} \right)^+ \right\rceil$$

implies that

$$\mathbb{E} \|x_1 - x_1^\star\|^2 \leq e^{-\frac{\mu \gamma_0}{4} T_0} \|x_0 - x_0^\star\|^2 + \epsilon_0 \leq \exp \left( -\frac{\mu^2}{4 \cdot 2L_a^2} \frac{8L_a^2}{\mu^2} \log \left( \frac{L_a^2 R^2}{\sigma_a^2} \right) \right) \|x_0 - x_0^\star\|^2 + \epsilon_0 \leq \frac{\sigma_a^2}{L_a^2} + \epsilon_0 \leq 2\epsilon_0.$$

Suppose that the claim holds for some $k \geq 1$—i.e., the estimate $\mathbb{E} \|x_k - x_k^\star\|^2 \leq 2\epsilon_{k-1}$ holds for some fixed $k$. Then, we have that $e^{-\mu \gamma_k T_k} = e^{-\mu \gamma_k \left\lceil \frac{4\log(4)}{\mu \gamma_k} \right\rceil} \leq \frac{1}{4}$. Further, its easy to deduce that $\frac{1}{4} \leq \frac{\epsilon_k}{2\epsilon_{k-1}}$. Hence, putting these facts together, we have that

$$\mathbb{E} \|x_{k+1} - x_{k+1}^\star\|^2 \leq e^{-\mu \gamma_k T_k} \mathbb{E} \|x_k - x_k^\star\|^2 + \epsilon_k \leq \frac{1}{4} \mathbb{E} \|x_k - x_k^\star\|^2 + \epsilon_k$$

$$\leq \frac{\epsilon_k}{2\epsilon_{k-1}} \mathbb{E} \|x_k - x_k^\star\|^2 + \epsilon_k \leq 2\epsilon_k,$$

by the induction hypothesis $\mathbb{E} \|x_k - x_k^\star\|^2 \leq 2\epsilon_{k-1}$. In particular, this implies that $\mathbb{E} \|x_K - \bar{x}_K\|^2 \leq 2\epsilon_{K-1}$.

Now, we need to show that the claimed efficiency estimate holds. That is, we need to show that $\epsilon_{K-1} \asymp \varepsilon_\star$. Observe that for

some constants $c$ that we will set later, the following is true:

$$\epsilon_{k-1} - c \left( \frac{\Delta_a \sigma_a^2}{\mu^2} \right)^{2/3} = \frac{8}{\mu} \left( \gamma_{k-1} \sigma_a^2 + 3 \frac{\Delta_a^2}{\mu \gamma_\star^2} \right) - c \left( \frac{\Delta_a \sigma_a^2}{\mu^2} \right)^{2/3}$$

$$= \frac{8}{\mu} \left( \gamma_{k-1} \sigma_a^2 + 3 \left( \frac{\Delta_a^2}{\mu \sigma_a^2} \right)^{1/3} \frac{\sigma_a^2}{6^{2/3}} \right) - c \left( \frac{\Delta_a \sigma_a^2}{\mu^2} \right)^{2/3}$$

$$= \frac{8 \sigma_a^2}{\mu} \left( \gamma_{k-1} + 3 \left( \frac{\Delta_a^2}{6^2 \sigma_a^2 \mu} \right)^{1/3} \right) - c \left( \frac{\Delta_a \sigma_a^2}{\mu^2} \right)^{2/3}$$

$$= \frac{8 \sigma_a^2}{\mu} \left( \gamma_{k-1} + 3 \left( \frac{\Delta_a^2}{6^2 \sigma_a^2 \mu} \right)^{1/3} - \frac{c}{8} \left( \frac{\Delta_a^2}{\mu \sigma_a^2} \right)^{1/3} \right).$$

Thus by setting $c := 12 \cdot 6^{1/3}$ we have that

$$\epsilon_{k-1} - 12 \cdot 6^{1/3} \left( \frac{\Delta_a \sigma_a^2}{\mu^2} \right)^{2/3} = \frac{8 \sigma_a^2}{\mu} \left( \gamma_{K-1} - \gamma_\star \right) = \frac{8 \sigma_a^2}{\mu} \cdot \frac{\gamma_0 - \gamma_\star}{2^{K-1}} \leq 4 \left( \frac{\Delta_a \sigma_a^2}{\mu^2} \right)^{2/3} = \varepsilon_\star.$$

Indeed, this inequality holds since

$$\frac{8 \sigma_a^2}{\mu} \cdot \frac{\gamma_0 - \gamma_\star}{2^{K-1}} = \frac{8 \sigma_a^2}{\mu} \cdot \frac{\frac{\mu}{2 L_a^2} - \left( 6 L_{\text{eq}}^2 \Delta^2 / (\mu \sigma_a^2) \right)^{1/3}}{\left( \frac{\mu}{L_a^2} \cdot \left( \frac{\sigma_a^2 \mu}{\Delta_a^2} \right)^{1/3} \right)}$$

$$= \frac{8 \sigma_a^2 L_a^2 \Delta_a^{2/3}}{\mu^2 \sigma_a^{2/3} \mu^{1/3}} \left( \frac{\mu}{2 L_a^2} - \frac{6^{1/3} \Delta_a^{2/3}}{\mu^{1/3} \sigma_a^{2/3}} \right)$$

$$= \left( \frac{\Delta_a \sigma_a^2}{\mu^2} \right)^{2/3} \frac{8 L_a^2}{\mu} \left( \frac{\mu}{2 L_a^2} - \frac{6^{1/3} \Delta_a^{2/3}}{\mu^{1/3} \sigma_a^{2/3}} \right)$$

$$= \left( \frac{\Delta_a \sigma_a^2}{\mu^2} \right)^{2/3} \left( 4 - \frac{6^{1/3} \cdot 8 \cdot L_a^2 \Delta_a^{2/3}}{\mu^{4/3} \sigma_a^{2/3}} \right)$$

$$\leq 4 \left( \frac{\Delta_a \sigma_a^2}{\mu^2} \right)^{2/3}$$

$$\leq \varepsilon_\star,$$

where we used the fact that

$$K = 1 + \left\lceil \log_2 \left( \frac{\mu}{L_a^2} \cdot \left( \frac{\sigma_a^2 \mu}{\Delta_a^2} \right)^{1/3} \right) \right\rceil.$$

Therefore, we have that

$$\mathbb{E} \left\| x_K - x_K^\star \right\|^2 \leq 2 (1 + 12 \cdot 6^{1/3}) \left( \frac{\Delta_a \sigma_a^2}{\mu^2} \right)^{2/3} \asymp \varepsilon_\star.$$

What remains is to show that the total time $T$ satisfies the claimed bound. Recall that we set

$$K = 1 + \left\lceil \log_2 \left( \frac{\mu}{L_a^2} \cdot \left( \frac{\sigma_a^2 \mu}{\Delta_a^2} \right)^{1/3} \right) \right\rceil.$$

Observe that

$$T \lesssim \frac{L_a^2}{\mu^2} \log \left( \frac{L_a^2 R}{\sigma_a^2} \right)^+ + \frac{1}{\mu} \sum_{k=1}^{K-1} \frac{1}{\gamma_k}.$$

We need to show that the sum on the left is asymptotically proportional to $\frac{\sigma_a^2}{\mu \varepsilon_\star}$. To this end, observe that

$$\sum_{k=1}^{K-1} \frac{1}{\gamma_k} \le \frac{2L_a^2}{\mu} \sum_{k=1}^{K-1} 2^k \le \frac{2L_a^2}{\mu} \cdot 2^K = \frac{2 \cdot 2L_a^2}{\mu} 2^{K-1}.$$

Using the definition of $K$, we have that

$$2^{K-1} = 2^{\log_2 \left( \frac{\mu}{L_a^2} \cdot \left( \frac{\sigma_a^2 \mu}{\Delta_a^2} \right)^{1/3} \right)} = \frac{\mu}{L_a^2} \cdot \left( \frac{\sigma_a^2 \mu}{\Delta_a^2} \right)^{1/3},$$

Hence, we deduce that

$$\frac{1}{\mu} \sum_{k=1}^{K-1} \frac{1}{\gamma_k} \le \frac{2 \cdot 2L_a^2}{\mu} 2^{K-1} \le \frac{2 \cdot 2L_a^2}{\mu} \cdot \frac{\mu}{L_a^2} \cdot \left( \frac{\sigma_a^2 \mu}{\Delta_a^2} \right)^{1/3} = 4 \left( \frac{\sigma_a^2 \mu}{\Delta_a^2} \right)^{1/3} = \frac{4\sigma_a^2}{\mu} \cdot \left( \frac{\Delta_a \sigma_a^2}{\mu^2} \right)^{-2/3} \asymp \frac{\sigma_a^2}{\mu \varepsilon_\star},$$

as claimed. This completes the proof. $\qquad\square$

## H.2. Beyond Worst Case Expected Tracking Error

The preceding results provide a "worst-case" bound in the sense that $\Delta = \max\{\|u_{t+1} - u_t\|^2\}$ is the largest difference in the decision maker's actions. Here, we want to understand what happens when we make "reasonable" assumptions on the behavior of $\Delta_t := \|u_{t+1} - u_t\|^2$. For instance, one reasonable assumption is that the decision-maker is employing some stochastic gradient method with a convergence guarantee of the form $\mathbb{E}\|u_t - u^*\|^2 \le \mathcal{O}((t+1)^{-2a})$. Here $u^*$ might be a locally optimal point for $\mathcal{L}$ or $\operatorname{argmin}_{u \in \mathcal{U}} \mathcal{L}(u)$ given that players are playing a Nash $x^*(u) \in \operatorname{Eq}(\mathcal{G}_u)$ or even some other solution concept—e.g., in Section 4.1 we introduce the notion of a performatively stable Stackelberg equilibrium. Note that $\mathbb{E}\|u_{t+1} - u_t\|^2 \le \mathcal{O}((t+1)^{-2a})$ means there exists a constant $c_d > 0$ such that $\mathbb{E}\|u_{t+1} - u_t\|^2 \le \frac{c_d}{(t+1)^{2a}}$.

**Proposition H.6.** *Suppose that Assumptions 2.1, H.1, and H.3 hold and that the decision-maker deploys a sequence of actions such that $\mathbb{E}\|u_{t+1} - u_t\|^2 \le \frac{c_d}{(t+1)^{-2a}}$ for some $a \in (0, 1/2]$ and absolute constant $c_d > 0$. Set $\gamma_t = \frac{8}{\mu(t+t_0)^b}$ for some $b \in (0, 1]$ and integer $t_0 \ge 1$ and consider agents running stochastic gradient play with time varying stepsize $\gamma_t$. Then, the iterates satisfy*

$$\mathbb{E}\|x_t - x_t^\star\|^2 \le \max\{(1 + t_0)\|x_0 - x_0^\star\|^2, c_a\} \cdot \begin{cases} (t + t_0)^{-b}, & \text{if } b > \frac{2}{3}a, \\ (t + t_0)^{b/2 - a}, & \text{otherwise,} \end{cases}$$

*where $c_a := \frac{8\sigma_a^2}{\mu^2} + \frac{5L_{eq}^2 c_d^2}{8}$.*

*Proof.* Let $\Delta_{a,t} := \|x_{t+1}^\star - x_t^\star\|$. We know from the proof of Lemma G.4, that stochastic gradient play is $\rho$-contracting. Moreover, for a fixed $u_t$ which induces $x_t^\star \in \operatorname{Eq}(\mathcal{G}_{u_t})$, we have that

$$\mathbb{E}_t\|x_{t+1} - x_t^\star\|^2 \le \frac{1}{1 + \gamma\mu}\|x_t - x_t^\star\|^2 + \frac{2\gamma^2 \sigma_a^2}{1 + \gamma\mu}.$$

Now observe that

$$\begin{aligned} \|x_{t+1} - x_{t+1}^\star\|^2 &= \|x_{t+1} - x_t^\star\|^2 + \|x_t^\star - x_{t+1}^\star\|^2 + 2\langle x_{t+1} - x_t^\star, x_t^\star - x_{t+1}^\star \rangle \\ &\le \|x_{t+1} - x_t^\star\|^2 + \|x_t^\star - x_{t+1}^\star\|^2 + 2\|x_{t+1} - x_t^\star\|\|x_t^\star - x_{t+1}^\star\| \\ &\le \left(1 + \frac{\mu\gamma}{4}\right)\|x_{t+1} - x_t^\star\|^2 + \left(1 + \frac{4}{\mu\gamma}\right)\|x_t^\star - x_{t+1}^\star\|^2 \end{aligned}$$

where the last inequality follows from Young's inequality. Since $1 - \frac{\gamma\mu}{1+\gamma\mu} \leq 1 - \frac{\mu\gamma}{2}$, we have that

$$\mathbb{E}_t\|x_{t+1} - x_{t+1}^\star\|^2 \leq \left(1 + \frac{\gamma\mu}{4}\right)\left(\left(1 - \frac{\gamma\mu}{2}\right)\|x_t - x_t^\star\|^2 + 2\gamma^2\sigma_a^2\left(1 - \frac{\gamma\mu}{2}\right)\right)$$
$$+ \left(1 + \frac{4}{\gamma\mu}\right)\|x_t^\star - x_{t+1}^\star\|^2$$
$$\leq \left(1 - \frac{\gamma\mu}{4}\right)\|x_t - x_t^\star\|^2 + 2\gamma^2\sigma_a^2\left(1 - \frac{\gamma\mu}{4}\right) + \left(1 + \frac{4}{\gamma\mu}\right)\Delta_{a,t}^2$$
$$\leq \left(1 - \frac{\gamma\mu}{4}\right)\|x_t - x_t^\star\| + 2\gamma^2\sigma_a^2 + \frac{5}{\mu\gamma}L_{eq}^2\|u_{t+1} - u_t\|^2.$$

The agents are engaging in stochastic gradient play given the induced sequence of games $\mathcal{G}_{u_t}$ with $\gamma_t = \frac{8}{\mu(t+t_0)^b}$. Hence, plugging $\gamma_t$ in to the above bound, we have that

$$\mathbb{E}_t\|x_{t+1} - x_{t+1}^\star\|^2 \leq \left(1 - \frac{2}{(t+t_0)^b}\right)\|x_t - x_t^\star\|^2 + \frac{8\sigma_a^2}{\mu^2(t+t_0)^{2b}} + \frac{5(t+t_0)^b}{8}L_{eq}^2\|u_{t+1} - u_t\|^2$$
$$\leq \left(1 - \frac{2}{(t+t_0)^b}\right)\|x_t - x_t^\star\|^2 + \frac{8\sigma_a^2}{\mu^2(t+t_0)^{2b}} + \frac{5(t+t_0)^b}{8}L_{eq}^2\frac{c_d^2}{(t+t_0)^{2a}}$$
$$\leq \left(1 - \frac{2}{(t+t_0)^b}\right)\|x_t - x_t^\star\|^2 + \frac{8\sigma_a^2}{\mu^2(t+t_0)^{2b}} + \frac{5}{8}L_{eq}^2\frac{c_d^2}{(t+t_0)^{2a-b}}.$$

Define $D_t := \mathbb{E}\|x_t - x_t^\star\|$. Then there are two cases to analyze.

**Case 1:** If $b > \frac{2}{3}a$, then the above bound reduces to

$$\mathbb{E}_t\|x_{t+1} - x_{t+1}^\star\|^2 \leq \left(1 - \frac{2}{(t+t_0)^b}\right)\|x_t - x_t^\star\|^2 + \frac{c_a}{(t+t_0)^{2b}}$$

Then we claim that

$$D_t \leq \frac{\max\{(1+t_0)D_0, c_a\}}{(t+t_0)^b}.$$

Indeed, it clearly holds for $t = 0$. Hence we may use induction to conclude the argument. Suppose it holds for some fixed $t \geq 1$. Then, we have that

$$D_{t+1} \leq \left(1 - \frac{2}{(t+t_0)^b}\right)\frac{c_a}{(t+t_0)^b} + \frac{c_a}{(t+t_0)^{2b}}$$
$$\leq c_a\left(\frac{1}{(t+t_0)^b} - \frac{2}{(t+t_0)^{2b}}\right) + \frac{c_a}{(t+t_0)^{2b}}$$
$$\leq c_a\left(\frac{1}{(t+t_0)^b} - \frac{1}{(t+t_0)^{2b}}\right)$$
$$\leq \frac{c_a}{(t+1+t_0)^b},$$

where the last inequality holds since $\frac{1}{(t+t_0)^b} - \frac{1}{(t+t_0)^{2b}} \leq \left(\frac{1}{(t+t_0)} - \frac{1}{(t+t_0)^2}\right)^b \leq \frac{1}{(t+1+t_0)^b}$ for any $t \geq 1$ and $b \in (0, 1]$. This verifies the claim.

**Case 2:** Suppose now that $b \leq \frac{2}{3}a$. Then the bound on $D_{t+1}$ reduces to

$$D_{t+1} \leq \left(1 - \frac{2}{(t+t_0)^b}\right)D_t + \frac{c_a}{(t+t_0)^{2a-b}} \leq \left(1 - \frac{2}{(t+t_0)^{a-b/2}}\right)D_t + \frac{c_a}{(t+t_0)^{2a-b}},$$

where the last inequality holds since $b \leq \frac{2}{3}a$. Using a completely analogous argument to case 1, we have that

$$D_t \leq \frac{\max\{(1+t_0)D_1, c_a\}}{(t+t_0)^{a-b/2}}.$$

Therefore, putting the two cases together, concludes the proof. $\qquad\square$

As noted in the main, this proposition shows that if the decision-maker is employing a reasonably well-behaved sequence of actions (i.e., that is stabilizing at a sufficient rate), then the agents can utilize time varying step-sizes to control the drift and obtain an expected tracking error bound that is decaying in time. The rate of decay however highly depends on the behavior of the decision maker's sequence. For instance, if $a = 1/2$, then choosing $b = 1$ leads to a rate of $\mathcal{O}(1/t)$. Here, $a = 1/2$ is not just a reasonable rate for a stochastic gradient method for the decision-maker as we will see in Section 4, but likely the best we could hope for. However, if the agents choose a much slower rate such as $b < 1/3$, then even with $a = 1/2$ the tracking error decays at a rate of $\mathcal{O}(t^{b/2-1/2})$ so that, somewhat counter intuitively, the rate is much slower as $b \to \frac{1}{3}$. This is because the rate of the decision-maker dominates.

### H.3. High-Probability Guarantees

The above results are characterized in terms of the *expected* tracking error; accordingly, characterizing the guarantees of the algorithm are only meaningful if it is run multiple times. Instead, if our algorithm were deployed in real-time with *irreversible* drift, we would like high-probability efficiency results to characterize the performance of our algorithm if it were executed only once. Here, we present high-probability guarantees for the tracking error. We require the following tail assumptions on the equilibrium drift and gradient noise.

**Assumption H.7** (Sub-Gaussian drift and noise). There exist constants $\Delta_\mathsf{a}, \sigma_\mathsf{a} > 0$ such that the following two conditions hold for all $t \geq 0$:

(a) The drift $\Delta_{\mathsf{a},t}^2$ is sub-exponential conditioned on $\mathcal{F}_t$ with parameter $\Delta_\mathsf{a}^2$:

$$\mathbb{E}[\exp(\lambda \Delta_{\mathsf{a},t}^2) | \mathcal{F}_t] \leq \exp(\lambda \Delta_\mathsf{a}^2) \quad \text{for all} \quad 0 \leq \lambda \leq \Delta_\mathsf{a}^2$$

(b) The gradient noise $\xi_t$ is norm sub-Gaussian conditioned on $\mathcal{F}_t$ with parameter $\sigma_\mathsf{a}^2$:

$$\mathbb{P}(\|\xi_t\| \geq \zeta | \mathcal{F}_t) \leq 2\exp(-2\zeta^2/\sigma_\mathsf{a}^2) \quad \text{for all} \quad \zeta > 0.$$

Note that Assumption H.7 implies Assumption H.3 under the with the same constants $\Delta_\mathsf{a}, \sigma_\mathsf{a}$. We need the following (albeit simplified) proposition from (Cutler et al., 2023), which is an extension of Claim D.1 from (Harvey et al., 2019).

**Proposition H.8** (Simplified version of Proposition 29, (Cutler et al., 2023)). *Consider a scalar stochastic process $\{V_t, X_t\}$ on a probability space with filtration $\mathcal{H}_t$ such that $V_t$ is nonnegative and $\mathcal{H}_t$-measurable, and satisfies*

$$V_{t+1} \leq \alpha_t V_t + X_t + \kappa_t$$

*for deterministic constant $\alpha_t \in (-\infty, 1]$. Suppose that the moment generating functions of $X_t$ conditioned on $\mathcal{H}_t$ satisfy*

$$\mathbb{E}[\exp(\lambda X_t) | \mathcal{H}_t] \leq \exp(\lambda \nu_t) \quad \forall\, 0 \leq \lambda \leq \frac{1}{\nu_t}.$$

*for constant $\nu_t > 0$. Then the inequality*

$$\mathbb{E}[\exp(\lambda V_{t+1})] \leq \exp(\lambda \cdot \nu_t) \mathbb{E}\left[\exp(\lambda \alpha_t V_t)\right],$$

*holds for all $0 \leq \lambda \leq \frac{1-\alpha_t}{2\sigma_t^2}$.*

*Proof.* For any index $t$ and scalar $\lambda \geq 0$, the tower rule of expectations implies that

$$\mathbb{E}[\exp(\lambda V_{t+1})] \leq \mathbb{E}[\exp(\lambda(\alpha_t V_t + X_t)] = \mathbb{E}\left[\exp(\lambda \alpha_t V_t) \mathbb{E}[\exp(\lambda X_t) | \mathcal{H}_t]\right].$$

By assumption, we have that $\mathbb{E}[\exp(\lambda X_t) | \mathcal{H}_t] \leq \exp(\lambda \nu_t)$ for $0 \leq \lambda \leq \frac{1}{2\nu_t}$. Thus, we have that

$$\mathbb{E}[\exp(\lambda V_{t+1})] \leq \mathbb{E}\left[\exp(\lambda \alpha_t V_t) \mathbb{E}[\exp(\lambda X_t) | \mathcal{H}_t]\right] \leq \exp(\lambda v_t) \mathbb{E}[\exp(\lambda \alpha_t V_t)],$$

which completes the proof. $\qquad\square$

Given this proposition, we have the following high probability bounds.

**Theorem H.9** (High probability tracking error.). *Suppose that Assumptions 2.1, H.1, and H.7 hold and that the decision-maker deploys a sequence $\{u_s\}_{s=0}^t$ satisfying $\frac{\Delta_a}{\sigma_a} < \frac{\mu^2}{4\sqrt{3}L_a^3}$ so that the agents are in the low drift-to-noise regime. Let $\{x_t\}$ be the iterates produced by stochastic gradient play (SGP) with $\gamma \leq \frac{\mu}{2L_a^2}$. There exists an absolute constant $c > 0$ such that for any specified $t \in \mathbb{N}$ and $\delta \in (0,1)$, the following estimate holds with probability at least $1 - \delta$:*

$$\|x_t - x_t^\star\|^2 \leq \left(1 - \frac{\mu\gamma}{4}\right)^t \|x_0 - x_0^\star\|^2 + c\left(\frac{\sigma_a^2\gamma}{\mu} + \left(\frac{\Delta_a}{\mu\gamma}\right)^2\right)\log\left(\frac{e}{\delta}\right).$$

*Proof.* By Young's inequality, we have that

$$\begin{aligned}
\|x_{t+1} - x_{t+1}^\star\|^2 &= \|x_{t+1} - x_t^\star\|^2 + \|x_t^\star - x_{t+1}^\star\|^2 + 2\langle x_{t+1} - x_t^\star, x_t^\star - x_{t+1}^\star\rangle, \\
&\leq \|x_{t+1} - x_t^\star\|^2 + \|x_t^\star - x_{t+1}^\star\|^2 + 2\|x_{t+1} - x_t^\star\|\|x_t^\star - x_{t+1}^\star\|, \\
&\leq (1 + \lambda)\|x_{t+1} - x_t^\star\|^2 + (1 + \lambda^{-1})\|x_t^\star - x_{t+1}^\star\|^2,
\end{aligned}$$

for some $\lambda$. Observe that $x \mapsto \frac{1}{2}\|x_t - \gamma\widehat{\omega}(x_t) - x\|^2$ is a 1-strongly convex function over $\mathcal{X}$. Therefore, we deduce that

$$\begin{aligned}
\frac{1}{2}\|x_{t+1} - x_t^\star\|^2 &\leq \frac{1}{2}\|x_t - \gamma\widehat{\omega}(x_t) - x_t^\star\|^2 - \frac{1}{2}\|x_t - \gamma\widehat{\omega}(x_t) - x_{t+1}\|^2, \\
&\leq \frac{1}{2}\|x_t - x_t^\star\|^2 - \gamma\langle\widehat{\omega}(x_t), x_{t+1} - x_t^\star\rangle - \frac{1}{2}\|x_{t+1} - x_t\|^2, \\
&= \frac{1}{2}\|x_t - x_t^\star\|^2 - \gamma\langle\widehat{\omega}(x_t), x_t - x_t^\star\rangle - \frac{1}{2}\|x_{t+1} - x_t\|^2 - \gamma\langle\widehat{\omega}(x_t), x_{t+1} - x_t\rangle.
\end{aligned}$$

Next, we have that

$$\begin{aligned}
\frac{1}{2}\|x_{t+1} - x_t^\star\|^2 &\leq \frac{1}{2}\|x_t - x^*\|^2 - \gamma\langle\widehat{\omega}(x_t), x_t - x_t^\star\rangle - \frac{1}{2}\|x_{t+1} - x_t\|^2 - \gamma\langle\widehat{\omega}(x_t), x_{t+1} - x_t\rangle, \\
&\leq \frac{1}{2}\|x_t - x_t^\star\|^2 - \gamma\langle\omega(x_t), x_t - x_t^\star\rangle - \frac{1}{2}\|x_{t+1} - x^*\|^2 - \gamma\langle\widehat{\omega}(x_t), x_{t+1} - x_t\rangle, \\
&= \frac{1}{2}\|x_t - x_t^\star\|^2 - \gamma\langle\omega(x_{t+1}), x_{t+1} - x_t^\star\rangle - \frac{1}{2}\|x_{t+1} - x_t\|^2, \\
&\quad + \gamma\underbrace{\langle\widehat{\omega}(x_t) - \omega(x_t), x_t - x_{t+1}\rangle}_{=:P_1} + \gamma\underbrace{\langle\omega(x_t) - \omega(x_{t+1}), x_t^\star - x_{t+1}\rangle}_{=:P_2}.
\end{aligned}$$

Since each induced game $\mathcal{G}_u$ is is $\mu$–strongly monotone, we have that

$$\langle\omega(x_{t+1}), x_{t+1} - x_t^\star\rangle \geq \langle\omega(x_{t+1}) - \omega(x_t^\star), x_{t+1} - x_t^\star\rangle \geq \mu\|x_{t+1} - x_t^\star\|^2.$$

This in turn implies that

$$\frac{1 + 2\gamma\mu}{2}\|x_{t+1} - x_t^\star\|^2 \leq \frac{1}{2}\|x_t - x^*\|^2 - \frac{1}{2}\|x_{t+1} - x_t\|^2 + \gamma(P_1 + P_2).$$

Applying Young's inequality, we have that

$$P_1 \leq \frac{\|\xi_t\|^2}{2\nu_1} + \frac{\nu_1\|x_{t+1} - x_t\|^2}{2},$$

and analogously, we have that

$$\begin{aligned}
P_2 &\leq \frac{\|\omega(x_t) - \omega(x_{t+1})\|^2}{2\nu_2} + \frac{\nu_2\|x_{t+1} - x_t^\star\|^2}{2}, \\
&\leq \frac{L_a^2\|x_t - x_{t+1}\|^2}{2\nu_2} + \frac{\nu_2\|x_{t+1} - x_t^\star\|^2}{2}.
\end{aligned}$$

Combining these two bounds, we have that

$$\frac{1 + 2\gamma\mu - \gamma\nu_2}{2}\|x_{t+1} - x_t^\star\|^2 \le \frac{1}{2}\|x_t - x_t^\star\|^2 + \frac{\|\xi_t\|^2}{2\nu_1} - \frac{(1 - \gamma L_a^2 \nu_2^{-1} - \gamma\nu_1)}{2}\|x_{t+1} - x_t\|^2.$$

Setting $\nu_2 = \mu$ and $\nu_1 = \gamma^{-1} - L_a^2/\mu$, we have that the last term on the right hand side is zero, and since $\gamma \le \frac{\mu}{2L_a^2}$ we have that $\nu_1 \ge \frac{1}{2\gamma}$; indeed, $-\frac{1}{2\gamma} \le -\frac{2L_a^2}{\mu}$ so that $\nu_1 = \frac{1}{\gamma} - \frac{2L_a^2}{\mu} \ge \frac{1}{\gamma} - \frac{1}{2\gamma} = \frac{1}{2\gamma}$. Applying these bounds on the constants $\nu_1$ and $\nu_2$, we have that

$$\|x_{t+1} - x_t^\star\|^2 \le \frac{1}{1 + \gamma\mu}\|x_t - x^*\|^2 + \frac{2\gamma^2\|\xi_t\|^2}{1 + \mu\gamma}.$$

Thus, we have that

$$
\begin{aligned}
\|x_{t+1} - x_{t+1}^\star\|^2 &= \|x_{t+1} - x_t^\star\|^2 + \|x_t^\star - x_{t+1}^\star\|^2 + 2\langle x_{t+1} - x_t^\star, x_t^\star - x_{t+1}^\star\rangle \\
&\le \|x_{t+1} - x_t^\star\|^2 + \|x_t^\star - x_{t+1}^\star\|^2 + 2\|x_{t+1} - x_t^\star\|\|x_t^\star - x_{t+1}^\star\| \\
&\le (1 + \lambda)\left(\frac{1}{1 + \gamma\mu}\|x_t - x_t^\star\|^2 + \frac{2\gamma^2\|\xi_t\|^2}{1 + \mu\gamma}\right) + (1 + \lambda^{-1})\|x_t^\star - x_{t+1}^\star\|^2 \\
&\le (1 + \lambda)\left(\left(1 - \frac{\mu\gamma}{2}\right)\|x_t - x_t^\star\|^2 + 2\left(1 - \frac{\mu\gamma}{2}\right)\gamma^2\|\xi_t\|^2\right) + (1 + \lambda^{-1})\|x_t^\star - x_{t+1}^\star\|^2 \\
&\le \left(1 - \frac{\mu\gamma}{4}\right)\|x_t - x_t^\star\|^2 + 2\left(1 - \frac{\mu\gamma}{4}\right)\gamma^2\|\xi_t\|^2 + \left(1 + \frac{4}{\mu\gamma}\right)\Delta_{a,t}^2,
\end{aligned}
$$

where we have set $\lambda = \frac{\mu\gamma}{4}$, and $\Delta_{a,t} := \|x_t^\star - x_{t+1}^\star\|$. Bounding the last two terms, we have that

$$\|x_{t+1} - x_{t+1}^\star\|^2 \le \left(1 - \frac{\mu\gamma}{4}\right)\|x_t - x_t^\star\|^2 + 2\gamma^2\|\xi_t\|^2 + \frac{5}{\mu\gamma}\Delta_{a,t}^2. \tag{13}$$

Under Assumption H.7, there exists an absolute constant $c \ge 1$ such that $\|\xi_t\|^2$ is sub-exponential conditioned on $\mathcal{F}_t$ with parameter $c\sigma_a^2$ and $\xi_t$ is mean-zero sub-Gaussian conditioned on $\mathcal{F}_t$ with parameter $c\sigma_a$ for all $t$ (cf. Lemma 3 from Jin et al. (2019)). Assumption H.7 also implies that $\Delta_{a,t}^2$ is sub-exponential conditioned on $\mathcal{F}_t$ with parameter $\Delta_a^2$. Given (13), we apply Proposition H.8 with parameters

$$V_t = \|x_t - x_t^\star\|^2, \quad D_t = 0, \quad X_t = 2\gamma^2\|\xi_t\|^2 + \frac{5}{\mu\gamma}\Delta_{a,t}^2, \quad \alpha_t = 1 - \frac{\mu\gamma}{4}, \quad \kappa_t = 0, \quad \text{and} \quad \nu_t = 2\gamma^2 c\sigma_a^2 + \frac{5}{\mu\gamma}\Delta_a^2.$$

This yields the estimate

$$\mathbb{E}[\exp(\lambda\|x_{t+1} - x_{t+1}^\star\|^2)] \le \exp\left(\lambda\left(2\gamma^2 c\sigma_a^2 + \frac{5\Delta_a^2}{\mu\gamma}\right)\right)\mathbb{E}\left[\exp\left(\lambda\left(1 - \frac{\mu\gamma}{4}\right)\|x_t - x_t^\star\|^2\right)\right], \tag{14}$$

for all

$$0 \le \lambda \le \frac{1}{2(2\gamma^2 c\sigma_a^2 + 5\Delta_a^2/(\mu\gamma))}. \tag{15}$$

Since $\left(1 - \frac{\mu\gamma}{4}\right) \in (0, 1]$ and $V_t$ is a non-negative random variable (almost surely), Jensen's inequality implies that

$$
\begin{aligned}
\mathbb{E}[\exp(\lambda V_{t+1})] &\le \exp\left(\lambda\nu_t\right)\mathbb{E}\left[\exp\left(\lambda V_t\right)^{\left(1 - \frac{\mu\gamma}{4}\right)}\right] \\
&\le \exp\left(\lambda\nu_t\right)\mathbb{E}\left[\exp\left(\lambda V_t\right)\right]^{\left(1 - \frac{\mu\gamma}{4}\right)}.
\end{aligned}
$$

Iterating this expression, we have that

$$\mathbb{E}\left[\exp(\lambda\|x_t - x_t^\star\|^2)\right] \leq \exp\left(\left(2c\gamma^2\sigma_a^2 + \frac{5\Delta_a^2}{\mu\gamma}\right)\sum_{s=0}^{t-1}\left(1 - \frac{\mu\gamma}{4}\right)^s\right)\left(\mathbb{E}[\exp(\lambda\|x_0 - x_0^\star\|^2)]\right)^{(1-\mu\gamma/4)^t}$$

$$= \exp(\lambda\nu)\exp\left(\lambda\left(\left(1 - \frac{\mu\gamma}{4}\right)^t\|x_0 - x_0^\star\|^2 + \left(2c\gamma^2\sigma_a^2 + \frac{5\Delta_a^2}{\mu\gamma}\right)\sum_{s=1}^{t-1}\left(1 - \frac{\mu\gamma}{4}\right)^s\right)\right)$$

$$\leq \exp\left(\lambda\left(\left(1 - \frac{\mu\gamma}{4}\right)^t\|x_0 - x_0^\star\|^2 + \left(2c\gamma^2\sigma_a^2 + \frac{5\Delta_a^2}{\mu\gamma}\right)\sum_{s=0}^{t-1}\left(1 - \frac{\mu\gamma}{4}\right)^s\right)\right)$$

$$\leq \exp\left(\lambda\left(\left(1 - \frac{\mu\gamma}{4}\right)^t\|x_0 - x_0^\star\|^2 + \left(\frac{8c\gamma\sigma_a^2}{\mu} + \frac{20\Delta_a^2}{(\mu\gamma)^2}\right)\right)\right)$$

for all $\lambda$ satisfying (33), where the equality holds since $\|x_0 - x_0^\star\|^2$ is a constant. Let $\nu := \frac{32(c\sigma_a)^2\gamma}{\mu} + 20\left(\frac{\Delta_a}{\mu\gamma}\right)^2$. Recall that $c \geq 1$ and $\mu\gamma \leq 1$ so that

$$\left(\frac{8c\gamma\sigma_a^2}{\mu} + \frac{20\Delta_a^2}{(\mu\gamma)^2}\right) \leq \nu$$

and

$$\frac{1}{\nu} = \frac{\mu}{32\gamma(c\sigma_a)^2 + 20\Delta_a^2/(\mu\gamma^2)} \leq \min\left\{\frac{\mu}{32 \cdot c^2\gamma\sigma_a^2}, \frac{1}{2(2\gamma^2 c\sigma_a^2 + 5\Delta_a^2/(\mu\gamma))}\right\}.$$

Hence, we have that

$$\mathbb{E}\left[\exp\left(\lambda\left(\|x_t - x_t^\star\|^2 - \left(1 - \frac{\mu\gamma}{4}\right)^t\|x_0 - x_0^\star\|^2\right)\right)\right] \leq \exp(\lambda\nu) \quad \forall \ 0 \leq \lambda \leq \frac{1}{\nu}.$$

Rewriting this expression, we have that

$$\frac{\mathbb{E}\left[\exp\left(\lambda\left(\|x_t - x_t^\star\|^2 - \left(1 - \frac{\mu\gamma}{4}\right)^t\|x_0 - x_0^\star\|^2\right)\right)\right]}{\exp(\lambda\nu)} \leq 1.$$

Applying Markov's inequality, we have that

$$\Pr\left(\exp\left(\lambda\left(\|x_t - x_t^\star\|^2 - \left(1 - \frac{\mu\gamma}{4}\right)^t\|x_0 - x_0^\star\|^2\right)\right) \geq \frac{\exp(\lambda\nu)}{\delta}\right)$$

$$\leq \frac{\mathbb{E}\left[\exp\left(\lambda\left(\|x_t - x_t^\star\|^2 - \left(1 - \frac{\mu\gamma}{4}\right)^t\|x_0 - x_0^\star\|^2\right)\right)\right]}{\exp(\lambda\nu)/\delta} \leq \delta$$

Therefore, setting $\lambda = \frac{1}{\nu}$, with probability $1 - \delta$, we have that

$$\|x_t - x_t^\star\|^2 \leq \left(1 - \frac{\mu\gamma}{4}\right)^t\|x_0 - x_0^\star\|^2 + \left(\frac{32(c\sigma_a)^2\gamma}{\mu} + 20\left(\frac{\Delta_a}{\mu\gamma}\right)^2\right)\log\left(\frac{e}{\delta}\right), \tag{16}$$

as claimed. $\square$

The above theorem can be translated to a time-to-track high probability result.

**Corollary H.10** (Time to track with high probability.)**.** *Suppose that the assumptions of Theorem H.9 hold so that we are in the low drift-to-noise regime, and there is a constant $R$ available such that $R \geq \|x_0 - x_0^\star\|^2$. Suppose that the agents are running* SGP *in $k = 0, \ldots, K - 1$ stages (cf. Lemma E.2) with constants constants*

$$K = 1 + \left\lceil\log_2\left(\frac{\mu}{L_a^2} \cdot \left(\frac{\sigma_a^2\mu}{\Delta_a^2}\right)^{1/3}\right)\right\rceil, \quad \gamma_0 = \frac{\mu}{2L_a^2},$$

*and*

$$T_0 = \left\lceil \frac{8L_{\mathrm{a}}^2}{\mu^2} \log\left(\frac{L_{\mathrm{a}}^2 R}{\sigma_{\mathrm{a}}^2}\right)^+ \right\rceil, \quad \gamma_k = \frac{\gamma_{k-1} + \gamma_\star}{2}, \quad T_k = \left\lceil \frac{4\log(12)}{\mu\gamma_k} \right\rceil \quad \textit{for all } k \geq 1.$$

*Then* $T = T_0 + \cdots + T_{K-1}$ *satisfies*

$$T \lesssim \frac{L_{\mathrm{a}}^2}{\mu^2} \log\left(\frac{L_{\mathrm{a}}^2 R}{\sigma_{\mathrm{a}}^2}\right)^+ + \frac{\sigma_{\mathrm{a}}^2}{\mu^2 \varepsilon_\star} \leq \frac{L_{\mathrm{a}}^2}{\mu^2} \log\left(\frac{R}{\varepsilon_\star}\right)^+ + \frac{\sigma_{\mathrm{a}}^2}{\mu^2 \varepsilon_\star},$$

*and for any given $\delta \in (0,1)$, the tracking error satisfies $\|x_K - x_K^\star\|^2 \lesssim \varepsilon_\star \log\left(\frac{e}{\delta}\right)$ with probability at least $1 - \delta$.*

*Proof.* Set $t_0 := 0$. For each $k$, let $t_k := T_0 + \cdots + T_{k-1}$, let $x_t^\star$ be the Nash equilibrium of the game $\mathcal{G}_{u_t}$, and set

$$E_k := c\left(\frac{\gamma_k \sigma_{\mathrm{a}}^2}{\mu} + \left(\frac{\Delta_{\mathrm{a}}}{\mu\gamma_\star}\right)^2\right)$$

where $c \geq 1$ is an absolute constant satisfying the bound (16). Since $\gamma_k \geq \gamma_\star$, Theorem H.9 implies that for any specified index $k$ and $\delta \in (0,1)$, the following estimate holds with probability at least $1 - \delta$:

$$\|x_{k+1} - x_{k+1}^\star\|^2 \leq \left(1 - \frac{\mu\gamma_k}{4}\right)^{T_k} \|x_0 - x_0^\star\|^2 + c\left(\frac{\sigma_{\mathrm{a}}^2 \gamma_k}{\mu} + \left(\frac{\Delta_{\mathrm{a}}}{\mu\gamma_k}\right)^2\right) \log\left(\frac{e}{\delta}\right),$$

$$\leq e^{-\mu\gamma_k T_k/4} \|x_k - x_k^\star\|^2 + E_k \log\left(\frac{e}{\delta}\right).$$

We claim that an induction-based argument yields the following: for each $k \geq 1$, the estimate $\|x_k - x_k^\star\|^2 \leq A E_{k-1} \log(e/\delta)$ holds with probability at least $1 - \delta$ for all $\delta \in (0,1)$. To see the base case, observe that

$$e^{-\mu\gamma_0 T_0/4}\|x_0 - x_0^\star\|^2 \leq \exp\left(-\frac{\mu^2}{4 \cdot 2L_{\mathrm{a}}^2}\frac{8L_{\mathrm{a}}^2}{\mu^2}\log\left(\frac{L_{\mathrm{a}}^2 R}{\sigma_{\mathrm{a}}^2}\right)\right)\|x_0 - x_0^\star\|^2 = \frac{\sigma_{\mathrm{a}}^2}{L_{\mathrm{a}}^2 R}\|x_0 - x_0^\star\|^2 \leq \frac{\sigma_{\mathrm{a}}^2}{L_{\mathrm{a}}^2}$$

and

$$E_0 = c\left(\frac{\gamma_0 \sigma_{\mathrm{a}}^2}{\mu} + \left(\frac{\Delta_{\mathrm{a}}}{\mu\gamma_\star}\right)^2\right) \geq c\left(\frac{\sigma_{\mathrm{a}}^2}{2L_{\mathrm{a}}^2} + \left(\frac{\Delta_{\mathrm{a}}}{\mu\gamma_0}\right)^2\right) \geq c\left(\frac{3\sigma_{\mathrm{a}}^2}{4L_{\mathrm{a}}^2}\right) \geq \left(\frac{3\sigma_{\mathrm{a}}^2}{4L_{\mathrm{a}}^2}\right)$$

since $\gamma_k \geq \gamma_\star$ and $c \geq 1$, and we are in the regime where $\frac{\Delta_{\mathrm{a}}}{\sigma_{\mathrm{a}}} < \frac{\mu^2}{4L_{\mathrm{a}}^3}$. Therefore we have that

$$e^{-\mu\gamma_0 T_0/4}\|x_0 - x_0^\star\|^2 \leq \frac{\sigma_{\mathrm{a}}^2}{L_{\mathrm{a}}^2} \leq \frac{4}{3}E_0.$$

Hence

$$\|x_1 - x_1^\star\|^2 \leq e^{-\mu\gamma_0 T_0/4}\|x_0 - x_0^\star\|^2 + E_0 \log\left(\frac{e}{\delta}\right) \leq \frac{7}{3}E_0 \log\left(\frac{e}{\delta}\right) \leq 3E_0 \log\left(\frac{e}{\delta}\right)$$

since $\log(e/\delta) \geq 1$, and where we take the bound in the last inequality to simplify constants.

Now, suppose the claim holds for some index $k \geq 1$ and let $\delta \in (0,1)$; then $\|x_k - x_k^\star\|^2 \leq 3E_{k-1}\log(2e/\delta)$ with probability at least $1 - \delta/2$. Since

$$e^{-\mu\gamma_k T_k/4} \leq \exp\left(-\mu\gamma_k \left\lceil\frac{4\log(12)}{\mu\gamma_k}\right\rceil \cdot \frac{1}{4}\right) \leq \frac{1}{12},$$

we also have that

$$\|x_{k+1} - x_{k+1}^\star\|^2 \leq e^{-\mu\gamma_k T_k/4}\|x_k - x_k^\star\|^2 + E_k \log\left(\frac{2e}{\delta}\right),$$

$$\leq \frac{1}{12}\|x_k - x_k^\star\|^2 + E_k \log\left(\frac{2e}{\delta}\right)$$

$$\leq \frac{E_k}{6E_{k-1}} + E_k \log\left(\frac{2e}{\delta}\right),$$

$$\leq \frac{3}{6}E_k \log\left(\frac{2e}{\delta}\right) + E_k \log\left(\frac{2e}{\delta}\right)$$

with probability at least $1 - \delta/2$. Taking a union bound, we have that

$$\|x_{k+1} - x_{k+1}\|^2 \le \frac{3}{2} E_k \log\left(\frac{2e}{\delta}\right) \le 3E_k \log\left(\frac{e}{\delta}\right),$$

with probability at least $1 - \delta$. This completes the inductive proof. Hence, fixing $\delta \in (0, 1)$, we have that $\|x_K - x_K^\star\|^2 \le 3E_{K-1} \log(e/\delta)$ with probability at least $1 - \delta$.

Recall that we are in the regime where $\frac{\Delta_a}{\sigma_a} < \frac{\mu^2}{4\sqrt{3}L_a^3}$ and we have set constants $\gamma_\star := \left(6\Delta_a^2/(\mu\sigma_a^2)\right)^{1/3}$ and $\varepsilon_\star := (\Delta_a\sigma_a^2/\mu^2)^{2/3}$. Observe that for some constant $C$, the following is true:

$$
\begin{aligned}
\frac{2}{c} E_{k-1} - C\left(\frac{\Delta_a\sigma_a^2}{\mu^2}\right)^{2/3} &= \frac{2}{\mu}\left(\gamma_{k-1}\sigma_a^2 + \frac{\Delta_a^2}{\mu\gamma_\star^2}\right) - C\left(\frac{\Delta_a\sigma_a^2}{\mu^2}\right)^{2/3} \\
&= \frac{2}{\mu}\left(\gamma_{k-1}\sigma_a^2 + \left(\frac{\Delta_a^2}{\mu\sigma_a^2}\right)^{1/3}\frac{\sigma_a^2}{6^{2/3}}\right) - C\left(\frac{\Delta_a\sigma_a^2}{\mu^2}\right)^{2/3} \\
&= \frac{2\sigma_a^2}{\mu}\left(\gamma_{k-1} + \left(\frac{\Delta_a^2}{6^2\sigma_a^2\mu}\right)^{1/3}\right) - C\left(\frac{\Delta_a\sigma_a^2}{\mu^2}\right)^{2/3} \\
&= \frac{2\sigma_a^2}{\mu}\left(\gamma_{k-1} + \left(\frac{\Delta_a^2}{6^2\sigma_a^2\mu}\right)^{1/3} - \frac{C}{2}\left(\frac{\Delta_a^2}{\mu\sigma_a^2}\right)^{1/3}\right).
\end{aligned}
$$

With this expression in hand, setting $C := 7 \cdot \left(\frac{2}{9}\right)^{1/3}$, we have that

$$\frac{2}{c} E_{K-1} - 7 \cdot \left(\frac{2}{9}\right)^{1/3}\left(\frac{\Delta_a\sigma_a^2}{\mu^2}\right)^{2/3} = \frac{2\sigma_a^2}{\mu}(\gamma_{K-1} - \gamma_\star) = \frac{2\sigma_a^2}{\mu}\cdot\frac{\gamma_0 - \gamma_\star}{2^{K-1}} \le \left(\frac{\Delta_a\sigma_a^2}{\mu^2}\right)^{2/3} = \varepsilon_\star.$$

Therefore, we deduce that

$$E_{K-1} \le \frac{c}{2}\left(1 + 7 \cdot \left(\frac{2}{9}\right)^{1/3}\right)\left(\frac{\Delta_a\sigma_a^2}{\mu^2}\right)^{2/3}\log\left(\frac{e}{\delta}\right)$$

so that

$$\mathbb{E}\|x_K - x_K^\star\|^2 \le 3E_{K-1}\log\left(\frac{e}{\delta}\right) \le \frac{3 \cdot c}{2}\left(1 + 7 \cdot \left(\frac{2}{9}\right)^{1/3}\right)\left(\frac{\Delta_a\sigma_a^2}{\mu^2}\right)^{2/3}\log\left(\frac{e}{\delta}\right) \asymp \varepsilon_\star \log\left(\frac{e}{\delta}\right).$$

What remains is to show that the total time $T$ satisfies the claimed bound. To this end, recall that

$$K = 1 + \left\lceil \log_2\left(\frac{\mu}{L_a^2}\cdot\left(\frac{\sigma_a^2\mu}{\Delta_a^2}\right)^{1/3}\right)\right\rceil.$$

and observe that

$$T \lesssim \frac{L_a^2}{\mu^2}\log\left(\frac{L_a^2 R}{\sigma_a^2}\right)^+ + \frac{1}{\mu}\sum_{k=1}^{K-1}\frac{1}{\gamma_k}.$$

Here, we need to show that the sum on the right is asymptotically proportional to $\sigma_a^2/(\mu\varepsilon_\star)$. Indeed, we have that

$$\sum_{k=1}^{K-1}\frac{1}{\gamma_k} \le \frac{2L_a^2}{\mu}\sum_{k=1}^{K-1}2^k \le \frac{2L_a^2}{\mu}\cdot 2^K = \frac{2 \cdot 2L_a^2}{\mu}2^{K-1},$$

so that by using the definition of $K$, we have

$$2^{K-1} = 2^{\log_2\left(\frac{\mu}{L_a^2}\cdot\left(\frac{\sigma_a^2\mu}{\Delta_a^2}\right)^{1/3}\right)} = \frac{\mu}{L_a^2}\cdot\left(\frac{\sigma_a^2\mu}{\Delta_a^2}\right)^{1/3}.$$

Hence, we deduce that

$$\sum_{k=1}^{K-1} \frac{1}{\gamma_k} \leq \frac{2 \cdot 2L_{\mathsf{a}}^2}{\mu} 2^{K-1} \leq \frac{2 \cdot 2L_{\mathsf{a}}^2}{\mu} \cdot \frac{\mu}{L_{\mathsf{a}}^2} \cdot \left(\frac{\sigma_{\mathsf{a}}^2 \mu}{\Delta_{\mathsf{a}}^2}\right)^{1/3} = 4 \left(\frac{\sigma_{\mathsf{a}}^2 \mu}{\Delta_{\mathsf{a}}^2}\right)^{1/3} = \frac{4\sigma_{\mathsf{a}}^2}{\mu} \cdot \left(\frac{\Delta_{\mathsf{a}}\sigma_{\mathsf{a}}^2}{\mu^2}\right)^{-2/3} \asymp \frac{\sigma_{\mathsf{a}}^2}{\mu\varepsilon_\star},$$

as claimed. This completes the proof. $\qquad\square$

---

**Algorithm 4** Epoch-Based Algorithm Framework for Stochastic Stackelberg Games

---

1: **Input:** decision maker algorithm $\mathtt{Alg_{dm}}$, horizon $T$, stepsize schedule $\{\eta_t\}$, initial parameter $u_1 \in \mathcal{U}$, and query radius $\delta > 0$ (if $\mathtt{Alg_{dm}} = \mathtt{DFM}$)
2: **for** $t = 1, \ldots, T$ **do**
3:    **if** $\mathtt{Alg_{dm}} = \mathtt{DFM}$ **then**
4:       Sample $v_t \sim \mathbb{S}^d$ uniformly at random                       `/* i.e., derivative free method */`
5:       Set $\tilde{u}_t = u_t + \delta v_t$ and $\widetilde{\mathcal{U}} = (1-\delta)\mathcal{U}$
6:    **else if** $\mathtt{Alg_{dm}} = \mathtt{RGM}$ **then**
7:       Set $\tilde{u}_t = u_t$ and $\widetilde{\mathcal{U}} = \mathcal{U}$                           `/* i.e., repeated gradient method`
8:    **end if**
9:    **for** $k = 1, \ldots, \tau_t$ **do**
10:       Query agents with $\tilde{u}_t$                `/* i.e., agents update with any $\rho$ contracting method`
11:    **end for**
12:    Decision-maker observes $x_t^{\tau_t}(\tilde{u}_t)$
13:    **if** $\mathtt{Alg_{dm}} = \mathtt{DFM}$ **then**
14:       Set $\widehat{g}_t = \frac{d}{\delta}\ell(\tilde{u}_t, (x_t^{\tau_t}(\tilde{u}_t), \xi))v_t$
15:    **else if** $\mathtt{Alg_{dm}} = \mathtt{RGM}$ **then**
16:       Set $\widehat{g}_t = \nabla_u \ell(\tilde{u}_t, (x_t^{\tau_t}(\tilde{u}_t), \xi))$
17:    **end if**
18:    Update $u_{t+1} = \mathrm{proj}_{\widetilde{\mathcal{U}}}(u_t - \eta_t \widehat{g}_t)$
19: **end for**

---

# I. Naïve Decision-Maker

In Algorithm 4 we state the main algorithm structure for the epoch based methods including the *naïve decision-maker*. In this appendix section, we provide the formal statements for the results in Section 4.1 and the proofs. To reduce notation in places, we let $L_\ell$ denote the overall Lipschitz parameter for $(u, z) \mapsto (\nabla_u \ell(u, z), \nabla_z \ell(u, z))$.

## I.1. Existence of Performatively Stable Stackelberg Equilibrium

Recall that performatively stable Stackelberg equilibrium are precisely the fixed points of the map

$$\mathtt{pseq}(u') := \left\{ u \in \mathcal{U} : u \text{ is optimal for } \mathbb{E}_{\xi \sim \mathcal{D}_e(u')} \ell(u, (x^*(u'), \xi)) \text{ and } x^*(u') \in \mathtt{Eq}(\mathcal{G}_{u'}) \right\}.$$

**Lemma I.1.** *Fix a function $\ell : \mathbb{R}^d \times \mathcal{Z} \to \mathbb{R}$ such that $\ell(\cdot, z)$ is $C^1$ for all $z \in \mathcal{Z}$ and the map $z \mapsto \nabla_u \ell(u, z)$ is $L_z$-Lipschitz continuous for any $u \in \mathbb{R}^d$. Fix now any measures $\nu_1, \nu_2 \in \mathbb{P}(\mathcal{Z})$ such that $\ell(u, \cdot)$ is both $\nu_1$ and $\nu_2$ integrable for all $u$. Then we may exchange differentiation and integration $\nabla_u \mathbb{E}_{z \sim \nu} \ell(u, z) = \mathbb{E}_{z \sim \nu} \nabla_u \ell(u, z)$ and the estimate holds:*

$$\sup_u \|\nabla_u \mathbb{E}_{z \sim \nu_1} \ell(u, z) - \nabla_u \mathbb{E}_{z \sim \nu_2} \ell(u, z)\| \leq L_z \cdot W_1(\nu_1, \nu_2).$$

**Assumption I.2** (Lipschitz Distributions). *There exists $L_{\mathsf{en}} > 0$ satisfying*

$$W_1(\mathcal{D}_e(u), \mathcal{D}_e(w)) \leq L_{\mathsf{en}} \cdot \|u - w\| \quad \text{for all } u, w \in \mathcal{U}.$$

**Theorem I.3** (Existence & Uniqueness of Performatively Stable Equilibrium). *Under Assumptions 2.1, 4.1 and I.2 and when $1 < \alpha/(L_z(L_{\text{eq}} + L_{\text{en}}))$, there exists a unique performatively stable equilibrium. When the non-strategic decision-dependent component is stationary—i.e., $\mathcal{D}_e(u) \equiv \mathcal{D}_e$ for stationary distribution $\mathcal{D}_e$—existence and uniqueness is guaranteed if $1 < \alpha/(L_z L_{\text{eq}})$ and Assumption I.2 is no longer required.*

*Proof.* We show that $\text{pseq}(\cdot)$ is Lipschitz continuous with parameter $\lambda$. Since the induced game $\mathcal{G}_u$ is $\mu$ strongly monotone for any $u \in \mathcal{U}$ by assumption, there is a unique induced Nash equilibrium $x^*(u) \in \text{Eq}(\mathcal{G}_u)$ for each $u$. Consider two points $u$ and $u'$ and set $w := \text{pseq}(u)$ and $w' := \text{pseq}(u')$. First order optimality conditions for $w$ and $w'$ guarantee

$$\langle g_u(w), w - w' \rangle \leq 0 \quad \text{and} \quad \langle g_{u'}(w'), w' - w \rangle \leq 0,$$

where $g_v(v') = \mathbb{E}_{\xi \sim \mathcal{D}_e(v)} \nabla_u \ell(v', (x^*(v), \xi))$. Since the loss $\ell(\cdot, z)$ is $\alpha$-strongly convex for any $z \in \mathcal{Z}$, we have that

$$
\begin{aligned}
\alpha \cdot \|w - w'\|^2 &\leq \langle g_u(w) - g_u(w'), w - w' \rangle \\
&\leq \langle g_{u'}(w') - g_u(w'), w - w' \rangle \\
&\leq \|g_{u'}(w') - g_u(w')\| \cdot \|w - w'\| \\
&= \left\| \mathbb{E}_{\xi \sim \mathcal{D}_e(u')} \nabla_u \ell(w', (x^*(u'), \xi)) - \mathbb{E}_{\xi \sim \mathcal{D}_e(u)} \nabla_u \ell(w', (x^*(u), \xi)) \right\| \cdot \|w - w'\| \\
&\leq L_z(L_{\text{eq}} + L_{\text{en}})\|u' - u\| \cdot \|w - w'\|.
\end{aligned}
$$

Dividing through by $\|w - w'\|$ guarantees

$$\|w - w'\| = \|\text{pseq}(u) - \text{pseq}(u')\| \leq \frac{L_z(L_{\text{eq}} + L_{\text{en}})}{\alpha}\|u - u'\|.$$

Observe that $L_{\text{eq}} + L_{\text{en}}$ characterizes the total reactivity of the environment due to decision-dependence from *both* the induced agent behavior *and* environmental stochasticity. In the regime where $(L_{\text{eq}} + L_{\text{en}}) < \frac{\alpha}{L_z}$, then $\lambda \in [0, 1)$ so that that $\text{pseq}(\cdot)$ is indeed a contraction as claimed. The result follows immediately from the Banach fixed point theorem. $\qquad\square$

## I.2. Characterization of Performative Gap

In this section, we characterize the notion of the *performative gap*—i.e., the gap between the performatively stable equilibrium and the Stackelberg equilibrium. We also introduce a toy example to see the implications of this gap on the losses for the decision-maker.

**Proposition I.4** (Performative Equilibrium Gap.). *Under the assumptions of Theorem I.3, if $\ell(u, z)$ is $L_z$ Lipschitz continuous in $z$, then*

$$\|u^* - u^{\text{ps}}\| + \|x^*(u^*) - x^*(u^{\text{ps}})\| \leq (1 + L_{\text{eq}})\frac{L_z(L_{\text{eq}} + L_{\text{en}})}{\alpha - L_z(L_{\text{eq}} + L_{\text{en}})}.$$

*When the non-strategic decision-dependent component is stationary—i.e., $\mathcal{D}_e(u) \equiv \mathcal{D}_e$ for stationary distribution $\mathcal{D}_e$—the bound reduces to*

$$\|u^* - u^{\text{ps}}\| + \|x^*(u^*) - x^*(u^{\text{ps}})\| \leq (1 + L_{\text{eq}})\frac{L_z L_{\text{eq}}}{\alpha - L_z L_{\text{eq}}},$$

*and Assumption I.2 is no longer required.*

*Proof of Proposition I.4.* First observe that

$$\|u^* - u^{\text{ps}}\| + \|x^*(u^*) - x^*(u^{\text{ps}})\| \leq (1 + L_{\text{eq}})\|u^* - u^{\text{ps}}\|. \tag{17}$$

Recall that

$$G_{u'}(u) = \mathbb{E}_{\xi \sim \mathcal{D}_e(u')} \nabla_u \ell(u, (x^*(u'), \xi)).$$

Next, since $\ell$ is $\alpha$–strongly convex in $u$, we have that

$$
\begin{aligned}
\alpha\|u^* - u^{\mathrm{ps}}\|^2 &\leq \langle G_{u^{\mathrm{ps}}}(u^*) - G_{u^{\mathrm{ps}}}(u^{\mathrm{ps}}), u^* - u^{\mathrm{ps}} \rangle \\
&\leq \langle G_{u^{\mathrm{ps}}}(u^*), u^* - u^{\mathrm{ps}} \rangle \\
&\leq \langle G_{u^{\mathrm{ps}}}(u^*) - G_{u^*}(u^*) + G_{u^*}(u^*), u^* - u^{\mathrm{ps}} \rangle \\
&\leq \langle G_{u^*}(u^*), u^* - u^{\mathrm{ps}} \rangle + \| \underset{\xi \sim \mathcal{D}_e(u^{\mathrm{ps}})}{\mathbb{E}} \nabla_u \ell(u^*, (x^*(u^{\mathrm{ps}}), \xi)) - \underset{\xi \sim \mathcal{D}_e(u^*)}{\mathbb{E}} \nabla_u \ell(u^*, (x^*(u^*), \xi)) \| \|u^* - u^{\mathrm{ps}}\| \\
&\leq \langle G_{u^*}(u^*), u^* - u^{\mathrm{ps}} \rangle + L_z(L_{\mathrm{eq}} + L_{\mathrm{en}})\|u^* - u^{\mathrm{ps}}\|^2.
\end{aligned}
$$

Now, since $u^*$ is optimal for $\min_u \mathcal{L}(u)$, we have that

$$
0 \in G_{u^*}(u^*) + \underbrace{\frac{d}{dw}\left( \underset{\xi \sim \mathcal{D}_e(w)}{\mathbb{E}} \ell(u^*, (x^*(w), \xi)) \right)\Big|_{w=u^*}}_{:=Q} + N_{\mathcal{U}}(u^*).
$$

By Lipschitz conitnuity of $\ell$ in $z$ and of $x^*(\cdot)$, we have that $\|Q\| \leq L_z(L_{\mathrm{eq}} + L_{\mathrm{en}})$. Therefore, combining the above results, we have that

$$
\|u^* - u^{\mathrm{ps}}\| \leq \frac{L_z(L_{\mathrm{eq}} + L_{\mathrm{en}})}{\alpha - L_z(L_{\mathrm{eq}} + L_{\mathrm{en}})}
$$

so that

$$
\|u^* - u^{\mathrm{ps}}\| + \|x^*(u^*) - x^*(u^{\mathrm{ps}})\| \leq (1 + L_{\mathrm{eq}})\frac{L_z(L_{\mathrm{eq}} + L_{\mathrm{en}})}{\alpha - L_z(L_{\mathrm{eq}} + L_{\mathrm{en}})},
$$

as claimed. □

**Informative Toy Example: Implications of the Performative Gap in Quadratic Games.** Figure 2 in the main paper explores the sample complexity and performative gap tradeoff in terms of the equilibrium strategies. Indeed, if the reactivity of the agents is small and the dependence of the cost for the decision-maker on the agents is also "small", then the performatively stable equilibrium will be near the Stackelberg equilibrium. It is also natural to ask what the implications are for the utility of the decision-maker. Here we provide a concrete example of Stackelberg games where the gap in utility between the Stackelberg equilibrium and performative stable equilibrium is large or small.

Consider a quadratic game with a linear "tariff" $\phi_i(x, u)$:

$$
f_1(x_i, x_{-i}) = \frac{1}{2}x_i^2 + b_i x_i x_{-i} + c_i x_i + \phi_i(x, u) \quad \text{and} \quad \phi_i(x, u) = u_i \quad \text{for } i = 1, 2
$$

And let the decision-maker cost is something simple like

$$
\ell(x, u) = \|u - u^d\|^2 + \|x^*(u) - x^d\|^2
$$

for some "desired" $(u^d, x^d)$. Supposing the game constants are such that the game Jacobian is invertible, the Nash is given by

$$
\omega(x, u) = (x_1 + b_1 x_2 + c_1, x_2 + b_2 x_1 + c_2) = (0, 0) \implies x^* = -\begin{bmatrix} 1 & b_1 \\ b_2 & 1 \end{bmatrix}^{-1} \begin{bmatrix} c_1 \\ c_2 \end{bmatrix}
$$

Then no matter what $u$ is the Nash equilibrium stays the same. And, we have that $u^{\mathrm{ps}} = u^* = u^d$ since $x^*(u)$ is constant. So not only are the performatively stable equilibrium and Stackelberg equilibrium "close" but the change in decision-maker cost is also small (zero in this case).

Now, consider a quadratic game with a slightly different linear "tariff": $\phi_i(x, u) = u_i x_i + c_i$. Thus the Nash equilibrium is given by

$$
\omega(x, u) = (x_1 + b_1 x_2 + u_1, x_2 + b_2 x_1 + u_2) = (0, 0) \implies x^*(u) = -\underbrace{\begin{bmatrix} 1 & b_1 \\ b_2 & 1 \end{bmatrix}^{-1}}_{:=A} \begin{bmatrix} u_1 \\ u_2 \end{bmatrix}
$$

Hence, if

$$
\|A\| \leq \frac{1}{b_1 b_2 - 1} \max\{1 - \sqrt{b_1 b_2}, 1 + \sqrt{b_1 b_2}\}
$$

is small $x^*(u)$ does not change much if $u$ does not change much. We again have that $u^{\text{ps}} = u^d$ for the performatively stable equilibrium and for the Stackelberg equilibrium we have that

$$2(u - u^d) + 2A^\top(Au - x^d) = 0 \implies u^* = (I + A^\top A)^{-1}(u^d + A^\top x^d)$$

so that $u^* \neq u^{\text{ps}}$. Then $\ell(x^{\text{ps}}, u^{\text{ps}}) = \|Au^d - x^d\|^2$ and

$$\ell(x^*(u^*), u^*) = \|(I + A^\top A)^{-1}(u^d + A^\top x^d) - u^d\|^2 + \|A(I + A^\top A)^{-1}(u^d + A^\top x^d) - x^d\|^2.$$

We know that $\ell(x^{\text{ps}}, u^{\text{ps}}) \geq \ell(x^*(u^*), u^*)$ just from the basics of optimization. Depending on the size of $A$ (i.e., the size of the agent reactivity $L_{\text{eq}} = \|A\|$) the losses can be very close or very far apart. So in this case $A$ determines both the closeness of the equilibrium and the differences in losses.

If the decision-maker places more or less emphasis on the $u$ term via a weighting term $\lambda$, then that can cause the decision-makers utility to change more significantly. For example, consider $(u^d, x^d) = ((1, 1), (1, 1))$ for simplicity. And let $b_1 = b_2 = 0.5$ and $\ell_\lambda(x, u) = \lambda\|u - u^d\|^2 + \|x - x^d\|^2$. Then $u^{\text{ps}} = (1, 1)$. Then (relatively speaking) we have a small change in both $\ell$ and $x^*$ when $\lambda = 1$:

$$\ell_1(x^{\text{ps}}, u^{\text{ps}}) - \ell_1(x^*, u^*) = 1.71 \quad \text{and} \quad \|x^*(u^{\text{ps}}) - x^*(u^*)\|^2 = 0.72.$$

On the other hand, we have a large change in $\ell$ with a small change in $x^*$ when $\lambda = 5$:

$$\ell_5(x^{\text{ps}}, u^{\text{ps}}) - \ell_5(x^*, u^*) = 22.67 \quad \text{and} \quad \|x^*(u^{\text{ps}}) - x^*(u^*)\|^2 = 0.19.$$

Thus, even in simple quadratic settings it is possible to get a variety of outcomes in terms of the decision-maker's loss depending on the reactivity of the agents' ($L_{\text{eq}}$) and how reactive the decision-maker's loss is to changes in the agents' behavior ($L_z$).

### I.3. Naïve Decision-Maker: Stationary Non-strategic Environment

Given the preceding technical lemma, we know prove Theorem 4.4. Let's us restate it more formally.

**Theorem I.5** (Formal Statement of Theorem 4.4). *Suppose that Assumptions 2.1, 4.1, and 4.3 hold, that we have available constants $R > \|x_{-1} - x^*(u_0)\|^2$ and $B > \|u_0 - u^{\text{ps}}\|^2$, and that we are in the regime where $\alpha > L_z L_{\text{eq}}$ so that there is a unique performatively stable equilibrium. Further, suppose the decision-maker runs Algorithm 4 with* `Alg := RGM` *using step-size $\eta \leq \frac{\bar{\alpha}}{4L_\ell^2(1+L_{\text{eq}}^2)}$ where $\bar{\alpha} := \alpha - L_z L_{\text{eq}}$, and the agents employ a $\rho$–contracting algorithm $\mathcal{A}$ with $\rho \in [0, 1)$ and $\sigma_{\text{a}} \in (0, \infty)$. Suppose the agents run their $\rho$-contracting algorithm stage-wise via Algorithm 2. In this case, set the epoch length to*

$$\tau = \sum_{k=0}^{K} T_k = \left\lceil \frac{1}{1 - \rho^2} \cdot \log\left(\frac{2\bar{R}}{\epsilon_\tau}\right) \right\rceil + \sum_{k=1}^{K} \left\lceil \left(\frac{1}{1 - 2^{-k}\rho^2}\right) \log(4) \right\rceil, \tag{18}$$

*and tolerance $\epsilon_\tau = \eta^2\sigma^2$ where $K = \left\lceil 1 + \log_2\left(\frac{\rho^2 c^2 \sigma_{\text{a}}^2}{(1-\rho^2)\epsilon_\tau}\right) \right\rceil$ and*

$$\bar{R} := R + \frac{2c^2\sigma_{\text{a}}^2}{1 - \rho^2} + 6\left(\frac{L_{\text{eq}}^2}{(1 - \rho^2)^2}\left(4B + \frac{4\sigma^2}{L_\ell^2(1 + L_{\text{eq}}^2)}\right)\right), \tag{19}$$

*where $\beta = (1 - \rho^2)$. Then, the following estimate holds:*

$$\mathbb{E}_t\|u_{t+1} - u^{\text{ps}}\|^2 \leq \left(1 - \frac{\bar{\alpha}\eta}{2}\right)^{t+1}\|u_0 - u^{\text{ps}}\|^2 + \frac{4\eta\sigma^2}{\bar{\alpha}}.$$

Recall from Corollary G.5, that if the agents run stochastic gradient play in stages then we are able to characterize precisely the number of iterations required to hit a particular specified error tolerance. This is where the epoch length in (41) is derived.

*Proof of Theorem I.5.* Define the following objects:

$$g_t := \nabla_u \ell(u_t, (\mathcal{A}(x_{t-1}, u_t), \xi)), \text{ where } \xi \sim \mathcal{D}_e;$$
$$G_t(u_t) := \mathop{\mathbb{E}}_{\xi \sim \mathcal{D}_e} \nabla_u \ell(u_t, (\mathcal{A}(x_{t-1}, u_t), \xi));$$
$$G_\star(u_t) := \mathop{\mathbb{E}}_{\xi \sim \mathcal{D}_e} \nabla_u \ell(u_t, (x^*(u_t), \xi)); \tag{20}$$
$$G_{\mathrm{ps}}(u_t) := \mathop{\mathbb{E}}_{\xi \sim \mathcal{D}_e} \nabla_u \ell(u_t, (x^*(u^{\mathrm{ps}}), \xi)).$$

Also note that $\mathbb{E}_t[g_t] = G_t(u_t)$—i.e., the gradient estimate $g_t$ is an unbiased estimate of the time varying expected gradient $G_t$—and

$$u^{\mathrm{ps}} = \operatorname*{argmin}_{u \in \mathcal{U}} \mathop{\mathbb{E}}_{z \sim \mathcal{D}(u^{\mathrm{ps}})} \ell(u, z) \quad \text{so that} \quad \langle G_{\mathrm{ps}}(u^{\mathrm{ps}}), u - u^{\mathrm{ps}} \rangle \geq 0 \ \forall \ u \in \mathcal{U}.$$

Fix two constants $\nu_1, \nu_2 > 0$ to be specified later. Noting that $u_{t+1}$ is the minimizer of the 1-strongly convex function $u \mapsto \frac{1}{2}\|u_t - \eta g_t - u\|^2$ over $\mathcal{U}$, we deduce that

$$\frac{1}{2}\|u_{t+1} - u^{\mathrm{ps}}\|^2 \leq \frac{1}{2}\|u_t - \eta g_t - u^{\mathrm{ps}}\|^2 - \frac{1}{2}\|u_t - \eta_t g_t - u_{t+1}\|^2.$$

Expanding the squares on the right hand side and combining terms yields

$$\frac{1}{2}\|u_{t+1} - u^{\mathrm{ps}}\|^2 \leq \frac{1}{2}\|u_t - u^{\mathrm{ps}}\|^2 - \eta_t \langle g_t, u_{t+1} - u^{\mathrm{ps}} \rangle - \frac{1}{2}\|u_{t+1} - u_t\|^2$$
$$= \frac{1}{2}\|u_t - u^{\mathrm{ps}}\|^2 - \eta \langle g_t, u_t - u^{\mathrm{ps}} \rangle - \frac{1}{2}\|u_{t+1} - u_t\|^2 - \eta \langle g_t, u_{t+1} - u_t \rangle.$$

Using the fact that $\mathbb{E}_t[g_t] = G_t(u_t)$, we successively compute

$$\frac{1}{2}\mathbb{E}_t\|u_{t+1} - u^{\mathrm{ps}}\|^2 \leq \frac{1}{2}\|u_t - u^{\mathrm{ps}}\|^2 - \eta \langle \mathbb{E}_t g_t, u_t - u^{\mathrm{ps}} \rangle - \frac{1}{2}\mathbb{E}_t\|u_{t+1} - u_t\|^2 - \eta \mathbb{E}_t \langle g_t, u_{t+1} - u_t \rangle,$$
$$\leq \frac{1}{2}\|u_t - u^{\mathrm{ps}}\|^2 - \eta \langle G_t(u_t), u_t - u^{\mathrm{ps}} \rangle - \frac{1}{2}\mathbb{E}_t\|u_{t+1} - u_t\|^2 - \eta \mathbb{E}_t \langle g_t, u_{t+1} - u_t \rangle,$$
$$= \frac{1}{2}\|u_t - u^{\mathrm{ps}}\|^2 - \eta \mathbb{E}_t \langle G_\star(u_{t+1}), u_{t+1} - u^{\mathrm{ps}} \rangle - \frac{1}{2}\mathbb{E}_t\|u_{t+1} - u_t\|^2$$
$$+ \eta \underbrace{\mathbb{E}_t \langle g_t - G_t(u_t), u_t - u_{t+1} \rangle}_{P_1} + \eta_t \underbrace{\mathbb{E}_t \langle G_t(u_t) - G_\star(u_{t+1}), u^{\mathrm{ps}} - u_{t+1} \rangle}_{P_2}.$$

Recall that for any $z$, the loss $\ell(u, z)$ is $\alpha$–strongly convex in $u$ so that

$$\langle G_{\mathrm{ps}}(u_{t+1}), u_{t+1} - u^{\mathrm{ps}} \rangle \geq \langle G_{\mathrm{ps}}(u_{t+1}) - G_{\mathrm{ps}}(u^{\mathrm{ps}}), u_{t+1} - u^{\mathrm{ps}} \rangle \geq \alpha\|u_{t+1} - u^{\mathrm{ps}}\|^2.$$

Therefore, adding and subtracting $G_{\mathrm{ps}}(u_{t+1})$, we have that

$$\mathbb{E}_t \langle G_\star(u_{t+1}), u_{t+1} - u^{\mathrm{ps}} \rangle = \mathbb{E}_t \langle G_{\mathrm{ps}}(u_{t+1}), u_{t+1} - u^{\mathrm{ps}} \rangle + \mathbb{E}_t \langle G_\star(u_{t+1}) - G_{\mathrm{ps}}(u_{t+1}), u_{t+1} - u^{\mathrm{ps}} \rangle$$

Now, the second term is upper bounded as follows:

$$\mathbb{E}_t \langle G_\star(u_{t+1}) - G_{\mathrm{ps}}(u_{t+1}), u_{t+1} - u^{\mathrm{ps}} \rangle \leq L_z L_{\mathrm{eq}}\|u_{t+1} - u^{\mathrm{ps}}\|^2.$$

Then, rearranging the above expression, we have that

$$\frac{1 + 2\eta\bar{\alpha}}{2}\mathbb{E}_t\|u_{t+1} - u^{\mathrm{ps}}\|^2 \leq \frac{1}{2}\|u_t - u^{\mathrm{ps}}\|^2 - \frac{1}{2}\mathbb{E}_t\|u_{t+1} - u_t\|^2 + \eta(P_1 + P_2). \tag{21}$$

Applying Young's inequality to $P_1$, we have that

$$P_1 \leq \frac{\mathbb{E}_t\|g_t - G_t(u_t)\|^2}{2\nu_1} + \frac{\nu_1 \mathbb{E}_t\|u_{t+1} - u_t\|^2}{2} \leq \frac{\sigma^2}{2\nu_1} + \frac{\nu_1 \mathbb{E}_t\|u_{t+1} - u_t\|^2}{2}. \tag{22}$$

We have the following upper bound for $P_2$:

$$P_2 \leq \frac{\mathbb{E}_t \|G_t(u_t) - G_\star(u_{t+1})\|^2}{2\nu_2} + \frac{\nu_2 \mathbb{E}_t \|u^{\mathrm{ps}} - u_{t+1}\|^2}{2}$$

$$\leq \frac{2\mathbb{E}_t \|G_t(u_t) - G_\star(u_t)\|^2 + 2\mathbb{E}_t \|G_\star(u_t) - G_\star(u_{t+1})\|^2}{2\nu_2} + \frac{\nu_2 \mathbb{E}_t \|u^{\mathrm{ps}} - u_{t+1}\|^2}{2}$$

$$\leq \frac{2\mathbb{E}_t \|G_t(u_t) - G_\star(u_t)\|^2 + 2L_\ell^2(1 + L_{\mathrm{eq}}^2)\|u_t - u_{t+1}\|^2}{2\nu_2} + \frac{\nu_2 \mathbb{E}_t \|u^{\mathrm{ps}} - u_{t+1}\|^2}{2}.$$

The first term in the first fraction can be bounded as follows:

$$\mathbb{E}_t \|G_t(u_t) - G_\star(u_t)\|^2 = \mathbb{E}_t \| \mathop{\mathbb{E}}_{\xi \sim \mathcal{D}_e} \nabla_u \ell(u_t, (\mathcal{A}(x_{t-1}, u_t), \xi)) - \mathop{\mathbb{E}}_{\xi \sim \mathcal{D}_e} \nabla_u \ell(u_t, (x^*(u_t), \xi))\|^2$$

$$\leq L_\ell^2 \mathbb{E}_t \|\mathcal{A}(x_{t-1}, u_t) - x^*(u_t)\|^2. \tag{23}$$

This shows we have a time varying bias component in our gradient estimator. Here, we aim to show that $\mathbb{E}_t \|\mathcal{A}(x_{t-1}, u_t) - x^*(u_t)\|^2 \leq \epsilon_\tau = \eta^2 \sigma^2$ where in each epoch agents are running a $\rho$ contracting algorithms for $\tau$ steps. In order to obtain the $\epsilon_\tau$ target accuracy bound in epoch $t$, we need that the agents' initial condition to be bounded at the start of this epoch. To obtain a bound on the initial condition in expectation we need to perform an inductive argument to show that $\mathbb{E}[\|x_{t-1} - x^*(u_t)\|^2] \leq \bar{R}$ where $\bar{R}$ is defined in (19). Let us suppose for the time that this bound holds for each $t$.

Recall that

$$\tau = \sum_{k=0}^{K} T_k = \left\lceil \frac{1}{1 - \rho^2} \cdot \log\left(\frac{2\bar{R}}{\epsilon_\tau}\right) \right\rceil + \sum_{k=1}^{K} \left\lceil \left(\frac{1}{1 - 2^{-k}\rho^2}\right) \log(4) \right\rceil$$

total iterations where $K := \left\lceil 1 + \log_2\left(\frac{\rho^2 c^2 \sigma_a^2}{(1-\rho^2)\epsilon_\tau}\right) \right\rceil$. In this case, we deduce that

$$P_2 \leq \frac{L_\ell^2 \epsilon_\tau}{\nu_2} + \frac{2L_\ell^2(1 + L_{\mathrm{eq}}^2)\|u_t - u_{t+1}\|^2}{2\nu_2} + \frac{\nu_2 \mathbb{E}_t \|u^{\mathrm{ps}} - u_{t+1}\|^2}{2}.$$

Coming back to the bound in (37), we have that

$$\frac{1 + 2\eta\bar{\alpha}}{2} \mathbb{E}_t \|u_{t+1} - u^{\mathrm{ps}}\|^2 \leq \frac{1}{2}\|u_t - u^{\mathrm{ps}}\|^2 - \frac{1}{2}\mathbb{E}_t \|u_{t+1} - u_t\|^2 + \eta\left(\frac{\sigma^2}{2\nu_1} + \frac{\nu_1 \mathbb{E}_t \|u_{t+1} - u_t\|^2}{2}\right)$$

$$+ \eta\left(\frac{L_\ell^2 \epsilon_\tau}{\nu_2} + \frac{2L_\ell^2(1 + L_{\mathrm{eq}}^2)\|u_t - u_{t+1}\|^2}{2\nu_2} + \frac{\nu_2 \mathbb{E}_t \|u^{\mathrm{ps}} - u_{t+1}\|^2}{2}\right)$$

so that

$$\frac{1 + 2\eta\bar{\alpha} - \eta\nu_2}{2} \mathbb{E}_t \|u_{t+1} - u^{\mathrm{ps}}\|^2 \leq \frac{1}{2}\|u_t - u^{\mathrm{ps}}\|^2 + \frac{\eta\sigma^2}{2\nu_1} + \eta\frac{L_\ell^2 \epsilon_\tau}{\nu_2} - \frac{1 - 2L_\ell^2(1 + L_{\mathrm{eq}}^2)\eta\nu_2^{-1} - \eta\nu_1}{2} \mathbb{E}_t \|u_{t+1} - u_t\|^2.$$

Letting $\nu_1 = \eta^{-1} - \frac{2L_\ell^2(1 + L_{\mathrm{eq}}^2)}{\bar{\alpha}}$ and $\nu_2 = \bar{\alpha}$ ensures that the last term on the right is zero. By our assumption that $\eta \leq \frac{\bar{\alpha}}{4L_\ell^2(1 + L_{\mathrm{eq}}^2)}$ we have that $\frac{1}{\eta} \geq \frac{4L_\ell^2(1 + L_{\mathrm{eq}}^2)}{\bar{\alpha}}$ so that $\nu_1 \geq \frac{1}{2\eta}$; indeed, we claim that

$$\frac{1}{\eta} - \frac{2L_\ell^2(1 + L_{\mathrm{eq}}^2)}{\bar{\alpha}} \geq \frac{1}{2\eta}.$$

Rearranging, this is equivalent to showing that

$$1 - \eta\frac{2L_\ell^2(1 + L_{\mathrm{eq}}^2)}{\bar{\alpha}} \geq \frac{1}{2}.$$

Now we can lower bound the left-hand side as follows:

$$1 - \eta\frac{2L_\ell^2(1 + L_{\mathrm{eq}}^2)}{\bar{\alpha}} \geq 1 - \frac{\bar{\alpha}}{4L_\ell^2(1 + L_{\mathrm{eq}}^2)}\frac{2L_\ell^2(1 + L_{\mathrm{eq}}^2)}{\bar{\alpha}} = 1 - \frac{1}{2} = \frac{1}{2}.$$

That shows that claim. Hence we have that

$$\frac{1 + \eta\bar{\alpha}}{2}\mathbb{E}_t\|u_{t+1} - u^{\text{ps}}\|^2 \le \frac{1}{2}\|u_t - u^{\text{ps}}\|^2 + \eta^2\sigma^2 + \frac{\epsilon_\tau}{4(1 + L_{\text{eq}}^2)}$$

$$\le \frac{1}{2}\|u_t - u^{\text{ps}}\|^2 + \left(1 + \frac{1}{4(1 + L_{\text{eq}}^2)}\right)\eta^2\sigma^2,$$

$$\le \frac{1}{2}\|u_t - u^{\text{ps}}\|^2 + 2\eta^2\sigma^2,$$

where the second to last inequality holds since $\epsilon_\tau = \eta^2\sigma^2$. Thus, we have that

$$\mathbb{E}_t\|u_{t+1} - u^{\text{ps}}\|^2 \le \frac{1}{1 + \eta\bar{\alpha}}\|u_t - u^{\text{ps}}\|^2 + \frac{4}{1 + \eta\bar{\alpha}}\eta^2\sigma^2$$

Recursively iterating the above expression, we have that

$$\mathbb{E}_t\|u_{t+1} - u^{\text{ps}}\|^2 \le \frac{1}{1 + \eta\bar{\alpha}}\left(\frac{1}{1 + \eta\bar{\alpha}}(\|u_{t-1} - u^{\text{ps}}\|^2 + \frac{4}{1 + \eta\bar{\alpha}}\eta^2\sigma^2) + \frac{4}{1 + \eta\bar{\alpha}}\eta^2\sigma^2\right)$$

$$\le \left(\frac{1}{1 + \eta\bar{\alpha}}\right)^{t+1}\|u_0 - u^{\text{ps}}\|^2 + 4\eta^2\sigma^2\sum_{s=1}^{t+1}\left(\frac{1}{1 + \eta\bar{\alpha}}\right)^s$$

$$\le \left(\frac{1}{1 + \eta\bar{\alpha}}\right)^{t+1}\|u_0 - u^{\text{ps}}\|^2 + 4\eta^2\sigma^2\frac{1}{\eta\bar{\alpha}}$$

Given the choice of $\eta \le \frac{\bar{\alpha}}{4L_\ell^2}$, we have that

$$\mathbb{E}_t\|u_{t+1} - u^{\text{ps}}\|^2 \le \left(1 - \frac{\bar{\alpha}\eta}{2}\right)^{t+1}\|u_0 - u^{\text{ps}}\|^2 + \frac{4\eta\sigma^2}{\bar{\alpha}}.$$

**Bounding the initial condition via constructing the constant $\bar{R}$.** Recall from (29), that for each epoch $t$, the error decomposition for the decision maker contains a term $\mathbb{E}\|\mathcal{A}(x_{t-1}, u_t) - x^*(u_t)\|^2$. This means that we need a bound on the per epoch tracking error which as noted above depends on the initial condition being bounded. Let us argue by induction that the value of $\bar{R}$ as defined in (19) is such that $\mathbb{E}[\|x_{t-1} - x^*(u_t)\|^2] \le \bar{R}$ as long as $\|x_{-1} - x^*(u_0)\|^2 \le R$.

Consider $\|u_0 - u^{\text{ps}}\|^2 \le B$ and $\|x_{-1} - x^*(u_0)\|^2 \le R$. Set $\epsilon_\tau = \eta^2\sigma^2$ throughout. We will construct a sequence $\bar{R}_t$ that determines the epoch length

$$\tau_t = \left\lceil \frac{1}{1 - \rho^2} \cdot \log\left(\frac{2\bar{R}_t}{\epsilon_\tau}\right) \right\rceil + \sum_{k=1}^{K}\left\lceil\left(\frac{1}{1 - 2^{-k}\rho^2}\right)\log(4)\right\rceil$$

needed in order to hit $\epsilon_\tau$ target accuracy given the per-epoch initial condition bound $\mathbb{E}\|x_{t-1} - x^*(u_t)\|^2 \le \bar{R}_t$. Our goal is to determine $\bar{R}_t$ inductively and then show that there is in fact $\bar{R}_t \equiv \bar{R}$, i.e. an absolute bound based on problem constants.

***Base Case.*** Starting with $t = 0$, the aim is to choose $(\bar{R}_0, \tau_0)$ such that $\mathbb{E}\|\mathcal{A}(x_{-1}, u_0) - x^*(u_0)\|^2 = \mathbb{E}\|x_0 - x^*(u_0)\|^2 \le \epsilon_\tau$. Indeed, given that the agents run a $\rho$-contracting algorithm in stages for $\tau_0$ total iterations where

$$\tau_0 = \left\lceil \frac{1}{1 - \rho^2} \cdot \log\left(\frac{2R}{\epsilon_\tau}\right) \right\rceil + \sum_{k=1}^{K}\left\lceil\left(\frac{1}{1 - 2^{-k}\rho^2}\right)\log(4)\right\rceil$$

with $\bar{R}_0 = R$ we have that

$$\mathbb{E}\|\mathcal{A}(x_{-1}, u_0) - x^*(u_0)\|^2 = \mathbb{E}\|x_0 - x^*(u_0)\|^2 \le \epsilon_\tau.$$

***Warm-up to Inductive Step.*** Let us examine the expected tracking error for the decision maker. For $t = 1$, we have that

$$\frac{1 + \eta\bar{\alpha}}{2}\mathbb{E}\|u_1 - u^{\text{ps}}\|^2 \le \frac{1}{2}\|u_0 - u^{\text{ps}}\|^2 + \eta^2\sigma^2 + \frac{\mathbb{E}\|x_0 - x^*(u_0)\|^2}{4(1 + L_{\text{eq}}^2)}$$

$$\le \frac{1}{2}\|u_0 - u^{\text{ps}}\|^2 + \eta^2\sigma^2 + \frac{\epsilon_\tau}{4(1 + L_{\text{eq}}^2)}$$

$$\le \frac{1}{2}\|u_0 - u^{\text{ps}}\|^2 + \left(1 + \frac{1}{4(1 + L_{\text{eq}}^2)}\right)\eta^2\sigma^2,$$

$$\le \frac{1}{2}\|u_0 - u^{\text{ps}}\|^2 + 2\eta^2\sigma^2,$$

which implies that

$$\mathbb{E}[\|u_1 - u^{\text{ps}}\|^2] \le \frac{1}{1 + \eta\bar{\alpha}}\|u_0 - u^{\text{ps}}\|^2 + \frac{\eta^2\sigma^2}{1 + \eta\bar{\alpha}}. \tag{24}$$

More generally, from the above analysis, we have that

$$\frac{1 + \eta\bar{\alpha}}{2}\mathbb{E}\|u_{t+1} - u^{\text{ps}}\|^2 \le \frac{1}{2}\mathbb{E}\|u_t - u^{\text{ps}}\|^2 + \eta^2\sigma^2 + \frac{\mathbb{E}\|x_t - x^*(u_t)\|^2}{4(1 + L_{\text{eq}}^2)}.$$

For $t = 2$, if $\mathbb{E}[\|x_1 - x^*(u_1)\|^2|\mathcal{E}_1] \le \epsilon_\tau$ with $\mathcal{E}_1 = \{\|x_0 - x^*(u_1)\|^2 \le \bar{R}_1\}$, we have that

$$\frac{1 + \eta\bar{\alpha}}{2}\mathbb{E}\|u_2 - u^{\text{ps}}\|^2 \le \frac{1}{2}\mathbb{E}\|u_1 - u^{\text{ps}}\|^2 + \eta^2\sigma^2 + \frac{\mathbb{E}\|x_1 - x^*(u_1)\|^2}{4(1 + L_{\text{eq}}^2)}$$

$$\le \frac{1}{2}\mathbb{E}\|u_1 - u^{\text{ps}}\|^2 + \left(1 + \frac{1}{4(1 + L_{\text{eq}}^2)}\right)\eta^2\sigma^2,$$

$$\le \frac{1}{2}\mathbb{E}\|u_1 - u^{\text{ps}}\|^2 + 2\eta^2\sigma^2,$$

Hence, we need to select $(\bar{R}_1, \tau_1)$ such that this holds. Here we appeal to the drift-to-noise analysis in Section 3. Shifting indices in that analysis as appropriate for this setting, we have that

$$\mathbb{E}\|x_0 - x^*(u_1)\|^2 \le \left(1 - \frac{1 - \rho^2}{2}\right)\|x_{-1} - x^*(u_0)\|^2 + \frac{2c^2\sigma_{\text{a}}^2}{1 - \rho^2} + 6\left(\frac{L_{\text{eq}}^2\Delta_1^2}{(1 - \rho^2)^2}\right)$$

$$\le \left(1 - \frac{1 - \rho^2}{2}\right)R + \frac{2c^2\sigma_{\text{a}}^2}{1 - \rho^2} + 6\left(\frac{L_{\text{eq}}^2\Delta_1^2}{(1 - \rho^2)^2}\right)$$

$$\le R + \frac{2c^2\sigma_{\text{a}}^2}{1 - \rho^2} + 6\left(\frac{L_{\text{eq}}^2\Delta_1^2}{(1 - \rho^2)^2}\right)$$

where $\Delta_1^2 := \mathbb{E}_1\|u_1 - u_0\|^2$ and $x_{-1}$ is the given initial joint action profile of the agents. Using (24), we have that

$$\mathbb{E}\|u_1 - u_0\|^2 \le 2\|u_0 - u^{\text{ps}}\|^2 + 2\mathbb{E}\|u_1 - u^{\text{ps}}\|^2 \le 2B + 2\left(B + \frac{4\eta^2\sigma^2}{1 + \eta\bar{\alpha}}\right)$$

so that it suffices to set

$$\bar{R}_1 = R + \frac{2c^2\sigma_{\text{a}}^2}{1 - \rho^2} + 6\left(\frac{L_{\text{eq}}^2}{(1 - \rho^2)^2}\left(4B + 16\frac{\eta\sigma^2}{\bar{\alpha}}\right)\right) \ge R + \frac{2c^2\sigma_{\text{a}}^2}{1 - \rho^2} + 6\left(\frac{L_{\text{eq}}^2}{(1 - \rho^2)^2}\left(4B + 8\frac{\eta\sigma^2}{\bar{\alpha}}\right)\right),$$

where we upper bounded $2\eta^2\sigma^2/(1 + \eta\bar{\alpha}) \le 2\eta^2\sigma^2\sum_{s=1}^\infty 1/(1 + \eta\bar{\alpha})^s \le 2\frac{\eta\sigma^2}{\bar{\alpha}}$ and also multiplied the variance term by two (the reason for which will be come clear shortly). Moreover, note that $\eta \le \bar{\alpha}/(4(L_\ell(1 + L_{\text{eq}}^2))$ and $\bar{R}_1 \ge \bar{R}_0 = R$.

For $t = 3$, we have that

$$\frac{1 + \eta\bar{\alpha}}{2}\mathbb{E}\|u_3 - u^{\text{ps}}\|^2 \le \frac{1}{2}\mathbb{E}\|u_2 - u^{\text{ps}}\|^2 + \eta^2\sigma^2 + \frac{\mathbb{E}\|x_2 - x^*(u_2)\|^2}{4(1 + L_{\text{eq}}^2)}$$

and

$$\mathbb{E}\|x_1 - x^*(u_2)\|^2 \le \left(1 - \frac{1-\rho^2}{2}\right)\|x_{-1} - x^*(u_0)\|^2 + \frac{2c^2\sigma_{\mathsf{a}}^2}{1-\rho^2} + 6\left(\frac{L_{\mathsf{eq}}^2 \max_{k\le 2}\Delta_k^2}{(1-\rho^2)^2}\right)$$

$$\le R + \frac{2c^2\sigma_{\mathsf{a}}^2}{1-\rho^2} + 6\left(\frac{L_{\mathsf{eq}}^2 \max_{k\le 2}\Delta_k^2}{(1-\rho^2)^2}\right), \quad \text{where } \Delta_k^2 = \mathbb{E}\|u_k - u_{k-1}\|^2.$$

This means we need a bound on $\Delta_k^2$ for each $k \le t-1$. Let's examine the $t=3$ case in which we have

$$\mathbb{E}\|u_2 - u_1\|^2 \le 2\,\mathbb{E}\|u_2 - u^{\mathsf{ps}}\|^2 + 2\,\mathbb{E}\|u_1 - u^{\mathsf{ps}}\|^2$$

$$\le 2\left(\frac{1}{1+\eta\bar{\alpha}}\mathbb{E}\|u_1 - u^{\mathsf{ps}}\|^2 + \frac{4}{1+\eta\bar{\alpha}}\eta^2\sigma^2\right) + \frac{2}{1+\eta\bar{\alpha}}\|u_0 - u^{\mathsf{ps}}\|^2 + \frac{2\cdot 4\eta^2\sigma^2}{1+\eta\bar{\alpha}}$$

$$\le 2\left(\frac{1}{1+\eta\bar{\alpha}}\left(\frac{1}{1+\eta\bar{\alpha}}\|u_0 - u^{\mathsf{ps}}\|^2 + \frac{4\eta^2\sigma^2}{1+\eta\bar{\alpha}}\right) + \frac{4}{1+\eta\bar{\alpha}}\eta^2\sigma^2\right) + 2\left(\frac{1}{1+\eta\bar{\alpha}}\|u_0 - u^{\mathsf{ps}}\|^2 + \frac{4\eta^2\sigma^2}{1+\eta\bar{\alpha}}\right)$$

$$= 2B\sum_{s=1}^2 \frac{1}{(1+\bar{\alpha}\eta)^s} + 8\eta^2\sigma^2\sum_{s=1}^2 \frac{1}{(1+\bar{\alpha}\eta)^s} + 2\frac{4\eta^2\sigma^2}{(1+\bar{\alpha}\eta)}$$

$$\le 4B + \frac{8\eta\sigma^2}{\bar{\alpha}} + \frac{8\eta^2\sigma^2}{(1+\bar{\alpha}\eta)}$$

$$\le 4B + \frac{16\eta\sigma^2}{\bar{\alpha}}$$

where the second inequality holds by (24) (since we showed that $\mathbb{E}\|x_0 - x^*(u_0)\|^2 \le \epsilon_\tau = \eta^2\sigma^2$) and $\bar{\alpha}\eta \le \bar{\alpha}^2/(4L_\ell^2)$.

For good measure, let us consider $t=3$. Here, we need to bound $\max_{k\le 3}\mathbb{E}\|u_k - u_{k-1}\|^2$ which in turn means we need to bound the following term:

$$\mathbb{E}\|u_3 - u_2\|^2 \le 2\,\mathbb{E}\|u_3 - u^{\mathsf{ps}}\|^2 + 2\,\mathbb{E}\|u_2 - u^{\mathsf{ps}}\|^2$$

$$\le 2\left(\frac{1}{1+\eta\bar{\alpha}}\mathbb{E}\|u_2 - u^{\mathsf{ps}}\|^2 + \frac{4\eta^2\sigma^2}{1+\eta\bar{\alpha}}\right) + 2\,\mathbb{E}\|u_2 - u^{\mathsf{ps}}\|^2$$

$$\le \frac{2}{1+\eta\bar{\alpha}}\left(\frac{1}{1+\eta\bar{\alpha}}\mathbb{E}\|u_1 - u^{\mathsf{ps}}\|^2 + \frac{4\eta^2\sigma^2}{1+\eta\bar{\alpha}}\right) + \frac{2\eta^2\sigma^2}{1+\eta\bar{\alpha}} + \frac{2}{1+\eta\bar{\alpha}}\mathbb{E}\|u_1 - u^{\mathsf{ps}}\|^2 + \frac{2\cdot 4\eta^2\sigma^2}{1+\eta\bar{\alpha}}$$

$$\le 2\left(\frac{1}{1+\eta\bar{\alpha}}\right)^3\|u_0 - u^{\mathsf{ps}}\|^2 + 2\cdot 4\sum_{s=1}^3 \frac{\eta^2\sigma^2}{(1+\eta\bar{\alpha})^s} + 2\left(\frac{1}{1+\eta\bar{\alpha}}\right)^2\|u_0 - u^{\mathsf{ps}}\|^2 + 2\cdot 4\sum_{s=1}^2 \frac{\eta^2\sigma^2}{(1+\eta\bar{\alpha})^s}$$

$$\le 4B + 16\frac{\eta\sigma^2}{\bar{\alpha}}$$

so that we set $\bar{R}_2 = R + \frac{2c^2\sigma_{\mathsf{a}}^2}{1-\rho^2} + 6\left(\frac{L_{\mathsf{eq}}^2}{(1-\rho^2)^2}\left(4B + 16\frac{\eta\sigma^2}{\bar{\alpha}}\right)\right) = \bar{R}_1$. We claim at this point that $\bar{R}_t \equiv \bar{R}$ for all $t$, and we argue this claim holds via induction.

***Induction Step.*** For any $t$, to obtain a bound on the $\mathbb{E}\|x_{t-1} - x^*(u_t)\|^2$, we simply observe that

$$\mathbb{E}\|u_t - u_{t-1}\|^2 \le 2\left(\frac{1}{1+\eta\bar{\alpha}}\right)^t\|u_0 - u^{\mathsf{ps}}\|^2 + 2\sum_{s=1}^t \frac{4\eta^2\sigma^2}{(1+\eta\bar{\alpha})^s} + 2\left(\frac{1}{1+\eta\bar{\alpha}}\right)^{t-1}\|u_0 - u^{\mathsf{ps}}\|^2 + 2\sum_{s=1}^{t-1}\frac{4\eta^2\sigma^2}{(1+\eta\bar{\alpha})^s}$$

$$\le 4B + 16\frac{\eta\sigma^2}{\bar{\alpha}} \quad \text{for any } t.$$

Moreover, we have that $\max_{k\le t}\mathbb{E}\|u_k - u_{k-1}\|^2 \le 4B + 16\frac{\eta\sigma^2}{\bar{\alpha}}$ for all $t$. Now since $\eta \le \bar{\alpha}/(4L_\ell^2(1 + L_{\mathsf{eq}}^2))$, setting

$$\bar{R} := R + \frac{2c^2\sigma_{\mathsf{a}}^2}{1-\rho^2} + 6\left(\frac{L_{\mathsf{eq}}^2}{(1-\rho^2)^2}\left(4B + \frac{4\sigma^2}{L_\ell^2(1+L_{\mathsf{eq}}^2)}\right)\right)$$

results in the expected tracking error being bounded by $\epsilon_\tau = \eta^2\sigma^2$ as claimed. This completes the proof. $\qquad\square$

This proof utilizes an arbitrary $\rho$-contracting stochastic method for the agents. It useful to see what the statement is for some particular methods. Let us start with stochastic gradient play.

**Proposition I.6.** *Suppose that Assumptions 2.1, 4.1, and 4.3 hold, that we have available a constant $R > \|x_0 - x^*(u_0)\|$, and that we are in the regime where $\alpha > L_z L_{eq}$ so that there is a unique performatively stable equilibrium. Further, suppose the decision-maker runs Algorithm 4 with* `Alg := RGM` *using step-size $\eta \le \bar{\alpha}/(4L_\ell^2(1 + L_{eq}^2))$ where $\bar{\alpha} := \alpha - L_z L_{eq}$, and the agents employ a stochastic gradient player as $\mathcal{A}$ with $\rho \in [0, 1)$ and $\sigma_a \in (0, \infty)$. Suppose the agents run stochastic gradient play stage-wise via Algorithm 2. Set the epoch length to*

$$\tau = \sum_{k=0}^{K} T_k = \left\lceil \left(1 + \frac{2L_a^2}{\mu^2}\right) \log\left(\frac{2\bar{R}}{\epsilon_\tau}\right) \right\rceil + \sum_{k=1}^{K} \left\lceil \left(1 + \frac{2^{k+1}L_a^2}{\mu^2}\right) \log(4) \right\rceil, \tag{25}$$

*and tolerance $\epsilon_\tau = \eta^2 \sigma^2$ where $K = \left\lceil 1 + \log_2(\frac{\sigma_a^2}{\epsilon_\tau L_a^2}) \right\rceil$ and*

$$\bar{R} := R + \frac{4\gamma \sigma_a^2}{\mu} + 6\left(\frac{4L_{eq}^2}{(\mu\gamma)^2}\left(4B + \frac{4\sigma^2}{L_\ell^2(1 + L_{eq}^2)}\right)\right).$$

*Then the following estimate holds:*

$$\mathbb{E}_t \|u_t - u^{ps}\|^2 \le \left(1 - \frac{\bar{\alpha}\eta}{2}\right)^t \|u_0 - u^{ps}\|^2 + \frac{4\eta \sigma^2}{\bar{\alpha}}.$$

Notice the only change is the constants for the stage-based algorithm.

As noted in the main it possible to employ any number of stage-based methods from stochastic optimization in order to obtain convergence to an $\varepsilon$-performatively stable equilibrium. The following is a more formal statement of Corollary 4.5 from the main body.

**Corollary I.7** (Formal Statement of Corollary 4.5)**.** *Under the assumptions of Theorem 4.4, consider running the stochastic repeated gradient method in $k = 0, \ldots, K$ super-epochs, for $T_k$ epochs each with constant step-size $\eta_k = 2^{-k}\eta_0$, and such that the last iterate of each epoch $k$ is used as the first iterate in stage $k + 1$. Fix a target accuracy $\varepsilon > 0$ and suppose the decision-maker has $B \ge \|u_0 - u^{ps}\|$. Set $\eta_0 := \bar{\alpha}/(4L_\ell^2(1 + L_{eq}^2))$,*

$$T_0 = \left\lceil \frac{2}{\bar{\alpha}\eta_0} \log\left(\frac{2B^2}{\varepsilon}\right) \right\rceil, \quad T_k = \left\lceil \frac{2\log(4)}{\bar{\alpha}\eta_k} \right\rceil \quad \text{for } k \ge 1, \text{ and} \quad K = \left\lceil 1 + \log_2\left(\frac{\sigma^2}{L_\ell^2(1 + L_{eq}^2)\varepsilon}\right) \right\rceil.$$

*Then $\mathbb{E}\|u_T - u^{ps}\|^2 \le \varepsilon$ and $\mathbb{E}\|x_T - x^*(u^{ps})\|^2 \le 2(\epsilon_\tau + L_{eq}\varepsilon)$ in a total number of epochs*

$$T = \sum_{k=1}^{K} T_k \lesssim \mathcal{O}\left(\frac{L_\ell^2(1 + L_{eq}^2)}{\bar{\alpha}^2} \log\left(\frac{2B^2}{\varepsilon}\right) + \frac{\sigma^2}{\bar{\alpha}^2\varepsilon}\right).$$

*Proof.* The proof follows immediately from applying Lemma E.3 with $\mathcal{A} \equiv$ RGM in Algorithm 2. Indeed, we set constants

$$\psi(\eta) = \frac{\bar{\alpha}\eta}{2}, \quad C = 1, \quad D = \frac{4\sigma^2}{\bar{\alpha}}, \quad \eta_0 = \frac{\bar{\alpha}}{4L_\ell^2(1 + L_{eq}^2)}$$

so that

$$T = \sum_{k=0}^{K} T_k = \left\lceil \frac{8L_\ell^2(1 + L_{eq}^2)}{\bar{\alpha}^2} \cdot \log\left(\frac{2R^2}{\varepsilon}\right) \right\rceil + \sum_{k=1}^{K} \left\lceil \frac{8L_\ell^2(1 + L_{eq}^2)\log(4)}{2^{-k} \cdot \bar{\alpha}^2} \right\rceil$$

and $K$ is given as in the corollary statement. Applying Lemma E.3 gives us that $\mathbb{E}\|u_T - u^{ps}\|^2 \le \varepsilon$ in $T$ total stages. Then

$$\mathbb{E}\|x_T - x^*(u^{ps})\|^2 \le 2\mathbb{E}\|x_T - x^*(u_T)\|^2 + 2\mathbb{E}\|x^*(u_T) - x^*(u^{ps})\|^2 \le 2(\epsilon_\tau + L_{eq}\varepsilon),$$

where $\epsilon_\tau$ is given in the Theorem I.5. $\qquad\square$

### I.3.1. High Probability Results for Naïve Decision-Maker

In the preceding analysis we utilized the expected tracking error bounds for the agents problem from the oblivious setting (Section 3, Appendix H). These results hold only in *expectation* meaning that the decision-maker would need to be able to deploy $u_t$ several times to be confidence in each epoch the results hold in the practice. On the other hand, it is more reasonable to leverage the high probability results from Appendix H.3 since these convergence results state that with probability $(1 - \delta)$ that a single deployment of $u_t$ in epoch $t$ ensures that $\|x_t - x_t^*\|^2 \leq \epsilon \log(e/\delta)$. Let us state such a theorem. Indeed, if the decision-maker deploys their algorithms in real-time with *irreversible* drift, high-probability efficiency results are desired in order to characterize the performance of the algorithm if it were executed only once.

In addition to Assumption H.7 for the agents, we require the following tail assumptions on the equilibrium drift and gradient noise for the decision-maker.

**Assumption I.8** (Sub-Gaussian drift and noise). There exist constants $\Delta, \sigma > 0$ such that the following two conditions hold for all $t \geq 0$:

(a) The drift $\Delta_t^2$ is sub-exponential conditioned on $\mathcal{F}_t$ with parameter $\Delta^2$:

$$\mathbb{E}[\exp(\lambda \Delta_t^2)|\,\mathcal{F}_t] \leq \exp(\lambda \Delta^2) \quad \text{for all} \quad 0 \leq \lambda \leq \Delta^2$$

(b) The gradient noise $\phi_t$ is norm sub-Gaussian conditioned on $\mathcal{F}_t$ with parameter $\sigma^2$:

$$\mathbb{P}(\|\phi_t\| \geq \zeta|\,\mathcal{F}_t) \leq 2\exp(-2\zeta^2/\sigma^2) \quad \text{for all} \quad \zeta > 0.$$

Note that Assumption H.7 implies Assumption H.3 with the same constants $\Delta$ and $\sigma$.

**Theorem I.9.** *Suppose that Assumptions 2.1, 4.1, 4.3, H.7, and I.8 all hold, there exists $b, B > 0$ such that $b\mathbb{B} \subseteq \mathcal{U} \subseteq B\mathbb{B}$ where $\mathbb{B} = \{u \in \mathbb{R}^d |\, \|u\| \leq 1\}$, we have available constants $R_x > \|x_{-1} - x^*(u_0)\|^2$ and $R_u > \|u_0 - u^{\text{ps}}\|^2$, and we are in the regime where $\alpha > L_z L_{\text{eq}}$ so that there is a unique performatively stable equilibrium. Further, suppose the decision-maker runs Algorithm 4 with $\text{Alg} := \text{RGM}$ using step-size $\eta \leq \bar{\alpha}/(4L_\ell^2(1 + L_{\text{eq}}^2))$ where $\bar{\alpha} := \alpha - L_z L_{\text{eq}}$, and the agents employ stochastic gradient play (via Algorithm 2) $\mathcal{A}$ with $\sigma_{\text{a}} \in (0, \infty)$. There exists an absolute constant $\tilde{c} > 0$ such that for any specified $t \in \mathbb{N}$ and $\delta \in (0, 1)$, the equilibrium tracking error across epochs satisfies $\|x_t - x_t^\star\|^2 \lesssim \epsilon_\tau \log(e/\delta)$ with probability at least $1 - \delta$ where $\epsilon_\tau = \eta^2\sigma^2$, the epoch length is*

$$\tau = \sum_{k=0}^{K} T_k = \left\lceil \left(1 + \frac{2L_{\text{a}}^2}{\mu^2}\right) \log\left(\frac{2\bar{R}}{\epsilon_\tau}\right) \right\rceil + \sum_{k=1}^{K} \left\lceil \left(1 + \frac{2^{k+1}L_{\text{a}}^2}{\mu^2}\right) \log(4) \right\rceil, \tag{26}$$

*with $K = \left\lceil 1 + \log_2\left(\frac{\sigma_{\text{a}}^2}{\epsilon_\tau \bar{L}_{\text{a}}^2}\right) \right\rceil$ and*

$$\bar{R} := R_x + \left(\frac{32(\tilde{c}\sigma_{\text{a}})^2\gamma}{\mu} + 20\left(\frac{2L_{\text{eq}}B}{\mu\gamma}\right)^2\right) \log\left(\frac{e}{\delta_x}\right)$$

*so that the following estimate holds with probability at least $(1 - \delta)^3$:*

$$\|u_t - u^{\text{ps}}\|^2 \leq \left(1 - \frac{\eta\bar{\alpha}}{2}\right)^t \|u_0 - u^{\text{ps}}\|^2 + \left(\frac{(16\tilde{c} + 1)\eta\sigma^2}{\bar{\alpha}}\left(1 + \frac{1}{2(1 + L_{\text{eq}}^2)}\right)\right) \log^2\left(\frac{e}{\delta}\right),$$

The proof of this theorem follows Theorem I.5, replacing the expected bounds on the equilibrium tracking error for the agent initializations—i.e., $\mathbb{E}\|x_{t-1} - x_t^\star\|^2$—with high probability statements. Analogous statements to Proposition I.6 and Corollary I.7 hold as well in this high probability setting, simply by replacing the appropriate constants.

Below, let us highlight the key steps.

*Proof of Theorem I.9.* Recall the gradient definitions in (20) from the proof of Theorem I.5. Fix two constants $\nu_1, \nu_2 > 0$ to be specified later. Noting that $u_{t+1}$ is the minimizer of the 1-strongly convex function $u \mapsto \frac{1}{2}\|u_t - \eta g_t - u\|^2$ over $\mathcal{U}$, we deduce that

$$\frac{1}{2}\|u_{t+1} - u^{\text{ps}}\|^2 \leq \frac{1}{2}\|u_t - \eta g_t - u^{\text{ps}}\|^2 - \frac{1}{2}\|u_t - \eta_t g_t - u_{t+1}\|^2.$$

Expanding the squares on the right hand side and combining terms yields

$$\frac{1}{2}\|u_{t+1} - u^{\text{ps}}\|^2 \leq \frac{1}{2}\|u_t - u^{\text{ps}}\|^2 - \eta_t\langle g_t, u_{t+1} - u^{\text{ps}}\rangle - \frac{1}{2}\|u_{t+1} - u_t\|^2$$

$$= \frac{1}{2}\|u_t - u^{\text{ps}}\|^2 - \eta\langle g_t, u_t - u^{\text{ps}}\rangle - \frac{1}{2}\|u_{t+1} - u_t\|^2 - \eta\langle g_t, u_{t+1} - u_t\rangle.$$

Using the fact that $\mathbb{E}_t[g_t] = G_t(u_t)$, we successively compute

$$\frac{1}{2}\|u_{t+1} - u^{\text{ps}}\|^2 \leq \frac{1}{2}\|u_t - u^{\text{ps}}\|^2 - \eta\langle g_t, u_t - u^{\text{ps}}\rangle - \frac{1}{2}\|u_{t+1} - u_t\|^2 - \eta\mathbb{E}_t\langle g_t, u_{t+1} - u_t\rangle,$$

$$\leq \frac{1}{2}\|u_t - u^{\text{ps}}\|^2 - \eta\langle G_t(u_t), u_t - u^{\text{ps}}\rangle - \frac{1}{2}\|u_{t+1} - u_t\|^2 - \eta\langle g_t, u_{t+1} - u_t\rangle,$$

$$= \frac{1}{2}\|u_t - u^{\text{ps}}\|^2 - \eta\langle G_\star(u_{t+1}), u_{t+1} - u^{\text{ps}}\rangle - \frac{1}{2}\|u_{t+1} - u_t\|^2$$

$$+ \eta\underbrace{\langle g_t - G_t(u_t), u_t - u_{t+1}\rangle}_{P_1} + \eta_t\underbrace{\langle G_t(u_t) - G_\star(u_{t+1}), u^{\text{ps}} - u_{t+1}\rangle}_{P_2}.$$

Recall that for any $z$, the loss $\ell(u, z)$ is $\alpha$–strongly convex in $u$ so that

$$\langle G_{\text{ps}}(u_{t+1}), u_{t+1} - u^{\text{ps}}\rangle \geq \langle G_{\text{ps}}(u_{t+1}) - G_{\text{ps}}(u^{\text{ps}}), u_{t+1} - u^{\text{ps}}\rangle \geq \alpha\|u_{t+1} - u^{\text{ps}}\|^2.$$

Therefore, adding and subtracting $G_{\text{ps}}(u_{t+1})$, we have that

$$\langle G_\star(u_{t+1}), u_{t+1} - u^{\text{ps}}\rangle = \langle G_{\text{ps}}(u_{t+1}), u_{t+1} - u^{\text{ps}}\rangle + \langle G_\star(u_{t+1}) - G_{\text{ps}}(u_{t+1}), u_{t+1} - u^{\text{ps}}\rangle$$

Now, the second term is upper bounded as follows:

$$\langle G_\star(u_{t+1}) - G_{\text{ps}}(u_{t+1}), u_{t+1} - u^{\text{ps}}\rangle \leq L_z L_{\text{eq}}\|u_{t+1} - u^{\text{ps}}\|^2.$$

Then, rearranging the above expression, we have that

$$\frac{1 + 2\eta\bar{\alpha}}{2}\|u_{t+1} - u^{\text{ps}}\|^2 \leq \frac{1}{2}\|u_t - u^{\text{ps}}\|^2 - \frac{1}{2}\|u_{t+1} - u_t\|^2 + \eta(P_1 + P_2). \tag{27}$$

Applying Young's inequality to $P_1$ and invoking Assumption I.8, we have that

$$P_1 \leq \frac{\|g_t - G_t(u_t)\|^2}{2\nu_1} + \frac{\nu_1\|u_{t+1} - u_t\|^2}{2} \leq \frac{\|\phi_t\|^2}{2\nu_1} + \frac{\nu_1\|u_{t+1} - u_t\|^2}{2}. \tag{28}$$

We have the following upper bound for $P_2$:

$$P_2 \leq \frac{\|G_t(u_t) - G_\star(u_{t+1})\|^2}{2\nu_2} + \frac{\nu_2\|u^{\text{ps}} - u_{t+1}\|^2}{2}$$

$$\leq \frac{2\|G_t(u_t) - G_\star(u_t)\|^2 + 2\|G_\star(u_t) - G_\star(u_{t+1})\|^2}{2\nu_2} + \frac{\nu_2\|u^{\text{ps}} - u_{t+1}\|^2}{2}$$

$$\leq \frac{2\|G_t(u_t) - G_\star(u_t)\|^2 + 2L_\ell^2(1 + L_{\text{eq}}^2)\|u_t - u_{t+1}\|^2}{2\nu_2} + \frac{\nu_2\|u^{\text{ps}} - u_{t+1}\|^2}{2}.$$

The first term in the first fraction can be bounded as follows:

$$\|G_t(u_t) - G_\star(u_t)\|^2 \leq L_\ell^2\|\mathcal{A}(x_{t-1}, u_t) - x^*(u_t)\|^2. \tag{29}$$

This shows there is a time varying bias component in the gradient estimator. Here, we need to show that $\|\mathcal{A}(x_{t-1}, u_t) - x^*(u_t)\|^2 \leq \epsilon_\tau \log(e/\delta_x)$ with probability $1 - \delta_x$ for any $\delta_x \in [0, 1]$.

This is a good point to lay out the proof structure from here forward. Define the events

$$\mathcal{E}_t = \{\|\mathcal{A}(x_{t-1}, u_t) - x^*(u_t)\|^2 \leq \epsilon_\tau \log(e/\delta_x)\} \quad \text{and} \quad \mathcal{E}_0 = \{\|x_{t-1} - x^*(u_t)\|^2 \leq \bar{R}\},$$

where we will specify what $\bar{R}$ is shortly. The aim here is then to lower bound

$$\Pr(\mathcal{E}_t) = \Pr(\mathcal{E}_t|\mathcal{E}_0)\Pr(\mathcal{E}_0).$$

We know from the analysis of $\rho$ contracting algorithms, that within a $\tau$-length epoch conditioned on $\mathcal{E}_0$, we will have $\Pr(\mathcal{E}_t|\mathcal{E}_0) \geq 1 - \delta_x$ where $\tau$ is defined with respect to $\bar{R}$. Hence, we need to bound the probability of the initial condition event $\mathcal{E}_0$. This is where we use the high probability tracking error bound analysis from Appendix H.3. Indeed, for some absolute constant $c_x$, with probability $1 - \delta_x$, we have that

$$\|x_{t-1} - x^*(u_t)\|^2 \leq \left(1 - \frac{\mu\gamma}{4}\right)^t \|x_{-1} - x^*(u_0)\|^2 + \left(\frac{32(c_x\sigma_\mathsf{a})^2\gamma}{\mu} + 20\left(\frac{2L_\mathsf{eq}B}{\mu\gamma}\right)^2\right)\log\left(\frac{e}{\delta_x}\right), \qquad (30)$$

where we have bounded the drift $\|x^*(u_t) - x^*(u_{t-1})\| \leq L_\mathsf{eq}\|u_t - u_{t-1}\| \leq 2L_\mathsf{eq}B$ due to the fact that $u_t \in \mathcal{U} \subseteq B\mathbb{B}$ for all $t$ Immediately[7], we can see that we can set

$$\bar{R} := R_x + \left(\frac{32(c_x\sigma_\mathsf{a})^2\gamma}{\mu} + 20\left(\frac{2L_\mathsf{eq}B}{\mu\gamma}\right)^2\right)\log\left(\frac{e}{\delta_x}\right).$$

Now, coming back to the analysis of $\|u_t - u^\mathsf{ps}\|$, we now have that $\Pr(\mathcal{E}_t) = \Pr(\mathcal{E}_t|\mathcal{E}_0)\Pr(\mathcal{E}_0) \geq (1 - \delta_x)^2$. In this case, we deduce that

$$P_2 \leq \frac{L_\ell^2\epsilon_\tau\log(e/\delta)}{\nu_2} + \frac{2L_\ell^2(1+L_\mathsf{eq}^2)\|u_t - u_{t+1}\|^2}{2\nu_2} + \frac{\nu_2\|u^\mathsf{ps} - u_{t+1}\|^2}{2}.$$

Coming back to the bound in (37), we have that

$$\frac{1 + 2\eta\bar{\alpha}}{2}\|u_{t+1} - u^\mathsf{ps}\|^2 \leq \frac{1}{2}\|u_t - u^\mathsf{ps}\|^2 - \frac{1}{2}\|u_{t+1} - u_t\|^2 + \eta\left(\frac{\|\phi_t\|^2}{2\nu_1} + \frac{\nu_1\|u_{t+1} - u_t\|^2}{2}\right)$$
$$+ \eta\left(\frac{L_\ell^2\epsilon_\tau\log(e/\delta)}{\nu_2} + \frac{2L_\ell^2(1+L_\mathsf{eq}^2)\|u_t - u_{t+1}\|^2}{2\nu_2} + \frac{\nu_2\|u^\mathsf{ps} - u_{t+1}\|^2}{2}\right)$$

so that

$$\frac{1 + 2\eta\bar{\alpha} - \eta\nu_2}{2}\|u_{t+1} - u^\mathsf{ps}\|^2 \leq \frac{1}{2}\|u_t - u^\mathsf{ps}\|^2 + \frac{\eta\sigma^2}{2\nu_1} + \frac{\eta L_\ell^2\epsilon_\tau\log(\frac{e}{\delta})}{\nu_2} - \frac{1 - 2L_\ell^2(1+L_\mathsf{eq}^2)\eta\nu_2^{-1} - \eta\nu_1}{2}\|u_{t+1} - u_t\|^2.$$

Letting $\nu_1 = \eta^{-1} - 2L_\ell^2(1 + L_\mathsf{eq}^2)/\bar{\alpha}$ and $\nu_2 = \bar{\alpha}$ ensures that the last term on the right is zero. By the assumption $\eta \leq \bar{\alpha}/(4L_\ell^2(1 + L_\mathsf{eq}^2))$, we have that $\frac{1}{\eta} \geq 4L_\ell^2(1 + L_\mathsf{eq}^2)/\bar{\alpha}$ so that $\nu_1 \geq \frac{1}{2\eta}$. Hence we have that

$$\frac{1 + \eta\bar{\alpha}}{2}\|u_{t+1} - u^\mathsf{ps}\|^2 \leq \frac{1}{2}\|u_t - u^\mathsf{ps}\|^2 + \eta^2\|\phi_t\|^2 + \frac{\epsilon_\tau\log(e/\delta_x)}{4(1 + L_\mathsf{eq}^2)}$$

so that

$$\|u_{t+1} - u^\mathsf{ps}\|^2 = \frac{1}{1 + \eta\bar{\alpha}}\|u_t - u^\mathsf{ps}\|^2 + \frac{2\eta^2}{1 + \eta\bar{\alpha}}\|\phi_t\|^2 + \frac{1}{1 + \eta\bar{\alpha}}\frac{\epsilon_\tau\log(e/\delta_x)}{4(1 + L_\mathsf{eq}^2)}$$
$$\leq \left(1 - \frac{\eta\bar{\alpha}}{2}\right)\|u_t - u^\mathsf{ps}\|^2 + 2\eta^2\|\phi_t\|^2 + \frac{1}{1 + \eta\bar{\alpha}}\frac{\epsilon_\tau\log(e/\delta_x)}{4(1 + L_\mathsf{eq}^2)}. \qquad (31)$$

Under Assumption I.8, there exists an absolute constant $\tilde{c} \geq 1$ such that $\|\phi_t\|^2$ is sub-exponential conditioned on $\mathcal{F}_t$ with parameter $\tilde{c}\sigma^2$ and $\phi_t$ is mean-zero sub-Gaussian conditioned on $\mathcal{F}_t$ with parameter $\tilde{c}\sigma$ for all $t$ (cf. Lemma 3 from Jin et al. (2019)). Assumption I.8 also implies that $\Delta_t^2$ is sub-exponential conditioned on $\mathcal{F}_t$ with parameter $\Delta^2$.

---

[7]Note that we could reduce this term by more specifically bounding $\left(1 - \frac{\mu\gamma}{4}\right)^t\|x_{-1} - x^*(u_0)\|^2$ thereby allowing $\bar{R}$ to be a function of the current iteration $t$ instead of using the fixed bound $\|x_{-1} - x^*(u_0)\|^2 < R_x$.

Given (31), we apply Proposition H.8 with

$$V_t = \|u_t - u^{\mathrm{ps}}\|^2, \quad X_t = 2\eta^2\|\phi_t\|^2 + \frac{1}{1+\eta\bar{\alpha}}\frac{\epsilon_\tau \log(\frac{e}{\delta_x})}{4(1+L_{\mathrm{eq}}^2)}, \quad \alpha_t = 1 - \frac{\eta\bar{\alpha}}{2}, \quad \nu_t = 2\tilde{c}\eta^2\sigma^2 + \frac{1}{1+\eta\bar{\alpha}}\frac{\epsilon_\tau \log(\frac{e}{\delta_x})}{4(1+L_{\mathrm{eq}}^2)},$$

$D_t = 0$, and $\kappa_t = 0$. This yields the estimate

$$\mathbb{E}[\exp(\lambda\|u_{t+1} - u^{\mathrm{ps}}\|^2)] \le \exp\left(\lambda\left(2\tilde{c}\eta^2\sigma^2 + \frac{1}{1+\eta\bar{\alpha}}\frac{\epsilon_\tau \log(\frac{e}{\delta_x})}{4(1+L_{\mathrm{eq}}^2)}\right)\right)\mathbb{E}\left[\exp\left(\lambda\left(1 - \frac{\eta\bar{\alpha}}{2}\right)\|u_t - u^{\mathrm{ps}}\|^2\right)\right], \quad (32)$$

for all

$$0 \le \lambda \le \frac{1}{2(2\eta^2\tilde{c}\sigma^2 + \frac{1}{1+\eta\bar{\alpha}}\frac{\epsilon_\tau \log(e/\delta_x)}{4(1+L_{\mathrm{eq}}^2)})}. \quad (33)$$

Since $\left(1 - \frac{\eta\bar{\alpha}}{2}\right) \in (0,1]$ and $V_t$ is a non-negative random variable (almost surely), Jensen's inequality implies that

$$\mathbb{E}[\exp(\lambda V_{t+1})] \le \exp\left(\lambda\nu_t\right)\mathbb{E}\left[\exp\left(\lambda V_t\right)^{\left(1-\frac{\eta\bar{\alpha}}{2}\right)}\right]$$

$$\le \exp\left(\lambda\nu_t\right)\mathbb{E}\left[\exp\left(\lambda V_t\right)\right]^{\left(1-\frac{\eta\bar{\alpha}}{2}\right)}.$$

Iterating this expression, we have that

$$\mathbb{E}\left[\exp(\lambda V_t)\right] \le \exp\left(\left(2\tilde{c}\eta^2\sigma^2 + \frac{1}{1+\eta\bar{\alpha}}\frac{\epsilon_\tau \log(\frac{e}{\delta_x})}{4(1+L_{\mathrm{eq}}^2)}\right)\sum_{s=0}^{t-1}\left(1 - \frac{\eta\bar{\alpha}}{2}\right)^s\right)(\mathbb{E}[\exp(\lambda V_0)])^{(1-\frac{\eta\bar{\alpha}}{2})^t}$$

$$= \exp(\lambda\nu)\exp\left(\lambda\left(\left(1 - \frac{\eta\bar{\alpha}}{2}\right)^t V_0 + \left(2\tilde{c}\eta^2\sigma^2 + \frac{1}{1+\eta\bar{\alpha}}\frac{\epsilon_\tau \log(\frac{e}{\delta_x})}{4(1+L_{\mathrm{eq}}^2)}\right)\sum_{s=1}^{t-1}\left(1 - \frac{\eta\bar{\alpha}}{2}\right)^s\right)\right)$$

$$\le \exp\left(\lambda\left(\left(1 - \frac{\eta\bar{\alpha}}{2}\right)^t V_0 + \left(2\eta^2\tilde{c}\sigma^2 + \frac{1}{1+\eta\bar{\alpha}}\frac{\epsilon_\tau \log(\frac{e}{\delta_x})}{4(1+L_{\mathrm{eq}}^2)}\right)\sum_{s=0}^{t-1}\left(1 - \frac{\eta\bar{\alpha}}{2}\right)^s\right)\right)$$

$$\le \exp\left(\lambda\left(\left(1 - \frac{\eta\bar{\alpha}}{2}\right)^t V_0 + \left(\frac{4\tilde{c}\eta\sigma^2}{\bar{\alpha}} + \frac{2}{(\eta\bar{\alpha})^2}\frac{\epsilon_\tau \log(\frac{e}{\delta_x})}{4(1+L_{\mathrm{eq}}^2)}\right)\right)\right)$$

for all $\lambda$ satisfying (33), where the equality holds since $\|u_0 - u^{\mathrm{ps}}\|^2$ is a constant. Recall that $\tilde{c} \ge 1$ and $\eta\bar{\alpha} \le 1$ so that

$$\frac{4\eta\sigma^2}{\bar{\alpha}} + \frac{2}{(\eta\bar{\alpha})^2}\frac{\epsilon_\tau \log(\frac{e}{\delta_x})}{4(1+L_{\mathrm{eq}}^2)} \le \nu := \frac{16\tilde{c}\eta\sigma^2}{\bar{\alpha}} + \frac{2}{(\eta\bar{\alpha})^2}\frac{\epsilon_\tau \log(\frac{e}{\delta_x})}{4(1+L_{\mathrm{eq}}^2)}$$

and

$$\frac{1}{\bar{\alpha}} = \frac{\bar{\alpha}}{16\tilde{c}\eta\sigma^2 + \frac{2}{\eta^2\bar{\alpha}}\frac{\epsilon_\tau \log(\frac{e}{\delta_x})}{4(1+L_{\mathrm{eq}}^2)}} \le \min\left\{\frac{\bar{\alpha}}{16\cdot\tilde{c}^2\eta\sigma_{\mathrm{a}}^2}, \frac{1}{2\tilde{c}\eta^2\sigma^2 + \frac{2}{\eta\bar{\alpha}}\frac{\epsilon_\tau \log(\frac{e}{\delta_x})}{4(1+L_{\mathrm{eq}}^2)}}\right\}$$

Hence, we have that

$$\mathbb{E}\left[\exp\left(\lambda\left(\|u_t - u^{\mathrm{ps}}\|^2 - \left(1 - \frac{\eta\bar{\alpha}}{2}\right)^t\|u_0 - u^{\mathrm{ps}}\|^2\right)\right)\right] \le \exp(\lambda\nu) \quad \forall\ 0 \le \lambda \le \frac{1}{\nu}.$$

Rewriting this expression, we have that

$$\frac{\mathbb{E}\left[\exp\left(\lambda\left(\|u_t - u^{\mathrm{ps}}\|^2 - \left(1 - \frac{\eta\bar{\alpha}}{2}\right)^t\|u_0 - u^{\mathrm{ps}}\|^2\right)\right)\right]}{\exp(\lambda\nu)} \le 1.$$

Applying Markov's inequality, we have that

$$\Pr\left(\exp\left(\lambda\left(\|u_t - u^{\mathrm{ps}}\|^2 - \left(1 - \frac{\eta\bar{\alpha}}{2}\right)^t \|u_0 - u^{\mathrm{ps}}\|^2\right)\right) \geq \frac{\exp(\lambda\nu)}{\delta_u}\right)$$

$$\leq \frac{\mathbb{E}\left[\exp\left(\lambda\left(\|u_t - u^{\mathrm{ps}}\|^2 - \left(1 - \frac{\eta\bar{\alpha}}{2}\right)^t \|u_0 - u^{\mathrm{ps}}\|^2\right)\right)\right]}{\exp(\lambda\nu)/\delta_u} \leq \delta_u.$$

Therefore, setting $\lambda = \frac{1}{\nu}$, with probability $1 - \delta_u$, we have that

$$\|u_t - u^{\mathrm{ps}}\|^2 \leq \left(1 - \frac{\eta\bar{\alpha}}{2}\right)^t \|u_0 - u^{\mathrm{ps}}\|^2 + \left(\frac{16\tilde{c}\eta\sigma^2}{\bar{\alpha}} + \frac{2}{(\eta\bar{\alpha})^2}\frac{\epsilon_\tau \log(\frac{e}{\delta_x})}{4(1 + L_{\mathrm{eq}}^2)}\right)\log\left(\frac{e}{\delta_u}\right), \tag{34}$$

conditioned on the event $\mathcal{E}_t$ holding. Setting $\epsilon_\tau = \eta^2\sigma^2$ (34) becomes

$$\|u_t - u^{\mathrm{ps}}\|^2 \leq \left(1 - \frac{\eta\bar{\alpha}}{2}\right)^t \|u_0 - u^{\mathrm{ps}}\|^2 + \left(\frac{(16\tilde{c}+1)\eta\sigma^2}{\bar{\alpha}}\left(1 + \frac{1}{2(1 + L_{\mathrm{eq}}^2)}\right)\right)\max\left\{1, \log\left(\frac{e}{\delta_x}\right)\right\}\log\left(\frac{e}{\delta_u}\right), \tag{35}$$

Then all together we have we have that (35) holds with probability $(1 - \delta_u)(1 - \delta_x)^2$. With $\delta_u = \delta_x = \delta$ for some $\delta \in [0, 1]$ the claim holds. $\qquad\square$

### I.3.2. BOUNDING THE TIME TO THE LOW DRIFT-TO-NOISE REGIME IN EXPECTATION

Reflecting back to Figure 3, the target accuracy can be better optimized if the agents switch their step-size to the optimal $\gamma_\star$ once in the low drift-to-noise regime. Hence, it is interesting to characterize the time $T$ after which $\max_{k \leq T} \mathbb{E}\|u_{k-1} - u_k\|^2$ ensures the agents are in the low drift-to-noise regime in expectation.

The following is the formal statement of Proposition 4.6.

**Proposition I.10.** *Under the assumptions of Corollary I.7, the estimate* $\max_{k \leq T} \mathbb{E}\|u_k - u_{k-1}\|^2 \lesssim \left(\frac{\mu^2\sigma_{\mathsf{a}}}{4\cdot\sqrt{3}L_{\mathrm{eq}}L_{\mathsf{a}}^2}\right)^2$ *holds after* $T = \sum_{k=1}^K T_k \lesssim \mathcal{O}\left(\frac{L^2}{\bar{\alpha}^2}\log\left(\frac{2B^2}{\varepsilon}\right) + \frac{\sigma^2}{\bar{\alpha}^2\varepsilon}\right)$ *epochs where* $\varepsilon = \frac{1}{6}\left(\mu^2\sigma_{\mathsf{a}}/(4\sqrt{3}\cdot L_{\mathrm{eq}}L_{\mathsf{a}}^2)\right)^2$.

Once in this region the agents are naturally incentivized to optimize their learning rates (i.e., selecting $\gamma_\star$) as it will enable them to more effectively stabilize the learning process.

*Proof.* We aim to show that

$$\max_t \mathbb{E}\|u_t - u_{t-1}\|^2 \leq \left(\frac{\mu^2\sigma_{\mathsf{a}}}{4\sqrt{3}\cdot L_{\mathrm{eq}}L_{\mathsf{a}}^2}\right)^2.$$

Let us first bound the sequence of differences for any particular $t$. Observe that

$$\|u_t - u_{t-1}\|^2 \leq 2(\|u_t - u^{\mathrm{ps}}\|^2 + \|u_{t-1} - u^{\mathrm{ps}}\|^2) \quad \text{for any } t \geq 1.$$

Choose $(\varepsilon, T)$ such that $\|u_T - u^{\mathrm{ps}}\|^2 \leq \varepsilon$. Then, we know that since $u_{T+1}$ is an update from the $(K+1)$-th stage, we have that

$$\mathbb{E}\|u_{T+1} - u^{\mathrm{ps}}\|^2 \leq \left(1 - \frac{\bar{\alpha}\eta}{2}\right)\mathbb{E}\|u_T - u^{\mathrm{ps}}\|^2 + \frac{4\eta_K^2\sigma^2}{1 + \bar{\alpha}\eta_{K+1}}$$

$$\leq \varepsilon + \frac{4\sigma^2}{1 + \frac{\bar{\alpha}^2}{2^{K+1}\cdot 4\cdot L_\ell^2(1+L_{\mathrm{eq}}^2)}}\frac{\bar{\alpha}^2}{2^{2K}(4\cdot L_\ell^2(1+L_{\mathrm{eq}}^2))^2}$$

$$\leq \varepsilon + \frac{4\sigma^2 \cdot 2^{K+1}\cdot 4 \cdot L_\ell^2(1+L_{\mathrm{eq}}^2)}{\bar{\alpha}^2}\frac{\bar{\alpha}^2}{2^{2K}(4\cdot L_\ell^2(1+L_{\mathrm{eq}}^2))^2}$$

$$\leq \varepsilon + \frac{2\cdot 4\sigma^2}{1}\frac{1}{2^K\cdot 4\cdot L_\ell^2(1+L_{\mathrm{eq}}^2)}$$

$$\leq \varepsilon + \frac{4\sigma^2}{1}\frac{L_\ell^2(1+L_{\mathrm{eq}}^2)\epsilon}{\sigma^2\cdot 4\cdot L_\ell^2(1+L_{\mathrm{eq}}^2)}$$

$$\leq 2\varepsilon$$

Hence, by setting $\varepsilon := \frac{1}{6}\left(\frac{\mu^2\sigma_{\mathsf{a}}}{4\sqrt{3}\cdot L_{\mathsf{eq}}L_{\mathsf{a}}^2}\right)^2$, we have that

$$\mathbb{E}\|u_{T+1} - u_T\|^2 \leq \left(\frac{\mu^2\sigma_{\mathsf{a}}}{4\sqrt{3}\cdot L_{\mathsf{eq}}L_{\mathsf{a}}^2}\right)^2.$$

This holds for any $T$. Thus by setting

$$T = \sum_{k=1}^{K} T_k \lesssim \mathcal{O}\left(\frac{L_\ell^2(1+L_{\mathsf{eq}}^2)}{\bar{\alpha}^2}\log\left(\frac{2B^2}{\epsilon}\right) + \frac{\sigma^2}{\bar{\alpha}^2\epsilon}\right).$$

$$= \mathcal{O}\left(\frac{L_\ell^2(1+L_{\mathsf{eq}}^2)}{\bar{\alpha}^2}\log\left(\frac{12B^2\cdot(4\sqrt{3}\cdot L_{\mathsf{eq}}L_{\mathsf{a}}^2)^2}{(\mu^2\sigma_{\mathsf{a}})^2}\right) + \frac{6\sigma^2(4\sqrt{3}\cdot L_{\mathsf{eq}}L_{\mathsf{a}}^2)^2}{\bar{\alpha}^2(\mu^2\sigma_{\mathsf{a}})^2}\right)$$

we have that the drift-to-noise ratio is the low regime in expectation. $\qquad\square$

### I.3.3. NAÏVE DECISION-MAKER: DETERMINISTIC AGENT ALGORITHMS

Additionally, agents may run some deterministic algorithm. In this case the agents do not need to run a stage-wise algorithm since they do not introduce the additional bias due to stochasticity of their algorithm into the decision-makers problem.

**Proposition I.11.** *Suppose that Assumptions 2.1, 4.1, and 4.3 hold, that*

$$\sup_{(u,x)\in\mathcal{U}\times\mathcal{X}}\mathbb{E}[\|\nabla_u\ell(u,x+\xi)\|] \leq L_{\mathsf{u}},$$

*that we have available a constant $R > \|x_0 - x^*(u_0)\|$, and that we are in the regime where $\alpha > L_z L_{\mathsf{eq}}$ so that there is a unique performatively stable equilibrium. Further, suppose the decision-maker runs Algorithm 4 with Alg := RGM using step-size $\eta \leq \frac{\bar{\alpha}}{4L_\ell^2(1+L_{\mathsf{eq}}^2)}$ where $\bar{\alpha} := \alpha - L_z L_{\mathsf{eq}}$, and the agents employ a deterministic $\rho$-contracting algorithms (i.e. $\sigma_{\mathsf{a}} = 0$). Let the epoch length be given by*

$$\tau \geq \log\left(\frac{2L_\ell^2}{\bar{\alpha}\eta\sigma^2}\left(\rho^{t-1}R + \frac{\eta L_{\mathsf{eq}}L_{\mathsf{u}}}{1-\rho}\right)^2\right)\frac{1}{\log(1/\rho^2)}.$$

*Then the following estimate holds:*

$$\mathbb{E}_t\|u_{t+1} - u^{\mathsf{ps}}\|^2 \leq \left(1 - \frac{\bar{\alpha}\eta}{2}\right)^t\|u_0 - u^{\mathsf{ps}}\|^2 + \frac{4\eta\sigma^2}{\bar{\alpha}}.$$

To prove this proposition, we need a technical lemma on the contractive deterministic dynamics.

**Technical Lemma.** For both the naïve and strategic settings, we will need the following technical lemma on the behavior of the stochastic agents play. Recall that in each epoch $t$ the agents initialize their algorithm at $x_t^0 := x_{t-1}^\tau$ and that, by an abuse of notation, $x_0 = x_0^0$.

Recall that when the agents' algorithms are deterministic, Definition 2.3 reduces to $\|x_t^{k+1} - x^*(u_t)\|^2 \leq \rho^2\|x_t^k - x^*(u_t)\|^2$, so that

$$\|x_t^{k+1} - x^*(u_t)\| \leq \rho\|x_t^k - x^*(u_t)\|.$$

The following lemma will be used in the proof of Theorem I.5 when the agents are deterministic.

**Lemma I.12** (Deterministic Agent Contraction). *Suppose that the decision-maker is running Algorithm 4 with Alg := RGM using step-size $\eta$ and under the assumption that $\sup_{(u,x)\in\mathcal{U}\times\mathcal{X}}\|\nabla_u\ell(u,x)\| \leq L_{\mathsf{u}}$. Further, suppose agents use a $\rho$-contracting update (Definition 2.3) with $\rho \in [0,1)$ and $\sigma_{\mathsf{a}} = 0$. Under Assumption 2.1, the following bound holds:*

$$\|x_t^\tau(u_t) - x^*(u_t)\| \leq \rho^\tau\left(\rho^{t-1}\|x_0 - x^*(u_0)\| + \frac{\eta L_{\mathsf{eq}}L_{\mathsf{u}}}{1-\rho}\right).$$

*Proof.* Given Definition 2.3, we have that

$$\|x_t^\tau(u_t) - x^*(u_t)\| \leq \rho\|x_t^{\tau-1}(u_t) - x^*(u_t)\|.$$

Iterating this expression we have that

$$\|x_t^\tau(u_t) - x^*(u_t)\| \leq \rho^\tau\|x_{t-1}^\tau(u_{t-1}) - x^*(u_t)\|$$

Adding and subtracting appropriate terms we have that

$$\|x_t^\tau(u_t) - x^*(u_t)\| \leq \rho^\tau\|x_{t-1}^\tau(u_{t-1}) - x^*(u_{t-1}) + x^*(u_{t-1}) - x^*(u_t)\|$$
$$\leq \rho^\tau\|x_{t-1}^\tau(u_{t-1}) - x^*(u_{t-1})\| + \rho^\tau L_{\text{eq}}\|u_{t-1} - u_t\|$$

Continuing in this fashion we have that

$$\|x_t^\tau(u_t) - x^*(u_t)\| \leq \rho^\tau\|x_{t-1}^\tau(u_{t-1}) - x^*(u_{t-1})\| + \rho^\tau L_{\text{eq}}\|u_{t-1} - u_t\|$$
$$\leq \rho^\tau(\rho^\tau\|x_{t-2}^\tau(u_{t-2}) - x^*(u_{t-2})\| + \rho^\tau L_{\text{eq}}\|u_{t-2} - u_{t-1}\|) + \rho^\tau L_{\text{eq}}\|u_{t-1} - u_t\|$$
$$\leq \rho^\tau\rho^{t-1}\|x_0 - x^*(u_0)\| + L_{\text{eq}}\rho^\tau \sum_{s=1}^{t}\rho^s\|u_{t-s} - u_{t-s-1}\|$$
$$\leq \rho^\tau\rho^{t-1}\|x_0 - x^*(u_0)\| + L_{\text{eq}}L_{\text{u}}\eta\frac{\rho^\tau}{1-\rho}$$

where in the second to last inequality we use the fact that $\rho^\tau \leq \rho$ for any $\tau \geq 1$, and in the last inequality we use the fact that $u_t = u_{t-1} - \eta\nabla_u\ell(u_{t-1}, z_{t-1})$ and $\sup_{(u,x)\in\mathcal{U}\times\mathcal{X}}\|\nabla_u\ell(u,x)\| \leq L_{\text{u}}$. □

**Proof of Deterministic Case.** Now we are ready to prove the deterministic agent case.

*Proof of Proposition I.11.* The proof is the same as for Theorem I.5 up to bounding the bias due to the agents updates.

Given our assumption on the deterministic contractive dynamics of the followers, by Lemma I.12, we have that

$$\mathbb{E}_t\|G_t(u_t) - G_{\text{ps}}(u_t)\|^2 \leq L_\ell\mathbb{E}_t\|\mathcal{A}(x_{t-1}, u_t) - x^*(u_t)\|^2$$
$$\leq L_\ell^2\rho^{2\cdot\tau}\left(\rho^{t-1}\|x_0 - x^*(u_0)\| + \frac{\eta L_{\text{eq}}L_{\text{u}}}{1-\rho}\right)^2$$
$$\leq L_\ell^2\rho^{2\cdot\tau}\underbrace{\left(\rho\|x_0 - x^*(u_0)\| + \frac{\eta L_{\text{eq}}L_{\text{u}}}{1-\rho}\right)^2}_{:=C^2}$$

Therefore

$$P_2 \leq \frac{L_\ell^2}{\nu_2}\left(2\rho^{2\tau}C^2\right) + \frac{2L_\ell^2(1+L_{\text{eq}}^2)\|u_t - u_{t+1}\|^2}{2\nu_2} + \frac{\nu_2\mathbb{E}_t\|u^{\text{ps}} - u_{t+1}\|^2}{2}.$$

Coming back to the bound in (37), we have that

$$\frac{1+2\eta\bar{\alpha}}{2}\mathbb{E}_t\|u_{t+1} - u^{\text{ps}}\|^2 \leq \frac{1}{2}\|u_t - u^{\text{ps}}\|^2 - \frac{1}{2}\mathbb{E}_t\|u_{t+1} - u_t\|^2 + \eta\left(\frac{\sigma^2}{2\nu_1} + \frac{\nu_1\mathbb{E}_t\|u_{t+1} - u_t\|^2}{2}\right)$$
$$+ \eta\left(\frac{L_\ell^2}{\nu_2}\left(2\rho^{2\tau}C^2\right) + \frac{L_\ell^2(1+L_{\text{eq}}^2)\mathbb{E}_t\|u_t - u_{t+1}\|^2}{\nu_2} + \frac{\nu_2\mathbb{E}_t\|u^{\text{ps}} - u_{t+1}\|^2}{2}\right),$$

so that

$$\frac{1+2\eta\bar{\alpha} - \eta\nu_2}{2}\mathbb{E}_t\|u_{t+1} - u^{\text{ps}}\|^2 \leq \frac{1}{2}\|u_t - u^{\text{ps}}\|^2 + \frac{\eta\sigma^2}{2\nu_1} + \eta\frac{2L_\ell^2\rho^{2\tau}C^2}{\nu_2}$$
$$- \frac{1 - 2L_\ell^2(1+L_{\text{eq}}^2)\eta\nu_2^{-1} - \eta\nu_1}{2}\mathbb{E}_t\|u_{t+1} - u_t\|^2.$$

Letting $\nu_1 = \eta^{-1} - \frac{2L_\ell^2(1+L_{\mathrm{eq}}^2)}{\bar{\alpha}}$ and $\nu_2 = \bar{\alpha}$ ensures that the last term on the right is zero. By our assumption that $\eta \leq \frac{\bar{\alpha}}{4L_\ell^2(1+L_{\mathrm{eq}}^2)}$ we have that $\frac{1}{\eta} \geq \frac{4L_\ell^2(1+L_{\mathrm{eq}}^2)}{\bar{\alpha}}$ so that $\nu_1 \geq \frac{1}{2\eta}$; indeed,

$$\nu_1 = \eta^{-1} - \frac{2L_\ell^2(1+L_{\mathrm{eq}}^2)}{\bar{\alpha}} \geq \frac{4L_\ell^2(1+L_{\mathrm{eq}}^2)}{\bar{\alpha}} - \frac{2L_\ell^2(1+L_{\mathrm{eq}}^2)}{\bar{\alpha}} = \frac{2L_\ell^2(1+L_{\mathrm{eq}}^2)}{\bar{\alpha}} = \frac{1}{2\eta}.$$

Hence we have that

$$\frac{1+\eta\bar{\alpha}}{2}\mathbb{E}_t\|u_{t+1} - u^{\mathrm{ps}}\|^2 \leq \frac{1}{2}\|u_t - u^{\mathrm{ps}}\|^2 + \eta^2\sigma^2 + \eta\frac{2L_\ell^2\rho^{2\tau_t}C_t^2}{\bar{\alpha}}$$

Now, choose $\tau$ as stated in the theorem to ensure that $\eta\rho^{2\tau}\frac{2L_\ell^2C^2}{\bar{\alpha}} \leq \eta^2\sigma^2$. Indeed, this inequality is equivalent to

$$\tau\log\rho^2 \leq \log\left(\frac{\bar{\alpha}\eta\sigma^2}{2L_\ell^2C^2}\right) \iff \tau \geq \log\left(\frac{2L_\ell^2C^2}{\bar{\alpha}\eta\sigma^2}\right)\frac{1}{\log(1/\rho^2)},$$

which is precisely the stated lower bound on $\tau$. Hence, we have that

$$\mathbb{E}_t\|u_{t+1} - u^{\mathrm{ps}}\|^2 \leq \frac{1}{1+\eta\bar{\alpha}}\|u_t - u^{\mathrm{ps}}\|^2 + \frac{4}{1+\eta\bar{\alpha}}\eta^2\sigma^2.$$

Recursively iterating the above expression, we have that

$$\begin{aligned}
\mathbb{E}_t\|u_{t+1} - u^{\mathrm{ps}}\|^2 &\leq \frac{1}{1+\eta\bar{\alpha}}\left(\frac{1}{1+\eta\bar{\alpha}}(\|u_{t-1} - u^{\mathrm{ps}}\|^2 + \frac{4}{1+\eta\bar{\alpha}}\eta^2\sigma^2) + \frac{4}{1+\eta\bar{\alpha}}\eta^2\sigma^2\right.\\
&\leq \left(\frac{1}{1+\eta\bar{\alpha}}\right)^t\|u_0 - u^{\mathrm{ps}}\|^2 + 4\eta^2\sigma^2\sum_{s=1}^t\left(\frac{1}{1+\eta\bar{\alpha}}\right)^t\\
&\leq \left(\frac{1}{1+\eta\bar{\alpha}}\right)^t\|u_0 - u^{\mathrm{ps}}\|^2 + 4\eta^2\sigma^2\frac{1}{\eta\bar{\alpha}}
\end{aligned}$$

Given the choice of $\eta \leq \frac{\bar{\alpha}}{4L_\ell^2(1+L_{\mathrm{eq}}^2)}$, we have that

$$\mathbb{E}_t\|u_{t+1} - u^{\mathrm{ps}}\|^2 \leq \left(1 - \frac{\bar{\alpha}\eta}{2}\right)^t\|u_0 - u^{\mathrm{ps}}\|^2 + \frac{4\eta\sigma^2}{\bar{\alpha}},$$

as claimed. $\qquad\square$

### I.3.4. NAÏVE DECISION-MAKER: NON-STATIONARY NON-STRATEGIC ENVIRONMENT

Now we generalize to the case where $\mathcal{D}_e(u)$ now depends on $u$ so that the non-strategic component of the environment is also decision-dependent. This requires the additional assumption that the distribution $\mathcal{D}_e(\cdot)$ is $L_{\mathrm{en}}$-Lipschitz continuous (Assumption I.2), and we also need that $\alpha < L_z(L_{\mathrm{en}} + L_{\mathrm{eq}})$ which is already required for existence and uniqueness of the performatively stable equilibrium.

**Theorem I.13** (Naïve Repeated Gradient Method with Non-Stationary Non-Strategic Environment)**.** *Suppose that Assumptions 2.1, 4.1, I.2 and 4.3 hold, that we have available constant $R > \|x_0 - x^*(u_0)\|^2$ and $B > \|u_0 - u^{\mathrm{ps}}\|^2$, and that $\alpha < L_z(L_{\mathrm{en}} + L_{\mathrm{eq}})$ so that a unique performatively stable equilibrium exists. Further, suppose the decision-maker runs Algorithm 4 with* Alg := RGM *using step-size $\eta \leq \frac{\bar{\alpha}}{4L_\ell^2(1+L_{\mathrm{eq}}^2+L_{\mathrm{en}}^2)}$ where $\bar{\alpha} := \alpha - L_z(L_{\mathrm{en}} + L_{\mathrm{eq}})$, and the agents employ a $\rho$–contracting algorithm $\mathcal{A}$ with $\rho \in [0, 1)$ and $\sigma_{\mathrm{a}} \in (0, \infty)$. Suppose the agents run their $\rho$-contracting algorithm stage-wise via Algorithm 2. In this case, set the epoch length to*

$$\tau = \sum_{k=0}^K T_k = \left\lceil\frac{1}{1-\rho^2}\cdot\log\left(\frac{2\bar{R}}{\varepsilon}\right)\right\rceil + \sum_{k=1}^K\left\lceil\left(\frac{1}{1-2^{-k}\rho^2}\right)\log(4)\right\rceil, \tag{36}$$

*and tolerance $\epsilon_\tau = \eta^2 \sigma^2$ where $K = \left\lceil 1 + \log_2 \left( \frac{\rho^2 c^2 \sigma_a^2}{(1-\rho^2)\varepsilon} \right) \right\rceil$ and*

$$\bar{R} := R + \frac{2c^2 \sigma_a^2}{\beta} + 6 \left( \frac{L_{\text{eq}}^2}{\beta^2} \left( 4B + \frac{\sigma^2}{L_\ell^2 (1 + L_{\text{eq}}^2)} \right) \right).$$

*Then the following estimate holds:*

$$\mathbb{E}_t \| u_{t+1} - u^{\text{ps}} \|^2 \leq \left( 1 - \frac{\bar{\alpha} \eta}{2} \right)^t \| u_0 - u^{\text{ps}} \|^2 + \frac{4\eta \sigma^2}{\bar{\alpha}}.$$

Recall from Corollary G.5, that if the agents run stochastic gradient play in stages then we are able to characterize precisely the number of iterations required to hit a particular specified error tolerance. This is where the epoch length in (41) is derived.

*Proof of Theorem I.5.* Define the following objects:

$$g_t := \nabla_u \ell(u_t, (x_t^\tau(u_t), \xi)), \quad \text{where } \xi \sim \mathcal{D}_e(u_t);$$
$$G_t(u_t) := \mathop{\mathbb{E}}_{\xi \sim \mathcal{D}_e(u_t)} \nabla_u \ell(u_t, (x_t^\tau(u_t), \xi));$$
$$G_\star(u_t) := \mathop{\mathbb{E}}_{\xi \sim \mathcal{D}_e(u_t)} \nabla_u \ell(u_t, (x^*(u_t), \xi));$$
$$G_{\text{ps}}(u_t) := \mathop{\mathbb{E}}_{\xi \sim \mathcal{D}_e(u^{\text{ps}})} \nabla_u \ell(u_t, (x^*(u^{\text{ps}}), \xi))$$

Also note that $\mathbb{E}_t[g_t] = G_t(u_t)$—i.e., the gradient estimate $g_t$ is an unbiased estimate of the time varying expected gradient $G_t$—and

$$u^{\text{ps}} = \operatorname*{argmin}_{u \in \mathcal{U}} \mathop{\mathbb{E}}_{z \sim \mathcal{D}(u^{\text{ps}})} \ell(u, z) \quad \text{so that} \quad \langle G_{\text{ps}}(u^{\text{ps}}), u - u^{\text{ps}} \rangle \geq 0 \ \forall \, u \in \mathcal{U}.$$

Fix two constants $\nu_1, \nu_2 > 0$ to be specified later. Noting that $u_{t+1}$ is the minimizer of the 1-strongly convex function $u \mapsto \frac{1}{2} \| u_t - \eta g_t - u \|^2$ over $\mathcal{U}$, we deduce that

$$\frac{1}{2} \| u_{t+1} - u^{\text{ps}} \|^2 \leq \frac{1}{2} \| u_t - \eta g_t - u^{\text{ps}} \|^2 - \frac{1}{2} \| u_t - \eta_t g_t - u_{t+1} \|^2.$$

Expanding the squares on the right hand side and combining terms yields

$$\frac{1}{2} \| u_{t+1} - u^{\text{ps}} \|^2 \leq \frac{1}{2} \| u_t - u^{\text{ps}} \|^2 - \eta_t \langle g_t, u_{t+1} - u^{\text{ps}} \rangle - \frac{1}{2} \| u_{t+1} - u_t \|^2$$
$$= \frac{1}{2} \| u_t - u^{\text{ps}} \|^2 - \eta \langle g_t, u_t - u^{\text{ps}} \rangle - \frac{1}{2} \| u_{t+1} - u_t \|^2 - \eta \langle g_t, u_{t+1} - u_t \rangle.$$

Using the fact that $\mathbb{E}_t[g_t] = G_t(u_t)$, we successively compute

$$\frac{1}{2} \mathbb{E}_t \| u_{t+1} - u^{\text{ps}} \|^2 \leq \frac{1}{2} \| u_t - u^{\text{ps}} \|^2 - \eta \langle \mathbb{E}_t g_t, u_t - u^{\text{ps}} \rangle - \frac{1}{2} \mathbb{E}_t \| u_{t+1} - u_t \|^2 - \eta \mathbb{E}_t \langle g_t, u_{t+1} - u_t \rangle,$$
$$\leq \frac{1}{2} \| u_t - u^{\text{ps}} \|^2 - \eta \langle G_t(u_t), u_t - u^{\text{ps}} \rangle - \frac{1}{2} \mathbb{E}_t \| u_{t+1} - u_t \|^2 - \eta \mathbb{E}_t \langle g_t, u_{t+1} - u_t \rangle,$$
$$= \frac{1}{2} \| u_t - u^{\text{ps}} \|^2 - \eta \mathbb{E}_t \langle G_\star(u_{t+1}), u_{t+1} - u^{\text{ps}} \rangle - \frac{1}{2} \mathbb{E}_t \| u_{t+1} - u_t \|^2$$
$$+ \eta \underbrace{\mathbb{E}_t \langle g_t - G_t(u_t), u_t - u_{t+1} \rangle}_{P_1} + \eta_t \underbrace{\mathbb{E}_t \langle G_t(u_t) - G_\star(u_{t+1}), u^{\text{ps}} - u_{t+1} \rangle}_{P_2}.$$

Recall that for any $z$, the loss $\ell(u, z)$ is $\alpha$–strongly convex in $u$ so that

$$\langle G_{\text{ps}}(u_{t+1}), u_{t+1} - u^{\text{ps}} \rangle \geq \langle G_{\text{ps}}(u_{t+1}) - G_{\text{ps}}(u^{\text{ps}}), u_{t+1} - u^{\text{ps}} \rangle \geq \alpha \| u_{t+1} - u^{\text{ps}} \|^2$$

Hence, adding and subtracting appropriate terms, we have that

$$-\langle G_\star(u_{t+1}), u_{t+1} - u^{\mathrm{ps}} \rangle = -\langle G_\star(u_{t+1}) - G_{\mathrm{ps}}(u_{t+1}), u_{t+1} - u^{\mathrm{ps}} \rangle - \langle G_{\mathrm{ps}}(u_{t+1}), u_{t+1} - u^{\mathrm{ps}} \rangle.$$

The first term is upper bounded as follows:

$$-\langle G_\star(u_{t+1}) - G_{\mathrm{ps}}(u_{t+1}), u_{t+1} - u^{\mathrm{ps}} \rangle \le L_z(L_{\mathrm{en}} + L_{\mathrm{eq}}) \| u_{t+1} - u^{\mathrm{ps}} \|^2.$$

The second term is upper bounded using $\alpha$–strong convexity of $\ell$ in $u$ as noted above. Therefore, as in the proof for the stationary non-strategic environment, we have that

$$\frac{1 + 2\eta\bar{\alpha}}{2} \mathbb{E}_t \| u_{t+1} - u^{\mathrm{ps}} \|^2 \le \frac{1}{2} \| u_t - u^{\mathrm{ps}} \|^2 - \frac{1}{2} \mathbb{E}_t \| u_{t+1} - u_t \|^2 + \eta(P_1 + P_2). \tag{37}$$

Applying Young's inequality to $P_1$, we have that

$$P_1 \le \frac{\mathbb{E}_t \| g_t - G_t(u_t) \|^2}{2\nu_1} + \frac{\nu_1 \mathbb{E}_t \| u_{t+1} - u_t \|^2}{2} \le \frac{\sigma^2}{2\nu_1} + \frac{\nu_1 \mathbb{E}_t \| u_{t+1} - u_t \|^2}{2}. \tag{38}$$

We have the following upper bound for $P_2$:

$$
\begin{aligned}
P_2 &\le \frac{\mathbb{E}_t \| G_t(u_t) - G_\star(u_{t+1}) \|^2}{2\nu_2} + \frac{\nu_2 \mathbb{E}_t \| u^{\mathrm{ps}} - u_{t+1} \|^2}{2} \\
&\le \frac{2\mathbb{E}_t \| G_t(u_t) - G_\star(u_t) \|^2 + 2\mathbb{E}_t \| G_\star(u_t) - G_\star(u_{t+1}) \|^2}{2\nu_2} + \frac{\nu_2 \mathbb{E}_t \| u^{\mathrm{ps}} - u_{t+1} \|^2}{2} \\
&\le \frac{2\mathbb{E}_t \| G_t(u_t) - G_\star(u_t) \|^2 + 2L_\ell^2(1 + L_{\mathrm{eq}}^2 + L_{\mathrm{en}}^2)\| u_t - u_{t+1} \|^2}{2\nu_2} + \frac{\nu_2 \mathbb{E}_t \| u^{\mathrm{ps}} - u_{t+1} \|^2}{2}.
\end{aligned}
$$

The first term in the first fraction can be bounded as follows:

$$
\begin{aligned}
\mathbb{E}_t \| G_t(u_t) - G_\star(u_t) \|^2 &= \mathbb{E}_t \| \mathbb{E}_{\xi \sim \mathcal{D}_e(u_t)} \nabla_u \ell(u_t, (\mathcal{A}(x_{t-1}, u_t), \xi)) - \mathbb{E}_{\xi \sim \mathcal{D}_e(u_t)} \nabla_u \ell(u_t, (x^*(u_t), \xi)) \|^2 \\
&\le L_\ell^2 \mathbb{E}_t \| \mathcal{A}(x_{t-1}, u_t) - x^*(u_t) \|^2.
\end{aligned}
$$

This shows we have a time varying bias component in our gradient estimator. Recall that

$$\tau = \sum_{k=0}^{K} T_k = \left\lceil \frac{1}{1 - \rho^2} \cdot \log\left( \frac{2R}{\varepsilon} \right) \right\rceil + \sum_{k=1}^{K} \left\lceil \left( \frac{1}{1 - 2^{-k}\rho^2} \right) \log(4) \right\rceil$$

total iterations where $K := \left\lceil 1 + \log_2 \left( \frac{\rho^2 c^2 \sigma_a^2}{(1 - \rho^2)\varepsilon} \right) \right\rceil$. Moreover, the decision-maker sets $\epsilon_\tau = \eta^2 \sigma^2$. Therefore, we deduce that

$$P_2 \le \frac{L_\ell^2 \epsilon_\tau}{\nu_2} + \frac{2L_\ell^2(1 + L_{\mathrm{eq}}^2 + L_{\mathrm{en}}^2)\| u_t - u_{t+1} \|^2}{2\nu_2} + \frac{\nu_2 \mathbb{E}_t \| u^{\mathrm{ps}} - u_{t+1} \|^2}{2}.$$

Coming back to the bound in (37), we have that

$$
\begin{aligned}
\frac{1 + 2\eta\bar{\alpha}}{2} \mathbb{E}_t \| u_{t+1} - u^{\mathrm{ps}} \|^2 &\le \frac{1}{2} \| u_t - u^{\mathrm{ps}} \|^2 - \frac{1}{2} \mathbb{E}_t \| u_{t+1} - u_t \|^2 + \eta \left( \frac{\sigma^2}{2\nu_1} + \frac{\nu_1 \mathbb{E}_t \| u_{t+1} - u_t \|^2}{2} \right) \\
&\quad + \eta \left( \frac{L_\ell^2 \epsilon_\tau}{\nu_2} + \frac{2L_\ell^2(1 + L_{\mathrm{eq}}^2 + L_{\mathrm{en}}^2)\| u_t - u_{t+1} \|^2}{2\nu_2} + \frac{\nu_2 \mathbb{E}_t \| u^{\mathrm{ps}} - u_{t+1} \|^2}{2} \right)
\end{aligned}
$$

so that

$$
\begin{aligned}
\frac{1 + 2\eta\bar{\alpha} - \eta\nu_2}{2} \mathbb{E}_t \| u_{t+1} - u^{\mathrm{ps}} \|^2 &\le \frac{1}{2} \| u_t - u^{\mathrm{ps}} \|^2 + \frac{\eta\sigma^2}{2\nu_1} + \eta \frac{L_\ell^2 \epsilon_\tau}{\nu_2} \\
&\quad - \frac{1 - 2L_\ell^2(1 + L_{\mathrm{eq}}^2 + L_{\mathrm{en}}^2)\eta\nu_2^{-1} - \eta\nu_1}{2} \mathbb{E}_t \| u_{t+1} - u_t \|^2.
\end{aligned}
$$

Letting $\nu_1 = \eta^{-1} - \frac{2L_\ell^2(1+L_{\text{eq}}^2+L_{\text{en}}^2)}{\bar{\alpha}}$ and $\nu_2 = \bar{\alpha}$ ensures that the last term on the right is zero. By our assumption that $\eta \leq \frac{\alpha}{4L_\ell^2(1+L_{\text{eq}}^2+L_{\text{en}}^2)}$ we have that $\frac{1}{\eta} \geq \frac{4L_\ell^2(1+L_{\text{eq}}^2+L_{\text{en}}^2)}{\bar{\alpha}}$ so that $\nu_1 \geq \frac{1}{2\eta}$; indeed, we claim that

$$\frac{1}{\eta} - \frac{2L_\ell^2(1+L_{\text{eq}}^2+L_{\text{en}}^2)}{\bar{\alpha}} \geq \frac{1}{2\eta}.$$

Rearranging, this is equivalent to showing that

$$1 - \eta\frac{2L_\ell^2(1+L_{\text{eq}}^2+L_{\text{en}}^2)}{\bar{\alpha}} \geq \frac{1}{2}.$$

Now we can lower bound the left-hand side as follows:

$$1 - \eta\frac{2L_\ell^2(1+L_{\text{eq}}^2+L_{\text{en}}^2)}{\bar{\alpha}} \geq 1 - \frac{4L_\ell^2(1+L_{\text{eq}}^2+L_{\text{en}}^2)}{\bar{\alpha}}\frac{2L_\ell^2(1+L_{\text{eq}}^2+L_{\text{en}}^2)}{\bar{\alpha}} = 1 - \frac{1}{2} = \frac{1}{2}.$$

That shows that claim. Hence we have that

$$\frac{1+\eta\bar{\alpha}}{2}\mathbb{E}_t\|u_{t+1} - x^{\text{ps}}\|^2 \leq \frac{1}{2}\|u_t - x^{\text{ps}}\|^2 + \eta^2\sigma^2 + \frac{\epsilon_\tau}{2}$$

Then since $\epsilon_\tau = \eta^2\sigma^2$, we have that

$$\mathbb{E}_t\|u_{t+1} - x^{\text{ps}}\|^2 \leq \frac{1}{1+\eta\bar{\alpha}}\|u_t - x^{\text{ps}}\|^2 + \frac{4}{1+\eta\bar{\alpha}}\eta^2\sigma^2$$

Recursively iterating the above expression, we have that

$$\mathbb{E}_t\|u_{t+1} - u^{\text{ps}}\|^2 \leq \frac{1}{1+\eta\bar{\alpha}}\left(\frac{1}{1+\eta\bar{\alpha}}(\|u_{t-1} - u^{\text{ps}}\|^2 + \frac{4}{1+\eta\bar{\alpha}}\eta^2\sigma^2) + \frac{4}{1+\eta\bar{\alpha}}\eta^2\sigma^2\right)$$

$$\leq \left(\frac{1}{1+\eta\bar{\alpha}}\right)^t\|u_0 - u^{\text{ps}}\|^2 + 4\eta^2\sigma^2\sum_{s=1}^t\left(\frac{1}{1+\eta\bar{\alpha}}\right)^t$$

$$\leq \left(\frac{1}{1+\eta\bar{\alpha}}\right)^t\|u_0 - u^{\text{ps}}\|^2 + 4\eta^2\sigma^2\frac{1}{\eta\bar{\alpha}}$$

Given the choice of $\eta \leq \frac{\bar{\alpha}}{4L_\ell^2(1+L_{\text{eq}}^2+L_{\text{en}}^2)}$, we have that

$$\mathbb{E}_t\|u_{t+1} - u^{\text{ps}}\|^2 \leq \left(\frac{1}{1+\eta\bar{\alpha}}\right)^t\|u_0 - u^{\text{ps}}\|^2 + \frac{\sigma^2}{L_\ell^2(1+L_{\text{eq}}^2+L_{\text{en}}^2)} \leq \left(1 - \frac{\bar{\alpha}\eta}{2}\right)^t\|u_0 - u^{\text{ps}}\|^2 + \frac{4\eta\sigma^2}{\bar{\alpha}}.$$

The choice of constant $\bar{R}$ follows from the same proof as in Theorem I.5. This completes the proof. □

Note that $L_\ell^2(1+L_{\text{eq}}^2+L_{\text{en}}^2)$ can be replaced with $L_u^2 + L_z^2(L_{\text{eq}}^2+L_{\text{en}}^2)$ for more precise Lipschitz constants.

## J. Strategic Decision-Maker

In this appendix section, we put all the formal analysis for the strategic decision-maker. Let us introduce some needed notation. Let

$$\mathcal{L}_t^\delta(u_t) = \frac{d}{\delta}\mathbb{E}_{v\sim\mathbb{B}^d}\left[\mathbb{E}_{\xi\sim\mathcal{D}_e}[\ell(u_t + \delta v_t, \mathcal{A}(x_t, u_t + \delta v_t), +\xi)]\right]$$

denote the smoothed expected loss at time $t$, and let

$$\mathcal{L}^\delta(u) = \frac{d}{\delta}\mathbb{E}_{v\sim\mathbb{B}^d}\left[\mathbb{E}_{\xi\sim\mathcal{D}_e}[\ell(u_t + \delta v_t, x^*(u_t + \delta v_t) + \xi)]\right]$$

denote the smoothed expected risk. The smoothed expected risk is evaluated when the strategic agents are at the Nash equilibrium $x^*(u_t + \delta v_t)$ for the reported value $u_t + \delta v_t$. The estimate $\widehat{g}_t$ is an unbiased estimate of $\nabla\mathcal{L}_t^\delta$—i.e., $\mathbb{E}_{v\sim\mathbb{B}^d}[\widehat{g}_t] = \nabla\mathcal{L}_t^\delta(u_t)$.

## J.1. Technical Lemmas

The following series of lemmas allow us to bound the error between the true gradient of $\mathcal{L}(u)$ and the zeroth-order gradient estimate $g_t$.

**Lemma J.1.** *Suppose that Assumptions 2.1 and 4.1 hold. The smoothed expected risk $\mathcal{L}^\delta(u)$ satisfies $\|\nabla\mathcal{L}(u) - \nabla\mathcal{L}^\delta(u)\| \leq L\delta$, where $L := L_\ell(1 + L_{\text{eq}})$.*

*Proof.* For any points $u, u' \in \mathcal{U}$, we successively estimate

$$\|\nabla\mathcal{L}^\delta(u) - \nabla\mathcal{L}^\delta(u')\| \leq \underset{w\sim\mathbb{B}}{\mathbb{E}}\|\nabla\mathcal{L}(u + \delta w) - \nabla\mathcal{L}(u' + \delta w)\| \leq L\|u - u'\|.$$

Therefore $\nabla\mathcal{L}^\delta$ is $L$-Lipschitz continuous. Next, we have that

$$\|\nabla\mathcal{L}(u) - \nabla\mathcal{L}^\delta(u)\| \leq \underset{w\sim\mathbb{B}}{\mathbb{E}}\|\nabla\mathcal{L}(\delta w) - \nabla\mathcal{L}(u)\| \leq L\delta\underset{w\sim\mathbb{B}}{\mathbb{E}}\|w\| \leq L\delta,$$

which concludes the proof. $\qquad\square$

**Lemma J.2.** *Under Assumptions 4.1 and 4.8, by choosing $\delta \leq \frac{c\bar{\alpha}}{L_{\text{H}}}$, for any $c \in (0, 1)$ the smoothed decision-dependent risk $\mathcal{L}^\delta(u)$ is $(1 - c)\bar{\alpha}$–strongly convex.*

*Proof.* We first define $h(u) := \nabla\mathcal{L}^\delta(u) - \nabla\mathcal{L}(u)$. Observe that $\nabla h(u) = \mathbb{E}_{w\sim\mathbb{B}}[\nabla^2\mathcal{L}(u + \delta w) - \nabla^2\mathcal{L}(u)]$. Since $u \mapsto \nabla^2\mathcal{L}(u)$ is $L_{\text{H}}$-Lipschitz continuous, we deduce that

$$\|\nabla h(u)\|_{\text{op}} \leq \underset{w\sim\mathbb{B}}{\mathbb{E}}\|\nabla^2\mathcal{L}(u + \delta w) - \nabla^2\mathcal{L}(u)\|_{\text{op}} \leq \delta L_{\text{H}}\underset{w\sim\mathbb{B}}{\mathbb{E}}\|w\| \leq \delta L_{\text{H}}.$$

We therefore compute that

$$\langle\nabla\mathcal{L}^\delta(u) - \nabla\mathcal{L}^\delta(u'), u - u'\rangle = \langle\nabla\mathcal{L}(u) - \nabla\mathcal{L}(u')\rangle + \langle h(u) - h(u'), u - u'\rangle \geq (\bar{\alpha} - L_{\text{H}}\delta)\|u - u'\|^2,$$

which concludes the proof of the first statement. $\qquad\square$

Now, let $u^*$ be the optimal point for $\mathcal{L}$ over $\mathcal{U}$, and let $u^\delta$ be the optimal point of $\mathcal{L}^\delta$ on $(1 - \delta)\mathcal{U}$.

**Lemma J.3.** *Suppose Assumptions 4.1 and 4.8 hold. Choose any $\delta < \min\{r, \frac{\bar{\alpha}}{L_{\text{H}}}\}$. Then the estimate holds:*

$$\|u^\delta - u^*\| \leq \frac{\delta L}{\bar{\alpha}} + \left(\frac{\delta L}{\bar{\alpha}} + \delta\right)\|u^*\|.$$

*Proof.* There are two sources of perturbation: one replacing $\mathcal{U}$ with $(1 - \delta)\mathcal{U}$ and the other replacing $\mathcal{L}$ with $\mathcal{L}^\delta$.

Set $\phi = 1 - \delta$ and let $\tilde{u}$ be the optimal point for $\mathcal{L}$ on $\phi\mathcal{U}$. Thus $0 \in \nabla\mathcal{L}(\tilde{u}) + N_{\phi\mathcal{U}}(\tilde{u})$. Then

$$\|u^* - u^\delta\| \leq \|u^* - \tilde{u}\| + \|\tilde{u} - u^\delta\|.$$

The first term is bounded as

$$\bar{\alpha}\|\tilde{u} - \phi u^*\| \leq \text{dist}(0, \nabla\mathcal{L}(\phi u^*) + N_{\phi\mathcal{U}}(\phi u^*)$$

since $u \mapsto \nabla\mathcal{L}(u) + N_{\phi\mathcal{U}}(u)$ is $\bar{\alpha}$–strongly convex. For the second term, since $u^*$ is optimal, we have that $0 \in \nabla\mathcal{L}(u^*) + N_{\mathcal{U}}(u^*)$. Since $N_{\phi\mathcal{U}}(\phi u^*) = N_{\mathcal{U}}(u^*)$, we have that

$$\text{dist}(0, \nabla\mathcal{L}(\phi u^*) + N_{\phi\mathcal{U}}(\phi u^*)) = \text{dist}(0, \nabla\mathcal{L}(\phi u^*) + N_{\mathcal{U}}(u^*)) \leq \|\nabla\mathcal{L}(\phi u^*) - \nabla\mathcal{L}(u^*)\| \leq \delta L\|u^*\|.$$

We therefore have that

$$\|u^* - \tilde{u}\| \leq \|\tilde{u} - \phi u^*\| + \delta\|u^*\| \leq \delta\left(1 + \frac{L}{\bar{\alpha}}\right)\|u^*\|.$$

Since $\tilde{u}$ is optimal, we have that

$$\langle-\nabla\mathcal{L}(\tilde{u}), u - \tilde{u}\rangle \leq 0, \quad \forall u \in \phi\mathcal{U}.$$

Analogously, since $u^\delta$ is also optimal we have that

$$\langle -\nabla \mathcal{L}^\delta(u^\delta), u - u^\delta \rangle \leq 0, \quad \forall u \in \phi \mathcal{U}.$$

Therefore

$$
\begin{aligned}
\bar{\alpha} \|\tilde{u} - u^\delta\|^2 &\leq \langle \nabla \mathcal{L}(\tilde{u}) - \nabla \mathcal{L}(u^\delta), \tilde{u} - u^\delta \rangle \\
&\leq \langle \nabla \mathcal{L}^\delta(u^\delta) - \nabla \mathcal{L}(u^\delta), \tilde{u} - u^\delta \rangle \\
&\leq \|\nabla \mathcal{L}^\delta(u^\delta) - \nabla \mathcal{L}(u^\delta)\| \|\tilde{u} - u^\delta\| \\
&\leq L\delta \|\tilde{u} - u^\delta\|.
\end{aligned}
$$

Combining the bounds yields the claim. $\qquad \square$

The next lemma bounds the error between the converged strategies $x_t^\tau(u_t)$ and the equilibrium $x^*(u_t)$ as a function of the previous iterates.

**Lemma J.4.** *Suppose the agents are employing deterministic algorithms satisfying Definition 2.3 with $\rho \in [0, 1)$ and $\sigma_{\mathsf{a}} = 0$. Under Assumptions 2.1, 4.1, and 4.8, the estimate holds:*

$$\|\mathcal{A}(x_t, u_t) - x^*(u_t)\| \leq \rho^\tau \left( \rho^{t-1} \|x_0 - x^*(u_0)\| + \rho L_{\mathsf{eq}} \eta_0 \frac{\ell_* d}{\delta(1 - \rho)} \right)$$

*Proof.* Given Definition 2.3, we have that

$$\|x_t^\tau(u_t) - x^*(u_t)\| \leq \rho \|x_t^{\tau-1}(u_t) - x^*(u_t)\|.$$

Iterating this expression we have that

$$\|x_t^\tau(u_t) - x^*(u_t)\| \leq \rho^\tau \|x_{t-1}^{\tau_{t-1}}(u_{t-1}) - x^*(u_t)\|$$

Adding and subtracting appropriate terms we have that

$$
\begin{aligned}
\|x_t^\tau(u_t) - x^*(u_t)\| &\leq \rho^\tau \|x_{t-1}^{\tau_{t-1}}(u_{t-1}) - x^*(u_{t-1}) + x^*(u_{t-1}) - x^*(u_t)\| \\
&\leq \rho^\tau \|x_{t-1}^{\tau_{t-1}}(u_{t-1}) - x^*(u_{t-1})\| + \rho^\tau L_{\mathsf{eq}} \|u_{t-1} - u_t\|
\end{aligned}
$$

Continuing in this fashion we have that

$$
\begin{aligned}
\|x_t^\tau(u_t) - x^*(u_t)\| &\leq \rho^\tau \|x_{t-1}^{\tau_{t-1}}(u_{t-1}) - x^*(u_{t-1})\| + \rho^\tau L_{\mathsf{eq}} \|u_{t-1} - u_t\| \\
&\leq \rho^\tau (\rho^{\tau_{t-1}} \|x_{t-2}^{\tau_{t-2}}(u_{t-2}) - x^*(u_{t-2})\| + \rho^{\tau_{t-1}} L_{\mathsf{eq}} \|u_{t-2} - u_{t-1}\|) + \rho^\tau L_{\mathsf{eq}} \|u_{t-1} - u_t\| \\
&\leq \rho^\tau \rho^{t-1} \|x_0 - x^*(u_0)\| + L_{\mathsf{eq}} \rho^\tau \sum_{s=1}^t \rho^s \|u_{t-s} - u_{t-s-1}\|,
\end{aligned}
$$

where in the last inequality we use the fact that $\rho^\tau \leq \rho$ for any $\tau \geq 1$. Using the update for the decision maker, we have that

$$
\begin{aligned}
\|x_t^\tau(u_t) - x^*(u_t)\| &\leq \rho^\tau \left( \rho^{t-1} \|x_1 - x^*(u_1)\| + \rho L_{\mathsf{eq}} \sum_{s=0}^{t-1} \rho^s \eta_{t-1-s} \frac{\ell_* d}{\delta} \right) \\
&\leq \rho^\tau \left( \rho^{t-1} \|x_0 - x^*(u_0)\| + \rho L_{\mathsf{eq}} \eta_0 \frac{\ell_* d}{\delta(1 - \rho)} \right),
\end{aligned}
$$

where we used the fact that $\eta_0 \geq \eta_{t-i}$ for all $t \geq 1$. This concludes the proof. $\qquad \square$

Next, we use Lemma J.4 to bound the error between the gradient of the smoothed expected risk $\mathcal{L}^\delta(u)$ and the smoothed loss $\mathcal{L}_t^\delta(u_t)$.

**Lemma J.5.** *Suppose the agents are employing deterministic algorithms satisfying Definition 2.3 with $\rho \in [0, 1)$ and $\sigma_{\mathsf{a}} = 0$. Under Assumptions 2.1, 4.1, and 4.8, the smoothed expected risk and the smoothed loss satisfy*

$$\|\nabla \mathcal{L}_t^{\delta}(u_t) - \nabla \mathcal{L}^{\delta}(u_t)\|^2 \leq L_{\ell}^2 \left( \rho^{\tau} \left( \rho^{t-1} \|x_0 - x^*(u_0)\| + \frac{L_{\mathsf{eq}} \eta_0 \ell_* d}{\delta(1-\rho)} \right) \right)^2.$$

*Proof.* We have that

$$\|\nabla \mathcal{L}_t^{\delta}(u_t) - \nabla \mathcal{L}^{\delta}(u_t)\| \leq \mathop{\mathbb{E}}_{v \sim \mathbb{B}^d} \left\| \mathop{\mathbb{E}}_{\xi \sim \mathcal{D}_o} [\nabla \ell(u_t + \delta v_t, x_t^{\tau}(u_t + \delta v_t) + \xi) \right.$$
$$\left. - \nabla \ell(u_t + \delta v, x^*(u_t + \delta v) + \xi)] \right\|$$
$$\leq L_{\ell} \mathop{\mathbb{E}}_{v \sim \mathbb{B}^d} \|x_t^{\tau}(u_t + \delta v_t) - x^*(u_t + \delta v_t)\|.$$

Hence applying Lemma J.4 gives the result. $\square$

The above lemmas give us our main result, which establishes that the decision-maker's updates converge to the optimal parameter $u^* \in \mathcal{U}$ (and correspondingly, the agents' updates converge to the Nash equilibrium $x^*(u^*)$).

Let us define a useful quantity that we will use in the remaining proof:

$$C_t(\sigma_{\mathsf{a}}) := \left( \rho^{t-1} \|x_0 - x^*(u_0)\| + \frac{L_{\mathsf{eq}} \eta_0 \ell_* d}{\delta(1-\rho)} + \frac{\rho \sigma_{\mathsf{a}} c}{(1-\rho)^2} \right), \tag{39}$$

so that

$$\|\nabla \mathcal{L}_t^{\delta}(u_t) - \nabla \mathcal{L}^{\delta}(u_t)\|^2 \leq L_{\ell}^2 \left( \rho^{\tau} C_t(\sigma_{\mathsf{a}}) + \frac{\rho \sigma_{\mathsf{a}} c}{1-\rho} \right)^2.$$

### J.2. Derivative Free Method for Stochastic $\rho$-Contracting Agents

We now state a formal version of Theorem 4.9 and provide a proof.

**Theorem J.6** (Formal Statement of Theorem 4.9). *Suppose that Assumptions 2.1, 4.1, and 4.8 hold, and that we have available a constant $R > \|x_0 - x^*(u_0)\|^2$. Further, suppose the decision-maker runs Algorithm 1 with $\texttt{Alg} := \texttt{DFM}$ using step-size $\eta_t = \frac{4}{\bar{\alpha}(t+1)}$, query radius $\delta < \min\{b, \frac{\bar{\alpha}}{L_{\mathrm{H}}}\}$, and the agents employ a $\rho$–contracting algorithm $\mathcal{A}$ with $\rho \in [0, 1)$ and $\sigma_{\mathsf{a}} \in (0, \infty)$. Suppose the agents run their $\rho$-contracting algorithm stage-wise via Algorithm 2. In this case, set tolerance $\epsilon_{\tau} = \frac{1}{\delta(t+1)}$, constant $c = 16(\ell_*^2 d^2 + 1)$, and the epoch length to*

$$\tau = \sum_{k=0}^{K} T_k = \left\lceil \frac{1}{1-\rho^2} \cdot \log \left( \frac{2\bar{R}}{\epsilon_{\tau}} \right) \right\rceil + \sum_{k=1}^{K} \left\lceil \left( \frac{1}{1-2^{-k}\rho^2} \right) \log(4) \right\rceil, \tag{40}$$

*where $K = \left\lceil 1 + \log_2 \left( \frac{\rho^2 c^2 \sigma_{\mathsf{a}}^2}{(1-\rho^2)\epsilon_{\tau}} \right) \right\rceil$ and*

$$\bar{R} := R + \frac{2c^2 \sigma_{\mathsf{a}}^2}{\beta} + 6 \left( \frac{L_{\mathsf{eq}}^2}{\beta^2} \left( \frac{\max\{4\bar{\alpha}^2 \delta^2 B^2, 16(\ell_*^2 d^2 + 1)\}}{\delta^2 \bar{\alpha}^2} + 4\delta^2 \left( \left(1 + \frac{L}{\bar{\alpha}}\right) B + \frac{L}{\bar{\alpha}} \right) \right) \right).$$

*Then the following estimate holds:*

$$\mathbb{E}\|u_t - u^*\|^2 \leq \frac{\max\{2\bar{\alpha}^2 \delta^2 \|u_0 - u^*\|^2, c\}}{\delta^2 \bar{\alpha}^2 (t+1)^{\beta}} + 2\delta^2 \left( \left(1 + \frac{L}{\bar{\alpha}}\right) \|u^*\| + \frac{L}{\bar{\alpha}} \right).$$

The proof follows a similar structure to that of Theorem 4.4 (see Theorem I.5 for the longer version).

*Proof.* Consider the error $\|u_{t+1} - u^*\|^2$. Add and subtract $u^\delta$, and apply the triangle inequality and Lemma J.2 to get the following estimate:

$$\frac{1}{2}\|u_{t+1} - u^*\|^2 \le \|u_{t+1} - u^\delta\|^2 + \|u_\delta - u^*\|^2 \le \|u_{t+1} - u^\delta\|^2 + \left(\frac{\delta L}{\bar{\alpha}} + \left(\frac{\delta L}{\bar{\alpha}} + \delta\right)\|u^*\|\right)^2.$$

Now, to bound the error $\|u_{t+1} - u^\delta\|^2$, we note by the nonexpansiveness of the projection mapping that

$$\begin{aligned}
\mathbb{E}[\|u_{t+1} - u^\delta\|^2] &\le \mathbb{E}[\|u_t - u^\delta - \eta_t g_t\|^2] \\
&\le \mathbb{E}[\|u_t - u^\delta\|^2 - 2\eta_t \mathbb{E}\langle g_t, u_t - u^\delta\rangle + \eta_t^2 \mathbb{E}\|g_t\|^2] \\
&\le \mathbb{E}[\|u_t - u^\delta\|^2 - 2\eta_t \mathbb{E}\langle \nabla\mathcal{L}_t^\delta(u_t), u_t - u^\delta\rangle + \eta_t^2 \mathbb{E}\|g_t\|^2]
\end{aligned}$$

where we use the fact that $\mathbb{E}[g_t] = \nabla\mathcal{L}_t^\delta(u_t)$ in the last inequality, and the expectation is taken over the randomness in $\xi$ and $v_t$ up to time $t$.

Next, we add and subtract $\nabla\mathcal{L}^\delta(u_t)$ from the middle term to get that

$$\begin{aligned}
\langle \nabla\mathcal{L}_t^\delta(u_t), u_t - u^\delta\rangle &= \langle \nabla\mathcal{L}^\delta(u_t), u_t - u^\delta\rangle + \langle \nabla\mathcal{L}_t^\delta(u_t) - \nabla\mathcal{L}^\delta(u_t), u_t - u^\delta\rangle \\
&\le \frac{\bar{\alpha}}{2}\|u_t - u^\delta\|^2 + \langle \mathcal{L}_t^\delta(u_t) - \nabla\mathcal{L}^\delta(u_t), u_t - u^\delta\rangle,
\end{aligned}$$

where we have used the fact that $\mathcal{L}^\delta(x)$ is $(1-c)\bar{\alpha}$–strong convex with $c = 1/2$ (Lemma J.2). Hence, we deduce that

$$\begin{aligned}
\mathbb{E}[\|u_{t+1} - u^\delta\|^2] &\le \mathbb{E}[\|u_t - u^\delta - \eta_t g_t\|^2], \\
&\le \mathbb{E}[\|u_t - u^\delta\|^2] - 2\eta_t \, \mathbb{E}\langle \nabla\mathcal{L}^\delta(u_t), u_t - u^\delta\rangle \\
&\quad - 2\eta_t \, \mathbb{E}\langle \nabla\mathcal{L}_t^\delta(u_t) - \nabla\mathcal{L}^\delta(u_t), u_t - u^\delta\rangle + \eta_t^2 \, \mathbb{E}\|g_t\|^2, \\
&\le (1 - \eta_t\bar{\alpha})\, \mathbb{E}[\|u_t - u^\delta\|^2] - 2\eta_t \, \mathbb{E}\langle \nabla\mathcal{L}_t^\delta(u_t) - \nabla\mathcal{L}^\delta(u_t), u_t - u^\delta\rangle + \eta_t^2 \frac{\ell_*^2 d^2}{2\delta^2}.
\end{aligned}$$

The agents are running stage-based $\rho$–contracting algorithms. For the moment suppose our choice of $\bar{R}$ is correct. In this case, we set $\epsilon_\tau = (\delta^2(t+1))^{-1}$ and choose

$$\tau = \sum_{k=0}^{K} T_k = \left\lceil\left(1 + \frac{2L_a^2}{\mu^2}\right)\log\left(\frac{2\bar{R}}{\epsilon_\tau}\right)\right\rceil + \sum_{k=1}^{K}\left\lceil\left(1 + \frac{2^{k+1}L_a^2}{\mu^2}\right)\log(4)\right\rceil,$$

and $K = \left\lceil 1 + \log_2\left(\frac{\sigma_a^2}{L_a^2 \epsilon_\tau}\right)\right\rceil$. Then we have that

$$\begin{aligned}
\mathbb{E}\|u_{t+1} - u^\delta\|^2 &\le \left(1 - \frac{\bar{\alpha}\eta_t}{2}\right)\mathbb{E}[\|u_t - u^\delta\|^2] + \frac{8\ell_*^2 d^2}{\delta^2\alpha^2(t+1)^2} + \frac{8}{\bar{\alpha}^2}\frac{1}{\delta^2(t+1)^2} \\
&\le \left(1 - \frac{2}{t+1}\right)\mathbb{E}[\|u_t - u^\delta\|^2] + (\ell_*^2 d^2 + 1)\frac{8}{\bar{\alpha}^2\delta^2(t+1)^2}.
\end{aligned}$$

Next, we claim that

$$\mathbb{E}\|u_t - u^\delta\|^2 \le \frac{\max\{\bar{\alpha}^2\delta^2\|u_0 - u^\delta\|^2, 8(\ell_*^2 d^2 + 1)\}}{\delta^2\bar{\alpha}^2(t+1)}.$$

To see this, let $D_t = \mathbb{E}[\|u_t - u^\delta\|^2]$ so that we need to show the above claim given that

$$D_{t+1} \le \left(1 - \frac{2}{t+1}\right)D_t + (\ell_*^2 d^2 + 1)\frac{8}{\bar{\alpha}^2\delta^2(t+1)^2}.$$

Clearly the claim holds for $t = 1$. Suppose it holds for some fixed $t > 1$. Then we have that

$$
\begin{aligned}
D_{t+1} &\le \left(1 - \frac{2}{t+1}\right) D_t + (\ell_*^2 d^2 + 1) \frac{8}{\bar{\alpha}^2 \delta^2 (t+1)^2} \\
&\le \left(1 - \frac{2}{t+1}\right) \frac{8(\ell_*^2 d^2 + 1)}{\delta^2 \bar{\alpha}^2 (t+1)} + (\ell_*^2 d^2 + 1) \frac{8}{\bar{\alpha}^2 \delta^2 (t+1)^2} \\
&\le \frac{8(\ell_*^2 d^2 + 1)}{\delta^2 \bar{\alpha}^2} \left(\frac{1}{(t+1)} - \frac{1}{(t+1)^2}\right) \\
&\le \frac{8(\ell_*^2 d^2 + 1)}{\delta^2 \bar{\alpha}^2} \frac{1}{(t+2)}
\end{aligned}
$$

Therefore

$$
\mathbb{E} \|u_t - u^*\|^2 \le \frac{\max\{4\bar{\alpha}^2 \delta^2 \|u_0 - u^\delta\|^2, 16(\ell_*^2 d^2 + 1)\}}{\delta^2 \bar{\alpha}^2 (t+1)} + 2\delta^2 \left(\left(1 + \frac{L}{\bar{\alpha}}\right) \|u^*\| + \frac{L}{\bar{\alpha}}\right).
$$

**Finding the constant $\bar{R}$.**   What remains is to show that we set $\bar{R}$ correctly. The proof follows an analogous proof to the repeated gradient method (Theorem I.5). Recall that

$$
\mathbb{E} \|u_t - u^*\|^2 \le \frac{\max\{4\bar{\alpha}^2 \delta^2 B^2, 16(\ell_*^2 d^2 + 1)\}}{\delta^2 \bar{\alpha}^2 (t+1)} + 2\delta^2 \left(\left(1 + \frac{L}{\bar{\alpha}}\right) B + \frac{L}{\bar{\alpha}}\right).
$$

Hence, to obtain a bound on the $\mathbb{E} \|x_{t-1} - x^*(u_t)\|^2$, we simply observe that for any $t \ge 1$, we have

$$
\mathbb{E} \|u_t - u_{t+1}\|^2 \le \frac{\max\{4\bar{\alpha}^2 \delta^2 B^2, 16(\ell_*^2 d^2 + 1)\}}{\delta^2 \bar{\alpha}^2} + 4\delta^2 \left(\left(1 + \frac{L}{\bar{\alpha}}\right) B + \frac{L}{\bar{\alpha}}\right)
$$

Moreover, this implies that

$$
\max_{k \le t} \mathbb{E} \|u_k - u_{k-1}\|^2 \le \frac{\max\{4\bar{\alpha}^2 \delta^2 B^2, 16(\ell_*^2 d^2 + 1)\}}{\delta^2 \bar{\alpha}^2} + 4\delta^2 \left(\left(1 + \frac{L}{\bar{\alpha}}\right) B + \frac{L}{\bar{\alpha}}\right) \quad \text{for all } t,
$$

so that

$$
\bar{R} := R + \frac{2c^2 \sigma_a^2}{\beta} + 6 \left(\frac{L_{\mathsf{eq}}^2}{\beta^2} \left(\frac{\max\{4\bar{\alpha}^2 \delta^2 B^2, 16(\ell_*^2 d^2 + 1)\}}{\delta^2 \bar{\alpha}^2} + 4\delta^2 \left(\left(1 + \frac{L}{\bar{\alpha}}\right) B + \frac{L}{\bar{\alpha}}\right)\right)\right)
$$

where $\beta = (1 - \rho^2)$. This completes the proof.

$\square$

Observe that we can replace the expected value bound on the drift with a high probability statement as in Theorem I.9. Essentially, where we have bounds like $\mathbb{E} \|x_{t-1} - x^*(u_t)\|^2 \le \epsilon_\tau$, we replace them with bounds $\|x_{t-1} - x^*(u_t)\|^2 \le \epsilon_\tau \cdot \log(e/\delta)$ which hold with probability at least $(1 - \delta)$ for any selected $\delta \in (0, 1)$.

Theorem J.6 allows us to obtain the following convergence guarantee.

**Corollary J.7.** *Suppose the assumptions of Theorem 4.9 hold. Fix target accuracy $\varepsilon < 4b^2 \left(\left(1 + \frac{L}{\bar{\alpha}}\right) B + \frac{L}{\bar{\alpha}}\right)^2$ and set $\delta = \bar{\alpha}\sqrt{\varepsilon/4}/((\bar{\alpha} + L)B + L)$ and $\eta_t = 4/(\bar{\alpha}(t+1))$. The iterates $(u_t, x_t)$ converge to an approximate Stackelberg equilibrium: $\mathbb{E}[\|u_t - u^*\|^2] \le \varepsilon$ and $\mathbb{E}[\|x_t - x^*(u^*)\|^2] \le 2(\epsilon_\tau + L_{\mathsf{eq}}\varepsilon)$ hold for all $t \ge 16 \max\{\bar{\alpha}^4 \varepsilon B^2, 8(\ell_*^2 d^2 + 1)((\bar{\alpha} + L)B + L)^2\}/(\bar{\alpha}^4 \varepsilon^2)$.*

In the proceeding corollary, the lower bound on $t$ is in terms of the number of epochs. In terms of total iterations $\sum_{s=1}^t \tau_s$ the rate is $\mathcal{O}\left(\frac{d^2}{\varepsilon^2}\left(\log\left(1/\epsilon_\tau\right) + \sigma_a^2/\epsilon_\tau\right)\right)$. This rate is equivalent to $\widetilde{\mathcal{O}}(T^{-1/2})$ in terms of iteration complexity where $T = \sum_{s=1}^t \tau_s$.

*Proof of Corollary 4.10.* The assumed upper bound on $\varepsilon$ directly implies that $\delta \leq \bar{\alpha}/(2L_{\mathtt{H}})$ and $\delta < b$. Applying Theorem J.6 yields

$$\mathbb{E}[\|x_t - x^*\|^2] \leq \frac{\max\{2\bar{\alpha}^2\delta^2\|u_0 - u^\delta\|^2, c\}}{\delta^2\bar{\alpha}^2(t+1)} + 2\delta^2\left(\left(1 + \frac{L}{\bar{\alpha}}\right)\|u^*\| + \frac{L}{\bar{\alpha}}\right)^2$$
$$\leq \frac{\max\{8\bar{\alpha}^4\varepsilon B^2, 4c((\bar{\alpha}+L)B+L)^2\}}{\varepsilon\bar{\alpha}^4(t+1)} + \frac{\varepsilon}{2}.$$

Setting the right-hand side to $\varepsilon$ and solving for $t$, concludes the proof. $\qquad\square$

We can also specialize the result to the case where agents run stochastic gradient play (in order to better understand the constants).

**Corollary J.8.** *Suppose that Assumptions 2.1, 4.1, and 4.8 hold, and that we have available constants $R > \|x_0 - x^*(u_0)\|^2$ and $B > \|u_0 - u^*\|^2$. Further, suppose the decision-maker runs Algorithm 1 with $\mathtt{Alg} := \mathtt{DFM}$ using step-size $\eta_t = \frac{4}{\bar{\alpha}(t+1)}$, query radius $\delta < \min\{b, \frac{\bar{\alpha}}{L_{\mathtt{H}}}\}$, and the agents employ stochastic gradient play as $\mathcal{A}$ with $\rho \in [0,1)$ and $\sigma_{\mathtt{a}} \in (0,\infty)$. Suppose the agents run their algorithm stage-wise via Algorithm 2. In this case, set tolerance $\epsilon_\tau = \frac{1}{\delta(t+1)}$, constant $c = 16(\ell_*^2 d^2 + 1)$, and the epoch length to*

$$\tau = \sum_{k=0}^{K} T_k = \left\lceil \frac{1}{1-\rho^2} \cdot \log\left(\frac{2\bar{R}}{\epsilon_\tau}\right) \right\rceil + \sum_{k=1}^{K}\left\lceil \left(\frac{1}{1-2^{-k}\rho^2}\right)\log(4) \right\rceil, \tag{41}$$

*where $K = \left\lceil 1 + \log_2\left(\frac{\sigma_{\mathtt{a}}^2}{\epsilon_\tau \bar{L}_{\mathtt{a}}^2}\right) \right\rceil$. Then the following estimate holds:*

$$\mathbb{E}\|u_t - u^*\|^2 \leq \frac{\max\{2\bar{\alpha}^2\delta^2\|u_0 - u^*\|^2, c\}}{\delta^2\bar{\alpha}^2(t+1)^\beta} + 2\delta^2\left(\left(1 + \frac{L}{\bar{\alpha}}\right)\|u^*\| + \frac{L}{\bar{\alpha}}\right).$$

The proof is identical to that of Theorem J.6 and hence we omit it. Moreover, a corollary completely analogous to Corollary J.8 immediately follows.

### J.2.1. STRATEGIC DECISION-MAKER: DETERMINISTIC AGENT ALGORITHMS

We can also specialize to the case when the agents use a deterministic algorithm. In the latter case, much like the repeated gradient method, the agents do not need to run an stage based method.

**Proposition J.9.** *Suppose that Assumptions 2.1, 4.1, and 4.8 hold, and that we have available a constant $R > \|x_0 - x^*(u_0)\|$. Further, suppose the decision-maker runs Algorithm 1 with $\mathtt{Alg} := \mathtt{DFM}$ using step-size $\eta_t = \frac{4}{\bar{\alpha}(t+1)}$, query radius $\delta < \min\{b, \frac{\bar{\alpha}}{L_{\mathtt{H}}}\}$, and the agents employ a $\rho$–contracting algorithm $\mathcal{A}$ with $\rho \in [0,1)$ The agents employ deterministic algorithms (i.e., $\sigma_{\mathtt{a}} = 0$) and the decision-maker receives a noisy observation $\mathcal{A}(x_{t-1}, u_t) + \xi$ in each round where $\xi$ is zero mean and finite variance. In this case, set the epoch length such that $\tau \geq \log\left(\frac{2\delta L_\ell C_t(0)}{\sqrt{\eta_t}\bar{\alpha}\ell_* d}\right)\frac{1}{\log(1/\rho)}$, constant $c = 32\ell_*^2 d^2$ and agent tolerance $\epsilon_t = C_t(0)\rho^\tau$. Then the following estimate holds:*

$$\mathbb{E}\|u_t - u^*\|^2 \leq \frac{\max\{2\bar{\alpha}^2\delta^2\|u_0 - u^*\|^2, c\}}{\delta^2\bar{\alpha}^2(t+1)^\beta} + 2\delta^2\left(\left(1 + \frac{L}{\bar{\alpha}}\right)\|u^*\| + \frac{L}{\bar{\alpha}}\right).$$

*Proof.* The proof proceeds in a similar fashion to Theorem J.6.

Consider the error $\|u_{t+1} - u^*\|^2$. Add and subtract $u^\delta$, and apply the triangle inequality and Lemma J.2 to get the following estimate:

$$\frac{1}{2}\|u_{t+1} - u^*\|^2 \leq \|u_{t+1} - u^\delta\|^2 + \|u_\delta - u^*\|^2 \leq \|u_{t+1} - u^\delta\|^2 + \left(\frac{\delta L}{\bar{\alpha}} + \left(\frac{\delta L}{\bar{\alpha}} + \delta\right)\|u^*\|\right)^2.$$

Now, to bound the error $\|u_{t+1} - u^\delta\|^2$, we note by the nonexpansiveness of the projection mapping that

$$\mathbb{E}[\|u_{t+1} - u^\delta\|^2] \leq \mathbb{E}[\|u_t - u^\delta - \eta_t g_t\|^2]$$
$$\leq \mathbb{E}[\|u_t - u^\delta\|^2 - 2\eta_t\mathbb{E}\langle g_t, u_t - u^\delta\rangle + \eta_t^2\mathbb{E}\|g_t\|^2]$$
$$\leq \mathbb{E}[\|u_t - u^\delta\|^2 - 2\eta_t\mathbb{E}\langle\nabla\mathcal{L}_t^\delta(u_t), u_t - u^\delta\rangle + \eta_t^2\mathbb{E}\|g_t\|^2]$$

where we use the fact that $\mathbb{E}[g_t] = \nabla \mathcal{L}_t^\delta(u_t)$ in the last inequality, and the expectation is taken over the randomness in $\xi$ and $v_t$ up to time $t$.

Next, we add and subtract $\nabla \mathcal{L}^\delta(u_t)$ from the middle term to get that

$$\langle \nabla \mathcal{L}_t^\delta(u_t), u_t - u^\delta \rangle = \langle \nabla \mathcal{L}^\delta(u_t), u_t - u^\delta \rangle + \langle \nabla \mathcal{L}_t^\delta(u_t) - \nabla \mathcal{L}^\delta(u_t), u_t - u^\delta \rangle$$
$$\leq \frac{\bar{\alpha}}{2} \|u_t - u^\delta\|^2 + \langle \mathcal{L}_t^\delta(u_t) - \nabla \mathcal{L}^\delta(u_t), u_t - u^\delta \rangle,$$

where we have used the fact that $\mathcal{L}^\delta(x)$ is $(1-c)\bar{\alpha}$–strong convex with $c = 1/2$ (Lemma J.2). Hence, we deduce that

$$\mathbb{E}[\|u_{t+1} - u^\delta\|^2] \leq \mathbb{E}[\|u_t - u^\delta - \eta_t g_t\|^2],$$
$$\leq \mathbb{E}[\|u_t - u^\delta\|^2] - 2\eta_t \mathbb{E}\langle \nabla \mathcal{L}^\delta(u_t), u_t - u^\delta \rangle$$
$$- 2\eta_t \mathbb{E}\langle \nabla \mathcal{L}_t^\delta(u_t) - \nabla \mathcal{L}^\delta(u_t), u_t - u^\delta \rangle + \eta_t^2 \mathbb{E}\|g_t\|^2,$$
$$\leq (1 - \eta_t \bar{\alpha}) \mathbb{E}[\|u_t - u^\delta\|^2] - 2\eta_t \mathbb{E}\langle \nabla \mathcal{L}_t^\delta(u_t) - \nabla \mathcal{L}^\delta(u_t), u_t - u^\delta \rangle + \eta_t^2 \frac{\ell_*^2 d^2}{2\delta^2}.$$

Now, we bound the term $\langle \nabla \mathcal{L}_t^\delta(u_t) - \nabla \mathcal{L}^\delta(u_t), u_t - u^\delta \rangle$ and we apply Young's inequality[8] to this term to get that

$$\mathbb{E}|\langle \nabla \mathcal{L}_t^\delta(u_t) - \nabla \mathcal{L}^\delta(u_t), u_t - u^\delta \rangle| \leq \frac{1}{2\nu_1} \mathbb{E}\|\nabla \mathcal{L}_t^\delta(u_t) - \nabla \mathcal{L}^\delta(u_t)\|^2 + \frac{\nu_1}{2} \mathbb{E}\|u_t - u^\delta\|^2$$
$$\leq \frac{1}{2\nu_1} \left( L_\ell^2 (\rho^\tau C_t)^2 \right) + \frac{\nu_1}{2} \mathbb{E}\|u_t - u^\delta\|^2,$$

where we have that $\sigma_{\mathsf{a}} = 0$ so that

$$\bar{C}_t := \left( \rho^{t-1} \|x_0 - x^*(u_0)\| + \frac{L_{\mathsf{eq}} \eta_0 \ell_* d}{\delta(1-\rho)} \right).$$

Setting $\nu_1 := \bar{\alpha}/2$, we deduce that

$$\mathbb{E}\|u_{t+1} - u^\delta\|^2 \leq (1 - \eta_t \bar{\alpha}) \mathbb{E}[\|u_t - u^\delta\|^2] + \frac{\eta_t^2 \ell_*^2 d^2}{2\delta^2} + 2\eta_t \left( \frac{1}{2\Delta_1} \left( L_\ell^2 (\rho^\tau \bar{C}_t)^2 \right) + \frac{\Delta_1}{2} \|u_t - u^\delta\|^2 \right),$$
$$\leq \left( 1 - \frac{\bar{\alpha} \eta_t}{2} \right) \mathbb{E}[\|u_t - u^\delta\|^2] + \frac{\eta_t^2 \ell_*^2 d^2}{2\delta^2} + \frac{2\eta_t}{\bar{\alpha}} \left( L_\ell^2 (\rho^\tau \bar{C}_t)^2 \right).$$

Hence, if it is the case that

$$\frac{2\eta_t}{\bar{\alpha}} L_\ell^2 \rho^{2\tau} \bar{C}_t^2 \leq \frac{\eta_t^2 \ell_*^2 d^2}{2\delta^2}, \tag{42}$$

then we conclude

$$\mathbb{E}\|u_{t+1} - u^\delta\|^2 \leq \left( 1 - \eta_t \frac{\bar{\alpha}}{2} \right) \mathbb{E}[\|u_t - u^\delta\|^2] + \eta_t^2 \frac{\ell_*^2 d^2}{\delta^2} \tag{43}$$

Indeed, the bound in (42) is equivalent to

$$\tau \cdot \log(\rho^2) \leq \log \left( \frac{\eta_t \ell_*^2 d^2}{4\delta^2} \frac{\bar{\alpha}}{L_\ell^2 \bar{C}_t^2} \right) \iff \tau \geq \log \left( \frac{2\delta L_\ell \bar{C}_t}{\sqrt{\eta_t \bar{\alpha}} \ell_* d} \right) \frac{1}{\log(1/\rho)},$$

which is precisely the assumed bound on $\tau$.

Recall that $\eta_t = \frac{4}{\bar{\alpha}(t+1)}$. Hence we apply Lemma E.1 to obtain the final bound in this case. Indeed, we have that

$$\mathbb{E}\|u_t - u^*\|^2 \leq \frac{\max\{4\bar{\alpha}^2 \delta^2 \|u_0 - u^\delta\|^2, 32d^2 \ell_*^2\}}{\bar{\alpha}^2 \delta^2 (t+1)} + 2\delta^2 \left( \left( 1 + \frac{L}{\bar{\alpha}} \right) \|u^*\| + \frac{L}{\bar{\alpha}} \right) \tag{44}$$

as claimed. $\qquad \square$

A completely analogous corollary to Corollary H.4 directly follows.

---

[8]Young's inequality for inner product spaces says that for two vectors $u, v \in V$ where $V$ is an inner product space, we have $\langle u, v \rangle \leq \frac{\lambda}{2} \|u\|^2 + \frac{1}{2\lambda} \|v\|^2$ for any $\lambda > 0$.

## J.3. Strategic Decision-Maker: Non-Stationary Non-Strategic Environment

Recall the gradient deviation lemma (Lemma I.1) and the assumption on the environment distributions being Lipschitz continuous (Assumption I.2) from Appendix I.1. The following assumption implies a convex ordering on the non-strategic decision-dependent random variable on which the loss is dependent.

**Assumption J.10.** The probability measures $\mathcal{D}_e(u)$ and loss $\ell$ satisfy mixture dominance—i.e., for any $(u, x) \in \mathcal{U} \times \mathcal{X}$ and $\lambda \in [0, 1]$, the following inequality holds:

$$\mathop{\mathbb{E}}_{\xi \sim \mathcal{D}_e(\lambda v + (1-\lambda)w)} \ell(u, (x, \xi)) \leq \mathop{\mathbb{E}}_{\xi \sim \lambda \mathcal{D}_e(v) + (1-\lambda)\mathcal{D}_e(w)} \ell(u, (x, \xi)) \quad \text{for all } v, w \in \mathcal{U}.$$

Under these additional assumptions—namely Assumption I.2 and J.10—Theorem J.6 and Corollary J.8 immediately follow. The only change is to the gradient estimator and the Lipschitz constant for the gradient of the expected loss. Indeed, we have the modified costs

$$\mathcal{L}_t^\delta(u_t) = \frac{d}{\delta} \mathop{\mathbb{E}}_{v \sim \mathbb{B}^d} \left[ \mathop{\mathbb{E}}_{\xi \sim \mathcal{D}_e(u_t + \delta v_t)} [\ell(u_t + \delta v_t, (\mathcal{A}(x_t, u_t + \delta v_t), \xi))] \right]$$

and the modified smoothed expected loss at time $t$, and let

$$\mathcal{L}^\delta(u_t) = \frac{d}{\delta} \mathop{\mathbb{E}}_{v \sim \mathbb{B}^d} \left[ \mathop{\mathbb{E}}_{\xi \sim \mathcal{D}_e(u_t + \delta v_t)} [\ell(u_t + \delta v_t, (x^*(u_t + \delta v_t), \xi))] \right]$$

denote the smoothed expected risk. The smoothed expected risk is evaluated when the strategic agents are at the Nash equilibrium $x^*(u_t + \delta v_t)$ for the reported value $u_t + \delta v_t$. The estimate $\widehat{g}_t$ is an unbiased estimate of $\nabla \mathcal{L}_t^\delta$—i.e., $\mathbb{E}_{v \sim \mathbb{B}^d}[\widehat{g}_t] = \nabla \mathcal{L}_t^\delta(u_t)$. Further, we replace $L$ with $L := L_u + L_z(L_{\mathsf{eq}} + L_{\mathsf{en}})$.

