# OpenReview forum: "Finite-Time Convergence Rates in Stochastic Stackelberg Games with Smooth Algorithmic Agents"
_ICML.cc/2025/Conference — ICML 2025 poster_

### Official Review · Reviewer_dNEQ · 2025-03-05

**Overall Recommendation:** 4

**Summary:**

This paper studies the convergence problem of the stochastic Stackelberg games. Specifically, this paper makes the following contributions. It analyzes different scenarios that reflect the decision-maker's ability to reason about the agents' behavior via different estimates of how it could impact the gradient. The paper also conducts a thorough analysis of different drift-to-noise ratios in the game. Last but not least, it also analyzes the effect of induced drift.

**Claims And Evidence:**

Yes

**Essential References Not Discussed:**

N/A

**Experimental Designs Or Analyses:**

No empirical experiments.

**Methods And Evaluation Criteria:**

Yes

**Other Comments Or Suggestions:**

N/A

**Other Strengths And Weaknesses:**

Strength.
The theoretical analysis is thorough and solid. For example, in section 4, the paper conducts a theoretical analysis for the naive and strategic decision-makers.

Limitation.
Empirical experiments would significantly strengthen the paper.

**Questions For Authors:**

Have you tried to extend the proposed estimation to scenarios when the decision-maker could assume that the agent will also be strategic to the reverse direction of the decision-maker following certain constraints?

**Relation To Broader Scientific Literature:**

Yes, the problem abstraction (Stackelberg game) is deeply connected with the previous work.

**Theoretical Claims:**

Yes. I only checked the theoretical claims in the main paper, and they seem to be sound.

---

> ### Author Rebuttal · Authors · 2025-03-31
>
> Thank you for taking the time to review our paper. Below we respond to the topics you brought up in sections of the review.
>
> **Other Strengths And Weaknesses:**
> *Strength. The theoretical analysis is thorough and solid. For example, in section 4, the paper conducts a theoretical analysis for the naive and strategic decision-makers.
> Limitation. Empirical experiments would significantly strengthen the paper.*
>
> **Response**: Thank you for the acknowledgment of the theoretical strengths of the results. Regarding numerical experiments, we have included fairly extensive numerical experiments in the supplement. Due to page limitations and the theoretical nature of the paper, we ultimately decided to leave them to the supplement, however, given the additional page afforded by the final paper submission process, we would include some portion of the numerics in the main body.
>
> ---
>
> **Questions For Authors**:
> *Have you tried to extend the proposed estimation to scenarios when the decision-maker could assume that the agent will also be strategic to the reverse direction of the decision-maker following certain constraints?*
>
> **Response**: This is a very interesting question. We have not considered agents who are strategically adversarial against the decision-maker. This would go beyond the model and corresponding results we have in the paper. This is definitely challenging since one would want to know the unintended consequences (e.g., to which equilibrium do the agents and decision-maker converge if at all) and then design mitigation strategies on the part of the decision-maker. Also one would want to know what kinds of manipulation strategies are possible. This is an interesting future direction.

---

### Official Review · Reviewer_MjEv · 2025-03-06

**Overall Recommendation:** 2

**Summary:**

The paper studies the stochastic Stackelberg game and characterize the complex dynamics between the decision maker and learning agents. The drift, noise and optimization errors of the learning agents are decoupled and algorithms are proposed to control each components.

**Claims And Evidence:**

Yes

**Essential References Not Discussed:**

The paper should discuss existing works on online learning in Stackelberg games, e.g. Zhao et al 2023, Yu and Chen 2024 and their followup works.

Zhao, G., Zhu, B., Jiao, J., & Jordan, M. (2023, July). Online learning in stackelberg games with an omniscient follower. In International Conference on Machine Learning (pp. 42304-42316). PMLR.

Yu, Y., & Chen, H. (2024, July). Decentralized online learning in general-sum stackelberg games. In Proceedings of the Fortieth Conference on Uncertainty in Artificial Intelligence (pp. 4056-4077).

**Experimental Designs Or Analyses:**

Yes, I checked appendix D.

**Methods And Evaluation Criteria:**

Yes

**Other Comments Or Suggestions:**

While I appreciate the fact that the paper is very self-contained, I am not sure if the length of the paper will hinder the readability for ICML readers. It is hard to see such length is necessary for this work and the length of the paper  makes it hard for the reader to identify the technical contributions/ make followup contributions. I suggest to assume some appropriate prior knowledge of the readers and cite previous papers for relevant background (instead of restating them), e.g.  cite previous work that introduce examples of strongly monotone games, cite previous works for gradient estimators for bandit feedback.

**Other Strengths And Weaknesses:**

One strength is that the paper provides a general receipt for analyzing Stackelberg games. Different algorithmic approaches (naive decision maker and strategic decision maker) are provided. The assumptions (e.g. Definition 2.3) are also mild so that a large class of algorithms are included.

While the general framework provided by the paper is useful, it is not clear to me what are the technical contributions. For example, it is not clear whether Proposition 3.1 and Proposition 3.3 can be non-trivially extended from the analysis of Theorem 2 Duvocelle et al 2023. Or whether Theorem 4.9 is a mere combination of the analysis used in Lin et al. 2021 and Duvocelle et al 2023.

**Questions For Authors:**

1. Could you elaborate the technical contributions of this paper? Can we obtain the drift, noise and optimization error decomposition from existing works?
2. One significant difference between the Stackelberg game considered by this work and that considered by previous works (e.g. Zhao et al 2023, Yu and Chen 2024) is the introduction of the epoch length. While it seems reasonable (and realistic) to me introduce the epoch length, I am not sure whether the epoch length makes the problem much easier. As the epoch length is not controlled by the leader's algorithm, but rather a parameter for a centralized planner to choose, it seems that the epoch length would make the problem significantly easier. Due to this, I am also not sure if previous works did not choose to decompose the drift and noise because of this setting difference. Could you elaborate on the difference between the two settings?

I would be happy to raise my score if more discussion with existing papers on learning in Stackelberg games.

**Relation To Broader Scientific Literature:**

The paper studies the Stackelberg game and learning agents. The results helps advance the understanding of leader-follower dynamics in Stackelberg and and could have broader impacts on general machine learning algorithms that can be modeled with Stackelberg game dynamics.

**Theoretical Claims:**

No. But they seems to be valid to me.

---

> ### Author Rebuttal · Authors · 2025-04-01
>
> Thanks for the review. Given space constraints, we focus on related literature and technical novelties.
>
> **Referenced work**:
> The results don't exist in prior works, nor do they easily extend from analysis in referenced papers.
> - ZhaoEtAl2023, YuChen2024, different in game structure and theoretical approach, study **stationary** Stackelberg games with **side information** (i.e., omniscient follower) and are interested in manipulation strategies.  Additionally, YuChen2024 studies 2-player finite action games with bounded noisy rewards (i.e, $r_t\in[0,1]$).  Both study regret analysis for UCB style algorithms, i.e., very different than our setup where we study time-varying stochastic continuous action games with no side information. Yet, we will add these to related work.
> - [LinEtAl21], [DuvoEtAl23] study strongly monotone simultaneous play games and have no "leader" and thus a different eq concept; our Stackelberg game is not assumed to be strongly monotone (only the induced agent games $\mathcal{G}_u$ are):
>     -  [LinEtAl21] studies stationary games and employs self-concordant barrier function algorithm. We use a smoothed single point zeroth-order estimator for the DM. Hence the analysis doesn't apply to our algorithms.
>     - [DuvoEtAl23] employs an **asymptotic** analysis. We have a decision-maker **(DM)** which controls the sequence of games $\mathcal{G}_{u_t}$; they assume $\mathcal{G}\_t \to \mathcal{G}\_*$ or obtain asymptotic tracking results for Nash (!=Stack).
>
> **Specific Results**
> - **Prop 3.1, 3.3** don't follow from  the **asymptotic** analysis in [DuvoEtAl23], which doesn't imply a drift-to-noise decomposition (which is essential for our setting).  e.g., S1-S3 are asymptotic assumptions on stepsizes, gradient deviations (i.e. $R_{i,t}$), and bias and variance.  We provide a non-asymptotic analysis that determines the optimal error as a function of the drift-to-noise ratio (fundamentally different from [DuvoEtAl23]). This exposes how the DM can control the agents' dynamics via **choosing** $\tau$ and $\eta$.
>
> - **It is not possible for Thm 4.9 to result from analysis in [LinEtAl21], [DuvoEtAl23]**. We already noted differences to [LinEtAl21]. [DuvoEtAl23] have a similar zeroth-order alg (with the exception of a pivot point), yet, as noted they **rely on asymptotic assumptions**. These assumptions are not well-suited to our objective wherein one player aims to control the learning behavior of others in *finite time* and still converge to a Stack eq. (!=Nash). Their asymptotic convergence to Nash *assumes* $\mathcal{G}\_t\to\mathcal{G}\_{*}$ whereas for us the DM **controls** $\mathcal{G}\_t$ in finite time.
>
> **Technical Novelties**:
> - We study stochastic Stackelberg games s.t. the leader's problem is time-varying from follower agents' learning:  the agent eq $x_t^\star=x^\ast(u_t)$ are **induced by** DM actions (vs a random stablizing seq. as in [DuvoEtAl]).  Each induced $\mathcal{G}_u$ is strongly monotone but not necessarily the Stack game itself.
> - **Aim**: design an alg to control the drift-to-noise ratio to ensure finite time stabilization of agent behavior while converging to a static Stack eq (or PSE).: i.e., DM  chooses $\tau$ and $\eta$ to ensure finite time within epoch error $\epsilon_\tau \propto (\eta\sigma)^2$ and then control the noise. The epoch length (a DM design feature) **alleviates drift**;  while $\tau$ depends on agent regularity params, it can be estimated via adaptive algs as suggested in future work.
> -  To this end, we analyze $\Vert u_t-u^\star\Vert^2$ by decomposing  $\Vert x_{t}^\tau-x^\ast(u_t)\Vert^2$ into a stochastic optimization and a drift error.   The within epoch contraction gives
> $\mathbb{E}\Vert x_{t}^\tau-x^\ast(u_t)\Vert^2\lesssim \rho^{\tau}\mathbb{E}[\Vert x_{t}^0-x^\ast(u_t)\Vert^2]+\rho^2\sigma_a^2$.
> For the DM to set $\tau$ s.t. RHS$\lesssim \epsilon_\tau \propto \eta^2\sigma^2$, we bound the sequence of *epoch initial conditions* $\mathbb{E}[\Vert x_{t}^0-x^\ast(u_t)\Vert^2]$. The "trick" is to exploit the across epoch results (Sec 3) to bound
> $$\mathbb{E}\Vert x_{t}-x^\ast(u_t)\Vert^2\lesssim \left(1-\frac{1-\rho^2}{2}\right)^t\Vert x_{0}-x^\ast(u_0)\Vert^2+\frac{\sigma_a^2}{1-\rho^2}+\left(\frac{L_{eq}\Delta_u}{1-\rho^2}\right)^2, \text{where} \ \Delta_u:=\max_{s\leq t}\Vert u_{s}-u_{s-1}\Vert;$$  $\Delta_u$ is controlled by $\eta$.
> - Allowing for epoch-based algorithms doesn't make it easier: we conjecture it is necessary. To illustrate this point, see Rev GNfd on lower bounds and ref [1] therein.
>
> Overall, the approach is novel, not appearing in or simply derived from prior work, especially in Stackelberg games, to our knowledge.
> We exploit a drift-to-noise decomp (e.g., a need the [DuvoEtAl23] doesn't have as there is no leader) to design novel algs and prove convergence of $(u_t,x_t)$ to Stack eq.  **We will utilize the extra page to incorporate these comparisons and clarify technical novelties.**
>
> Happy to discuss further.

---

> > ### Comment · Reviewer_MjEv · 2025-04-02
> >
> > Thank you for your response. I understand that the epoch length is a design feature, which is used to alleviate drift, and the results are new. But within the epoch, the game is stationary, so I would expect prior methods on strongly monotone game to hold. It still seems like the problem is much simpler when you can control the epoch length, and I am not sufficiently convinced that the derivation of drift error is enough contribution.

---

> > > ### Author Response · Authors · 2025-04-03
> > >
> > > Thank you for your response!
> > >
> > > ---
> > >
> > > **$\tau$ lower-bound analysis**: we construct  randomly generated games that characterize the  necessity of $\tau$ for convergence with fixed step-sizes: https://imgur.com/qaDgEap and https://imgur.com/YHTalZO demonstrate the progression of system stability wrt $\tau$ for two different instances with the game structure described below. The top panel of each plot demonstrates that the evolution of $\tau$ is non-linear, and identifying the lower bound $\tau^*$, after which the system is stable, is **non-trivial and instance-dependent** as shown by the difference in the two plots. We expand more on this below.
> > >
> > > ---
> > >
> > > **$\tau$ is necessary for non-asymptotic rates**: without a monotonicity assumption on collective behavior, $\tau$ is required for providing non-asymptotic rates. Prior works on ZS Stack games (special case) have also shown theoretically that a time-scale separation is necessary for non-asymptotic rates [FiezEtAl21]. The hard question, which we address, is **what should $\tau$ be?**
> > >
> > > ---
> > >
> > > **Novelties**: Respectfully we reiterate that $\tau$ does not make the problem somehow simple. The technical challenge is constructing the decomposition as optimally as possible so we can control for the **across epoch** drift given the innate agent noise and the noise in the decision-maker's update.
> > >  - Given the necessity of $\tau$ for the non-asymptotic case, the question is **what should $\tau$ be**, especially in light of the objective of the decision-maker? Sec 3 determines the **optimal target accuracy** as a function of the drift (which DM will control) and noise, which consequently determines $\tau$. Sec 4 then optimizes these terms by designing $\eta$ wrt to $\tau$ to control the **size of the induced drift**.
> > >  - Indeed, as the reviewer notes, the drift-noise decomposition  is novel; this by itself provides **non-asymptotic rates** for time-varying strongly monotone games. This **extends [DuvoEtAl23]**, an entire paper on **asymptotic convergence** for the same class of games; our Sec 3 is a **non-asymptotic (non-trivial) extension that doesn't rely on their analysis**.
> > >  - Beyond these novelties,  the algs in Sec 4 are shown to converge **despite not having the typical monotonicity assumptions for the combined Stack. update**.
> > >  - And, *of course* results on strongly monotone games apply within epoch; we do not claim otherwise. In fact, our analysis allows for $\rho$-contracting updates beyond SGP (see Appendix). We exploit the regular behavior of this class to construct non-asymptotic convergence of the agent problem by constructing $\tau$ as a function of $\Delta_{\tt a}$, and an algorithm for the DM to control the joint behavior.
> > >
> > > ---
> > >
> > > **Related works that reinforce novelty**:
> > > - As for asymptotic rates (not the objective of our paper), it is not clear from any prior work that using asymptotic rates would work in the Stack. setting to obtain stationary asymptotic convergence, especially without the combined update being strongly monotone. It is our very strong conjecture that time scale separation is needed---meaning the sequence $\eta_t/\gamma_t\to \infty$ (cf [Ch 6, Borkar])---since the dynamics are general stochastic approximation updates.
> > > - Prior work on time-varying games focus on asymptotic rates and convergence to **Nash** under not just strong monotonicity but also asymptotic conditions on learning rates, such as asymptotic behavior of the bias and variance, and convergence to a stationary game, (e.g.,  [DuvoEtAl23]). We analyze finite-time convergence to the **Stack. eq.** without these assumptions (e.g. strong monotonicity).
> > >
> > > ---
> > >
> > > **Example**:
> > > Consider a two player game $(f_1,f_2)$ with $f_i:\mathbb{R}^{n\times m}\to \mathbb{R}$, with P1 the leader, and P2 the follower. Assuming SGP, the updates are $(x_1,x_2)^+=(x_1,x_2)-\eta g(x_1,x_2)$ where $g(x)=(g_1(x),\gamma/\eta D_2f_2(x))$ where $g_1:=D_1f_1$ for PSE and $g_1:=Df_1$ for Stack. For simplicity, let's take it to be quadratic (this is just a special class) with update $(x_1,x_2)^+=(I-\eta J)(x_1,x_2)$ where $J=[A, B; \frac{\gamma}{\eta}C,\frac{\gamma}{\eta} D]$ for matrices $(A,B,C,D)$. Then for some $\tau \in [0,\infty)$, the update is given by $(x_{1,t+1},x_{2,t+1})=(I-\eta J)(I-\eta V)^\tau (x_{1,t},x_{2,t})$ where $V=[0, 0; \frac{\gamma}{\eta}C,\frac{\gamma}{\eta} D]$.
> > > We generate examples such that
> > > - The equil. is not stable for $\tau=0$---i.e., $\text{spec}(I-\eta J)\not\subset [0,1]$ **for any $(\eta,\gamma)$** and $\text{Re}(\text{spec}(-J))>0$
> > > - The Stack. game is not strongly monotone---i.e., $\langle g(x)-g(y),x-y\rangle \leq 0$, and
> > > - There is a Stack. equil., i.e., $A-BD^{-1}C\succ 0$ and $D\succ 0$, (resp. a PSE  $A\succ 0$ and $D\succ 0$).
> > >
> > > Stability occurs at $\tau^*\geq 1$, where $\tau^*=\min$ {$\tau | \lambda((I-\eta J)(I-\eta V)^\tau)\subset [0,1)$}. We leave it to future work to understand this theoretically. Code: https://anonymous.4open.science/r/202504-StackExsTimeScale-F20D.

---

### Official Review · Reviewer_t2tf · 2025-03-14

**Overall Recommendation:** 2

**Summary:**

This paper focuses on learning in Stackelberg games under the setting where the agents might learn their equilibrium gradually. First, the author provides the equilibrium tracking error for the learning agents for a given sequence of the decision-maker’s actions. Then, a learning algorithm for the decision-maker, which can address this tracking error, is proposed. Convergence results for the proposed method are also provided, demonstrating the effectiveness of the approach.

**Claims And Evidence:**

The claims are generally supported by mathematical proofs.

**Essential References Not Discussed:**

The paper appropriately cites relevant prior works on learning in Stackelberg games.

**Experimental Designs Or Analyses:**

The experimental results are presented only in the supplementary material. While these results showcase the empirical performance of the proposed method, it is not clear if the selected benchmark problems are standard within the Stackelberg game community.

**Methods And Evaluation Criteria:**

The learning algorithm discussed in Corollary 4.5 and Proposition 4.6 involves three nested loops: super-epoch, epoch, and iteration for the agents. This triple-loop structure could potentially lead to slow convergence in practical applications. It would be beneficial to include experimental results in the main body of the paper to demonstrate the practical convergence rate and effectiveness of the proposed algorithm.

**Other Comments Or Suggestions:**

Please see “Theoretical Claims”.

**Other Strengths And Weaknesses:**

My main concerns and questions are outlined in “Theoretical Claims”.

**Questions For Authors:**

Please see “Theoretical Claims”.

**Relation To Broader Scientific Literature:**

The setting where the agents might learn and thus the objective function of the decision-maker can be time-varying seems novel.

**Theoretical Claims:**

The paper utilizes techniques from variational inequalities and stochastic approximation to establish the theoretical results.  However, some aspects of the theoretical analysis are unclear:

- In Section 2.2, the agents are assumed to update their actions for $\tau$ iterations. However, Propositions 3.1 and 3.2 seem independent of $\tau$, and the proof of Proposition H.2 utilizes an inequality that differs slightly from Definition 2.3, $\mathbb{E}\_t\\|x_{t+1} - x_t^{\ast}\\|^2 \leq \rho^2 \\|x_t - x_t^{\ast}\\| + \rho^2 (c\sigma_a)^2.$ Did the author assume $\tau=1$ in Proposition 3.1?
- Similarly, the (SPG) algorithm appears to update $x_t$ over $t$ rather than $k$. Clarification on the update rules and how they align with the theoretical results would strengthen the paper.
- Regarding Proposition 3.3, it is unclear whether it holds for any $\varepsilon$ or under specific conditions. Providing more details on this point would enhance the clarity of the theoretical contributions.

---

> ### Author Rebuttal · Authors · 2025-03-31
>
> Thank you for taking the time to review our paper, and pointing out some of the novelties in your review. Below we focus on the key queries in your review which we believe should address your concerns.   **Please let us know if there are further clarifications**.
>
> **Theoretical Claims:** We label each of these points 1,2,3.
>
> **Re 1.** For clarity, Def 2.3 is stating that **within an epoch** (i.e. since $u$ is fixed and therefore $t$ is fixed for the decision-maker), the agents contract at a rate $\rho$ to the equilibrium $x^*(u)$ of the induced game. That is $k$ here is indexing the steps within the epoch, and $t$ is the fixed index of the epoch.
>
> Now, we reflect on Prop 3.1 and *Cor* 3.2 and Prop. H.2: These are **across epoch results**. The confusion we believe comes from the fact that $\rho^\tau\leq \rho$ for $\tau\geq 0$ and $\rho\in(0,1]$. We have applied this to arrive at $\mathbb{E}\_t\Vert x_{t+1}-x_t^\ast\Vert^2\leq \rho^2\Vert x_t-x_t^\ast\Vert^2+\rho^2(c\sigma_a)^2$ from the expression in Def 2.3.
>
> In Section G.1 of the supplement, we show that a "one-step" within epoch contraction translates to a "one-step" across epoch contraction; here is where we use the fact that $\rho^t\leq \rho$ for any $t\in (0,1]$. Note that in this section, since it is stated for general update rules (independent of the existence of a decision-maker choosing $u$'s that influence the behavior), we have dropped the $k$ notation and are just using $t$ for an iteration. For clarity, the statements in G.1 would look like the following: fix epoch $t$ and action $u_t$, then
> $$\mathbb{E}[\Vert x_t^\tau-x^*(u_t)\Vert^2] \leq \rho^{2\tau}\Vert x_t^0-x^*(u_t)\Vert^2+c^2\sigma_a^2\frac{\rho^2}{1-\rho^2}\leq \rho^{2}\Vert x_t^0-x^*(u_t)\Vert^2+c^2\sigma_a^2\frac{\rho^2}{1-\rho^2}$$
>      Coming to the statement at the beginning of Proposition H.2: we have that there is a $\rho$ and $c$ such that $$\mathbb{E}\Vert x_{t+1}-x_t^\ast\Vert^2\leq \rho^2 \mathbb{E}\Vert x_t-x_t^\ast\Vert^2+c^2\cdot(\rho\sigma_a)^2\quad\text{where}\ \ x_t^\ast := x^*(u_t)$$
>      We were a little loose with the notation between the one-step within epoch and one-step across epoch results, and will update the statement (Prop H.2) in the appendix to make this clear; for instance, if Definition 2.3 holds for some say $c_1$, then the above across epoch inequality holds for  any constant $c$ satisfying $c^2\geq c_1^2/(1-\rho^2)$ -- e.g. for stochastic gradient play $c_1^2=2\gamma^2$ and $c^2=2c_1^2$. In particular, its just the constant $c$ that changes between the two.  Formally, nothing changes in the proposition and corollary in Section 3 as they are stated using the notation $\lesssim$ which absorbs constant scalings; though we will adjust this to make the relationship more clear in the main body.
>
> **Re 2:** As noted above, this is just context switching: since the results in Appendix G hold independent of the decision maker's existence, we dropped the epoch notation and are only expressing things in terms of iterations. That being said, we can switch the $t$ to a $k$ to make it more clear and add a note, as well as making the other suggested changes for clarity.
>
> **Re 3**: As noted in the paper, this is an informal statement stated as such for brevity. As we state below Prop 3.3, the formal statement is given in Prop H.5 which details all the assumptions. Given the context of the subsection, the statement holds under the assumption that the agents are running a $\rho$ contracting algorithm, and further (as detailed in Prop H.5) under assumptions on the constants of the decision-maker's algorithm. We will add clarifications to this point.
>
> Hopefully this clarifies the questions on notation and assumptions. Please also see the response to Reviewer MjEv for additional discussion/clarification on technical novelties.
>
> ---
>
> **Methods And Evaluation Criteria & Experimental Designs Or Analyses:**
>
> **Response:**
>
> - We will use the extra page in the final version to incorporate a subset of the numerics in the main body that explore the main theoretical results and their assumptions.
> - All the examples of monotone games (Kelly Auctions, quadratic games, and ride-sharing) used in experiments have appeared before in the prior literature, which we cite, just in different experiments. The nonlinear examples are intended to explore the boundaries of the theoretical results. That being said, we still utilize the Kelly Auction for the agent game, but incorporate non-convex social costs which are common in the literature (including social welfare and revenue maximization).

---

### Official Review · Reviewer_ffEi · 2025-03-17

**Overall Recommendation:** 3

**Summary:**

This paper proposes an algorithm for a game in which there is one leader and $n$ followers. The leader chooses an action and the followers play a Nash equilibrium which is influenced by the leader's action. The solution concept is that of  a Stackelberg equilibrium. The game is static but is repeated during the algorithm. The main challenge is the fact that the agents are adjusting their behavior to the leader's actions. The authors provide convergence guarantees for their algorithm.

**Claims And Evidence:**

The claims and the proofs are relatively clear.

**Essential References Not Discussed:**

Not as far as I know.

**Experimental Designs Or Analyses:**

I checked the ones in the main body. They seem fine to me.

**Methods And Evaluation Criteria:**

Yes, they make sense.

**Other Comments Or Suggestions:**

Nothing specific. There are a few typos such as "inquality".

**Other Strengths And Weaknesses:**

Nothing specific.

**Questions For Authors:**

1) Can you comment on the difference between Definition 4.2 and the notion of Stackelberg equilibrium on page 4 (left column)? How is this definition different and why is it necessary to introduce it?

2) Can you please provide sufficient conditions to guarantee that Assumption 4.3 holds?

**Relation To Broader Scientific Literature:**

It seems fine to me.

**Theoretical Claims:**

I did not check the proofs in detail but the ideas seem fine.

---

> ### Author Rebuttal · Authors · 2025-03-31
>
> Thank you for taking the time to review our paper. Below we respond to your questions. Please let us know if this has clarified things.
>
> **Questions For Authors:**
>
> 1. *Can you comment on the difference between Definition 4.2 and the notion of Stackelberg equilibrium on page 4 (left column)? How is this definition different and why is it necessary to introduce it?*
>
>     **Response**: The difference is that the decision maker is not optimizing through the dependence of $u$ in the agents best response $x^*(u)$; instead it is evaluated a the fixed point of the given $\arg\min$ expression. We introduce this because it is the natural equilibrium concept when the decision-maker does not optimize through  $x^*(u)$, but rather performs updates relative to the gradient of $\ell(u,z)$ (where $z\sim \mathcal{D}(u)$) with respect to the explicit dependence on $u$ -- namely stochastic samples of $\mathbb{E}\_{z\sim \mathcal{D}(u)}\nabla_u\ell(u,z)$ instead of the full gradient $\mathbb{E}\_{z\sim \mathcal{D}(u)}\nabla_u\ell(u,z)+\frac{d}{dv}\mathbb{E}\_{z\sim \mathcal{D}(v)}\ell(u,z)|_{u=v}$ as detailed in Section 2.3 (cf Equation (2) and discussion thereafter) as well as the discussion at the top of Section 4.1.
>
> 2. *Can you please provide sufficient conditions to guarantee that Assumption 4.3 holds?*
>
>     **Response**: This is a standard assumption in optimization for online stochastic gradient methods in games and otherwise (see [1--3]). For example, the assumption says that stochastic gradient methods that use gradient estimates $g_t=\nabla_u \ell(u_t,z_t)$ satisfy
>     $$\mathbb{E}_t[\Vert\nabla_u \ell(u_t,z_t)-\mathbb{E}\_{t}[g_t]\Vert^2]\leq \sigma^2<\infty,\quad\text{where}\ \mathbb{E}_t[\cdot\mid \mathcal{F}_t]$$ i.e., the estimator has finite variance conditioning on all the previously seen data (which is the filtration part in Assumption 4.3). This is just stating that the gradient noise has bounded variance, given data to time $t$.    One sufficient condition for this is that the environment distribution from which we sample $g_t$  has finite variance -- i.e. $\mathbb{E}[\Vert g_t\Vert^2]<\infty$.
>     As is standard the size of the variance can be controlled via stepsize choices, batching amongst other techniques.
>
> [1] Cutler et al "Stochastic Optimization under Distributional Drift" JMLR 2023
> [2] Narang et al "Multiplayer Performative Prediction: Learning in Decision-Dependent Games" JMLR 2023
> [3] Besbes et al "Non-stationary Stochastic Optimization", 2014

---

> > ### Comment · Reviewer_ffEi · 2025-04-05
> >
> > Thank you for your answers.

---

### Official Review · Reviewer_GNfd · 2025-03-23

**Overall Recommendation:** 3

**Summary:**

The paper explores a learning problem in the context of Stackelberg games with single leader (decision-maker) and multiple followers (agents). The authors consider a setting where both the agents and the decision-maker are learning. The agents are learning, for any action taken by the decision-maker, the Nash equilibrium of the game induced by such an action. The decision-maker aims at playing a sequence of actions ensuring the convergence to a Stackelberg equilibrium. Both the algorithm employed by the agents and the stochastic environment are unknown to the decision-maker.

First, the authors characterize the equilibrium tracking error of the agents, highlighting the contribution of the noise and the "drift" induced by the sequence of leader's actions. Then, they analyze the convergence rate of a simple decision-maker that ignores the fact that the agents are responding to it. Finally, they devise a more powerful decision-maker which adopts a derivative-free method that converges in $O(1/\epsilon^2)$ to an $\epsilon$-approximate equilibrium.

**Claims And Evidence:**

The claims in main paper are supported by proofs in the Appendix, albeit I did not check their correctness. The results are plausible given the assumptions, and the high-level idea of the algorithms employed is convincing.

**Essential References Not Discussed:**

The authors discuss all the relevant related works.

**Experimental Designs Or Analyses:**

I did not check the validity of the experiments.

**Methods And Evaluation Criteria:**

There are no typical benchmarks for this setting. Nonetheless, the authors provide experiments on some typical economics settings (e.g., Kelly auctions) and a quadratic game inspired by real-world data. They also investigate the performance of their algorithms in settings where the theoretical assumptions are not perfectly met.

**Other Comments Or Suggestions:**

Suggestion: after Corollary 4.5, the authors discuss the fact that the total number of iterations is $T\tau$. To complete this discussion, it would be useful to provide the formula to compute $\tau$ (up to constants) given the target accuracy $\epsilon$. The same holds for Corollary 4.10.

**Other Strengths And Weaknesses:**

Strengths: the paper addresses a novel learning problem, where both the leader and the followers are learning in a Stackelberg game. They provide an in-depth analysis of two different algorithms for the leader's problem and briefly discuss when one approach is preferable to the other.

Weaknesses: it is unclear what exactly the decision-maker has to learn, and what they know in advance (see questions).
The authors do not provide any lower bound on the convergence rate (in terms of iterations) achievable in this problem.

**Questions For Authors:**

1) Does the decision-maker know the agent's algorithm? Line 160 (right) states that the algorithm is unknown, yet it is employed at line 396 left. Similarly, is the noise distribution $\mathcal{D}_e$ known in advance?
2) At line 138 left, why is $x$ drawn from a distribution $\mathcal{D}_x(u)$, rather than being computed by the agent's algorithm based on both $u$ and the previous $x$?
3) At line 117 right, why does $x^{k+1}_{t+1}$ depend on the sequence of joint actions $x^1\_t,\dots, x^k_t $ rather than the sequence $x^\tau\_{t}, x^1\_{t+1}, \dots, x^k\_{t+1}  $?
4) Line 67 (right) states that the epoch complexity of your algorithms is optimal. Is it also optimal in terms of iterations?
6) Is the notion of performatively stable equilibrium  in Definition 4.2 equivalent to the one of (Narang et al., 2023)? Why is it the appropriate notion of equilibrium for Section 4.1?

**Relation To Broader Scientific Literature:**

Stackleberg games have been extensively studied in offline settings. This work considers a learning setting, and is built upon previous works on Stochastic Optimization, especially the work of Cutler et al. (2023). While in (Cutler et al. 2023) the loss function depends only on the decision-maker action, this paper considers also the presence of other strategic agents that influence the loss.

**Theoretical Claims:**

I did not check the correctness of the proofs.

---

> ### Author Rebuttal · Authors · 2025-03-31
>
> Thanks for reviewing our paper, and acknowledging some of the strengths of the paper.
>
> **Qs for Authors:**
>
> **Re1.**: No, the decision-maker **(DM)** doesn't know the algorithm or $\mathcal{D}\_e$ *a priori*.
> - **Algorithm**:  In Line 396, DM doesn't know $\mathcal{A}$. From the environment, it receives a sample of $\ell$ evaluated at the agents' response. In practice this means that the DM deploys action $u_t+\delta v_t$, observes $(\mathcal{A}(x_{t-1}, u_t+\delta v_t),\xi_t)$ and then computes $\ell(u_t+\delta v_t,(\mathcal{A}(x_{t-1}, u_t+\delta v_t),\xi_t))$ so as to compute $g_t$. It only needs to know its own loss and be able to query the environment.
> - **Environment**: Analogous to the above point, the decision-maker doesn't know $\mathcal{D}\_e$. We assume that they can stochastically query it.
> We will add further discussion to clarify.
>
> **Re 2.** Here, we are defining the problem the DM aims to solve if the the agents were not responding to $u_t$ but were at equilibrium. This forms the **in equilibrium** benchmark: i.e., the point to which the DM aims to converge depends on the environment stochasticity even when the agents are stochastically best responding. We show convergence to this stationary equilibrium so that we don't need a time-varying benchmark like many of the existing works on time-varying games. We exploit problem structure to get *stronger* convergence guarantees as compared to bounding stochastic tracking errors (e.g., we bound $\Vert u_t-u^\ast\Vert^2$ vs $\Vert u_t-u_t^\ast\Vert^2$ where $\{u_t^\ast\}$ is some time-varying "optimal" sequence given the drift induced by the agents).
>
> **Re 3.** Thanks for catching this: it is a typo that should read $x\_{i,t}^{k+1}=\mathcal{A}\_i(x_t^0,x_t^1,\ldots, x_t^k,u_t)$. Here, we should have the notation $x_{i,t}^{k+1}=\widetilde{\mathcal{A}}\_i(x_t^0,x_t^1,\ldots, x_t^k,u_t)$ where $\widetilde{A}\_i$ are $\rho$ contracting algorithms, and then define $x_t=\mathcal{A}(x_{t}^0,u_t)$ as the $\tau$ time joint algorithm since in the analysis that follows we treat $\mathcal{A}$ as the $\tau$ time response of the agents. We will update this notation to be consistent.
>
> **Re 4.** We are not aware of any lower bounds in terms of iterations for Stackelberg games employing epoch-based algorithms, and so it is not clear if it is optimal in terms of total iterations. We discuss lower bounds further below.
>
> **Re 5.** It is equivalent when Def 4.2 is restated in terms of the lifted $(n+1)$ player simultaneous play game as noted in the paper; e.g.,  consider $(\mathbb{E}_{\xi \sim \mathcal{D}_e(u^{ps})}\ell(u,(x,\xi)), f_1(x,u), \ldots, f_n(x,u))$ over the joint action $(u,x)\in \mathcal{U}\times \mathcal{X}$, then the performatively stable eq (PSE) $u^{ps}$ is a Nash eq of this game where the environment noise is fixed at $u^{ps}$. The difference between this equilibrium and Stackelberg (SE) is two-fold:  the DM is not optimizing through the dependence of $u$ in  the agents' best response $x^*(u)$ as in a SE, nor in the environment distribution $\mathcal{D}\_e(u)$. This is why it is the natural equilibrium concept for Sec 4.1  -- i.e., the DM is only using stochastic queries of $\mathbb{E}\_{z\sim \mathcal{D} (u)}$ $\nabla_u\ell(u,z)$ instead of the full gradient  $\mathbb{E}\_{z\sim \mathcal{D} (u)}$ $\nabla_u\ell(u,z)+$ $\frac{d}{dv}\mathbb{E}\_{z\sim \mathcal{D}(v)}\ell(u,z)$ as detailed in Sec 2.3  and the discussion at the top of Sec 4.1. To find an optimal point for $\mathbb{E}\_{z\sim \mathcal{D}(u)}\ell(u,z)$ the DM needs to optimize through $z(u)=(x^\ast(u),\xi(u))\sim \mathcal{D}_x(u)\times \mathcal{D}_e(u)$. Yet, it doesn't know the agents' preferences nor the environment distribution, but can query it. Sec 4.2, on the other hand, aims to estimate $\frac{d}{dv}\mathbb{E}\_{z\sim \mathcal{D}(v)}\ell(u,z)$ so that SE is the right benchmark.
>
> **Re: "Other... Weaknesses":**
> - **Lower Bounds** are open and interesting, but beyond the scope. We comment on a special sub-case:
>     - In zero-sum settings, which is a special case of our problem, the ODE method in stochastic approximation provides some lower bounds. The continuous-time limit translates to a time-scale separation $\tau$ [1]. Here, there are lower bounds in terms of eigenvalues of the game Jacobian block matrices; this could be any $\tau>0$. It's possible to extend this to general sum settings, however, the game Jacobian doesn't have a nice structure like zero-sum games so it's hard to meaningfully interpret these bounds. This suggests, however, that $\tau>0$ is necessary and that its precise value is problem dependent.
>
> Thanks for the suggestions; we will incorporate them.
>
> - [1] Fiez & Ratliff. "Local Convergence Analysis of Gradient Descent Ascent with Finite Timescale Separation", ICLR 2021

---

### Decision · Program_Chairs · 2025-05-01

**Decision:**

Accept (poster)

**Comment:**

The paper explores a learning problem in the context of Stackelberg games with single leader (decision-maker) and multiple followers (agents). The authors consider a setting where both the agents and the decision-maker are learning. The authors characterize the equilibrium tracking error of the agents, highlighting the contribution of the noise and the "drift" induced by the sequence of leader's actions. Then, they analyze the convergence rate of a simple decision-maker that ignores the fact that the agents are responding to it. Finally, they devise a more powerful decision-maker which adopts a derivative-free method.

The paper analyzes a novel setting and provides an in-depth analysis of the proposed algorithms. The main idea is clear. Anyway, the exposition is sometimes confusing and can be improved so that the readers can get the main message and understand the technical merits from the main pages, leading to a not full support for acceptance. Still, I also dive in the details of the paper, and in my opinion the merits of the paper overcome its drawbacks, and for this reason I feel myself confident in suggesting acceptance.